# Modern Hopfield Networks and Attention for Immune Repertoire Classification

**Michael Widrich**[*]   **Bernhard Schäfl**[*]   **Milena Pavlović**[†,‡]   **Hubert Ramsauer**[*]

**Lukas Gruber**[*]   **Markus Holzleitner**[*]   **Johannes Brandstetter**[*]   **Geir Kjetil Sandve**[‡]

**Victor Greiff**[†]                **Sepp Hochreiter**[*,§]

**Günter Klambauer**[*]
[*] ELLIS Unit Linz and LIT AI Lab,
Institute for Machine Learning,
Johannes Kepler University Linz, Austria
[†] Department of Immunology, University of Oslo, Norway
[‡] Department of Informatics, University of Oslo, Norway
[§] Institute of Advanced Research in Artificial Intelligence (IARAI)

## Abstract

A central mechanism in machine learning is to identify, store, and recognize patterns. How to learn, access, and retrieve such patterns is crucial in Hopfield networks and the more recent transformer architectures. We show that the attention mechanism of transformer architectures is actually the update rule of modern Hopfield networks that can store exponentially many patterns. We exploit this high storage capacity of modern Hopfield networks to solve a challenging multiple instance learning (MIL) problem in computational biology: immune repertoire classification. In immune repertoire classification, a vast number of immune receptors are used to predict the immune status of an individual. This constitutes a MIL problem with an unprecedentedly massive number of instances, two orders of magnitude larger than currently considered problems, and with an extremely low witness rate. Accurate and interpretable machine learning methods solving this problem could pave the way towards new vaccines and therapies, which is currently a very relevant research topic intensified by the COVID-19 crisis. In this work, we present our novel method DeepRC that integrates transformer-like attention, or equivalently modern Hopfield networks, into deep learning architectures for massive MIL such as immune repertoire classification. We demonstrate that DeepRC outperforms all other methods with respect to predictive performance on large-scale experiments including simulated and real-world virus infection data and enables the extraction of sequence motifs that are connected to a given disease class. Source code and datasets: *https://github.com/ml-jku/DeepRC*

## 1   Introduction

Transformer architectures (Vaswani et al., 2017) and their attention mechanisms are currently used in many applications, such as natural language processing (NLP), imaging, and also in multiple instance learning (MIL) problems (Lee et al., 2019). In MIL, a set or bag of objects is labelled rather than objects themselves as in standard supervised learning tasks (Dietterich et al., 1997). Examples for MIL

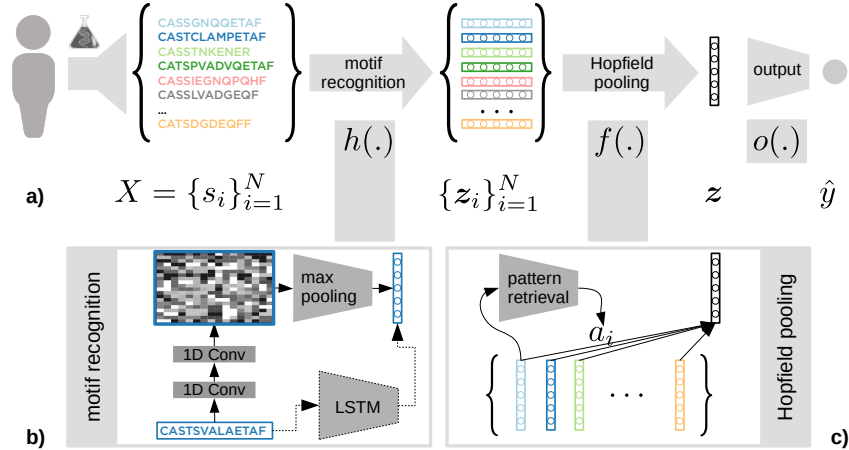

Figure 1: Schematic representation of the DeepRC approach. **a)** An immune repertoire $X$ is represented by large bags of immune receptor sequences (colored). A neural network (NN) $h$ serves to recognize patterns in each of the sequences $s_i$ and maps them to sequence-representations $z_i$. A pooling function $f$ is used to obtain a repertoire-representation $z$ for the input object. Finally, an output network $o$ predicts the class label $\hat{y}$. **b)** DeepRC uses stacked 1D convolutions for a parameterized function $h$ due to their computational efficiency. Potentially, millions of sequences have to be processed for each input object. In principle, also recurrent neural networks (RNNs), such as LSTMs (Hochreiter et al., 2007), or transformer networks (Vaswani et al., 2017) may be used but are currently computationally too costly. **c)** Hopfield-pooling is used to obtain a repertoire-representation $z$ for each input object. For this, DeepRC uses weighted averages of sequence-representations, where the weights are determined by an update rule of modern Hopfield networks that allows to retrieve exponentially many patterns.

problems are medical images, in which each sub-region of the image represents an instance, video classification, in which each frame is an instance, text classification, where words or sentences are instances of a text, point sets, where each point is an instance of a 3D object, and remote sensing data, where each sensor is an instance (Carbonneau et al., 2018; Uriot, 2019). Attention-based MIL has been successfully used for image data, for example to identify tiny objects in large images (Ilse et al., 2018; Pawlowski et al., 2019; Tomita et al., 2019; Kimeswenger et al., 2019) and transformer-like attention mechanisms for sets of points and images (Lee et al., 2019).

However, in MIL problems considered by machine learning methods up to now, the number of instances per bag is in the range of hundreds or few thousands (Carbonneau et al., 2018; Lee et al., 2019) (see also Suppl. Tab. A3). At the same time the witness rate (WR), the rate of discriminating instances per bag, is already considered low at $1\% - 5\%$. We will tackle the problem of *immune repertoire classification* in which large bags of immune receptor sequences have to be classified. This problem is characterized by hundreds of thousands of instances per bag without instance-level labels and with extremely low witness rates down to $0.01\%$. We will show that the attention mechanism of transformers is the update rule of modern Hopfield networks (Krotov & Hopfield, 2016, 2018; Demircigil et al., 2017) that are generalized to continuous states in contrast to classical Hopfield networks (Hopfield, 1982) (see Suppl. B). These novel continuous state Hopfield networks allow to store and retrieve exponentially (in the dimension of the association space) many patterns (see Section 2). Thus, modern Hopfield networks with their update rule allow us to approach MIL problems with large numbers of instances per bag, such as immune repertoire classification.

Immune repertoire classification, i.e. predicting the immune status based on the immune repertoire sequences, is essentially a text-book example for a *multiple instance learning* problem (Dietterich et al., 1997; Maron & Lozano-Pérez, 1998; Wang et al., 2018). Briefly, the immune repertoire of an individual consists of an immensely large bag of immune receptors, represented as amino acid sequences. Usually, the presence of only a small fraction of particular receptors determines the immune status with respect to a particular disease (Christophersen et al., 2014; Emerson et al., 2017). Therefore, classification of immune repertoires bears a high difficulty since each immune repertoire can contain millions of sequences as instances with only a few indicating the class. Further properties of the data that complicate the problem are: (a) The overlap of immune repertoires of

different individuals is low (in most cases, maximally low single-digit percentage values) (Greiff et al., 2017; Elhanati et al., 2018), (b) multiple different sequences can bind to the same pathogen (Wucherpfennig et al., 2007), and (c) only subsequences within the sequences determine whether binding to a pathogen is possible (Dash et al., 2017; Glanville et al., 2017; Akbar et al., 2019; Springer et al., 2020; Fischer et al., 2019). In summary, immune repertoire classification can be formulated as multiple instance learning with an extremely low witness rate and large numbers of instances, which represents a challenge for currently available machine learning methods. Furthermore, the methods should ideally be interpretable, since the extraction of class-associated sequence motifs is desired to gain crucial biological insights.

The acquisition of human immune repertoires has been enabled by immunosequencing technology (Georgiou et al., 2014; Brown et al., 2019) which allows to obtain the immune receptor sequences and immune repertoires of individuals. Each individual is uniquely characterized by their immune repertoire, which is acquired and changed during life. This repertoire may be influenced by all diseases that an individual is exposed to during their lives and hence contains highly valuable information about those diseases and the individual's immune status. Immune receptors enable the immune system to specifically recognize disease agents or pathogens. Each immune encounter is recorded as an immune event into immune memory by preserving and amplifying immune receptors in the repertoire used to fight a given disease. This is, for example, the working principle of vaccination. Each human has about $10^7$–$10^8$ unique immune receptors with low overlap across individuals and sampled from a potential diversity of $> 10^{14}$ receptors (Mora & Walczak, 2019). The ability to sequence and analyze human immune receptors at large scale has led to fundamental and mechanistic insights into the adaptive immune system and has also opened the opportunity for the development of novel diagnostics and therapy approaches (Georgiou et al., 2014; Brown et al., 2019).

Immunosequencing data have been analyzed with computational methods for a variety of different tasks (Greiff et al., 2015; Shugay et al., 2015; Miho et al., 2018; Yaari & Kleinstein, 2015; Wardemann & Busse, 2017). A large part of the available machine learning methods for immune receptor data has been focusing on the individual immune receptors in a repertoire, with the aim to, for example, predict the antigen or antigen portion (epitope) to which these sequences bind or to predict sharing of receptors across individuals (Gielis et al., 2019; Springer et al., 2020; Jurtz et al., 2018; Moris et al., 2019; Fischer et al., 2019; Greiff et al., 2017; Sidhom et al., 2019; Elhanati et al., 2018). Recently, Jurtz et al. (2018) used 1D convolutional neural networks (CNNs) to predict antigen binding of T-cell receptor (TCR) sequences (specifically, binding of TCR sequences to peptide-MHC complexes) and demonstrated that motifs can be extracted from these models. Similarly, Konishi et al. (2019) use CNNs, gradient boosting, and other machine learning techniques on B-cell receptor (BCR) sequences to distinguish tumor tissue from normal tissue. However, the methods presented so far predict a particular class, the epitope, based on a single input sequence.

Immune repertoire classification has been considered as a MIL problem in the following publications: A Deep Learning framework called DeepTCR (Sidhom et al., 2019) implements several Deep Learning approaches for immunosequencing data. The computational framework, inter alia, allows for attention-based MIL repertoire classifiers and implements a basic form of attention-based averaging. Ostmeyer et al. (2019) already suggested a MIL method for immune repertoire classification. This method considers 4-mers, fixed sub-sequences of length 4, as instances of an input object and trained a logistic regression model with these 4-mers as input. The predictions of the logistic regression model for each 4-mer were max-pooled to obtain one prediction per input object. This approach is characterized by (a) the rigidity of the k-mer features as compared to convolutional kernels (Alipanahi et al., 2015; Zhou & Troyanskaya, 2015; Zeng et al., 2016), (b) the max-pooling operation, which constrains the network to learn from a single, top-ranked k-mer for each iteration over the input object, and (c) the pooling of prediction scores rather than representations (Wang et al., 2018). Our experiments also support that these choices in the design of the method can lead to constraints on the predictive performance (see Table 1).

Our proposed method, DeepRC, also uses a MIL approach but considers sequences rather than k-mers as instances within an input object and a transformer-like attention mechanism. DeepRC sets out to avoid the above-mentioned constraints of current methods by (a) applying transformer-like attention-pooling instead of max-pooling and learning a classifier on the repertoire- rather than on the sequence-representation, (b) pooling learned representations rather than predictions, and (c) using less rigid feature extractors, such as 1D convolutions or LSTMs. *In this work, we contribute the following:* We demonstrate that continuous generalizations of binary modern Hopfield-networks (Krotov &

Hopfield, 2016, 2018; Demircigil et al., 2017) have an update rule that is known as the attention mechanisms in the transformer. We show that these modern Hopfield networks have exponential storage capacity, which allows them to extract patterns among a large set of instances (Section 2). Based on this result, we propose DeepRC, a deep MIL method based on modern Hopfield networks for large bags of complex sequences, as they occur in immune repertoire classification (Section 3). We evaluate the predictive performance of DeepRC and other machine learning approaches for the classification of immune repertoires in a large comparative study (Section 4).

## 2    Exponential storage capacity of continuous state modern Hopfield networks with transformer attention as update rule

In this section, we show that modern Hopfield networks have exponential storage capacity, which will later allow us to approach massive multiple instance learning problems. We assume *stored patterns* as fixed-size vectors $\boldsymbol{x}_1, \ldots, \boldsymbol{x}_N \in \mathbb{R}^d$ that are stacked as columns to the matrix $\boldsymbol{X} = (\boldsymbol{x}_1, \ldots, \boldsymbol{x}_N)$ and a query or *state pattern* $\boldsymbol{\xi}$ that represents the current state. The largest norm of a stored pattern is $M = \max_i \|\boldsymbol{x}_i\|$. The *separation* $\Delta_i$ of a pattern $\boldsymbol{x}_i$ is defined as its minimal dot product difference to any of the other patterns: $\Delta_i = \min_{j,j \neq i} \left( \boldsymbol{x}_i^T \boldsymbol{x}_i - \boldsymbol{x}_i^T \boldsymbol{x}_j \right)$. A pattern is *well-separated* from the data if $\Delta_i \geq \frac{2}{\beta N} + \frac{1}{\beta} \log \left( 2(N-1)N\beta M^2 \right)$. We consider a modern Hopfield network with current state $\boldsymbol{\xi}$ and the energy function $\mathrm{E} = -\beta^{-1} \log \left( \sum_{i=1}^N \exp(\beta \boldsymbol{x}_i^T \boldsymbol{\xi}) \right) + \beta^{-1} \log N + \frac{1}{2} \boldsymbol{\xi}^T \boldsymbol{\xi} + \frac{1}{2} M^2$. For energy E and state $\boldsymbol{\xi}$, the update rule

$$\boldsymbol{\xi}^{\mathrm{new}} = f(\boldsymbol{\xi}; \boldsymbol{X}, \beta) = \boldsymbol{X} \, \boldsymbol{p} = \boldsymbol{X} \, \mathrm{softmax}(\beta \boldsymbol{X}^T \boldsymbol{\xi}) \tag{1}$$

is proven to converge globally to stationary points of the energy E, which are local minima or saddle points (see Suppl. Theorem B2). *Surprisingly, the update rule Eq.* (1) *is also the formula of the well-known transformer attention mechanism.*

To see this more clearly, we simultaneously update several queries $\boldsymbol{\Xi} = (\boldsymbol{\xi}_1, \ldots, \boldsymbol{\xi}_N)$, which alters Eq.(1) to $\boldsymbol{\Xi}^{\mathrm{new}} = \boldsymbol{X} \, \mathrm{softmax}(\beta \boldsymbol{X}^T \boldsymbol{\Xi})$. Furthermore, the state patterns $\boldsymbol{\xi}_i$ and the stored patterns $\boldsymbol{x}_i$ are linear mappings of vectors $\boldsymbol{y}_i$ into the space $\mathbb{R}^d$. For matrix notation, we set $\boldsymbol{x}_i = \boldsymbol{W}_K^T \boldsymbol{y}_i$, $\boldsymbol{\xi}_i = \boldsymbol{W}_Q^T \boldsymbol{y}_i$ and multiply the result of our update rule with $\boldsymbol{W}_V$. Using $\boldsymbol{Y} = (\boldsymbol{y}_1, \ldots, \boldsymbol{y}_N)^T$, we define the matrices $\boldsymbol{X}^T = \boldsymbol{K} = \boldsymbol{Y} \boldsymbol{W}_K$, $\boldsymbol{Q} = \boldsymbol{\Xi}^T = \boldsymbol{Y} \boldsymbol{W}_Q$, and $\boldsymbol{V} = \boldsymbol{Y} \boldsymbol{W}_K \boldsymbol{W}_V = \boldsymbol{X}^T \boldsymbol{W}_V$, where $\boldsymbol{W}_K \in \mathbb{R}^{d_y \times d_k}, \boldsymbol{W}_Q \in \mathbb{R}^{d_y \times d_k}, \boldsymbol{W}_V \in \mathbb{R}^{d_k \times d_v}, \boldsymbol{K} \in \mathbb{R}^{N \times d_k}, \boldsymbol{Q} \in \mathbb{R}^{N \times d_k}, \boldsymbol{V} \in \mathbb{R}^{N \times d_v}$, and the patterns are now mapped to the Hopfield space with dimension $d = d_k$. We set $\beta = 1/\sqrt{d_k}$, where $\beta$ corresponds to the reciprocal of the *temperature* of the softmax function, and change softmax to a row vector. The update rule Eq. (1) multiplied by $\boldsymbol{W}_V$ and performed for all queries simultaneously becomes in row vector notation:

$$\mathrm{att}(\boldsymbol{Q}, \boldsymbol{K}, \boldsymbol{V}; \beta) = \mathrm{softmax} \left( \beta \, \boldsymbol{Q} \, \boldsymbol{K}^T \right) \boldsymbol{V} = \mathrm{softmax} \left( \left( 1/\sqrt{d_k} \right) \boldsymbol{Q} \, \boldsymbol{K}^T \right) \boldsymbol{V}, \tag{2}$$

which is the formula of the transformer attention mechanism.

If the patterns $\boldsymbol{x}_i$ are well separated, the iterate Eq. (1) converges to a fixed point close to a pattern to which the initial $\boldsymbol{\xi}$ is similar. If the patterns are not well separated the iterate Eq.(1) converges to a fixed point close to the arithmetic mean of the patterns. If some patterns are similar to each other but well separated from all other vectors, then a *metastable state* between the similar patterns exists. Iterates that start near a metastable state converge to this metastable state. For details see Suppl. Sect. B2. Typically, the update converges after one update step (see Suppl. Theorem B8) and has an exponentially small retrieval error (see Suppl. Theorem B9).

Our main concern for application to immune repertoire classification is the number of patterns that can be stored and retrieved by the modern Hopfield network, i.e. the attention mechanism. This storage capacity of an attention mechanism is critical for massive MIL problems. We first define what we mean by storing and retrieving patterns from the modern Hopfield network.

**Definition 1 (Pattern Stored and Retrieved)** *We assume that around every pattern $\boldsymbol{x}_i$ a sphere $\mathrm{S}_i$ is given. We say $\boldsymbol{x}_i$ is stored if there is a single fixed point $\boldsymbol{x}_i^* \in \mathrm{S}_i$ to which all points $\boldsymbol{\xi} \in \mathrm{S}_i$ converge, and $\mathrm{S}_i \cap \mathrm{S}_j = \emptyset$ for $i \neq j$. We say $\boldsymbol{x}_i$ is retrieved if the iteration Eq. (1) converged to the single fixed point $\boldsymbol{x}_i^* \in \mathrm{S}_i$.*

For randomly chosen patterns, the number of patterns that can be stored is exponential in the dimension $d$ of the space of the patterns ($\boldsymbol{x}_i \in \mathbb{R}^d$).

**Theorem 1** *We assume a failure probability $0 < p \leqslant 1$ and randomly chosen patterns on the sphere with radius $M = K\sqrt{d-1}$. We define $a := \frac{2}{d-1}\left(1 + \ln(2\,\beta\,K^2\,p\,(d-1))\right)$, $b := \frac{2\,K^2\,\beta}{5}$, and $c = \frac{b}{W_0(\exp(a + \ln(b)))}$, where $W_0$ is the upper branch of the Lambert $W$ function and ensure $c \geq \left(\frac{2}{\sqrt{p}}\right)^{\frac{4}{d-1}}$. Then with probability $1 - p$, the number of random patterns that can be stored is*

$$N \ \geq \ \sqrt{p}\,c^{\frac{d-1}{4}}\ . \tag{3}$$

*Examples are $c \geq 3.1546$ for $\beta = 1$, $K = 3$, $d = 20$ and $p = 0.001$ ($a + \ln(b) > 1.27$) and $c \geq 1.3718$ for $\beta = 1$ $K = 1$, $d = 75$, and $p = 0.001$ ($a + \ln(b) < -0.94$).*

See Suppl. Theorem B5 for a proof. We have established that a modern Hopfield network or a transformer attention mechanism can store and retrieve exponentially many patterns. This allows us to approach MIL problems with massive numbers of instances from which we have to retrieve a few with an attention mechanism.

## 3   Deep Repertoire Classification

**Problem setting and notation.** We consider a MIL problem, in which an input object $X$ is a *bag* of $N$ instances $X = \{s_1, \ldots, s_N\}$. The instances do not have dependencies nor orderings between them and $N$ can be different for every object. We assume that each instance $s_i$ is associated with a label $y_i \in \{0, 1\}$, assuming a binary classification task, to which we do not have access. We only have access to a label $Y = \max_i y_i$ for an input object or bag. Note that this poses a credit assignment problem, since the sequences that are responsible for the label $Y$ have to be identified and that the relation between instance-label and bag-label can be more complex (Foulds & Frank, 2010).

A model $\hat{y} = g(X)$ should be (a) invariant to permutations of the instances and (b) able to cope with the fact that $N$ varies across input objects (Ilse et al., 2018), which is a problem also posed by point sets (Qi et al., 2017). Two principled approaches exist. The first approach is to learn an instance-level scoring function $h : \mathcal{S} \mapsto [0, 1]$, which is then pooled across instances with a pooling function $f$, for example by average-pooling or max-pooling (see below). The second approach is to construct an instance representation $\boldsymbol{z}_i$ of each instance by $h : \mathcal{S} \mapsto \mathbb{R}^{d_v}$ and then encode the bag, or the input object, by pooling these instance representations (Wang et al., 2018; Yan et al., 2018) via a function $f$. An output function $o : \mathbb{R}^{d_v} \mapsto [0, 1]$ subsequently classifies the bag. The second approach, the pooling of representations rather than scoring functions, is currently best performing (Wang et al., 2018).

In the problem at hand, the input object $X$ is the immune repertoire of an individual that consists of a large set of immune receptor sequences (T-cell receptors or antibodies). Immune receptors are primarily represented as sequences $s_i$ from a space $s_i \in \mathcal{S}$. These sequences act as the instances in the MIL problem. Although immune repertoire classification can readily be formulated as MIL problem, it is yet unclear how well machine learning methods solve the above-described problem with a large number of instances $N \gg 10,000$ and with instances $s_i$ being complex sequences. Next we describe currently used pooling functions for MIL problems.

**Pooling functions for MIL problems.** Different pooling functions equip a model $g$ with the property to be invariant to permutations of instances and with the ability to process different numbers of instances. Typically, a neural network $h_{\boldsymbol{\theta}}$ with parameters $\boldsymbol{\theta}$ is trained to obtain a function that maps each instance onto a representation: $\boldsymbol{z}_i = h_{\boldsymbol{\theta}}(s_i)$ and then a pooling function $\boldsymbol{z} = f(\{\boldsymbol{z}_1, \ldots, \boldsymbol{z}_N\})$ supplies a representation $\boldsymbol{z}$ of the input object $X = \{s_1, \ldots, s_N\}$. The following pooling functions are typically used: *average-pooling*: $\boldsymbol{z} = \frac{1}{N}\sum_{i=1}^{N} \boldsymbol{z}_i$, *max-pooling*: $\boldsymbol{z} = \sum_{m=1}^{d_v} \boldsymbol{e}_m (\max_{i, 1 \leqslant i \leqslant N}\{z_{im}\})$, where $\boldsymbol{e}_m$ is the standard basis vector for dimension $m$ and *attention-pooling*: $\boldsymbol{z} = \sum_{i=1}^{N} a_i \boldsymbol{z}_i$ where $a_i$ are non-negative ($a_i \geq 0$), sum to one ($\sum_{i=1}^{N} a_i = 1$), and are determined by an attention mechanism. These pooling functions are invariant to permutations of $\{1, \ldots, N\}$ and are differentiable. Therefore, they are suited as building blocks for Deep Learning architectures. We employ attention-pooling in our DeepRC model as detailed in the following.

**Modern Hopfield networks viewed as transformer-like attention mechanisms.** The modern Hopfield networks from Section 2 have a storage capacity that is exponential in the dimension of the vector space and converge after just one update (see Suppl. B). Additionally, the update rule of modern Hopfield networks is known as key-value attention mechanism, which has been highly successful through the transformer (Vaswani et al., 2017) and BERT (Devlin et al., 2019) models in natural language processing. Therefore, using modern Hopfield networks with the key-value-attention mechanism as update rule is the natural choice for our task. In particular, modern Hopfield networks are theoretically justified for storing and retrieving the large number of vectors (sequence patterns) that appear in the immune repertoire classification task.

Instead of using the terminology of modern Hopfield networks, we explain our DeepRC architecture in terms of key-value-attention (the update rule of the modern Hopfield network), since it is well known in the deep learning community. The attention mechanism assumes a space of dimension $d_k$ in which keys and queries are compared. A set of $N$ key vectors are combined to the matrix $\boldsymbol{K}$. A set of $d_q$ query vectors are combined to the matrix $\boldsymbol{Q}$. Similarities between queries and keys are computed by inner products, therefore queries can search for similar keys that are stored. Another set of $N$ value vectors are combined to the matrix $\boldsymbol{V}$. The output of the attention mechanism is a weighted average of the value vectors for each query $\boldsymbol{q}$. The $i$-th vector $\boldsymbol{v}_i$ is weighted by the similarity between the $i$-th key $\boldsymbol{k}_i$ and the query $\boldsymbol{q}$. The similarity is given by the softmax of the inner products of the query $\boldsymbol{q}$ with the keys $\boldsymbol{k}_i$. All queries are calculated in parallel via matrix operations. Consequently, the attention function $\mathrm{att}(\boldsymbol{Q}, \boldsymbol{K}, \boldsymbol{V}; \beta)$ maps queries $\boldsymbol{Q}$, keys $\boldsymbol{K}$, and values $\boldsymbol{V}$ to $d_v$-dimensional outputs: $\mathrm{att}(\boldsymbol{Q}, \boldsymbol{K}, \boldsymbol{V}; \beta) = \mathrm{softmax}(\beta \boldsymbol{Q} \boldsymbol{K}^T) \boldsymbol{V}$ (see also Eq. (2)). While this attention mechanism has originally been developed for sequence tasks (Vaswani et al., 2017), it can be readily transferred to sets (Lee et al., 2019; Ye et al., 2018). This type of attention mechanism will be employed in DeepRC.

**The DeepRC method.** We propose a novel method **Deep Repertoire Classification** (**DeepRC**) for immune repertoire classification with attention-based deep massive multiple instance learning and compare it against other machine learning approaches. For DeepRC, we consider *immune repertoires as input objects*, which are represented as bags of instances. In a bag, *each instance is an immune receptor sequence* and each bag can contain a large number of sequences. Note that we will use $\boldsymbol{z}_i$ to denote the *sequence-representation* of the $i$-th sequence and $\boldsymbol{z}$ to denote the *repertoire-representation*. At the core, DeepRC consists of a transformer-like attention mechanism that extracts the most important information from each repertoire. We first give an overview of the attention mechanism and then provide details on each of the sub-networks $h_1$, $h_2$, and $o$ of DeepRC. (Overview: Fig. 1; Architecture: Fig. 2; Implementation details: Suppl. Sect. A3; DeepRC variations: Suppl. Sect. A10.)

**Attention mechanism in DeepRC.** This mechanism is based on the three matrices $\boldsymbol{K}$ (the keys), $\boldsymbol{Q}$ (the queries), and $\boldsymbol{V}$ (the values) together with a parameter $\beta$. *Values.* DeepRC uses a 1D convolutional network $h_1$ (LeCun et al., 1998; Hu et al., 2014; Kelley et al., 2016) that supplies a sequence-representation $\boldsymbol{z}_i = h_1(s_i)$, which acts as the values $\boldsymbol{V} = \boldsymbol{Z} = (\boldsymbol{z}_1, \dots, \boldsymbol{z}_N)$ in the attention mechanism (see Figure 2). *Keys.* A second neural network $h_2$, which shares its first layers with $h_1$, is used to obtain keys $\boldsymbol{K} \in \mathbb{R}^{N \times d_k}$ for each sequence in the repertoire. This network uses 2 self-normalizing layers (Klambauer et al., 2017) with 32 units per layer (see Figure 2). *Query.* We use a fixed $d_k$-dimensional query vector $\boldsymbol{\xi}$ which is learned via backpropagation. For more attention heads, each head has a fixed query vector. We note that applying a standard transformer, in which each instance would create a query, is not feasible due to the large number of instances per repertoire. With the quantities introduced above, the attention mechanism (Eq. (2)) of DeepRC is implemented as follows:

$$\boldsymbol{z} = \mathrm{att}(\boldsymbol{\xi}^T, \boldsymbol{K}, \boldsymbol{Z}; \frac{1}{\sqrt{d_k}}) = \mathrm{softmax}\left(\frac{\boldsymbol{\xi}^T \boldsymbol{K}^T}{\sqrt{d_k}}\right) \boldsymbol{Z}, \tag{4}$$

where $\boldsymbol{Z} \in \mathbb{R}^{N \times d_v}$ are the sequence–representations stacked row-wise, $\boldsymbol{K}$ are the keys, and $\boldsymbol{z}$ is the repertoire-representation and at the same time a weighted mean of sequence–representations $\boldsymbol{z}_i$. The attention mechanism can readily be extended to multiple queries, however, computational demand could constrain this depending on the application and dataset. Theorem 1 demonstrates that this mechanism is able to retrieve a single pattern out of several hundreds of thousands.

*Attention-pooling and interpretability.* Each input object, i.e. repertoire, consists of a large number $N$ of sequences, which are reduced to a single fixed-size feature vector of length $d_v$ representing the whole input object by an attention-pooling function. To this end, a transformer-like attention

mechanism adapted to sets is realized in DeepRC which supplies $a_i$ – the importance of the sequence $s_i$. This importance value is an interpretable quantity, which is highly desired for the immunological problem at hand. Thus, DeepRC allows for two forms of interpretability methods. (a) A trained DeepRC model can compute attention weights $a_i$, which directly indicate the importance of a sequence. (b) DeepRC furthermore allows for the usage of contribution analysis methods, such as Integrated Gradients (IG) (Sundararajan et al., 2017) or Layer-Wise Relevance Propagation (Montavon et al., 2018; Arras et al., 2019). See Suppl. Sect. A9 for details.

**Classification layer and network parameters.** The repertoire-representation $\boldsymbol{z}$ is then used as input for a fully-connected output network $\hat{y} = o(\boldsymbol{z})$ that predicts the immune status, where we found it sufficient to train single-layer networks. In the simplest case, DeepRC predicts a single target, the class label $y$, e.g. the immune status of an immune repertoire, using one output value. However, since DeepRC is an end-to-end deep learning model, multiple targets may be predicted simultaneously in classification or regression settings or a mix of both. This allows for the introduction of additional information into the system via auxiliary targets such as age, sex, or other metadata.

**Network parameters, training, and inference.** DeepRC is trained using standard gradient descent methods to minimize a cross-entropy loss. The network parameters are $\boldsymbol{\theta}_1, \boldsymbol{\theta}_2, \boldsymbol{\theta}_o$ for the sub-networks $h_1, h_2$, and $o$, respectively, and additionally $\boldsymbol{\xi}$. In more detail, we train DeepRC using Adam (Kingma & Ba, 2014) with a batch size of $4$ and dropout of input sequences (see Suppl. Sect. A3).

**Implementation.** To reduce computational time, the attention network first computes the attention weights $a_i$ for each sequence $s_i$ in a repertoire. Subsequently, the top $10\%$ of sequences with the highest $a_i$ per repertoire are used to compute the weight updates and prediction. Furthermore, computation of $\boldsymbol{z}_i$ is performed in 16-bit and other computations in 32-bit precision to ensure numerical stability in the softmax. See Suppl. Sect. A3 for details.

## 4 Experimental Results

In this section, we report and analyze the predictive power of DeepRC and the compared methods on several immunosequencing datasets. The ROC-AUC is used as the main metric for the predictive power.

**Methods compared.** We compared state-of-the-art methods for immune repertoire classification, (Ostmeyer et al., 2019) ("Log. MIL (KMER)", "Log. MIL (TCRB)") and a burden test (Emerson et al., 2017), as well as the methods Logistic Regression ("Log. Regr."), k-nearest neighbour ("KNN"), and Support Vector Machines ("SVM") with kernels designed for sets, such as the Jaccard kernel ("J") and the MinMax ("MM") kernel (Ralaivola et al., 2005). For the simulated data, we also added baseline methods that search for the implanted motif either in binary or continuous fashion ("Known motif b.", "Known motif c.") assuming that this motif was known (for details, see Suppl. Sect. A5).

**Datasets.** We aimed at constructing immune repertoire classification scenarios with varying degree of difficulties and realism in order to compare and analyze the suggested machine learning methods. To this end, we either use simulated or experimentally-observed immune receptor sequences and implant signals, specifically, sequence motifs or sets thereof (Akbar et al., 2019; Weber et al., 2020), at different frequencies into sequences of repertoires of the posi-

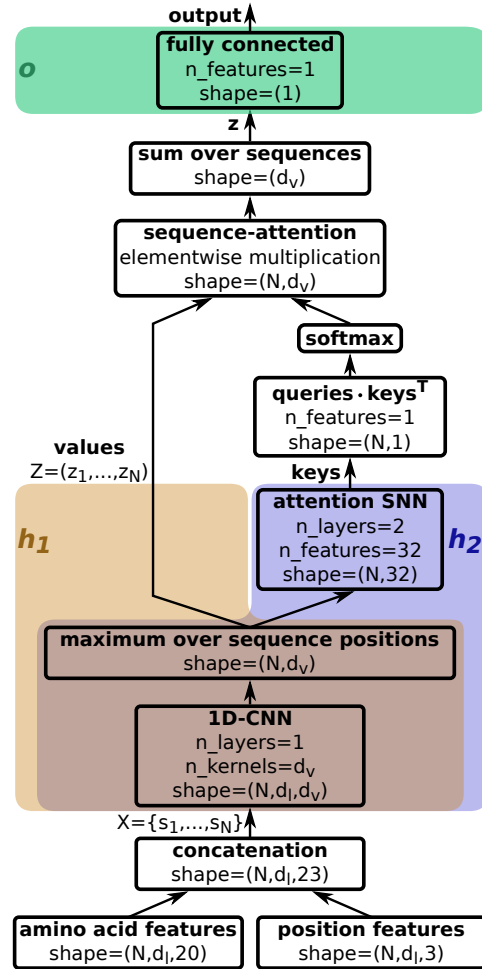

Figure 2: DeepRC architecture as used in Table 1 with sub-networks $h_1$, $h_2$, and $o$. $d_l$ indicates the sequence length.

tive class. These frequencies represent the *witness rates* and range from 0.01% to 10%. Overall, we compiled four categories of datasets: (a) simulated immunosequencing data with implanted signals, (b) LSTM-generated immunosequencing data with implanted signals, (c) real-world immunosequencing data with implanted signals, and (d) real-world immunosequencing data with known immune status, the *CMV dataset* (Emerson et al., 2017). The average number of instances per bag, which is the number of sequences per immune repertoire, is ≈300,000 except for category (c), in which we consider the scenario of low-coverage data with only 10,000 sequences per repertoire. The number of repertoires per dataset ranges from 785 to 5,000. In total, all datasets comprise ≈30 billion sequences or instances. This, to our knowledge, represents the largest comparative study on immune repertoire classification (see Suppl. Sect. A4).

**Hyperparameter selection.** We used a nested 5-fold cross validation (CV) procedure to estimate the performance of each of the methods. All methods could adjust their most important hyperparameters on a validation set in the inner loop of the procedure. See Suppl. Sect. A6 for details.

| | Real-world CMV | Real-world data with implanted signals | | | | LSTM-generated data | | | | | Simulated avg. |
|---|---|---|---|---|---|---|---|---|---|---|---|
| | | s.m. 1% | s.m. 0.1% | m.m. 1% | m.m. 0.1% | 10% | 1% | 0.5% | 0.1% | 0.05% | |
| DeepRC | **0.831** ± 0.002 | **1.000** ± 0.000 | **0.984** ± 0.008 | **0.999** ± 0.001 | **0.938** ± 0.009 | **1.000** ± 0.000 | **1.000** ± 0.000 | **1.000** ± 0.000 | **1.000** ± 0.000 | **0.998** ± 0.002 | **0.865** ± 0.211 |
| SVM (MM) | 0.825 ± 0.022 | **1.000** ± 0.000 | 0.578 ± 0.020 | **1.000** ± 0.000 | 0.531 ± 0.019 | **1.000** ± 0.000 | **1.000** ± 0.000 | 0.999 ± 0.001 | 0.999 ± 0.002 | 0.985 ± 0.014 | 0.832 ± 0.203 |
| SVM (J) | 0.546 ± 0.021 | 0.988 ± 0.003 | 0.527 ± 0.016 | **1.000** ± 0.000 | 0.574 ± 0.019 | 0.981 ± 0.041 | **1.000** ± 0.000 | **1.000** ± 0.000 | 0.904 ± 0.036 | 0.768 ± 0.068 | 0.543 ± 0.076 |
| KNN (MM) | 0.679 ± 0.076 | 0.744 ± 0.237 | 0.486 ± 0.031 | 0.674 ± 0.182 | 0.500 ± 0.022 | 0.699 ± 0.272 | 0.717 ± 0.263 | 0.732 ± 0.263 | 0.536 ± 0.156 | 0.516 ± 0.153 | 0.629 ± 0.126 |
| KNN (J) | 0.534 ± 0.039 | 0.652 ± 0.155 | 0.484 ± 0.025 | 0.695 ± 0.200 | 0.508 ± 0.025 | 0.698 ± 0.285 | 0.606 ± 0.237 | 0.698 ± 0.164 | 0.550 ± 0.186 | 0.539 ± 0.194 | 0.501 ± 0.007 |
| Log. Regr. | 0.613 ± 0.044 | **1.000** ± 0.000 | 0.585 ± 0.045 | **1.000** ± 0.000 | 0.512 ± 0.015 | **1.000** ± 0.000 | **1.000** ± 0.000 | **1.000** ± 0.000 | 0.697 ± 0.164 | 0.466 ± 0.103 | 0.832 ± 0.204 |
| Log. MIL (KMER) | 0.582 ± 0.065 | 0.541 ± 0.074 | 0.506 ± 0.034 | 0.994 ± 0.004 | 0.620 ± 0.153 | **0.997** ± 0.004 | 0.718 ± 0.112 | 0.637 ± 0.144 | 0.571 ± 0.146 | 0.528 ± 0.129 | 0.662 ± 0.216 |
| Log. MIL (TCRB) | 0.515 ± 0.073 | 0.503 ± 0.032 | 0.501 ± 0.016 | 0.992 ± 0.003 | 0.782 ± 0.030 | 0.541 ± 0.086 | 0.566 ± 0.162 | 0.468 ± 0.086 | 0.505 ± 0.067 | 0.500 ± 0.121 | 0.501 ± 0.015 |
| Burden test | 0.699 ± 0.041 | **1.000** ± 0.000 | 0.640 ± 0.048 | **1.000** ± 0.000 | 0.891 ± 0.016 | **1.000** ± 0.000 | **1.000** ± 0.000 | **1.000** ± 0.000 | 0.999 ± 0.003 | 0.792 ± 0.280 | 0.543 ± 0.070 |
| Motif binary | | 1.000 ± 0.000 | 0.704 ± 0.028 | 0.994 ± 0.003 | 0.620 ± 0.038 | 1.000 ± 0.000 | 1.000 ± 0.000 | 1.000 ± 0.000 | 0.999 ± 0.003 | 0.999 ± 0.003 | 0.899 ± 0.158 |
| Motif nonbinary | | 0.920 ± 0.004 | 0.562 ± 0.028 | 0.647 ± 0.030 | 0.515 ± 0.031 | 1.000 ± 0.000 | 1.000 ± 0.000 | 0.989 ± 0.011 | 0.722 ± 0.085 | 0.626 ± 0.094 | 0.727 ± 0.189 |

Table 1: Results in terms of AUC of the competing methods on all datasets. The reported errors are standard deviations across 5 cross-validation (CV) folds (except for the column "Simulated"). **Real-world CMV:** Average performance over 5 CV folds on the *CMV dataset* (Emerson et al., 2017). **Real-world data with implanted signals:** Average performance over 5 CV folds for each of the four datasets. A signal was implanted with a frequency (=witness rate) of 1% or 0.1%. Either a single motif ("s.m.") or multiple motifs ("m.m.") were implanted. **LSTM-generated data:** Average performance over 5 CV folds for each of the 5 datasets. In each dataset, a signal was implanted with a frequency of 10%, 1%, 0.5%, 0.1%, or 0.05%, respectively. **Simulated:** We report the arithmetic mean over 21 simulated datasets with implanted signals and varying difficulties (see Suppl. Tab. A10 for details). The error reported is the standard deviation of the AUC values across the 21 datasets.

**Results.** In each of the four categories, "real-world data", "real-world data with implanted signals", "LSTM-generated data", and "simulated immunosequencing data", DeepRC outperforms all competing methods with respect to average AUC. Across categories, the runner-up methods are either the SVM for MIL problems with MinMax kernel (SVM (MM)) or the burden test (see Table 1). The advantage of DeepRC over the other methods such as the SVM (MM) becomes apparent on datasets with more complex or noisy motifs (see Suppl. Sect. A7). DeepRC significantly outperforms the second best method, the SVM (MM), at a p-value of $4 \cdot 10^{-121}$ on the 4 categories (McNemar's test).

*Results on simulated immunosequencing data.* In this setting the complexity of the implanted signal is in focus and varies throughout 21 simulated datasets (Suppl. Sect. A4). Some datasets are challenging for the methods because the implanted motif is hidden by noise and others because only a small fraction of sequences carries the motif, resulting in a low witness rate. These difficulties become evident by the method called "known motif binary", which assumes the implanted motif is known. The performance of this method ranges from a perfect AUC of 1.000 in several datasets to an AUC of 0.532 in dataset '17' (see Suppl. Sect. A7). DeepRC outperforms all other methods with an average AUC of 0.865±0.211. The runner-up methods are the SVM (MM) and the Logistic Regression, which experience performance losses at higher motif complexities, with an average AUC of $0.832 \pm 0.203$ and $0.832 \pm 0.204$, respectively (see Suppl. Sect. A7). The predictive performance of all methods suffers if the signal occurs only in an extremely small fraction of sequences. DeepRC significantly outperforms the second best method, the SVM (MM), at a p-value of $3 \cdot 10^{-66}$ (McNemar's test). *Results on LSTM-generated data.* On the LSTM-generated data, in which we implanted noisy motifs with frequencies of 10%, 1%, 0.5%, 0.1%, and 0.05%, DeepRC yields almost perfect predictive performance with an average AUC of $1.000 \pm 0.001$ (see Suppl. Sect. A7 and A8). The second

best method, SVM with MinMax kernel, has a similar predictive performance to DeepRC on all datasets but the other competing methods have a lower predictive performance on datasets with low frequency of the signal ($0.05\%$). *Results on real-world data with implanted motifs.* In this dataset category, we used real immunosequences and implanted single or multiple noisy motifs. Again, DeepRC outperforms all other methods with an average AUC of $0.980 \pm 0.029$, with the second best method being the burden test with an average AUC of $0.883 \pm 0.170$. Notably, all methods except for DeepRC have difficulties with noisy motifs at a frequency of $0.1\%$ (see Suppl. Tab. A12). DeepRC significantly outperforms the second best method, the SVM (MM), at a p-value of $10^{-222}$ (McNemar's test). *Results on real-world data.* On the real-world dataset, in which the immune status of persons affected by the cytomegalovirus has to be predicted, the competing methods yield predictive AUCs between $0.515$ and $0.825$ (see Table 1). We note that this dataset is not the exact dataset that was used in Emerson et al. (2017). It differs in pre-processing and also comprises a different set of samples and a smaller training set due to the nested 5-fold cross-validation procedure, which leads to a more challenging dataset. The best performing method is DeepRC with an AUC of $0.831 \pm 0.002$, followed by the SVM (MM) (AUC $0.825 \pm 0.022$) and the burden test with an AUC of $0.699 \pm 0.041$. The top-ranked sequences by DeepRC significantly correspond to those detected by Emerson et al. (2017), which we tested by a Mann-Whitney U-test with the null hypothesis that the attention values of the sequences detected by Emerson et al. (2017) would be equal to the attention values of the remaining sequences ($p$-value of $1.3 \cdot 10^{-93}$). The sequence attention values are displayed in Suppl. Tab. A15.

**Conclusion.** We have demonstrated how modern Hopfield networks and attention mechanisms enable successful classification of the immune status of immune repertoires. For this task, methods have to identify the discriminating sequences amongst a large set of sequences in an immune repertoire. Specifically, even motifs within those sequences have to be identified. We have shown that DeepRC, a modern Hopfield network and an attention mechanism with a fixed query, can solve this difficult task despite the massive number of instances. DeepRC furthermore outperforms the compared methods across a range of different experimental conditions.

**Availability.** All datasets and code will be fully released at `https://github.com/ml-jku/DeepRC`. The *CMV dataset* is publicly available at `https://clients.adaptivebiotech.com/pub/Emerson-2017-NatGen`.

## Broader Impact

**Impact on machine learning and related scientific fields.** We envision that with (a) the increasing availability of large immunosequencing datasets (Kovaltsuk et al., 2018; Corrie et al., 2018; Christley et al., 2018; Zhang et al., 2020; Rosenfeld et al., 2018; Shugay et al., 2018), (b) further fine-tuning of ground-truth benchmarking immune receptor datasets (Weber et al., 2020; Olson et al., 2019; Marcou et al., 2018), (c) accounting for repertoire-impacting factors such as age, sex, ethnicity, and environment (potential confounding factors), and (d) increased GPU memory and increased computing power, it will be possible to identify discriminating immune receptor motifs for many diseases, potentially even for the current SARS-CoV-2 (COVID-19) pandemic (Raybould et al., 2020; Minervina et al., 2020; Galson et al., 2020). Such results would greatly benefit ongoing research on antibody and TCR-driven immunotherapies and immunodiagnostics as well as rational vaccine design (Brown et al., 2019).

In the course of this development, the experimental verification and interpretation of machine-learning-identified motifs could receive additional focus, as for most of the sequences within a repertoire the corresponding antigen is unknown. Nevertheless, recent technological breakthroughs in high-throughput antigen-labeled immunosequencing are beginning to generate large-scale antigen-labeled single-immune-receptor-sequence data, thus resolving this longstanding problem (Setliff et al., 2019).

From a machine learning perspective, the successful application of DeepRC on immune repertoires with their large numbers of instances per bag might encourage the application of modern Hopfield networks and attention mechanisms on new, previously unsolved or unconsidered, datasets and problems.

**Impact on society.** If the approach proves itself successful, it could lead to faster testing of individuals for their immune status w.r.t. a range of diseases based on blood samples. This might motivate changes in the pipeline of diagnostics and tracking of diseases, e.g. automated testing of the immune status in regular intervals. It would furthermore make the collection and screening of blood samples

for larger databases more attractive. In consequence, the improved testing of immune statuses might identify individuals that do not have a working immune response towards certain diseases to government or insurance companies, which could then push for targeted immunisation of the individual. Similarly to compulsory vaccination, such testing for the immune status could be made compulsory by governments, possibly violating privacy or personal self-determination in exchange for increased over-all health of a population.

Ultimately, if the approach proves itself successful, the insights gained from the screening of individuals that have successfully developed resistances against specific diseases could lead to faster targeted immunisation, once a certain number of individuals with resistances can be found. This might strongly decrease the harm done by e.g. pandemics and lead to a change in the societal perception of such diseases.

**Consequences of failures of the method.** As common with methods in machine learning, potential danger lies in the possibility that users rely too much on our new approach and use it without reflecting on the outcomes. However, the full pipeline in which our method would be used includes wet lab tests after its application to verify and investigate the results, gain insights, and possibly derive treatments. Failures of the proposed method would lead to unsuccessful wet lab validation and negative wet lab tests. Since the proposed algorithm does not directly suggest treatment or therapy, human beings are not directly at risk of being treated with a harmful therapy. Substantial wet lab and in-vitro testing and would indicate wrong decisions by the system.

**Leveraging of biases in the data and potential discrimination.** As for almost all machine learning methods, confounding factors, such as age or sex, could be used for classification. This, might lead to biases in predictions or uneven predictive performance across subgroups. As a result, failures in the wet lab would occur (see paragraph above). Moreover, insights into the relevance of the confounding factors could be gained, leading to possible therapies or counter-measures concerning said factors.

Furthermore, the amount of data available with respect to relevant confounding factors could lead to better or worse performance of our method. E.g. a dataset consisting mostly of data from individuals within a specific age group might yield better performance for that age group, possibly resulting in better or exclusive treatment methods for that specific group. Here, again, the application of DeepRC would be followed by in-vitro testing and development of a treatment, where all target groups for the treatment have to be considered accordingly.

## Acknowledgments and Disclosure of Funding

The ELLIS Unit Linz, the LIT AI Lab, the Institute for Machine Learning, are supported by the Federal State Upper Austria. IARAI is supported by Here Technologies. We thank the projects AI-MOTION (LIT-2018-6-YOU-212), DeepToxGen (LIT-2017-3-YOU-003), AI-SNN (LIT-2018-6-YOU-214), DeepFlood (LIT-2019-8-YOU-213), Medical Cognitive Computing Center (MC3), PRIMAL (FFG-873979), S3AI (FFG-872172), DL for granular flow (FFG-871302), ELISE (H2020-ICT-2019-3 ID: 951847), AIDD (MSCA-ITN-2020 ID: 956832). We thank Janssen Pharmaceutica, UCB Biopharma SRL, Merck Healthcare KGaA, Audi.JKU Deep Learning Center, TGW LOGISTICS GROUP GMBH, Silicon Austria Labs (SAL), FILL Gesellschaft mbH, Anyline GmbH, Google, ZF Friedrichshafen AG, Robert Bosch GmbH, Software Competence Center Hagenberg GmbH, TÜV Austria, and the NVIDIA Corporation. Victor Greiff (VG) and Geir Kjetil Sandve (GKS) are supported by The Helmsley Charitable Trust (#2019PG-T1D011, to VG), UiO World-Leading Research Community (to VG), UiO:LifeSciences Convergence Environment Immunolingo (to VG and GKS), EU Horizon 2020 iReceptorplus (#825821, to VG) and Stiftelsen Kristian Gerhard Jebsen (K.G. Jebsen Coeliac Disease Research Centre, to GKS).

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
