[Supplementary Material 1 · Supplement B.pdf]

# SUPPLEMENT B:
# Modern Hopfield Networks and Attention for Immune Repertoire Classification

**Sepp Hochreiter**[†]

**Markus Holzleitner**    **Lukas Gruber**    **Hubert Ramsauer**

**Günter Klambauer**    **Johannes Brandstetter**

ELLIS Unit Linz and LIT AI Lab
Institute for Machine Learning
Johannes Kepler University Linz, Austria
[†] also at Institute of Advanced Research in Artificial Intelligence (IARAI)

## Abstract

## Contents

## List of theorems

## List of definitions

# B1 Introduction

This document is a supplement to the paper "Modern Hopfield Networks and Attention for Immune Repertoire Classification".

In the next section (Section B2) our new modern Hopfield network is introduced. In Subsection B2.1 we present the new energy function. Then in Subsection B2.2, our new update rule is introduced. In Subsection B2.3, we show that this update rule ensures global convergence. We show that all the limit points of any sequence generated by the update rule are the stationary points (local minima or saddle points) of the energy function. In Section B2.4, we consider the local convergence of the update rule and see that it converges after one update. In Subsection B2.5, we consider the properties of the fixed points that are associated with the stored patterns. In Subsection B2.5.1, we show that exponentially many patterns can be stored. The main result is given in Theorem B5: for random patterns on a sphere we can store and retrieve exponentially (in the dimension of the space) many patterns. Subsection B2.5.2 reports that the update converges after one update step and that the retrieval error is exponentially small.

In Subsection B2.6, we consider how associations for the new Hopfield networks can be learned. In Subsection B2.6.1, we consider the initialization. In Subsection B2.6.2, we analyze if the association is learned directly by a bilinear form. In Subsection B2.6.3, we analyze if stored patterns and query patterns are mapped to the space of the Hopfield network. Therefore we treat the architecture of the transformer and BERT. In Subsection B2.7, we introduce a temporal component into the new Hopfield network that leads to a forgetting behavior. The forgetting allows us to treat infinite memory capacity in Subsection B2.7.1. In Subsection B2.7.2, we consider the controlled forgetting behavior. In Section B3, we provide the mathematical background that is needed for our proofs. In particular we give lemmas on properties of the softmax, the log-sum-exponential, the Legendre transform, and the Lambert $W$ function.

In Section B4, we review the new Hopfield network as introduced by Krotov and Hopfield in 2016. However in contrast to our new Hopfield network, Krotov and Hopfield' new Hopfield network is a binary, that is, a network with binary states. In Subsection B4.1, we give an introduction to neural networks equipped with associative memories and new Hopfield networks. In Subsection B4.1.1, we discuss neural networks that are enhanced by an additional external memory and by attention mechanisms. In Subsection B4.1.2, we give an overview over the modern Hopfield networks. Finally, in Subsection B4.2, we present the energy function and the update rule for the modern, binary Hopfield networks.

## B2 Modern Hopfield Networks: Continuous States (New Concept)

### B2.1 New Energy Function

We have patterns $\boldsymbol{x}_1, \ldots, \boldsymbol{x}_N$ that are represented by the matrix

$$\boldsymbol{X} = (\boldsymbol{x}_1, \ldots, \boldsymbol{x}_N) \ . \tag{1}$$

The largest norm of a pattern is

$$M = \max_i \|\boldsymbol{x}_i\| \ . \tag{2}$$

The query or state of the Hopfield network is $\boldsymbol{\xi}$.

The energy function E in the new type of Hopfield models of Krotov and Hopfield is $\mathrm{E} = -\sum_{i=1}^N F\left(\boldsymbol{\xi}^T \boldsymbol{x}_i\right)$ for binary patterns $\boldsymbol{x}_i$ and binary state $\boldsymbol{\xi}$ with interaction function $F(x) = x^n$, where $n = 2$ gives classical Hopfield model [28]. The storage capacity is proportional to $d^{n-1}$ [28]. This model was generalized by Demircigil et al. [18] to exponential interaction functions $F(x) = \exp(x)$ which gives the energy $\mathrm{E} = -\exp(\mathrm{lse}(1, \boldsymbol{X}^T \boldsymbol{\xi}))$. This energy leads to an exponential storage capacity of $N = 2^{d/2}$ for binary patterns. Furthermore with a single update the fixed point is recovered with high probability. See more details in Section B4.

In contrast to the these binary modern Hopfield networks, we focus on modern Hopfield networks with *continuous states* that can store *continuous patterns*. We generalize the energy of Demircigil et al. [18] to continuous states while keeping the lse properties which ensure high storage capacity and

fast convergence. Our new energy E for a continuous query or state $\boldsymbol{\xi}$ is defined as

$$\mathrm{E} \; = \; - \operatorname{lse}(\beta, \boldsymbol{X}^T \boldsymbol{\xi}) \; + \; \frac{1}{2} \boldsymbol{\xi}^T \boldsymbol{\xi} \; + \; \beta^{-1} \ln N \; + \; \frac{1}{2} M^2 \tag{3}$$

$$= \; - \beta^{-1} \ln \left( \sum_{i=1}^{N} \exp(\beta \boldsymbol{x}_i^T \boldsymbol{\xi}) \right) \; + \; \beta^{-1} \ln N \; + \; \frac{1}{2} \boldsymbol{\xi}^T \boldsymbol{\xi} \; + \; \frac{1}{2} M^2 \; . \tag{4}$$

First let us collect and prove some properties of E. The next lemma gives bounds on the energy E.

**Lemma 1.** *The energy* E *is larger than zero:*

$$0 \; \leqslant \; \mathrm{E} \; . \tag{5}$$

*For $\boldsymbol{\xi}$ in the simplex defined by the patterns, the energy* E *is upper bounded by:*

$$\mathrm{E} \; \leqslant \; \beta^{-1} \ln N \; + \; \frac{1}{2} M^2 \; , \tag{6}$$

$$\mathrm{E} \; \leqslant \; 2 M^2 \; . \tag{7}$$

*Proof.* We start by deriving the lower bound of zero. The pattern most similar to query or state $\boldsymbol{\xi}$ is $\boldsymbol{x}_{\boldsymbol{\xi}}$:

$$\boldsymbol{x}_{\boldsymbol{\xi}} \; = \; \boldsymbol{x}_k \; , \quad k \; = \; \arg\max_i \boldsymbol{\xi}^T \boldsymbol{x}_i \; . \tag{8}$$

We obtain

$$\mathrm{E} \; = \; - \beta^{-1} \ln \left( \sum_{i=1}^{N} \exp(\beta \boldsymbol{x}_i^T \boldsymbol{\xi}) \right) \; + \; \beta^{-1} \ln N \; + \; \frac{1}{2} \boldsymbol{\xi}^T \boldsymbol{\xi} \; + \; \frac{1}{2} M^2 \tag{9}$$

$$= \; - \beta^{-1} \ln \left( \frac{1}{N} \sum_{i=1}^{N} \exp(\beta \boldsymbol{x}_i^T \boldsymbol{\xi}) \right) \; + \; \frac{1}{2} \boldsymbol{\xi}^T \boldsymbol{\xi} \; + \; \frac{1}{2} M^2$$

$$\geq \; - \beta^{-1} \ln \left( \frac{1}{N} \sum_{i=1}^{N} \exp(\beta \boldsymbol{x}_i^T \boldsymbol{\xi}) \right) \; + \; \frac{1}{2} \boldsymbol{\xi}^T \boldsymbol{\xi} \; + \; \frac{1}{2} \boldsymbol{x}_{\boldsymbol{\xi}}^T \boldsymbol{x}_{\boldsymbol{\xi}}$$

$$\geq \; - \beta^{-1} \ln \left( \exp(\beta \boldsymbol{x}_{\boldsymbol{\xi}}^T \boldsymbol{\xi}) \right) \; + \; \frac{1}{2} \boldsymbol{\xi}^T \boldsymbol{\xi} \; + \; \frac{1}{2} \boldsymbol{x}_{\boldsymbol{\xi}}^T \boldsymbol{x}_{\boldsymbol{\xi}}$$

$$= \; - \boldsymbol{x}_{\boldsymbol{\xi}}^T \boldsymbol{\xi} \; + \; \frac{1}{2} \boldsymbol{\xi}^T \boldsymbol{\xi} \; + \; \frac{1}{2} \boldsymbol{x}_{\boldsymbol{\xi}}^T \boldsymbol{x}_{\boldsymbol{\xi}}$$

$$= \; \frac{1}{2} (\boldsymbol{\xi} - \boldsymbol{x}_{\boldsymbol{\xi}})^T (\boldsymbol{\xi} - \boldsymbol{x}_{\boldsymbol{\xi}}) \; = \; \frac{1}{2} \| \boldsymbol{\xi} - \boldsymbol{x}_{\boldsymbol{\xi}} \|^2 \; \geq \; 0 \; .$$

The energy is zero and, therefore, the bound attained, if all $\boldsymbol{x}_i$ are equal, that is, $\boldsymbol{x}_i = \boldsymbol{x}$ for all $i$ and $\boldsymbol{\xi} = \boldsymbol{x}$.

For deriving upper bounds on the energy E, we require the the query $\boldsymbol{\xi}$ to be in the simplex defined by the patterns, that is,

$$\boldsymbol{\xi} \; = \; \sum_{i=1}^{N} p_i \, \boldsymbol{x}_i \; , \quad \sum_{i=1}^{N} p_i \; = \; 1 \; , \quad \forall_i : \; 0 \; \leqslant \; p_i \; . \tag{10}$$

The first upper bound is.

$$\mathrm{E} \; = \; - \beta^{-1} \ln \left( \sum_{i=1}^{N} \exp(\beta \boldsymbol{x}_i^T \boldsymbol{\xi}) \right) \; + \; \frac{1}{2} \boldsymbol{\xi}^T \boldsymbol{\xi} \; + \; \beta^{-1} \ln N \; + \; \frac{1}{2} M^2 \tag{11}$$

$$\leqslant \; - \sum_{i=1}^{N} p_i \, (\boldsymbol{x}_i^T \boldsymbol{\xi}) \; + \; \frac{1}{2} \boldsymbol{\xi}^T \boldsymbol{\xi} \; + \; \beta^{-1} \ln N \; + \; \frac{1}{2} M^2$$

$$= \; - \frac{1}{2} \boldsymbol{\xi}^T \boldsymbol{\xi} \; + \; \beta^{-1} \ln N \; + \; \frac{1}{2} M^2 \; \leqslant \; \beta^{-1} \ln N \; + \; \frac{1}{2} M^2 \; .$$

For the first inequality we applied Lemma 19 to $-\mathrm{lse}(\beta, \boldsymbol{X}^T \boldsymbol{\xi})$ with $\boldsymbol{z} = \boldsymbol{p}$ giving

$$- \mathrm{lse}(\beta, \boldsymbol{X}^T \boldsymbol{\xi}) \;\leqslant\; - \sum_{i=1}^{N} p_i \, (\boldsymbol{x}_i^T \boldsymbol{\xi}) \;+\; \beta^{-1} \sum_{i=1}^{N} p_i \ln p_i \;\leqslant\; - \sum_{i=1}^{N} p_i \, (\boldsymbol{x}_i^T \boldsymbol{\xi}) \,, \qquad (12)$$

as the term involving the logarithm is non-positive.

Next we derive the second upper bound, for which we need the mean $\boldsymbol{m_x}$ of the patterns

$$\boldsymbol{m_x} \;=\; \frac{1}{N} \sum_{i=1}^{N} \boldsymbol{x}_i \,. \qquad (13)$$

We obtain

$$\mathrm{E} \;=\; - \beta^{-1} \ln \left( \sum_{i=1}^{N} \exp(\beta \boldsymbol{x}_i^T \boldsymbol{\xi}) \right) \;+\; \frac{1}{2}\, \boldsymbol{\xi}^T \boldsymbol{\xi} \;+\; \beta^{-1} \ln N \;+\; \frac{1}{2}\, M^2 \qquad (14)$$

$$\leqslant \; - \sum_{i=1}^{N} \frac{1}{N}\, \boldsymbol{x}_i^T \boldsymbol{\xi} \;+\; \frac{1}{2}\, \boldsymbol{\xi}^T \boldsymbol{\xi} \;+\; \frac{1}{2}\, M^2$$

$$= \; - \boldsymbol{m_x}^T \boldsymbol{\xi} \;+\; \frac{1}{2}\, \boldsymbol{\xi}^T \boldsymbol{\xi} \;+\; \frac{1}{2}\, M^2$$

$$\leqslant \; \|\boldsymbol{m_x}\| \, \|\boldsymbol{\xi}\| \;+\; \frac{1}{2}\, \|\boldsymbol{\xi}\|^2 \;+\; \frac{1}{2}\, M^2$$

$$\leqslant \; 2\, M^2 \,,$$

where for the first inequality we again applied Lemma 19 with $\boldsymbol{z} = (1/N, \ldots, 1/N)$ and $\beta^{-1} \sum_i 1/N \ln(1/N) = -\beta^{-1} \ln(N)$. This inequality also follows from Jensen's inequality. The second inequality uses the Cauchy-Schwarz inequality. The last inequality uses

$$\|\boldsymbol{\xi}\| \;=\; \left\| \sum_i p_i \, \boldsymbol{x}_i \right\| \;\leqslant\; \sum_i p_i \, \|\boldsymbol{x}_i\| \;\leqslant\; \sum_i p_i M \;=\; M \qquad (15)$$

and

$$\|\boldsymbol{m_x}\| \;=\; \left\| \sum_i (1/N) \, \boldsymbol{x}_i \right\| \;\leqslant\; \sum_i (1/N) \, \|\boldsymbol{x}_i\| \;\leqslant\; \sum_i (1/N) \, M \;=\; M \,. \qquad (16)$$

$$\square$$

## B2.2 New Update Rule

We now introduce an update rule for minimizing the energy function E. The new update rule is

$$\boldsymbol{\xi}^{\mathrm{new}} \;=\; \boldsymbol{X} \boldsymbol{p} \;=\; \boldsymbol{X} \mathrm{softmax}(\beta \boldsymbol{X}^T \boldsymbol{\xi}) \,, \qquad (17)$$

where we used

$$\boldsymbol{p} \;=\; \mathrm{softmax}(\beta \boldsymbol{X}^T \boldsymbol{\xi}) \,. \qquad (18)$$

The new state $\boldsymbol{\xi}^{\mathrm{new}}$ is in the simplex defined by the patterns, no matter what the previous state $\boldsymbol{\xi}$ was. In contrast, the synchronous update rule for the classical Hopfield network with threshold zero is

$$\boldsymbol{\xi}^{\mathrm{new}} \;=\; \mathrm{sgn}\left( \boldsymbol{X} \boldsymbol{X}^T \boldsymbol{\xi} \right) \,. \qquad (19)$$

Therefore instead of using the vector $\boldsymbol{X}^T \boldsymbol{\xi}$ as in the classical Hopfield network, its softmax version $\mathrm{softmax}(\beta \boldsymbol{X}^T \boldsymbol{\xi})$ is used.

In the next section (Section B2.3) we show that the update rule Eq. (17) ensures global convergence. We show that all the limit points of any sequence generated by the update rule are the stationary points (local minima or saddle points) of the energy function E. In Section B2.4 we consider the local convergence of the update rule Eq. (17) and see that it converges after one update.

## B2.3 Global Convergence of the Update Rule

We are interested in the *global convergence*, that is, convergence from each initial point, of the iterate

$$\boldsymbol{\xi}^{\text{new}} \; = \; f(\boldsymbol{\xi}) \; = \; \boldsymbol{X}\boldsymbol{p} \; = \; \boldsymbol{X}\text{softmax}(\beta \boldsymbol{X}^T \boldsymbol{\xi}) \,, \tag{20}$$

where we used

$$\boldsymbol{p} \; = \; \text{softmax}(\beta \boldsymbol{X}^T \boldsymbol{\xi}) \,. \tag{21}$$

We defined the energy function

$$\text{E} \; = \; - \text{lse}(\beta, \boldsymbol{X}^T \boldsymbol{\xi}) \; + \; \frac{1}{2}\boldsymbol{\xi}^T \boldsymbol{\xi} \; + \; \beta^{-1} \ln N \; + \; \frac{1}{2}M^2 \tag{22}$$

$$= \; - \beta^{-1} \ln \left( \sum_{i=1}^{N} \exp(\beta \boldsymbol{x}_i^T \boldsymbol{\xi}) \right) \; + \; \beta^{-1} \ln N \; + \; \frac{1}{2}\boldsymbol{\xi}^T \boldsymbol{\xi} \; + \; \frac{1}{2}M^2 \,. \tag{23}$$

We will show that the update rule in Eq. (20) is the Concave-Convex Procedure (CCCP) for minimizing the energy E. The CCCP is proven to converge globally.

**Theorem B1** (Global Convergence (Zangwill): Energy). *The update rule Eq. (20) converges globally: For $\boldsymbol{\xi}^{t+1} = f(\boldsymbol{\xi}^t)$, the energy $\text{E}(\boldsymbol{\xi}^t) \to \text{E}(\boldsymbol{\xi}^*)$ for $t \to \infty$ and a fixed point $\boldsymbol{\xi}^*$.*

*Proof.* The Concave-Convex Procedure (CCCP) [51, 52] minimizes a function that is the sum of a concave function and a convex function. CCCP is equivalent to Legendre minimization [36, 37] algorithms [52]. The Jacobian of the softmax is positive semi-definite according to Lemma 22. The Jacobian of the softmax is the Hessian of the lse, therefore lse is a convex and $-$lse a concave function. Therefore, the energy function $\text{E}(\boldsymbol{\xi})$ is the sum of the convex function $\text{E}_1(\boldsymbol{\xi}) = 1/2\boldsymbol{\xi}^T \boldsymbol{\xi} + C_1$ and the concave function $\text{E}_2(\boldsymbol{\xi}) = -$lse:

$$\text{E}(\boldsymbol{\xi}) \; = \; \text{E}_1(\boldsymbol{\xi}) \; + \; \text{E}_2(\boldsymbol{\xi}) \,, \tag{24}$$

$$\text{E}_1(\boldsymbol{\xi}) \; = \; \frac{1}{2}\boldsymbol{\xi}^T \boldsymbol{\xi} \; + \; \beta^{-1} \ln N \; + \; \frac{1}{2}M^2 \; = \; \frac{1}{2}\boldsymbol{\xi}^T \boldsymbol{\xi} \; + \; C_1 \,, \tag{25}$$

$$\text{E}_2(\boldsymbol{\xi}) \; = \; - \text{lse}(\beta, \boldsymbol{X}^T \boldsymbol{\xi}) \,, \tag{26}$$

where $C_1$ does not depend on $\boldsymbol{\xi}$.
The Concave-Convex Procedure (CCCP) [51, 52] applied to E is

$$\nabla_\xi \text{E}_1\left( \boldsymbol{\xi}^{t+1} \right) \; = \; - \nabla_\xi \text{E}_2\left( \boldsymbol{\xi}^t \right) \,, \tag{27}$$

which is

$$\nabla_\xi \left( \frac{1}{2} \left( \boldsymbol{\xi}^{t+1} \right)^T \boldsymbol{\xi}^{t+1} \; + \; C_1 \right) \; = \; \nabla_\xi \text{lse}(\beta, \boldsymbol{X}^T \boldsymbol{\xi}^t) \,. \tag{28}$$

The resulting update rule is

$$\boldsymbol{\xi}^{t+1} \; = \; \boldsymbol{X}\boldsymbol{p}^t \; = \; \boldsymbol{X}\text{softmax}(\beta \boldsymbol{X}^T \boldsymbol{\xi}^t) \tag{29}$$

using

$$\boldsymbol{p}^t \; = \; \text{softmax}(\beta \boldsymbol{X}^T \boldsymbol{\xi}^t) \,. \tag{30}$$

This is the update rule in Eq. (20).
Theorem 2 in [51] and Theorem 2 in [52] state that the update rule Eq. (20) is guaranteed to monotonically decrease the energy E as a function of time. See also Theorem 2 in [39]. □

Although the objective converges in all cases, it does not necessarily converge to a local minimum [30].
However the convergence proof of CCCP in [51, 52] was not as rigorous as required. In [39] a rigorous analysis of the convergence of CCCP is performed using Zangwill's global convergence theory of iterative algorithms.
In [39] the minimization problem

$$\min_{\boldsymbol{\xi}} \; \text{E}_1 \; + \; \text{E}_2 \tag{31}$$

$$\text{s.t.} \;\; \boldsymbol{c}(\boldsymbol{\xi}) \leqslant \boldsymbol{0} \,, \quad \boldsymbol{d}(\boldsymbol{\xi}) \; = \; \boldsymbol{0}$$

is considered with $E_1$ convex, $-E_2$ convex, $\boldsymbol{c}$ component-wise convex function, and $\boldsymbol{d}$ an affine function. The CCCP algorithm solves this minimization problem by linearization of the concave part and is defined in [39] as

$$\boldsymbol{\xi}^{t+1} \in \arg\min_{\boldsymbol{\xi}} \; E_1(\boldsymbol{\xi}) + \boldsymbol{\xi}^T \nabla_\xi E_2(\boldsymbol{\xi}^t) \tag{32}$$

$$\text{s.t.} \quad \boldsymbol{c}(\boldsymbol{\xi}) \leqslant \boldsymbol{0}, \quad \boldsymbol{d}(\boldsymbol{\xi}) = \boldsymbol{0} .$$

We define the upper bound $E_C$ on the energy:

$$E_C(\boldsymbol{\xi}, \boldsymbol{\xi}^t) := E_1(\boldsymbol{\xi}) + E_2(\boldsymbol{\xi}^t) + (\boldsymbol{\xi} - \boldsymbol{\xi}^t)^T \nabla_\xi E_2(\boldsymbol{\xi}^t) . \tag{33}$$

$E_C$ is equal to the energy $E(\boldsymbol{\xi}^t)$ for $\boldsymbol{\xi} = \boldsymbol{\xi}^t$:

$$E_C(\boldsymbol{\xi}^t, \boldsymbol{\xi}^t) = E_1(\boldsymbol{\xi}^t) + E_2(\boldsymbol{\xi}^t) = E(\boldsymbol{\xi}^t) . \tag{34}$$

Since $-E_2$ is convex, the first order characterization of convexity holds (Eq. 3.2 in [9]):

$$- E_2(\boldsymbol{\xi}) \geq - E_2(\boldsymbol{\xi}^t) - (\boldsymbol{\xi} - \boldsymbol{\xi}^t)^T \nabla_\xi E_2(\boldsymbol{\xi}^t) , \tag{35}$$

that is

$$E_2(\boldsymbol{\xi}) \leqslant E_2(\boldsymbol{\xi}^t) + (\boldsymbol{\xi} - \boldsymbol{\xi}^t)^T \nabla_\xi E_2(\boldsymbol{\xi}^t) . \tag{36}$$

Therefore, for $\boldsymbol{\xi} \neq \boldsymbol{\xi}^t$ the function $E_C$ is an upper bound on the energy:

$$E(\boldsymbol{\xi}) \leqslant E_C(\boldsymbol{\xi}, \boldsymbol{\xi}^t) = E_1(\boldsymbol{\xi}) + E_2(\boldsymbol{\xi}^t) + (\boldsymbol{\xi} - \boldsymbol{\xi}^t)^T \nabla_\xi E_2(\boldsymbol{\xi}^t) \tag{37}$$

$$= E_1(\boldsymbol{\xi}) + \boldsymbol{\xi}^T \nabla_\xi E_2(\boldsymbol{\xi}^t) + C_2 ,$$

where $C_2$ does not depend on $\boldsymbol{\xi}$. Since we do not have constraints, $\boldsymbol{\xi}^{t+1}$ is defined as

$$\boldsymbol{\xi}^{t+1} \in \arg\min_{\boldsymbol{\xi}} \; E_C(\boldsymbol{\xi}, \boldsymbol{\xi}^t) , \tag{38}$$

hence $E_C(\boldsymbol{\xi}^{t+1}, \boldsymbol{\xi}^t) \leqslant E_C(\boldsymbol{\xi}^t, \boldsymbol{\xi}^t)$. Combining the inequalities gives:

$$E(\boldsymbol{\xi}^{t+1}) \leqslant E_C(\boldsymbol{\xi}^{t+1}, \boldsymbol{\xi}^t) \leqslant E_C(\boldsymbol{\xi}^t, \boldsymbol{\xi}^t) = E(\boldsymbol{\xi}^t) . \tag{39}$$

Since we do not have constraints, $\boldsymbol{\xi}^{t+1}$ is the minimum of

$$E_C(\boldsymbol{\xi}, \boldsymbol{\xi}^t) = E_1(\boldsymbol{\xi}) + \boldsymbol{\xi}^T \nabla_\xi E_2(\boldsymbol{\xi}^t) + C_2 \tag{40}$$

as a function of $\boldsymbol{\xi}$.
For a minimum not at the border, the derivative has to be the zero vector

$$\frac{\partial E_C(\boldsymbol{\xi}, \boldsymbol{\xi}^t)}{\partial \boldsymbol{\xi}} = \boldsymbol{\xi} + \nabla_\xi E_2(\boldsymbol{\xi}^t) = \boldsymbol{\xi} - \boldsymbol{X}\text{softmax}(\beta \boldsymbol{X}^T \boldsymbol{\xi}^t) = \boldsymbol{0} \tag{41}$$

and the Hessian must be positive semi-definite

$$\frac{\partial^2 E_C(\boldsymbol{\xi}, \boldsymbol{\xi}^t)}{\partial \boldsymbol{\xi}^2} = \boldsymbol{I} . \tag{42}$$

The Hessian is strict positive definite everywhere, therefore the optimization problem is strict convex (if the domain is convex) and there exist only one minimum, which is a global minimum. $E_C$ can even be written as a quadratic form:

$$E_C(\boldsymbol{\xi}, \boldsymbol{\xi}^t) = \frac{1}{2} (\boldsymbol{\xi} + \nabla_\xi E_2(\boldsymbol{\xi}^t))^T (\boldsymbol{\xi} + \nabla_\xi E_2(\boldsymbol{\xi}^t)) + C_3 , \tag{43}$$

where $C_3$ does not depend on $\boldsymbol{\xi}$.
Therefore the minimum is

$$\boldsymbol{\xi}^{t+1} = - \nabla_\xi E_2(\boldsymbol{\xi}^t) = \boldsymbol{X}\text{softmax}(\beta \boldsymbol{X}^T \boldsymbol{\xi}^t) \tag{44}$$

if it is in the domain as we assume.
Using $M = \max_i \|\boldsymbol{x}_i\|$, $\boldsymbol{\xi}^{t+1}$ is in the sphere $S = \{\boldsymbol{x} \mid \|\boldsymbol{x}\| \leqslant M\}$ which is a convex and compact set. Hence, if $\boldsymbol{\xi}^0 \in S$, then the iterate is a mapping from S to S. Therefore the point-set-map defined by the iteration Eq. (44) is uniformly compact on S according to Remark 7 in [39]. Theorem 2 and

Theorem 4 in [39] states that all the limit points of the iteration Eq. (44) are stationary points. These theorems follow from Zangwill's global convergence theorem: Convergence Theorem A, page 91 in [53] and page 3 in [49].

The global convergence theorem only assures that for the sequence $\boldsymbol{\xi}^{t+1} = f(\boldsymbol{\xi}^t)$ and a function $\Phi$ we have $\Phi(\boldsymbol{\xi}^t) \to \Phi(\boldsymbol{\xi}^*)$ for $t \to \infty$ but not $\boldsymbol{\xi}^t \to \boldsymbol{\xi}^*$. However, if $f$ is strictly monotone with respect to $\Phi$, then we can strengthen Zangwill's global convergence theorem [34]. We set $\Phi = E$ and show $E(\boldsymbol{\xi}^{t+1}) < E(\boldsymbol{\xi}^t)$ if $\boldsymbol{\xi}^t$ is not a stationary point of E, that is, $f$ is strictly monotone with respect to E. The following theorem is similar to the convergence results for the expectation maximization (EM) algorithm in [49] which are given in theorems 1 to 6 in [49]. The following theorem is also very similar to Theorem 8 in [39].

**Theorem B2** (Global Convergence: Stationary Points). *For the iteration Eq. (44) we have* $E\left(\boldsymbol{\xi}^t\right) \to E\left(\boldsymbol{\xi}^*\right) = E^*$ *as* $t \to \infty$, *for some stationary point* $\boldsymbol{\xi}^*$. *Furthermore* $\left\|\boldsymbol{\xi}^{t+1} - \boldsymbol{\xi}^t\right\| \to 0$ *and either* $\{\boldsymbol{\xi}^t\}_{t=0}^\infty$ *converges or, in the other case, the set of limit points of* $\{\boldsymbol{\xi}^t\}_{t=0}^\infty$ *is a connected and compact subset of* $\mathcal{L}(E*)$, *where* $\mathcal{L}(a) = \{\boldsymbol{\xi} \in \mathcal{L} \mid E(\boldsymbol{\xi}) = a\}$ *and* $\mathcal{L}$ *is the set of stationary points of the iteration Eq. (44). If* $\mathcal{L}(E^*)$ *is finite, then any sequence* $\{\boldsymbol{\xi}^t\}_{t=0}^\infty$ *generated by the iteration Eq. (44) converges to some* $\boldsymbol{\xi}^* \in \mathcal{L}(E^*)$.

*Proof.* We have $E\left(\boldsymbol{\xi}^t\right) = E_1\left(\boldsymbol{\xi}^t\right) + E_2\left(\boldsymbol{\xi}^t\right)$. The gradient $\nabla_\xi E_2\left(\boldsymbol{\xi}^t\right) = -\nabla_\xi \mathrm{lse}(\beta, \boldsymbol{X}^T\boldsymbol{\xi})$ is continuous. Therefore Eq. (40) has minimum in the sphere S, which is a convex and compact set. If $\boldsymbol{\xi}^{t+1} \neq \boldsymbol{\xi}^t$, then $\boldsymbol{\xi}^t$ was not the minimum of Eq. (37) as the derivative at $\boldsymbol{\xi}^t$ is not equal to zero. Eq. (42) shows that the optimization problem Eq. (37) is strict convex, hence it has only one minimum, which is a global minimum. Eq. (43) shows that the optimization problem Eq. (37) is even a quadratic form. Therefore we have

$$E\left(\boldsymbol{\xi}^{t+1}\right) \leqslant E_C\left(\boldsymbol{\xi}^{t+1}, \boldsymbol{\xi}^t\right) < E_C\left(\boldsymbol{\xi}^t, \boldsymbol{\xi}^t\right) = E\left(\boldsymbol{\xi}^t\right) . \tag{45}$$

Therefore the point-set-map defined by the iteration Eq. (44) (for definitions see [39]) is strictly monotonic with respect to E. Therefore we can apply Theorem 3 in [39] or Theorem 3.1 and Corollary 3.2 in [34], which give the statements of the theorem.

$\square$

We showed global convergence of the iteration Eq. (20). We have shown that all the limit points of any sequence generated by the iteration Eq. (20) are the stationary points (local minima or saddle points) of the energy function E. Local maxima as stationary points are only possible if the iterations exactly hits a maximum. However, a local maximum as an accumulation of different iteration points is not possible because Eq. (45) ensures a strict decrease of the energy E. Therefore almost sure local maxima are not obtained as stationary points. Either the iteration converges or, in the second case, the set of limit points is a connected and compact set. But what happens if $\boldsymbol{\xi}^0$ is in an $\epsilon$-neighborhood around a local minimum $\boldsymbol{\xi}^*$? Will the iteration Eq. (20) converge to $\boldsymbol{\xi}^*$? What is the rate of convergence? These questions are about *local convergence* which will be treated in detail in next section.

### B2.4 Local Convergence of the Update Rule: Fixed Point Iteration

For the proof of local convergence to a fixed point we will apply Banach fixed point theorem. For the rate of convergence we will rely on properties of a contraction mapping.

### B2.4.1 General Bound on the Jacobian of the Iterate

We consider the iteration

$$\boldsymbol{\xi}^{\mathrm{new}} = f(\boldsymbol{\xi}) = \boldsymbol{X}\boldsymbol{p} = \boldsymbol{X}\mathrm{softmax}(\beta\boldsymbol{X}^T\boldsymbol{\xi}) \tag{46}$$

using

$$\boldsymbol{p} = \mathrm{softmax}(\beta\boldsymbol{X}^T\boldsymbol{\xi}) . \tag{47}$$

The Jacobian J is symmetric and has the following form:

$$J = \frac{\partial f(\boldsymbol{\xi})}{\partial \boldsymbol{\xi}} = \beta\,\boldsymbol{X}\left(\mathrm{diag}(\boldsymbol{p}) - \boldsymbol{p}\boldsymbol{p}^T\right)\boldsymbol{X}^T = \boldsymbol{X}J_s\boldsymbol{X}^T , \tag{48}$$

where $J_s$ is Jacobian of the softmax.

To analyze the local convergence of the iterate, we distinguish between the following three cases (see also Fig. B1). Here we only provide an informal discussion to give the reader some intuition. A rigorous formulation of the results can be found in the corresponding subsections.

a) If the patterns $\boldsymbol{x}_i$ are not well separated, the iterate goes to a fixed point close to the arithmetic mean of the vectors. In this case $\boldsymbol{p}$ is close to $p_i = 1/N$.

b) If the patterns $\boldsymbol{x}_i$ are well separated, then the iterate goes to the pattern to which the initial $\boldsymbol{\xi}$ is similar. If the initial $\boldsymbol{\xi}$ is similar to a vector $\boldsymbol{x}_i$ then it will converge to a vector close to $\boldsymbol{x}_i$ and $\boldsymbol{p}$ will converge to a vector close to $\boldsymbol{e}_i$.

c) If some vectors are similar to each other but well separated from all other vectors, then a so called metastable state between the similar vectors exists. Iterates that start near the metastable state converge to this metastable state.

● pattern     × fixed point     ◇ average pattern

Figure B1: The three cases of fixed points. **a) Stored patterns (fixed point is single pattern)**: patterns are stored if they are well separated. Each pattern $\boldsymbol{x}_i$ has a single fixed point $\boldsymbol{x}_i^*$ close to it. In the sphere $\mathrm{S}_i$, pattern $\boldsymbol{x}_i$ is the only pattern and $\boldsymbol{x}_i^*$ the only fixed point. **b) Metastable state (fixed point is average of similar patterns)**: $\boldsymbol{x}_i$ and $\boldsymbol{x}_j$ are similar to each other and not well separated. The fixed point $\boldsymbol{m}_{\boldsymbol{x}}^*$ is a metastable state that is close to the mean $\boldsymbol{m}_{\boldsymbol{x}}$ of the similar patterns. **c) Global fixed point (fixed point is average of all patterns)**: no pattern is well separated from the others. A single global fixed point $\boldsymbol{m}_{\boldsymbol{x}}^*$ exists that is close to the arithmetic mean $\boldsymbol{m}_{\boldsymbol{x}}$ of all patterns.

We begin with a bound on the Jacobian of the iterate, thereby heavily relying on the Jacobian of the softmax from Lemma 24.

**Lemma 2.** *For $N$ patterns $\boldsymbol{X} = (\boldsymbol{x}_1, \ldots, \boldsymbol{x}_N)$, $\boldsymbol{p} = \mathrm{softmax}(\beta \boldsymbol{X}^T \boldsymbol{\xi})$, $M = \max_i \|\boldsymbol{x}_i\|$, and $m = \max_i p_i(1 - p_i)$, the spectral norm of the Jacobian $\mathrm{J}$ of the fixed point iteration is bounded:*

$$\|\mathrm{J}\|_2 \; \leqslant \; 2\,\beta\,\|\boldsymbol{X}\|_2^2\, m \; \leqslant \; 2\,\beta\,N\,M^2\,m \;. \tag{49}$$

*If $p_{\max} = \max_i p_i \geq 1 - \epsilon$, then for the spectral norm of the Jacobian holds*

$$\|\mathrm{J}\|_2 \; \leqslant \; 2\,\beta\,N\,M^2\,\epsilon \; - \; 2\,\epsilon^2\,\beta\,N\,M^2 \; < \; 2\,\beta\,N\,M^2\,\epsilon \;. \tag{50}$$

*Proof.* With

$$\boldsymbol{p} \; = \; \mathrm{softmax}(\beta \boldsymbol{X}^T \boldsymbol{\xi}) \;, \tag{51}$$

the symmetric Jacobian J is

$$\mathrm{J} \; = \; \frac{\partial f(\boldsymbol{\xi})}{\partial \boldsymbol{\xi}} \; = \; \beta\,\boldsymbol{X}\left(\mathrm{diag}(\boldsymbol{p}) - \boldsymbol{p}\boldsymbol{p}^T\right)\boldsymbol{X}^T \; = \; \boldsymbol{X}\mathrm{J}_s\boldsymbol{X}^T \;, \tag{52}$$

where $\mathrm{J}_s$ is Jacobian of the softmax.
With $m = \max_i p_i(1 - p_i)$, Eq. (465) from Lemma 24 is

$$\|\mathrm{J}_s\|_2 \; = \; \beta\left\|\mathrm{diag}(\boldsymbol{p}) - \boldsymbol{p}\boldsymbol{p}^T\right\|_2 \; \leqslant \; 2\,m\,\beta \;. \tag{53}$$

Using this bound on $\|\mathrm{J}_s\|_2$, we obtain

$$\|\mathrm{J}\|_2 \; \leqslant \; \beta\left\|\boldsymbol{X}^T\right\|_2 \|\mathrm{J}_s\|_2 \|\boldsymbol{X}\|_2 \; \leqslant \; 2\,m\,\beta\,\|\boldsymbol{X}\|_2^2 \;. \tag{54}$$

The spectral norm $\|.\|_2$ is bounded by the Frobenius norm $\|.\|_F$ which can be expressed by the norm squared of its column vectors:

$$\|\boldsymbol{X}\|_2 \; \leqslant \; \|\boldsymbol{X}\|_F \; = \; \sqrt{\sum_i \|\boldsymbol{x}_i\|^2} \;. \tag{55}$$

Therefore, we obtain the first statement of the lemma:

$$\|\mathrm{J}\|_2 \ \leqslant \ 2\,\beta\,\|\boldsymbol{X}\|_2^2\,m \ \leqslant \ 2\,\beta\,N\,M^2\,m\,. \tag{56}$$

With $p_{\max} = \max_i p_i \geq 1 - \epsilon$ Eq. (469) in Lemma 24 is

$$\|\mathrm{J}_s\|_2 \ \leqslant \ 2\,\beta\,\epsilon \ - \ 2\,\epsilon^2\,\beta \ < \ 2\,\beta\,\epsilon\,. \tag{57}$$

Using this inequality, we obtain the second statement of the lemma:

$$\|\mathrm{J}\|_2 \ \leqslant \ 2\,\beta\,N\,M^2\,\epsilon \ - \ 2\,\epsilon^2\,\beta\,N\,M^2 \ < \ 2\,\beta\,N\,M^2\,\epsilon\,. \tag{58}$$

$$\square$$

We now define the "separation" $\Delta_i$ of a pattern $\boldsymbol{x}_i$ from data $\boldsymbol{X} = (\boldsymbol{x}_1, \ldots, \boldsymbol{x}_N)$ here, since it has an important role for the convergence properties of the iteration.

**Definition B1** (Separation of Patterns). *We define $\Delta_i$, i.e. the separation of pattern $\boldsymbol{x}_i$ from data $\boldsymbol{X} = (\boldsymbol{x}_1, \ldots, \boldsymbol{x}_N)$ as:*

$$\Delta_i \ = \ \min_{j,j\neq i}\left(\boldsymbol{x}_i^T\boldsymbol{x}_i \ - \ \boldsymbol{x}_i^T\boldsymbol{x}_j\right) \ = \ \boldsymbol{x}_i^T\boldsymbol{x}_i \ - \ \max_{j,j\neq i}\boldsymbol{x}_i^T\boldsymbol{x}_j\,. \tag{59}$$

*The pattern is separated from the other data if $0 < \Delta_i$. Using the parallelogram identity, $\Delta_i$ can also be expressed as*

$$\Delta_i \ = \ \min_{j,j\neq i} \frac{1}{2}\left(\|\boldsymbol{x}_i\|^2 \ - \ \|\boldsymbol{x}_j\|^2 \ + \ \|\boldsymbol{x}_i \ - \ \boldsymbol{x}_j\|^2\right) \tag{60}$$

$$= \ \frac{1}{2}\|\boldsymbol{x}_i\|^2 \ - \ \frac{1}{2}\max_{j,j\neq i}\left(\|\boldsymbol{x}_j\|^2 \ - \ \|\boldsymbol{x}_i \ - \ \boldsymbol{x}_j\|^2\right)\,.$$

*For $\|\boldsymbol{x}_i\| = \|\boldsymbol{x}_j\|$ we have $\Delta_i = 1/2\min_{j,j\neq i}\|\boldsymbol{x}_i \ - \ \boldsymbol{x}_j\|^2$.*
*Analog we say for a query $\boldsymbol{\xi}$ and data $\boldsymbol{X} = (\boldsymbol{x}_1, \ldots, \boldsymbol{x}_N)$, that $\boldsymbol{x}_i$ is least separated from $\boldsymbol{\xi}$ while being separated from other $\boldsymbol{x}_j$ with $j \neq i$ if*

$$i \ = \ \arg\max_k \min_{j,j\neq k}\left(\boldsymbol{\xi}^T\boldsymbol{x}_k \ - \ \boldsymbol{\xi}^T\boldsymbol{x}_j\right) \ = \ \arg\max_k\left(\boldsymbol{\xi}^T\boldsymbol{x}_k \ - \ \max_{j,j\neq k}\boldsymbol{\xi}^T\boldsymbol{x}_j\right) \tag{61}$$

$$0 \ \leqslant \ c \ = \ \max_k \min_{j,j\neq k}\left(\boldsymbol{\xi}^T\boldsymbol{x}_k \ - \ \boldsymbol{\xi}^T\boldsymbol{x}_j\right) \ = \ \max_k\left(\boldsymbol{\xi}^T\boldsymbol{x}_k \ - \ \max_{j,j\neq k}\boldsymbol{\xi}^T\boldsymbol{x}_j\right)\,. \tag{62}$$

Next we consider the case where the iteration has only one stable fixed point.

### B2.4.2   One Stable State: Fixed Point Near the Mean of the Patterns

We start with the case where no pattern is well separated from the others.

**Global Fixed Point Near the Global Mean: Analysis Using the Data Center.**   We revisit the bound on the Jacobian of the iterate by utilizing properties of pattern distributions. We begin with a probabilistic interpretation where we consider $p_i$ as the probability of selecting the vector $\boldsymbol{x}_i$. Consequently, we define expectations as $\mathrm{E}_{\boldsymbol{p}}[f(\boldsymbol{x})] = \sum_{i=1}^N p_i f(\boldsymbol{x}_i)$. In this setting the matrix

$$\boldsymbol{X}\left(\mathrm{diag}(\boldsymbol{p}) - \boldsymbol{p}\boldsymbol{p}^T\right)\boldsymbol{X}^T \tag{63}$$

is the covariance matrix of data $\boldsymbol{X}$ when its vectors are selected according to the probability $\boldsymbol{p}$:

$$\boldsymbol{X}\left(\mathrm{diag}(\boldsymbol{p}) - \boldsymbol{p}\boldsymbol{p}^T\right)\boldsymbol{X}^T \ = \ \boldsymbol{X}\mathrm{diag}(\boldsymbol{p})\boldsymbol{X}^T \ - \ \boldsymbol{X}\boldsymbol{p}\boldsymbol{p}^T\boldsymbol{X}^T \tag{64}$$

$$= \ \sum_{i=1}^N p_i\,\boldsymbol{x}_i\,\boldsymbol{x}_i^T \ - \ \left(\sum_{i=1}^N p_i\,\boldsymbol{x}_i\right)\left(\sum_{i=1}^N p_i\,\boldsymbol{x}_i\right)^T \tag{65}$$

$$= \ \mathrm{E}_{\boldsymbol{p}}[\boldsymbol{x}\,\boldsymbol{x}^T] \ - \ \mathrm{E}_{\boldsymbol{p}}[\boldsymbol{x}]\,\mathrm{E}_{\boldsymbol{p}}[\boldsymbol{x}]^T \ = \ \mathrm{Var}_{\boldsymbol{p}}[\boldsymbol{x}]\,, \tag{66}$$

therefore we have

$$\mathrm{J} \ = \ \beta\,\mathrm{Var}_{\boldsymbol{p}}[\boldsymbol{x}]\,. \tag{67}$$

The largest eigenvalue of the covariance matrix (equal to the largest singular value) is the variance in the direction of the eigenvector associated with the largest eigenvalue.

We define:

$$\boldsymbol{m_x} = \frac{1}{N} \sum_{i=1}^{N} \boldsymbol{x}_i \,, \tag{68}$$

$$m_{\max} = \max_{1 \leqslant i \leqslant N} \|\boldsymbol{x}_i - \boldsymbol{m_x}\|_2 \,. \tag{69}$$

$\boldsymbol{m_x}$ is the arithmetic mean (the center) of the patterns. $m_{\max}$ is the maximal distance of the patterns to the center $\boldsymbol{m_x}$ .

The variance of the patterns is

$$\mathrm{Var}_{\boldsymbol{p}}[\boldsymbol{x}] = \sum_{i=1}^{N} p_i \, \boldsymbol{x}_i \, \boldsymbol{x}_i^T - \left( \sum_{i=1}^{N} p_i \, \boldsymbol{x}_i \right) \left( \sum_{i=1}^{N} p_i \, \boldsymbol{x}_i \right)^T \tag{70}$$

$$= \sum_{i=1}^{N} p_i \left( \boldsymbol{x}_i - \sum_{i=1}^{N} p_i \boldsymbol{x}_i \right) \left( \boldsymbol{x}_i - \sum_{i=1}^{N} p_i \boldsymbol{x}_i \right)^T \,.$$

The maximal distance to the center $m_{\max}$ allows to derive a bound on the norm of the Jacobian. Next lemma gives a condition for a global fixed point.

**Lemma 3.** *The following bound on the norm $\|\mathrm{J}\|_2$ of the Jacobian of the fixed point iteration $f$ holds independent of $\boldsymbol{p}$ or the query $\boldsymbol{\xi}$.*

$$\|\mathrm{J}\|_2 \leqslant \beta \, m_{\max}^2 \,. \tag{71}$$

*For $\beta \, m_{\max}^2 < 1$ there exists a unique fixed point (global fixed point) of iteration $f$ in each compact set.*

*Proof.* In order to bound the variance we compute the vector $\boldsymbol{a}$ that minimizes

$$f(\boldsymbol{a}) = \sum_{i=1}^{N} p_i \|\boldsymbol{x}_i - \boldsymbol{a}\|^2 = \sum_{i=1}^{N} p_i (\boldsymbol{x}_i - \boldsymbol{a})^T (\boldsymbol{x}_i - \boldsymbol{a}) \,. \tag{72}$$

The solution to

$$\frac{\partial f(\boldsymbol{a})}{\partial \boldsymbol{a}} = 2 \sum_{i=1}^{N} p_i (\boldsymbol{a} - \boldsymbol{x}_i) = 0 \tag{73}$$

is

$$\boldsymbol{a} = \sum_{i=1}^{N} p_i \boldsymbol{x}_i \,. \tag{74}$$

The Hessian of $f$ is positive definite since

$$\frac{\partial^2 f(\boldsymbol{a})}{\partial \boldsymbol{a}^2} = 2 \sum_{i=1}^{N} p_i \, \boldsymbol{I} = 2 \, \boldsymbol{I} \tag{75}$$

and $f$ is a convex function. Hence, the mean

$$\bar{\boldsymbol{x}} := \sum_{i=1}^{N} p_i \, \boldsymbol{x}_i \tag{76}$$

minimizes $\sum_{i=1}^{N} p_i \|\boldsymbol{x}_i - \boldsymbol{a}\|^2$. Therefore we have

$$\sum_{i=1}^{N} p_i \|\boldsymbol{x}_i - \bar{\boldsymbol{x}}\|^2 \leqslant \sum_{i=1}^{N} p_i \|\boldsymbol{x}_i - \boldsymbol{m_x}\|^2 \leqslant m_{\max}^2 \,. \tag{77}$$

Let us quickly recall that the spectral norm of an outer product of two vectors is the product of the Euclidean norms of the vectors:

$$\|\boldsymbol{a}\boldsymbol{b}^T\|_2 = \sqrt{\lambda_{\max}(\boldsymbol{b}\boldsymbol{a}^T \boldsymbol{a}\boldsymbol{b}^T)} = \|\boldsymbol{a}\| \sqrt{\lambda_{\max}(\boldsymbol{b}\boldsymbol{b}^T)} = \|\boldsymbol{a}\| \, \|\boldsymbol{b}\| \,, \tag{78}$$

since $\boldsymbol{b}\boldsymbol{b}^T$ has eigenvector $\boldsymbol{b}/\|\boldsymbol{b}\|$ with eigenvalue $\|\boldsymbol{b}\|^2$ and otherwise zero eigenvalues.
We now bound the variance of the patterns:

$$\|\mathrm{Var}_{\boldsymbol{p}}[\boldsymbol{x}]\|_2 \; \leqslant \; \sum_{i=1}^{N} p_i \Big\| (\boldsymbol{x}_i \; - \; \bar{\boldsymbol{x}}) \; (\boldsymbol{x}_i \; - \; \bar{\boldsymbol{x}})^T \Big\|_2 \tag{79}$$

$$= \; \sum_{i=1}^{N} p_i \|\boldsymbol{x}_i \; - \; \bar{\boldsymbol{x}}\|^2 \; \leqslant \; \sum_{i=1}^{N} p_i \|\boldsymbol{x}_i \; - \; \boldsymbol{m}_{\boldsymbol{x}}\|^2 \; \leqslant \; m_{\max}^2 \; .$$

The bound of the lemma on $\|\mathrm{J}\|_2$ follows from Eq. (67).
For $\|\mathrm{J}\|_2 \leqslant \beta \, m_{\max}^2 < 1$ we have a contraction mapping on each compact set. Banach fixed point theorem says there is a unique fixed point in the compact set.

$\square$

Now let us further investigate the tightness of the bound on $\|\mathrm{Var}_{\boldsymbol{p}}[\boldsymbol{x}]\|_2$ via $\|\boldsymbol{x}_i - \bar{\boldsymbol{x}}\|^2$: we consider the trace, which is the sum $\sum_{k=1}^{d} e_k$ of the w.l.o.g. ordered nonnegative eigenvalues $e_k$ of $\mathrm{Var}_{\boldsymbol{p}}[\boldsymbol{x}]$ The spectral norm is equal to the largest eigenvalue $e_1$, which is equal to the largest singular value, as we have positive semidefinite matrices. We obtain:

$$\|\mathrm{Var}_{\boldsymbol{p}}[\boldsymbol{x}]\|_2 \; = \; \mathrm{Tr}\left( \sum_{i=1}^{N} p_i \, (\boldsymbol{x}_i \; - \; \bar{\boldsymbol{x}}) \; (\boldsymbol{x}_i \; - \; \bar{\boldsymbol{x}})^T \right) \; - \; \sum_{k=2}^{d} e_k \tag{80}$$

$$= \; \sum_{i=1}^{N} p_i \mathrm{Tr}\left( (\boldsymbol{x}_i \; - \; \bar{\boldsymbol{x}}) \; (\boldsymbol{x}_i \; - \; \bar{\boldsymbol{x}})^T \right) \; - \; \sum_{k=2}^{d} e_k$$

$$= \; \sum_{i=1}^{N} p_i \|\boldsymbol{x}_i \; - \; \bar{\boldsymbol{x}}\|^2 \; - \; \sum_{k=2}^{d} e_k \; .$$

Therefore the tightness of the bound depends on eigenvalues which are not the largest. Hence variations which are not along the largest variation weaken the bound.

Next we investigate the location of fixed points which existence is ensured by the global convergence stated in Theorem B2. For $N$ patterns $\boldsymbol{X} = (\boldsymbol{x}_1, \ldots, \boldsymbol{x}_N)$, we consider the iteration

$$\boldsymbol{\xi}^{\mathrm{new}} \; = \; f(\boldsymbol{\xi}) \; = \; \boldsymbol{X}\boldsymbol{p} \; = \; \boldsymbol{X}\mathrm{softmax}(\beta \boldsymbol{X}^T \boldsymbol{\xi}) \tag{81}$$

using

$$\boldsymbol{p} \; = \; \mathrm{softmax}(\beta \boldsymbol{X}^T \boldsymbol{\xi}) \; . \tag{82}$$

$\boldsymbol{\xi}^{\mathrm{new}}$ is in the simplex of the patterns, that is, $\boldsymbol{\xi}^{\mathrm{new}} = \sum_i p_i \boldsymbol{x}_i$ with $\sum_i p_i = 1$ and $0 \leqslant p_i$. Hence, after one update $\boldsymbol{\xi}$ is in the simplex of the pattern and stays there. If the center $\boldsymbol{m}_{\boldsymbol{x}}$ is the zero vector $\boldsymbol{m}_{\boldsymbol{x}} = \boldsymbol{0}$, that is, the data is centered, then the mean is a fixed point of the iteration. For $\boldsymbol{\xi} = \boldsymbol{m}_{\boldsymbol{x}} = \boldsymbol{0}$ we have

$$\boldsymbol{p} \; = \; 1/N \, \boldsymbol{1} \tag{83}$$

and

$$\boldsymbol{\xi}^{\mathrm{new}} \; = \; 1/N \, \boldsymbol{X} \, \boldsymbol{1} \; = \; \boldsymbol{m}_{\boldsymbol{x}} \; = \; \boldsymbol{\xi} \; . \tag{84}$$

In particular normalization methods like batch normalization would promote the mean as a fixed point.
We consider the differences of dot products for $\boldsymbol{x}_i$: $\boldsymbol{x}_i^T \boldsymbol{x}_i - \boldsymbol{x}_i^T \boldsymbol{x}_j = \boldsymbol{x}_i^T (\boldsymbol{x}_i - \boldsymbol{x}_j)$, for fixed point $\boldsymbol{m}_{\boldsymbol{x}}^*$: $(\boldsymbol{m}_{\boldsymbol{x}}^*)^T \boldsymbol{x}_i - (\boldsymbol{m}_{\boldsymbol{x}}^*)^T \boldsymbol{x}_j = (\boldsymbol{m}_{\boldsymbol{x}}^*)^T (\boldsymbol{x}_i - \boldsymbol{x}_j)$, and for the center $\boldsymbol{m}_{\boldsymbol{x}}$: $\boldsymbol{m}_{\boldsymbol{x}}^T \boldsymbol{x}_i - \boldsymbol{m}_{\boldsymbol{x}}^T \boldsymbol{x}_j = \boldsymbol{m}_{\boldsymbol{x}}^T (\boldsymbol{x}_i - \boldsymbol{x}_j)$.
Using the Cauchy-Schwarz inequality, we get

$$\left| \boldsymbol{\xi}^T (\boldsymbol{x}_i \; - \; \boldsymbol{x}_j) \right| \; \leqslant \; \|\boldsymbol{\xi}\| \, \|\boldsymbol{x}_i \; - \; \boldsymbol{x}_j\| \; \leqslant \; \|\boldsymbol{\xi}\| \, (\|\boldsymbol{x}_i \; - \; \boldsymbol{m}_{\boldsymbol{x}}\| \; + \; \|\boldsymbol{x}_j \; - \; \boldsymbol{m}_{\boldsymbol{x}}\|) \tag{85}$$

$$\leqslant \; 2 \, m_{\max} \, \|\boldsymbol{\xi}\| \; .$$

This inequality gives:

$$\left|\boldsymbol{\xi}^T(\boldsymbol{x}_i - \boldsymbol{x}_j)\right| \leqslant 2\, m_{\max}\,(m_{\max} + \|\boldsymbol{m_x}\|)\,, \tag{86}$$

$$\left|\boldsymbol{\xi}^T(\boldsymbol{x}_i - \boldsymbol{x}_j)\right| \leqslant 2\, m_{\max}\, M\,,$$

where we used $\|\boldsymbol{\xi} - \boldsymbol{0}\| \leqslant \|\boldsymbol{\xi} - \boldsymbol{m_x}\| + \|\boldsymbol{m_x} - \boldsymbol{0}\|$, $\|\boldsymbol{\xi} - \boldsymbol{m_x}\| = \|\sum_i p_i \boldsymbol{x}_i - \boldsymbol{m_x}\| \leqslant \sum_i p_i \|\boldsymbol{x}_i - \boldsymbol{m_x}\| \leqslant m_{\max}$, and $M = \max_i \|\boldsymbol{x}_i\|$. In particular

$$\beta\left|\boldsymbol{m_x}^T(\boldsymbol{x}_i - \boldsymbol{x}_j)\right| \leqslant 2\,\beta\, m_{\max}\,\|\boldsymbol{m_x}\|\,, \tag{87}$$

$$\beta\left|(\boldsymbol{m_x^*})^T(\boldsymbol{x}_i - \boldsymbol{x}_j)\right| \leqslant 2\,\beta\, m_{\max}\,\|\boldsymbol{m_x^*}\| \leqslant 2\,\beta\, m_{\max}\,(m_{\max} + \|\boldsymbol{m_x}\|)\,, \tag{88}$$

$$\beta\left|\boldsymbol{x}_i^T(\boldsymbol{x}_i - \boldsymbol{x}_j)\right| \leqslant 2\,\beta\, m_{\max}\,\|\boldsymbol{x}_i\| \leqslant 2\,\beta\, m_{\max}\,(m_{\max} + \|\boldsymbol{m_x}\|)\,. \tag{89}$$

Let $i = \arg\max_j \boldsymbol{\xi}^T \boldsymbol{x}_j$, therefore the maximal softmax component is $i$. For the maximal softmax component $i$ we have:

$$[\text{softmax}(\beta\, \boldsymbol{X}^T \boldsymbol{\xi})]_i = \frac{1}{1 + \sum_{j\neq i}\exp(-\beta\,(\boldsymbol{\xi}^T \boldsymbol{x}_i - \boldsymbol{\xi}^T \boldsymbol{x}_j))} \tag{90}$$

$$\leqslant \frac{1}{1 + \sum_{j\neq i}\exp(-2\,\beta\, m_{\max}\,(m_{\max} + \|\boldsymbol{m_x}\|))}$$

$$= \frac{1}{1 + (N-1)\exp(-2\,\beta\, m_{\max}\,(m_{\max} + \|\boldsymbol{m_x}\|))}$$

$$= \frac{\exp(2\,\beta\, m_{\max}\,(m_{\max} + \|\boldsymbol{m_x}\|))}{\exp(2\,\beta\, m_{\max}\,(m_{\max} + \|\boldsymbol{m_x}\|)) + (N-1)}$$

$$\leqslant 1/N \,\exp(2\,\beta\, m_{\max}\,(m_{\max} + \|\boldsymbol{m_x}\|))\,.$$

Analogously we obtain for $i = \arg\max_j \boldsymbol{m_x}^T \boldsymbol{x}_j$, a bound on the maximal softmax component $i$ if the center is put into the iteration:

$$[\text{softmax}(\beta\, \boldsymbol{X}^T \boldsymbol{m_x})]_i \leqslant 1/N \,\exp(2\,\beta\, m_{\max}\,\|\boldsymbol{m_x}\|)\,. \tag{91}$$

Analog we obtain a bound for $i = \arg\max_j (\boldsymbol{m_x^*})^T \boldsymbol{x}_j$ on the maximal softmax component $i$ of the fixed point:

$$[\text{softmax}(\beta\, \boldsymbol{X}^T \boldsymbol{m_x^*})]_i \leqslant 1/N \,\exp(2\,\beta\, m_{\max}\,\|\boldsymbol{m_x^*}\|) \tag{92}$$

$$\leqslant 1/N \,\exp(2\,\beta\, m_{\max}\,(m_{\max} + \|\boldsymbol{m_x}\|))\,.$$

The two important terms are $m_{\max}$, the variance or spread of the data and $\|\boldsymbol{m_x}\|$, which tells how well the data is centered. For a contraction mapping we already required $\beta m_{\max}^2 < 1$, therefore the first term in the exponent is $2\beta m_{\max}^2 < 2$. The second term $2\beta m_{\max}\|\boldsymbol{m_x}\|$ is small if the data is centered.

**Global Fixed Point Near the Global Mean: Analysis Using Softmax Values.** If $\boldsymbol{\xi}^T \boldsymbol{x}_i \approx \boldsymbol{\xi}^T \boldsymbol{x}_j$ for all $i$ and $j$, then $p_i \approx 1/N$ and we have $m = \max_i p_i(1 - p_i) < 1/N$. For $M \leqslant 1/\sqrt{2\beta}$ we obtain from Lemma 2:

$$\|\text{J}\|_2 < 1\,. \tag{93}$$

The local fixed point is $\boldsymbol{m_x^*} \approx \boldsymbol{m_x} = (1/N)\sum_{i=1}^N \boldsymbol{x}_i$ with $p_i \approx 1/N$.

We now treat this case more formally. First we discuss conditions that ensure that the iteration is a contraction mapping. We consider the iteration Eq. (46) in the variable $\boldsymbol{p}$:

$$\boldsymbol{p}^{\text{new}} = g(\boldsymbol{p}) = \text{softmax}(\beta \boldsymbol{X}^T \boldsymbol{X} \boldsymbol{p})\,. \tag{94}$$

The Jacobian is

$$\text{J}(\boldsymbol{p}) = \frac{\partial g(\boldsymbol{p})}{\partial \boldsymbol{p}} = \boldsymbol{X}^T \boldsymbol{X}\, \text{J}_s \tag{95}$$

with

$$\text{J}_s(\boldsymbol{p}^{\text{new}}) = \beta\left(\text{diag}(\boldsymbol{p}^{\text{new}}) - \boldsymbol{p}^{\text{new}}(\boldsymbol{p}^{\text{new}})^T\right)\,. \tag{96}$$

The mean value theorem states for $\mathrm{J}^m = \int_0^1 \mathrm{J}(\lambda \boldsymbol{p}) \, \mathrm{d}\lambda = \boldsymbol{X}^T \boldsymbol{X} \mathrm{J}_s^m$ with the symmetric matrix $\mathrm{J}_s^m = \int_0^1 \mathrm{J}_s(\lambda \boldsymbol{p}) \, \mathrm{d}\lambda$:

$$\boldsymbol{p}^{\text{new}} \;=\; g(\boldsymbol{p}) \;=\; g(\boldsymbol{0}) \,+\, (\mathrm{J}^m)^T \boldsymbol{p} \;=\; g(\boldsymbol{0}) \,+\, \mathrm{J}_s^m \, \boldsymbol{X}^T \boldsymbol{X} \, \boldsymbol{p} \;=\; 1/N \, \boldsymbol{1} \,+\, \mathrm{J}_s^m \, \boldsymbol{X}^T \boldsymbol{X} \, \boldsymbol{p} \,. \quad (97)$$

With $m = \max_i p_i(1 - p_i)$, Eq. (465) from Lemma 24 is

$$\|\mathrm{J}_s(\boldsymbol{p})\|_2 \;=\; \beta \, \left\|\operatorname{diag}(\boldsymbol{p}) - \boldsymbol{p}\boldsymbol{p}^T\right\|_2 \;\leqslant\; 2 \, m \, \beta \,. \quad (98)$$

First observe that $\lambda p_i(1 - \lambda p_i) \leqslant p_i(1 - p_i)$ for $p_i \leqslant 0.5$ and $\lambda \in [0, 1]$, since $p_i(1 - p_i) - \lambda p_i(1 - \lambda p_i) = (1 - \lambda)p_i(1 - (1 + \lambda)p_i) \geq 0$. For $\max_i p_i \leqslant 0.5$ this observation leads to the following bound for $\mathrm{J}_s^m$:

$$\|\mathrm{J}_s^m\|_2 \;\leqslant\; 2 \, m \, \beta \,. \quad (99)$$

Eq. (468) in Lemma 24 states that every $\mathrm{J}_s$ is bounded by $1/2\beta$, therefore also the mean:

$$\|\mathrm{J}_s^m\|_2 \;\leqslant\; 0.5 \, \beta \,. \quad (100)$$

Since $m = \max_i p_i(1 - p_i) < \max_i p_i = p_{\max}$, the previous bounds can be combined as follows:

$$\|\mathrm{J}_s^m\|_2 \;\leqslant\; 2 \, \min\{0.25, p_{\max}\} \, \beta \,. \quad (101)$$

Consequently,

$$\|\mathrm{J}^m\|_2 \;\leqslant\; N \, M^2 \, 2 \, \min\{0.25, p_{\max}\} \, \beta \,, \quad (102)$$

where we used Eq. (159). $\left\|\boldsymbol{X}^T \boldsymbol{X}\right\|_2 = \left\|\boldsymbol{X}\boldsymbol{X}^T\right\|_2$, therefore $\left\|\boldsymbol{X}^T \boldsymbol{X}\right\|_2$ is $N$ times the maximal second moment of the data squared.

Obviously, $g(\boldsymbol{p})$ is a contraction mapping in compact sets, where

$$N \, M^2 \, 2 \, \min\{0.25, p_{\max}\} \, \beta \;<\; 1 \,. \quad (103)$$

S is the sphere around the origin $\boldsymbol{0}$ with radius one. For

$$\boldsymbol{p}^{\text{new}} \;=\; g(\boldsymbol{p}) \;=\; 1/N \, \boldsymbol{1} \,+\, \mathrm{J}^m \, \boldsymbol{p} \,, \quad (104)$$

we have $\|\boldsymbol{p}\| \leqslant \|\boldsymbol{p}\|_1 = 1$ and $\|\boldsymbol{p}^{\text{new}}\| \leqslant \|\boldsymbol{p}^{\text{new}}\|_1 = 1$. Therefore, $g$ maps points from S into S. $g$ is a contraction mapping for

$$\|\mathrm{J}^m\|_2 \;\leqslant\; N \, M^2 \, 2 \, \min\{0.25, p_{\max}\} \, \beta \;=\; c \;<\; 1 \,. \quad (105)$$

According to Banach fixed point theorem $g$ has a fixed point in the sphere S.

Hölder's inequality gives:

$$\|\boldsymbol{p}\|^2 \;=\; \boldsymbol{p}^T \boldsymbol{p} \;\leqslant\; \|\boldsymbol{p}\|_1 \|\boldsymbol{p}\|_\infty \;=\; \|\boldsymbol{p}\|_\infty \;=\; p_{\max} \,. \quad (106)$$

Alternatively:

$$\|\boldsymbol{p}\|^2 \;=\; \sum_i p_i^2 \;=\; p_{\max} \sum_i \frac{p_i}{p_{\max}} \, p_i \;\leqslant\; p_{\max} \sum_i p_i \;=\; p_{\max} \,. \quad (107)$$

Let now S be the sphere around the origin $\boldsymbol{0}$ with radius $1/\sqrt{N} + \sqrt{p_{\max}}$ and let $\|\mathrm{J}^m(\boldsymbol{p})\|_2 \leqslant c < 1$ for $\boldsymbol{p} \in$ S. The old $\boldsymbol{p}$ is in the sphere S ($\boldsymbol{p} \in$ S) since $p_{\max} < \sqrt{p_{\max}}$ for $p_{\max} < 1$. We have

$$\|\boldsymbol{p}^{\text{new}}\| \;\leqslant\; 1/\sqrt{N} \,+\, \|\mathrm{J}^m\|_2 \, \|\boldsymbol{p}\| \;\leqslant\; 1/\sqrt{N} \,+\, \sqrt{p_{\max}} \,. \quad (108)$$

Therefore $g$ is a mapping from S into S and a contraction mapping. According to Banach fixed point theorem, a fixed point exists in S.

For the 1-norm, we use Lemma 24 and $\|\boldsymbol{p}\|_1 = 1$ to obtain from Eq. (104):

$$\|\boldsymbol{p}^{\text{new}} \,-\, 1/N \, \boldsymbol{1}\|_1 \;\leqslant\; \|\mathrm{J}^m\|_1 \;\leqslant\; 2 \, \beta \, m \, \|\boldsymbol{X}\|_\infty \, M_1 \,, \quad (109)$$

$$\|\boldsymbol{p}^{\text{new}} \,-\, 1/N \, \boldsymbol{1}\|_1 \;\leqslant\; \|\mathrm{J}^m\|_1 \;\leqslant\; 2 \, \beta \, m \, N \, M_\infty \, M_1 \,, \quad (110)$$

$$\|\boldsymbol{p}^{\text{new}} \,-\, 1/N \, \boldsymbol{1}\|_1 \;\leqslant\; \|\mathrm{J}^m\|_1 \;\leqslant\; 2 \, \beta \, m \, N \, M^2 \,, \quad (111)$$

where $m = \max_i p_i(1 - p_i)$, $M_1 = \|\boldsymbol{X}\|_1 = \max_i \|\boldsymbol{x}_i\|_1$, $M = \max_i \|\boldsymbol{x}_i\|$, $\|\boldsymbol{X}\|_\infty = \|\boldsymbol{X}^T\|_1 = \max_i \|[X^T]_i\|_1$ (maximal absolute row sum norm), and $M_\infty = \max_i \|\boldsymbol{x}_i\|_\infty$. Let us quickly mention some auxiliary estimates related to $\boldsymbol{X}^T\boldsymbol{X}$:

$$\left\|\boldsymbol{X}^T\boldsymbol{X}\right\|_1 = \max_i \sum_{j=1}^{N} \left|\boldsymbol{x}_i^T \boldsymbol{x}_j\right| \leqslant \max_i \sum_{j=1}^{N} \|\boldsymbol{x}_i\|_\infty \|\boldsymbol{x}_j\|_1 \tag{112}$$

$$\leqslant M_\infty \sum_{j=1}^{N} M_1 = N\, M_\infty\, M_1\,,$$

where the first inequaltiy is from Hölder's inequality. We used

$$\left\|\boldsymbol{X}^T\boldsymbol{X}\right\|_1 = \max_i \sum_{j=1}^{N} \left|\boldsymbol{x}_i^T \boldsymbol{x}_j\right| \leqslant \max_i \sum_{j=1}^{N} \|\boldsymbol{x}_i\| \|\boldsymbol{x}_j\| \tag{113}$$

$$\leqslant M \sum_{j=1}^{N} M = N\, M^2\,,$$

where the first inequality is from Hölder's inequality (here the same as the Cauchy-Schwarz inequality). See proof of Lemma 24 for the 1-norm bound on $J_s$. Everything else follows from the fact that the 1-norm is sub-multiplicative as induced matrix norm.

We consider the minimal $\|\boldsymbol{p}\|$.

$$\min_{\boldsymbol{p}} \; \|\boldsymbol{p}\|^2 \tag{114}$$

$$\text{s.t.} \quad \sum_i p_i = 1$$

$$\forall_i : \; p_i \geq 0\,.$$

The solution to this minimization problem is $\boldsymbol{p} = (1/N)\mathbf{1}$. Therefore we have $1/\sqrt{N} \leqslant \|\boldsymbol{p}\|$ and $1/N \leqslant \|\boldsymbol{p}\|^2$ Using Eq. (108) we obtain

$$1/\sqrt{N} \leqslant \|\boldsymbol{p}^{\text{new}}\| \leqslant 1/\sqrt{N} + \sqrt{p_{\max}}\,. \tag{115}$$

Moreover

$$\|\boldsymbol{p}^{\text{new}}\|^2 = (\boldsymbol{p}^{\text{new}})^T \boldsymbol{p}^{\text{new}} = 1/N + (\boldsymbol{p}^{\text{new}})^T \mathrm{J}^m\, \boldsymbol{p} \leqslant 1/N + \|\mathrm{J}^m\|_2 \|\boldsymbol{p}\| \tag{116}$$

$$\leqslant 1/N + \|\mathrm{J}^m\|_2\,,$$

since $\boldsymbol{p}^{\text{new}} \in \mathrm{S}$ and $\boldsymbol{p} \in \mathrm{S}$.
For the fixed point, we have

$$\|\boldsymbol{p}^*\|^2 = (\boldsymbol{p}^*)^T \boldsymbol{p}^* = 1/N + (\boldsymbol{p}^*)^T \mathrm{J}^m\, \boldsymbol{p}^* \leqslant 1/N + \|\mathrm{J}^m\|_2 \|\boldsymbol{p}^*\|^2\,, \tag{117}$$

and hence

$$1/N \leqslant \|\boldsymbol{p}^*\|^2 \leqslant 1/N \frac{1}{1 - \|\mathrm{J}^m\|_2} = 1/N\left(1 + \frac{\|\mathrm{J}^m\|_2}{1 - \|\mathrm{J}^m\|_2}\right)\,. \tag{118}$$

Therefore, for small $\|\mathrm{J}^m\|_2$ we have $\boldsymbol{p}^* \approx (1/N)\mathbf{1}$.

### B2.4.3 Many Stable States: Fixed Points Near Stored Patterns

We move on to the next case, where the patterns $\boldsymbol{x}_i$ are well separated. In this case the iterate goes to the pattern to which the initial $\boldsymbol{\xi}$ is most similar. If the initial $\boldsymbol{\xi}$ is similar to a vector $\boldsymbol{x}_i$ then it will converge to $\boldsymbol{x}_i$ and $\boldsymbol{p}$ will be $\boldsymbol{e}_i$. The main ingredients are again Banach's Theorem and estimates on the Jacobian norm.

**Proof of a Fixed Point by Banach Fixed Point Theorem** *Mapped Vectors Stay in a Compact Environment.* We show that if $\boldsymbol{x}_i$ is sufficient dissimilar to other $\boldsymbol{x}_j$ then there is an compact environment of $\boldsymbol{x}_i$ (a sphere) where the fixed point iteration maps this environment into itself. The idea of the proof is to define a sphere around $\boldsymbol{x}_i$ for which points from the sphere are mapped by $f$ into the sphere.

We first need following lemma which bounds the distance $\|\boldsymbol{x}_i - f(\boldsymbol{\xi})\|$, where $\boldsymbol{x}_i$ is the pattern that is least separated from $\boldsymbol{\xi}$ but separated from other patterns.

**Lemma 4.** *For a query $\boldsymbol{\xi}$ and data $\boldsymbol{X} = (\boldsymbol{x}_1, \ldots, \boldsymbol{x}_N)$, there exists a $\boldsymbol{x}_i$ that is least separated from $\boldsymbol{\xi}$ while being separated from other $\boldsymbol{x}_j$ with $j \neq i$:*

$$i = \arg\max_k \min_{j, j \neq k} \left( \boldsymbol{\xi}^T \boldsymbol{x}_k - \boldsymbol{\xi}^T \boldsymbol{x}_j \right) = \arg\max_k \left( \boldsymbol{\xi}^T \boldsymbol{x}_k - \max_{j, j \neq k} \boldsymbol{\xi}^T \boldsymbol{x}_j \right) \tag{119}$$

$$0 \leqslant c = \max_k \min_{j, j \neq k} \left( \boldsymbol{\xi}^T \boldsymbol{x}_k - \boldsymbol{\xi}^T \boldsymbol{x}_j \right) = \max_k \left( \boldsymbol{\xi}^T \boldsymbol{x}_k - \max_{j, j \neq k} \boldsymbol{\xi}^T \boldsymbol{x}_j \right) . \tag{120}$$

*For $\boldsymbol{x}_i$, the following holds:*

$$\|\boldsymbol{x}_i - f(\boldsymbol{\xi})\| \leqslant 2\,\epsilon\,M \,, \tag{121}$$

*where*

$$M = \max_i \|\boldsymbol{x}_i\| \,, \tag{122}$$

$$\epsilon = (N-1)\,\exp(-\beta\,c) \,. \tag{123}$$

*Proof.* For the softmax component $i$ we have:

$$[\text{softmax}(\beta\,\boldsymbol{X}^T\boldsymbol{\xi})]_i = \frac{1}{1 + \sum_{j \neq i} \exp(\beta\,(\boldsymbol{\xi}^T\boldsymbol{x}_j - \boldsymbol{\xi}^T\boldsymbol{x}_i))} \geq \frac{1}{1 + \sum_{j \neq i} \exp(-\beta\,c)} \tag{124}$$

$$= \frac{1}{1 + (N-1)\exp(-\beta\,c)} = 1 - \frac{(N-1)\exp(-\beta\,c)}{1 + (N-1)\exp(-\beta\,c)}$$

$$\geq 1 - (N-1)\exp(-\beta\,c) = 1 - \epsilon$$

For softmax components $k \neq i$ we have

$$[\text{softmax}(\beta\boldsymbol{X}^T\boldsymbol{\xi})]_k = \frac{\exp(\beta\,(\boldsymbol{\xi}^T\boldsymbol{x}_k - \boldsymbol{\xi}^T\boldsymbol{x}_i))}{1 + \sum_{j \neq i} \exp(\beta\,(\boldsymbol{\xi}^T\boldsymbol{x}_j - \boldsymbol{\xi}^T\boldsymbol{x}_i))} \leqslant \exp(-\beta\,c) = \frac{\epsilon}{N-1} \,. \tag{125}$$

The iteration $f$ can be written as

$$f(\boldsymbol{\xi}) = \boldsymbol{X}\,\text{softmax}(\beta\boldsymbol{X}^T\boldsymbol{\xi}) = \sum_{j=1}^N \boldsymbol{x}_j\,[\text{softmax}(\beta\boldsymbol{X}^T\boldsymbol{\xi})]_j \,. \tag{126}$$

We now can bound $\|\boldsymbol{x}_i - f(\boldsymbol{\xi})\|$:

$$\|\boldsymbol{x}_i - f(\boldsymbol{\xi})\| = \left\| \boldsymbol{x}_i - \sum_{j=1}^N [\text{softmax}(\beta\boldsymbol{X}^T\boldsymbol{\xi})]_j\,\boldsymbol{x}_j \right\| \tag{127}$$

$$= \left\| (1 - [\text{softmax}(\beta\boldsymbol{X}^T\boldsymbol{\xi})]_i)\,\boldsymbol{x}_i - \sum_{j=1, j \neq i}^N [\text{softmax}(\beta\boldsymbol{X}^T\boldsymbol{\xi})]_j\,\boldsymbol{x}_j \right\|$$

$$\leqslant \epsilon\,\|\boldsymbol{x}_i\| + \frac{\epsilon}{N-1} \sum_{j=1, j \neq i}^N \|\boldsymbol{x}_j\|$$

$$\leqslant \epsilon\,M + \frac{\epsilon}{N-1} \sum_{j=1, j \neq i}^N M = 2\,\epsilon\,M \,.$$

$\square$

We define $\Delta_i$, i.e. the separation of pattern $\boldsymbol{x}_i$ from data $\boldsymbol{X} = (\boldsymbol{x}_1, \ldots, \boldsymbol{x}_N)$ as:

$$\Delta_i \;=\; \min_{j,j \neq i} \left( \boldsymbol{x}_i^T \boldsymbol{x}_i \;-\; \boldsymbol{x}_i^T \boldsymbol{x}_j \right) \;=\; \boldsymbol{x}_i^T \boldsymbol{x}_i \;-\; \max_{j,j \neq i} \boldsymbol{x}_i^T \boldsymbol{x}_j \;. \tag{128}$$

The pattern is separated from the other data if $0 < \Delta_i$. Using the parallelogram identity, $\Delta_i$ can also be expressed as

$$\Delta_i \;=\; \min_{j,j \neq i} \frac{1}{2} \left( \|\boldsymbol{x}_i\|^2 \;-\; \|\boldsymbol{x}_j\|^2 \;+\; \|\boldsymbol{x}_i \;-\; \boldsymbol{x}_j\|^2 \right) \tag{129}$$

$$= \; \frac{1}{2} \|\boldsymbol{x}_i\|^2 \;-\; \frac{1}{2} \max_{j,j \neq i} \left( \|\boldsymbol{x}_j\|^2 \;-\; \|\boldsymbol{x}_i \;-\; \boldsymbol{x}_j\|^2 \right) \;.$$

For $\|\boldsymbol{x}_i\| = \|\boldsymbol{x}_j\|$ we have $\Delta_i = 1/2 \min_{j,j \neq i} \|\boldsymbol{x}_i \;-\; \boldsymbol{x}_j\|^2$.
Next we define the sphere where we want to apply Banach fixed point theorem.

**Definition B2** (Sphere $S_i$). *The sphere $S_i$ is defined as*

$$S_i \;:=\; \left\{ \boldsymbol{\xi} \mid \|\boldsymbol{\xi} \;-\; \boldsymbol{x}_i\| \;\leqslant\; \frac{1}{\beta \, N \, M} \right\} \;. \tag{130}$$

**Lemma 5.** *With $\boldsymbol{\xi}$ given, if the assumptions*

    *A1: $\boldsymbol{\xi}$ is inside sphere: $\boldsymbol{\xi} \in S_i$,*

    *A2: data point $\boldsymbol{x}_i$ is well separated from the other data:*

$$\Delta_i \;\geqslant\; \frac{2}{\beta \, N} \;+\; \frac{1}{\beta} \, \ln \left( 2 \, (N-1) \, N \, \beta \, M^2 \right) \tag{131}$$

*hold, then $f(\boldsymbol{\xi})$ is inside the sphere: $f(\boldsymbol{\xi}) \in S_i$. Therefore with assumption (A2), $f$ is a mapping from $S_i$ into $S_i$.*

*Proof.* We need the separation $\tilde{\Delta}_i$ of $\boldsymbol{\xi}$ from the data.

$$\tilde{\Delta}_i \;=\; \min_{j,j \neq i} \left( \boldsymbol{\xi}^T \boldsymbol{x}_i \;-\; \boldsymbol{\xi}^T \boldsymbol{x}_j \right) \;. \tag{132}$$

Using the Cauchy-Schwarz inequality, we obtain for $1 \leqslant j \leqslant N$:

$$\left| \boldsymbol{\xi}^T \boldsymbol{x}_j \;-\; \boldsymbol{x}_i^T \boldsymbol{x}_j \right| \;\leqslant\; \|\boldsymbol{\xi} \;-\; \boldsymbol{x}_i\| \, \|\boldsymbol{x}_j\| \;\leqslant\; \|\boldsymbol{\xi} \;-\; \boldsymbol{x}_i\| \, M \;. \tag{133}$$

We have the lower bound

$$\tilde{\Delta}_i \;\geqslant\; \min_{j,j \neq i} \left( \left( \boldsymbol{x}_i^T \boldsymbol{x}_i \;-\; \|\boldsymbol{\xi} \;-\; \boldsymbol{x}_i\| \, M \right) \;-\; \left( \boldsymbol{x}_i^T \boldsymbol{x}_j \;+\; \|\boldsymbol{\xi} \;-\; \boldsymbol{x}_i\| \, M \right) \right) \tag{134}$$

$$= \; -2 \, \|\boldsymbol{\xi} \;-\; \boldsymbol{x}_i\| \, M \;+\; \min_{j,j \neq i} \left( \boldsymbol{x}_i^T \boldsymbol{x}_i \;-\; \boldsymbol{x}_i^T \boldsymbol{x}_j \right) \;=\; \Delta_i \;-\; 2 \, \|\boldsymbol{\xi} \;-\; \boldsymbol{x}_i\| \, M$$

$$\geqslant \; \Delta_i \;-\; \frac{2}{\beta \, N} \;,$$

where we used the assumption (A1) of the lemma.
From the proof in Lemma 4 we have

$$p_{\max} \;=\; [\mathrm{softmax}(\beta \boldsymbol{X}^T \boldsymbol{\xi})]_i \;\geqslant\; 1 \;-\; (N-1) \, \exp(-\, \beta \, \tilde{\Delta}_i) \;=\; 1 \;-\; \tilde{\epsilon} \;. \tag{135}$$

Lemma 4 states that

$$\|\boldsymbol{x}_i \;-\; f(\boldsymbol{\xi})\| \;\leqslant\; 2 \, \tilde{\epsilon} \, M \;=\; 2 \, (N-1) \, \exp(-\, \beta \, \tilde{\Delta}_i) \, M \tag{136}$$

$$\leqslant \; 2 \, (N-1) \, \exp(-\, \beta \, (\Delta_i \;-\; \frac{2}{\beta \, N})) \, M \;.$$

We have

$$\|\boldsymbol{x}_i \;-\; f(\boldsymbol{\xi})\| \tag{137}$$

$$\leqslant \; 2 \, (N-1) \, \exp(-\, \beta \, (\frac{2}{\beta \, N} \;+\; \frac{1}{\beta} \, \ln \left( 2 \, (N-1) \, N \, \beta \, M^2 \right) \;-\; \frac{2}{\beta \, N})) \, M$$

$$= \; 2 \, (N-1) \, \exp(-\, \ln \left( 2 \, (N-1) \, N \, \beta \, M^2 \right)) \, M$$

$$= \; \frac{1}{N \, \beta \, M} \;,$$

where we used assumption (A2) of the lemma. Therefore, $f(\boldsymbol{\xi})$ is a mapping from the sphere $S_i$ into the sphere $S_i$: If $\boldsymbol{\xi} \in S_i$ then $f(\boldsymbol{\xi}) \in S_i$. $\qquad\square$

**Contraction Mapping.** For applying Banach fixed point theorem we need to show that $f$ is contraction in the compact environment $S_i$.

**Lemma 6.** *Assume that*

*A1:*

$$\Delta_i \;\geqslant\; \frac{2}{\beta\,N} \;+\; \frac{1}{\beta}\,\ln\left(2\,(N-1)\,N\,\beta\,M^2\right)\,,\tag{138}$$

*then $f$ is a contraction mapping in $S_i$.*

*Proof.* The mean value theorem states for $\mathrm{J}^m = \int_0^1 \mathrm{J}(\lambda\boldsymbol{\xi} + (1-\lambda)\boldsymbol{x}_i)\,\mathrm{d}\lambda$:

$$f(\boldsymbol{\xi}) \;=\; f(\boldsymbol{x}_i) \;+\; \mathrm{J}^m\,(\boldsymbol{\xi} \;-\; \boldsymbol{x}_i)\,.\tag{139}$$

Therefore

$$\|f(\boldsymbol{\xi}) \;-\; f(\boldsymbol{x}_i)\| \;\leqslant\; \|\mathrm{J}^m\|_2\,\|\boldsymbol{\xi} \;-\; \boldsymbol{x}_i\|\,.\tag{140}$$

We define $\tilde{\boldsymbol{\xi}} = \lambda\boldsymbol{\xi} + (1-\lambda)\boldsymbol{x}_i$ for some $\lambda \in [0,1]$. From the proof in Lemma 4 we have

$$p_{\max}(\tilde{\boldsymbol{\xi}}) \;=\; [\mathrm{softmax}(\beta\,\boldsymbol{X}^T\,\tilde{\boldsymbol{\xi}})]_i \;\geqslant\; 1 \;-\; (N-1)\,\exp(-\,\beta\,\tilde{\Delta}_i) \;=\; 1 \;-\; \tilde{\epsilon}\,,\tag{141}$$

$$\tilde{\epsilon} \;=\; (N-1)\,\exp(-\,\beta\,\tilde{\Delta}_i)\,,\tag{142}$$

$$\tilde{\Delta}_i \;=\; \min_{j,j\neq i}\left(\tilde{\boldsymbol{\xi}}^T\boldsymbol{x}_i \;-\; \tilde{\boldsymbol{\xi}}^T\boldsymbol{x}_j\right)\,.\tag{143}$$

First we compute an upper bound on $\tilde{\epsilon}$. We need the separation $\tilde{\Delta}_i$ of $\boldsymbol{\xi}$ from the data. Using the Cauchy-Schwarz inequality, we obtain for $1 \leqslant j \leqslant N$:

$$\left|\tilde{\boldsymbol{\xi}}^T\boldsymbol{x}_j \;-\; \boldsymbol{x}_i^T\boldsymbol{x}_j\right| \;\leqslant\; \left\|\tilde{\boldsymbol{\xi}} \;-\; \boldsymbol{x}_i\right\|\,\|\boldsymbol{x}_j\| \;\leqslant\; \left\|\tilde{\boldsymbol{\xi}} \;-\; \boldsymbol{x}_i\right\|\,M\,.\tag{144}$$

We have the lower bound on $\tilde{\Delta}_i$:

$$\begin{aligned}
\tilde{\Delta}_i \;&\geqslant\; \min_{j,j\neq i}\left(\left(\boldsymbol{x}_i^T\boldsymbol{x}_i \;-\; \left\|\tilde{\boldsymbol{\xi}} \;-\; \boldsymbol{x}_i\right\|\,M\right) \;-\; \left(\boldsymbol{x}_i^T\boldsymbol{x}_j \;+\; \left\|\tilde{\boldsymbol{\xi}} \;-\; \boldsymbol{x}_i\right\|\,M\right)\right)\\
&=\; -\,2\,\left\|\tilde{\boldsymbol{\xi}} \;-\; \boldsymbol{x}_i\right\|\,M \;+\; \min_{j,j\neq i}\left(\boldsymbol{x}_i^T\boldsymbol{x}_i \;-\; \boldsymbol{x}_i^T\boldsymbol{x}_j\right) \;=\; \Delta_i \;-\; 2\,\left\|\tilde{\boldsymbol{\xi}} \;-\; \boldsymbol{x}_i\right\|\,M\\
&\geqslant\; \Delta_i \;-\; 2\,\|\boldsymbol{\xi} \;-\; \boldsymbol{x}_i\|\,M\,,
\end{aligned}\tag{145}$$

where we used $\left\|\tilde{\boldsymbol{\xi}} - \boldsymbol{x}_i\right\| = \lambda\|\boldsymbol{\xi} - \boldsymbol{x}_i\| \leqslant \|\boldsymbol{\xi} - \boldsymbol{x}_i\|$. From the definition of $\tilde{\epsilon}$ in Eq. (141) we have

$$\begin{aligned}
\tilde{\epsilon} \;&=\; (N-1)\,\exp(-\,\beta\,\tilde{\Delta}_i)\\
&\leqslant\; (N-1)\,\exp\left(-\,\beta\,(\Delta_i \;-\; 2\,\|\boldsymbol{\xi} \;-\; \boldsymbol{x}_i\|\,M)\right)\\
&\leqslant\; (N-1)\,\exp\left(-\,\beta\,\left(\Delta_i \;-\; \frac{2}{\beta\,N}\right)\right)\,,
\end{aligned}\tag{146}$$

where we used $\boldsymbol{\xi} \in S_i$, therefore $\|\boldsymbol{\xi} \;-\; \boldsymbol{x}_i\| \;\leqslant\; \frac{1}{\beta\,N\,M}$.

Next we compute a lower bound on $\tilde{\epsilon}$. We start with an upper on $\tilde{\Delta}_i$:

$$\begin{aligned}
\tilde{\Delta}_i \;&\leqslant\; \min_{j,j\neq i}\left(\left(\boldsymbol{x}_i^T\boldsymbol{x}_i \;+\; \left\|\tilde{\boldsymbol{\xi}} \;-\; \boldsymbol{x}_i\right\|\,M\right) \;-\; \left(\boldsymbol{x}_i^T\boldsymbol{x}_j \;-\; \left\|\tilde{\boldsymbol{\xi}} \;-\; \boldsymbol{x}_i\right\|\,M\right)\right)\\
&=\; 2\,\left\|\tilde{\boldsymbol{\xi}} \;-\; \boldsymbol{x}_i\right\|\,M \;+\; \min_{j,j\neq i}\left(\boldsymbol{x}_i^T\boldsymbol{x}_i \;-\; \boldsymbol{x}_i^T\boldsymbol{x}_j\right) \;=\; \Delta_i \;+\; 2\,\left\|\tilde{\boldsymbol{\xi}} \;-\; \boldsymbol{x}_i\right\|\,M\\
&\leqslant\; \Delta_i \;+\; 2\,\|\boldsymbol{\xi} \;-\; \boldsymbol{x}_i\|\,M\,,
\end{aligned}\tag{147}$$

where we used $\left\|\tilde{\boldsymbol{\xi}} - \boldsymbol{x}_i\right\| = \lambda\|\boldsymbol{\xi} - \boldsymbol{x}_i\| \leqslant \|\boldsymbol{\xi} - \boldsymbol{x}_i\|$. From the definition of $\tilde{\epsilon}$ in Eq. (141) we have

$$\begin{aligned}
\tilde{\epsilon} \;&=\; (N-1)\,\exp(-\,\beta\,\tilde{\Delta}_i)\\
&\geqslant\; (N-1)\,\exp\left(-\,\beta\,(\Delta_i \;+\; 2\,\|\boldsymbol{\xi} \;-\; \boldsymbol{x}_i\|\,M)\right)\\
&\geqslant\; (N-1)\,\exp\left(-\,\beta\,\left(\Delta_i \;+\; \frac{2}{\beta\,N}\right)\right)\,,
\end{aligned}\tag{148}$$

where we used $\boldsymbol{\xi} \in \mathrm{S}_i$, therefore $\|\boldsymbol{\xi} - \boldsymbol{x}_i\| \leqslant \frac{1}{\beta \, N \, M}$.

Now we bound the Jacobian. We can assume $\tilde{\epsilon} \leqslant 0.5$ otherwise $(1 - \tilde{\epsilon}) \leqslant 0.5$ in the following. From the proof of Lemma 24 we know for $p_{\max}(\tilde{\boldsymbol{\xi}}) \geq 1 - \tilde{\epsilon}$, then $p_i(\tilde{\boldsymbol{\xi}}) \leqslant \tilde{\epsilon}$ for $p_i(\tilde{\boldsymbol{\xi}}) \neq p_{\max}(\tilde{\boldsymbol{\xi}})$. Therefore $p_i(\tilde{\boldsymbol{\xi}})(1 - p_i(\tilde{\boldsymbol{\xi}})) \leqslant m \leqslant \tilde{\epsilon}(1 - \tilde{\epsilon})$ for all $i$. Next we use the derived upper and lower bound on $\tilde{\epsilon}$ in previous Eq. (50) in Lemma 2:

$$\left\| \mathrm{J}(\tilde{\boldsymbol{\xi}}) \right\|_2 \leqslant 2 \, \beta \, N \, M^2 \, \tilde{\epsilon} \, - \, 2 \, \tilde{\epsilon}^2 \, \beta \, N \, M^2 \tag{149}$$

$$\leqslant 2 \, \beta \, N \, M^2 \, (N-1) \, \exp\left( -\beta \left( \Delta_i - \frac{2}{\beta \, N} \right) \right) -$$
$$2 \, (N-1)^2 \, \exp\left( -2 \, \beta \left( \Delta_i + \frac{2}{\beta \, N} \right) \right) \beta \, N \, M^2 \, .$$

The bound Eq. (149) holds for the mean $\mathrm{J}^m$, too, since it averages over $\mathrm{J}(\tilde{\boldsymbol{\xi}})$:

$$\| \mathrm{J}^m \|_2 \leqslant 2 \, \beta \, N \, M^2 \, (N-1) \, \exp\left( -\beta \left( \Delta_i - \frac{2}{\beta \, N} \right) \right) - \tag{150}$$
$$2 \, (N-1)^2 \, \exp\left( -2 \, \beta \left( \Delta_i + \frac{2}{\beta \, N} \right) \right) \beta \, N \, M^2 \, .$$

The assumption of the lemma is

$$\Delta_i \geq \frac{2}{\beta \, N} + \frac{1}{\beta} \, \ln\left( 2 \, (N-1) \, N \, \beta \, M^2 \right) , \tag{151}$$

This is

$$\Delta_i - \frac{2}{\beta \, N} \geq \frac{1}{\beta} \, \ln\left( 2 \, (N-1) \, N \, \beta \, M^2 \right) , \tag{152}$$

Therefore the spectral norm $\|\mathrm{J}\|_2$ can be bounded by:

$$\| \mathrm{J}^m \|_2 \leqslant 2 \, \beta \, (N-1) \, \exp\left( -\beta \, \frac{1}{\beta} \, \ln\left( 2 \, (N-1) \, N \, \beta \, M^2 \right) \right) N \, M^2 - \tag{153}$$
$$2 \, (N-1)^2 \, \exp\left( -2 \, \beta \left( \Delta_i + \frac{2}{\beta \, N} \right) \right) \beta \, N \, M^2$$
$$= 2 \, \beta \, (N-1) \, \frac{1}{2 \, (N-1) \, N \, \beta \, M^2} \, N \, M^2 -$$
$$2 \, (N-1)^2 \, \exp\left( -2 \, \beta \left( \Delta_i + \frac{2}{\beta \, N} \right) \right) \beta \, N \, M^2$$
$$= 1 - 2 \, (N-1)^2 \, \exp\left( -2 \, \beta \left( \Delta_i + \frac{2}{\beta \, N} \right) \right) \beta \, N \, M^2 \, < 1 \, .$$

Therefore $f$ is a contraction mapping in $\mathrm{S}_i$. $\qquad\square$

**Banach Fixed Point Theorem.** Now we have all ingredients to apply Banach fixed point theorem.

**Lemma 7.** *Assume that*

*A1:*

$$\Delta_i \geq \frac{2}{\beta \, N} + \frac{1}{\beta} \, \ln\left( 2 \, (N-1) \, N \, \beta \, M^2 \right) , \tag{154}$$

*then $f$ has a fixed point in $\mathrm{S}_i$.*

*Proof.* We use Banach fixed point theorem: Lemma 5 says that $f$ maps from $\mathrm{S}_i$ into $\mathrm{S}_i$. Lemma 6 says that $f$ is a contraction mapping in $\mathrm{S}_i$. $\qquad\square$

**Contraction Mapping with a Fixed Point** We have shown that a fixed point exists. We want to know how fast the iteration converges to the fixed point. Let $\boldsymbol{x}_i^*$ be the fixed point of the iteration $f$ in the sphere $\mathrm{S}_i$. Using the mean value theorem, we have with $\mathrm{J}^m = \int_0^1 \mathrm{J}(\lambda \boldsymbol{\xi} + (1 - \lambda)\boldsymbol{x}_i^*) \, \mathrm{d}\lambda$:

$$\|f(\boldsymbol{\xi}) - \boldsymbol{x}_i^*\| = \|f(\boldsymbol{\xi}) - f(\boldsymbol{x}_i^*)\| \leqslant \|\mathrm{J}^m\|_2 \, \|\boldsymbol{\xi} - \boldsymbol{x}_i^*\| \tag{155}$$

According to Lemma 24, if $p_{\max} = \max_i p_i \geq 1 - \epsilon$ for all $\tilde{\boldsymbol{x}} = \lambda \boldsymbol{\xi} + (1 - \lambda)\boldsymbol{x}_i^*$, then the spectral norm of the Jacobian is bounded by

$$\|\mathrm{J}_s(\tilde{\boldsymbol{x}})\|_2 < 2\,\epsilon\,\beta \ . \tag{156}$$

The norm of Jacobian at $\tilde{\boldsymbol{x}}$ is bounded

$$\|\mathrm{J}(\tilde{\boldsymbol{x}})\|_2 \leqslant 2\,\beta\,\|\boldsymbol{X}\|_2^2\,\epsilon \leqslant 2\,\beta\,NM^2\,\epsilon \ . \tag{157}$$

We used that the spectral norm $\|.\|_2$ is bounded by the Frobenius norm $\|.\|_F$ which can be expressed by the norm squared of its column vectors:

$$\|\boldsymbol{X}\|_2 \leqslant \|\boldsymbol{X}\|_F = \sqrt{\sum_i \|\boldsymbol{x}_i\|^2} \ . \tag{158}$$

Therefore

$$\|\boldsymbol{X}\|_2^2 \leqslant N\,M^2 \ . \tag{159}$$

The norm of Jacobian of the fixed point iteration is bounded

$$\|\mathrm{J}^m\|_2 \leqslant 2\,\beta\,\|\boldsymbol{X}\|_2^2\,\epsilon \leqslant 2\,\beta\,NM^2\,\epsilon \ . \tag{160}$$

The separation of pattern $\boldsymbol{x}_i$ from data $\boldsymbol{X} = (\boldsymbol{x}_1, \ldots, \boldsymbol{x}_N)$ is

$$\Delta_i = \min_{j, j \neq i} \left(\boldsymbol{x}_i^T \boldsymbol{x}_i - \boldsymbol{x}_i^T \boldsymbol{x}_j\right) = \boldsymbol{x}_i^T \boldsymbol{x}_i - \max_{j, j \neq i} \boldsymbol{x}_i^T \boldsymbol{x}_j \ . \tag{161}$$

We need the separation $\tilde{\Delta}_i$ of $\tilde{\boldsymbol{x}} = \lambda \boldsymbol{\xi} + (1 - \lambda)\boldsymbol{x}_i^*$ from the data:

$$\tilde{\Delta}_i = \min_{j, j \neq i} \left(\tilde{\boldsymbol{x}}^T \boldsymbol{x}_i - \tilde{\boldsymbol{x}}^T \boldsymbol{x}_j\right) \ . \tag{162}$$

We compute a lower bound on $\tilde{\Delta}_i$. Using the Cauchy-Schwarz inequality, we obtain for $1 \leqslant j \leqslant N$:

$$\left|\tilde{\boldsymbol{x}}^T \boldsymbol{x}_j - \boldsymbol{x}_i^T \boldsymbol{x}_j\right| \leqslant \|\tilde{\boldsymbol{x}} - \boldsymbol{x}_i\| \, \|\boldsymbol{x}_j\| \leqslant \|\tilde{\boldsymbol{x}} - \boldsymbol{x}_i\| \, M \ . \tag{163}$$

We have the lower bound

$$\tilde{\Delta}_i \geq \min_{j, j \neq i} \left(\left(\boldsymbol{x}_i^T \boldsymbol{x}_i - \|\tilde{\boldsymbol{x}} - \boldsymbol{x}_i\| \, M\right) - \left(\boldsymbol{x}_i^T \boldsymbol{x}_j + \|\tilde{\boldsymbol{x}} - \boldsymbol{x}_i\| \, M\right)\right) \tag{164}$$

$$= -2\,\|\tilde{\boldsymbol{x}} - \boldsymbol{x}_i\| \, M + \min_{j, j \neq i} \left(\boldsymbol{x}_i^T \boldsymbol{x}_i - \boldsymbol{x}_i^T \boldsymbol{x}_j\right) = \Delta_i - 2\,\|\tilde{\boldsymbol{x}} - \boldsymbol{x}_i\| \, M \ .$$

Since

$$\|\tilde{\boldsymbol{x}} - \boldsymbol{x}_i\| = \|\lambda \boldsymbol{\xi} + (1 - \lambda)\boldsymbol{x}_i^* - \boldsymbol{x}_i\| \tag{165}$$

$$\leqslant \lambda \, \|\boldsymbol{\xi} - \boldsymbol{x}_i\| + (1 - \lambda) \, \|\boldsymbol{x}_i^* - \boldsymbol{x}_i\|$$

$$\leqslant \max\{\|\boldsymbol{\xi} - \boldsymbol{x}_i\|, \|\boldsymbol{x}_i^* - \boldsymbol{x}_i\|\} \ ,$$

we have

$$\tilde{\Delta}_i \geq \Delta_i - 2\,\max\{\|\boldsymbol{\xi} - \boldsymbol{x}_i\|, \|\boldsymbol{x}_i^* - \boldsymbol{x}_i\|\} \, M \ . \tag{166}$$

For the softmax component $i$ we have:

$$[\mathrm{softmax}(\beta \, \boldsymbol{X}^T \tilde{\boldsymbol{\xi}})]_i = \frac{1}{1 + \sum_{j \neq i} \exp(\beta \, (\tilde{\boldsymbol{\xi}}^T \boldsymbol{x}_j - \tilde{\boldsymbol{\xi}}^T \boldsymbol{x}_i))} \tag{167}$$

$$\geq \frac{1}{1 + \sum_{j \neq i} \exp(-\beta \, (\Delta_i - 2\,\max\{\|\boldsymbol{\xi} - \boldsymbol{x}_i\|, \|\boldsymbol{x}_i^* - \boldsymbol{x}_i\|\} \, M))}$$

$$= \frac{1}{1 + (N - 1)\exp(-\beta \, (\Delta_i - 2\,\max\{\|\boldsymbol{\xi} - \boldsymbol{x}_i\|, \|\boldsymbol{x}_i^* - \boldsymbol{x}_i\|\} \, M))}$$

$$= 1 - \frac{(N - 1)\exp(-\beta \, (\Delta_i - 2\,\max\{\|\boldsymbol{\xi} - \boldsymbol{x}_i\|, \|\boldsymbol{x}_i^* - \boldsymbol{x}_i\|\} \, M))}{1 + (N - 1)\exp(-\beta \, (\Delta_i - 2\,\max\{\|\boldsymbol{\xi} - \boldsymbol{x}_i\|, \|\boldsymbol{x}_i^* - \boldsymbol{x}_i\|\} \, M))}$$

$$\geq 1 - (N - 1)\exp(-\beta \, (\Delta_i - 2\,\max\{\|\boldsymbol{\xi} - \boldsymbol{x}_i\|, \|\boldsymbol{x}_i^* - \boldsymbol{x}_i\|\} \, M))$$

$$= 1 - \epsilon \ .$$

Therefore

$$\epsilon \;=\; (N-1)\exp(-\,\beta\,(\Delta_i \;-\; 2\,\max\{\|\boldsymbol{\xi}\;-\;\boldsymbol{x}_i\|, \|\boldsymbol{x}_i^*\;-\;\boldsymbol{x}_i\|\}\,M))\,. \tag{168}$$

We can bound the spectral norm of the Jacobian, which upper bounds the Lipschitz constant:

$$\|\mathrm{J}^m\|_2 \;\leqslant\; 2\,\beta\,N\,M^2\,(N-1)\exp(-\,\beta\,(\Delta_i\;-\;2\,\max\{\|\boldsymbol{\xi}\;-\;\boldsymbol{x}_i\|,\|\boldsymbol{x}_i^*\;-\;\boldsymbol{x}_i\|\}\,M))\,. \tag{169}$$

For a contraction mapping we require

$$\|\mathrm{J}^m\|_2 \;<\; 1\,, \tag{170}$$

which can be ensured by

$$2\,\beta\,NM^2\,(N-1)\exp(-\,\beta\,(\Delta_i\;-\;2\,\max\{\|\boldsymbol{\xi}\;-\;\boldsymbol{x}_i\|,\|\boldsymbol{x}_i^*\;-\;\boldsymbol{x}_i\|\}\,M))\;<\;1\,. \tag{171}$$

Solving this inequality for $\Delta_i$ gives

$$\Delta_i \;>\; 2\,\max\{\|\boldsymbol{\xi}\;-\;\boldsymbol{x}_i\|,\|\boldsymbol{x}_i^*\;-\;\boldsymbol{x}_i\|\}\,M \;+\; \frac{1}{\beta}\,\ln\left(2\,(N-1)\,N\,\beta\,M^2\right)\,. \tag{172}$$

In an environment around $\boldsymbol{x}_i^*$ in which Eq. (172) holds, $f$ is a contraction mapping and every point converges under the iteration $f$ to $\boldsymbol{x}_i^*$ when the iteration stays in the environment. After every iteration the mapped point $f(\boldsymbol{\xi})$ is closer to the fixed point $\boldsymbol{x}_i^*$ than the original point $\boldsymbol{x}_i$:

$$\|f(\boldsymbol{\xi})\;-\;\boldsymbol{x}_i^*\| \;\leqslant\; \|\mathrm{J}^m\|_2\,\|\boldsymbol{\xi}\;-\;\boldsymbol{x}_i^*\| \;<\; \|\boldsymbol{\xi}\;-\;\boldsymbol{x}_i^*\|\,. \tag{173}$$

Using

$$\|f(\boldsymbol{\xi})\;-\;\boldsymbol{x}_i^*\| \;\leqslant\; \|\mathrm{J}^m\|_2\,\|\boldsymbol{\xi}\;-\;\boldsymbol{x}_i^*\| \;\leqslant\; \|\mathrm{J}^m\|_2\,\|\boldsymbol{\xi}\;-\;f(\boldsymbol{\xi})\| \;+\; \|\mathrm{J}^m\|_2\,\|f(\boldsymbol{\xi})\;-\;\boldsymbol{x}_i^*\|\,, \tag{174}$$

we obtain

$$\|f(\boldsymbol{\xi})\;-\;\boldsymbol{x}_i^*\| \;\leqslant\; \frac{\|\mathrm{J}^m\|_2}{1\;-\;\|\mathrm{J}^m\|_2}\,\|\boldsymbol{\xi}\;-\;f(\boldsymbol{\xi})\|\,. \tag{175}$$

For large $\Delta_i$ the iteration is close to the fixed point even after one update. This has been confirmed in several experiments.

### B2.4.4 Metastable States: Fixed Points Near Mean of Similar Patterns

The proof concept is the same as for a single pattern but now for the arithmetic mean of similar patterns.

**Bound on the Jacobian.** The Jacobian of the fixed point iteration is

$$\mathrm{J} \;=\; \beta\,\boldsymbol{X}\left(\mathrm{diag}(\boldsymbol{p}) - \boldsymbol{p}\boldsymbol{p}^T\right)\boldsymbol{X}^T \;=\; \boldsymbol{X}\mathrm{J}_s\boldsymbol{X}^T\,. \tag{176}$$

If we consider $p_i$ as the probability of selecting the vector $\boldsymbol{x}_i$, then we can define expectations as $\mathrm{E}_{\boldsymbol{p}}[f(\boldsymbol{x})] = \sum_{i=1}^N p_i f(\boldsymbol{x}_i)$. In this setting the matrix

$$\boldsymbol{X}\left(\mathrm{diag}(\boldsymbol{p}) - \boldsymbol{p}\boldsymbol{p}^T\right)\boldsymbol{X}^T \tag{177}$$

is the covariance matrix of data $\boldsymbol{X}$ when its vectors are selected according to the probability $\boldsymbol{p}$:

$$\boldsymbol{X}\left(\mathrm{diag}(\boldsymbol{p}) \;-\; \boldsymbol{p}\boldsymbol{p}^T\right)\boldsymbol{X}^T \;=\; \boldsymbol{X}\mathrm{diag}(\boldsymbol{p})\boldsymbol{X}^T \;-\; \boldsymbol{X}\boldsymbol{p}\boldsymbol{p}^T\boldsymbol{X}^T \tag{178}$$

$$=\; \sum_{i=1}^N p_i\,\boldsymbol{x}_i\,\boldsymbol{x}_i^T \;-\; \left(\sum_{i=1}^N p_i\,\boldsymbol{x}_i\right)\left(\sum_{i=1}^N p_i\,\boldsymbol{x}_i\right)^T \tag{179}$$

$$=\; \mathrm{E}_{\boldsymbol{p}}[\boldsymbol{x}\,\boldsymbol{x}^T] \;-\; \mathrm{E}_{\boldsymbol{p}}[\boldsymbol{x}]\,\mathrm{E}_{\boldsymbol{p}}[\boldsymbol{x}]^T \;=\; \mathrm{Var}_{\boldsymbol{p}}[\boldsymbol{x}]\,, \tag{180}$$

therefore we have

$$\mathrm{J} \;=\; \beta\,\mathrm{Var}_{\boldsymbol{p}}[\boldsymbol{x}]\,. \tag{181}$$

We now elaborate more on this interpretation as variance. Specifically the singular values of J (or in other words: the covariance) should be reasonably small. The singular values are the key to ensure convergence of the iteration Eq. (46). Next we present some thoughts.

1. It's clear that the largest eigenvalue of the covariance matrix (equal to the largest singular value) is the variance in the direction of the eigenvector associated with the largest eigenvalue.

2. Furthermore the variance goes to zero as one $p_i$ goes to one, since only one pattern is chosen and there is no variance.

3. The variance is reasonable small if all patterns are chosen with equal probability.

4. The variance is small if few similar patterns are chosen with high probability. If the patterns are sufficient similar, then the spectral norm of the covariance matrix is smaller than one.

The first three issues have already been adressed. Now we focus on the last one in greater detail. We assume that the first $l$ patterns are much more probable (and similar to one another) than the other patterns. Therefore we define:

$$M \; := \; \max_i \|\boldsymbol{x}_i\| \,, \tag{182}$$

$$\gamma \; = \; \sum_{i=l+1}^{N} p_i \; \leqslant \; \epsilon \,, \tag{183}$$

$$1 - \gamma \; = \; \sum_{i=1}^{l} p_i \; \geq \; 1 - \epsilon \,, \tag{184}$$

$$\tilde{p}_i \; := \; \frac{p_i}{1 - \gamma} \; \leqslant \; p_i/(1 - \epsilon) \,, \tag{185}$$

$$\sum_{i=1}^{l} \tilde{p}_i \; = \; 1 \,, \tag{186}$$

$$\boldsymbol{m_x} \; = \; \frac{1}{l} \sum_{i=1}^{l} \boldsymbol{x}_i \,, \tag{187}$$

$$m_{\max} \; = \; \max_{1 \leqslant i \leqslant l} \|\boldsymbol{x}_i - \boldsymbol{m_x}\| \,. \tag{188}$$

$M$ is an upper bound on the Euclidean norm of the patterns, which are vectors. $\epsilon$ is an upper bound on the probability $\gamma$ of not choosing one of the first $l$ patterns, while $1 - \epsilon$ is a lower bound the probability $(1 - \gamma)$ of choosing one of the first $l$ patterns. $\boldsymbol{m_x}$ is the arithmetic mean (the center) of the first $l$ patterns. $m_{\max}$ is the maximal distance of the patterns to the center $\boldsymbol{m_x}$. $\tilde{\boldsymbol{p}}$ is the probability $\boldsymbol{p}$ normalized for the first $l$ patterns.

The variance of the first $l$ patterns is

$$\mathrm{Var}_{\tilde{p}}[\boldsymbol{x}_{1:l}] \; = \; \sum_{i=1}^{l} \tilde{p}_i \, \boldsymbol{x}_i \, \boldsymbol{x}_i^T \; - \; \left( \sum_{i=1}^{l} \tilde{p}_i \, \boldsymbol{x}_i \right) \left( \sum_{i=1}^{l} \tilde{p}_i \, \boldsymbol{x}_i \right)^T \tag{189}$$

$$= \; \sum_{i=1}^{l} \tilde{p}_i \left( \boldsymbol{x}_i - \sum_{i=1}^{l} \tilde{p}_i \boldsymbol{x}_i \right) \left( \boldsymbol{x}_i - \sum_{i=1}^{l} \tilde{p}_i \boldsymbol{x}_i \right)^T \,.$$

**Lemma 8.** *With the definitions in Eq.* (182) *to Eq.* (189)*, the following bounds on the norm* $\|\mathrm{J}\|_2$ *of the Jacobian of the fixed point iteration hold. The $\gamma$-bound for $\|\mathrm{J}\|_2$ is*

$$\|\mathrm{J}\|_2 \; \leqslant \; \beta \left( (1 - \gamma) \, m_{\max}^2 \; + \; \gamma \, 2 \, (2 - \gamma) \, M^2 \right) \tag{190}$$

*and the $\epsilon$-bound for $\|\mathrm{J}\|_2$ is:*

$$\|\mathrm{J}\|_2 \; \leqslant \; \beta \left( m_{\max}^2 \; + \; \epsilon \, 2 \, (2 - \epsilon) \, M^2 \right) \,. \tag{191}$$

*Proof.* The variance $\mathrm{Var}_{\tilde{p}}[\boldsymbol{x}_{1:l}]$ can be expressed as:

$$(1-\gamma)\,\mathrm{Var}_{\tilde{p}}[\boldsymbol{x}_{1:l}] \;=\; \sum_{i=1}^{l} p_i \left(\boldsymbol{x}_i \;-\; \frac{1}{1-\gamma}\sum_{i=1}^{l} p_i\,\boldsymbol{x}_i\right)\left(\boldsymbol{x}_i \;-\; \frac{1}{1-\gamma}\sum_{i=1}^{l} p_i\,\boldsymbol{x}_i\right)^{T} \tag{192}$$

$$= \sum_{i=1}^{l} p_i\,\boldsymbol{x}_i\,\boldsymbol{x}_i^{T} \;-\; \left(\sum_{i=1}^{l} p_i\,\boldsymbol{x}_i\right)\frac{1}{1-\gamma}\left(\sum_{i=1}^{l} p_i\,\boldsymbol{x}_i\right)^{T}$$

$$-\; \frac{1}{1-\gamma}\left(\sum_{i=1}^{l} p_i\,\boldsymbol{x}_i\right)\left(\sum_{i=1}^{l} p_i\,\boldsymbol{x}_i\right)^{T}$$

$$+\; \frac{\sum_{i=1}^{l} p_i}{(1-\gamma)^2}\left(\sum_{i=1}^{l} p_i\,\boldsymbol{x}_i\right)\left(\sum_{i=1}^{l} p_i\,\boldsymbol{x}_i\right)^{T}$$

$$= \sum_{i=1}^{l} p_i\,\boldsymbol{x}_i\,\boldsymbol{x}_i^{T} \;-\; \frac{1}{1-\gamma}\left(\sum_{i=1}^{l} p_i\,\boldsymbol{x}_i\right)\left(\sum_{i=1}^{l} p_i\,\boldsymbol{x}_i\right)^{T}$$

$$= \sum_{i=1}^{l} p_i\,\boldsymbol{x}_i\,\boldsymbol{x}_i^{T} \;-\; \left(\sum_{i=1}^{l} p_i\,\boldsymbol{x}_i\right)\left(\sum_{i=1}^{l} p_i\,\boldsymbol{x}_i\right)^{T} + \left(1 - \frac{1}{1-\gamma}\right)\left(\sum_{i=1}^{l} p_i\,\boldsymbol{x}_i\right)\left(\sum_{i=1}^{l} p_i\,\boldsymbol{x}_i\right)^{T}$$

$$= \sum_{i=1}^{l} p_i\,\boldsymbol{x}_i\,\boldsymbol{x}_i^{T} \;-\; \left(\sum_{i=1}^{l} p_i\,\boldsymbol{x}_i\right)\left(\sum_{i=1}^{l} p_i\,\boldsymbol{x}_i\right)^{T} - \frac{\gamma}{1-\gamma}\left(\sum_{i=1}^{l} p_i\,\boldsymbol{x}_i\right)\left(\sum_{i=1}^{l} p_i\,\boldsymbol{x}_i\right)^{T}.$$

Therefore we have

$$\sum_{i=1}^{l} p_i\,\boldsymbol{x}_i\,\boldsymbol{x}_i^{T} \;-\; \left(\sum_{i=1}^{l} p_i\,\boldsymbol{x}_i\right)\left(\sum_{i=1}^{l} p_i\,\boldsymbol{x}_i\right)^{T} \tag{193}$$

$$= (1-\gamma)\,\mathrm{Var}_{\tilde{p}}[\boldsymbol{x}_{1:l}] \;+\; \frac{\gamma}{1-\gamma}\left(\sum_{i=1}^{l} p_i\,\boldsymbol{x}_i\right)\left(\sum_{i=1}^{l} p_i\,\boldsymbol{x}_i\right)^{T}.$$

We now can reformulate the Jacobian J:

$$J = \beta \left( \sum_{i=1}^{l} p_i \, \boldsymbol{x}_i \, \boldsymbol{x}_i^T \; + \; \sum_{i=l+1}^{N} p_i \, \boldsymbol{x}_i \, \boldsymbol{x}_i^T \right. \tag{194}$$

$$- \left( \sum_{i=1}^{l} p_i \, \boldsymbol{x}_i \; + \; \sum_{i=l+1}^{N} p_i \, \boldsymbol{x}_i \right) \left( \sum_{i=1}^{l} p_i \, \boldsymbol{x}_i \; + \; \sum_{i=l+1}^{N} p_i \, \boldsymbol{x}_i \right)^T \bigg)$$

$$= \beta \left( \sum_{i=1}^{l} p_i \, \boldsymbol{x}_i \, \boldsymbol{x}_i^T \; - \; \left( \sum_{i=1}^{l} p_i \, \boldsymbol{x}_i \right) \left( \sum_{i=1}^{l} p_i \, \boldsymbol{x}_i \right)^T \right.$$

$$+ \sum_{i=l+1}^{N} p_i \, \boldsymbol{x}_i \, \boldsymbol{x}_i^T \; - \; \left( \sum_{i=l+1}^{N} p_i \, \boldsymbol{x}_i \right) \left( \sum_{i=l+1}^{N} p_i \, \boldsymbol{x}_i \right)^T$$

$$- \left( \sum_{i=1}^{l} p_i \, \boldsymbol{x}_i \right) \left( \sum_{i=l+1}^{N} p_i \, \boldsymbol{x}_i \right)^T \; - \; \left( \sum_{i=l+1}^{N} p_i \, \boldsymbol{x}_i \right) \left( \sum_{i=1}^{l} p_i \, \boldsymbol{x}_i \right)^T \bigg)$$

$$= \beta \left( (1 - \gamma) \, \mathrm{Var}_{\tilde{p}}[\boldsymbol{x}_{1:l}] \; + \; \frac{\gamma}{1 - \gamma} \left( \sum_{i=1}^{l} p_i \, \boldsymbol{x}_i \right) \left( \sum_{i=1}^{l} p_i \, \boldsymbol{x}_i \right)^T \right.$$

$$+ \sum_{i=l+1}^{N} p_i \, \boldsymbol{x}_i \, \boldsymbol{x}_i^T \; - \; \left( \sum_{i=l+1}^{N} p_i \, \boldsymbol{x}_i \right) \left( \sum_{i=l+1}^{N} p_i \, \boldsymbol{x}_i \right)^T$$

$$- \left( \sum_{i=1}^{l} p_i \, \boldsymbol{x}_i \right) \left( \sum_{i=l+1}^{N} p_i \, \boldsymbol{x}_i \right)^T \; - \; \left( \sum_{i=l+1}^{N} p_i \, \boldsymbol{x}_i \right) \left( \sum_{i=1}^{l} p_i \, \boldsymbol{x}_i \right)^T \bigg) \, .$$

The spectral norm of an outer product of two vectors is the product of the Euclidean norms of the vectors:

$$\left\| \boldsymbol{a} \boldsymbol{b}^T \right\|_2 \; = \; \sqrt{\lambda_{\max}(\boldsymbol{b} \boldsymbol{a}^T \boldsymbol{a} \boldsymbol{b}^T)} \; = \; \|\boldsymbol{a}\| \, \sqrt{\lambda_{\max}(\boldsymbol{b} \boldsymbol{b}^T)} \; = \; \|\boldsymbol{a}\| \, \|\boldsymbol{b}\| \, , \tag{195}$$

since $\boldsymbol{b} \boldsymbol{b}^T$ has eigenvector $\boldsymbol{b}/\|\boldsymbol{b}\|$ with eigenvalue $\|\boldsymbol{b}\|^2$ and otherwise zero eigenvalues.
We now bound the norms of some matrices and vectors:

$$\left\| \sum_{i=1}^{l} p_i \, \boldsymbol{x}_i \right\| \; \leqslant \; \sum_{i=1}^{l} p_i \, \|\boldsymbol{x}_i\| \; \leqslant \; (1 - \gamma) \, M \, , \tag{196}$$

$$\left\| \sum_{i=l+1}^{N} p_i \, \boldsymbol{x}_i \right\| \; \leqslant \; \sum_{i=l+1}^{N} p_i \, \|\boldsymbol{x}_i\| \; \leqslant \; \gamma \, M \, , \tag{197}$$

$$\left\| \sum_{i=l+1}^{N} p_i \, \boldsymbol{x}_i \, \boldsymbol{x}_i^T \right\|_2 \; \leqslant \; \sum_{i=l+1}^{N} p_i \, \left\| \boldsymbol{x}_i \, \boldsymbol{x}_i^T \right\|_2 \; = \; \sum_{i=l+1}^{N} p_i \, \|\boldsymbol{x}_i\|^2 \; \leqslant \; \sum_{i=l+1}^{N} p_i \, M^2 \; = \; \gamma \, M^2 \, . \tag{198}$$

In order to bound the variance of the first $l$ patterns, we compute the vector $\boldsymbol{a}$ that minimizes

$$f(\boldsymbol{a}) \; = \; \sum_{i=1}^{l} p_i \|\boldsymbol{x}_i \; - \; \boldsymbol{a}\|^2 \; = \; \sum_{i=1}^{l} p_i (\boldsymbol{x}_i \; - \; \boldsymbol{a})^T (\boldsymbol{x}_i \; - \; \boldsymbol{a}) \, . \tag{199}$$

The solution to

$$\frac{\partial f(\boldsymbol{a})}{\partial \boldsymbol{a}} \; = \; 2 \sum_{i=1}^{N} p_i (\boldsymbol{a} \; - \; \boldsymbol{x}_i) \; = \; 0 \tag{200}$$

is

$$\boldsymbol{a} \; = \; \sum_{i=1}^{N} p_i \boldsymbol{x}_i \, . \tag{201}$$

The Hessian of $f$ is positive definite since

$$\frac{\partial^2 f(\boldsymbol{a})}{\partial \boldsymbol{a}^2} \;=\; 2 \sum_{i=1}^{N} p_i \, \boldsymbol{I} \;=\; 2\, \boldsymbol{I} \tag{202}$$

and $f$ is a convex function. Hence, the mean

$$\bar{\boldsymbol{x}} \;:=\; \sum_{i=1}^{N} p_i \, \boldsymbol{x}_i \tag{203}$$

minimizes $\sum_{i=1}^{N} p_i \|\boldsymbol{x}_i - \boldsymbol{a}\|^2$. Therefore we have

$$\sum_{i=1}^{l} p_i \|\boldsymbol{x}_i \,-\, \bar{\boldsymbol{x}}\|^2 \;\leqslant\; \sum_{i=1}^{l} p_i \|\boldsymbol{x}_i \,-\, \boldsymbol{m_x}\|^2 \;\leqslant\; (1 \,-\, \gamma)\, m_{\max}^2 \;. \tag{204}$$

We now bound the variance on the first $l$ patterns:

$$(1 - \gamma)\, \|\mathrm{Var}_{\tilde{p}}[\boldsymbol{x}_{1:l}]\|_2 \;\leqslant\; \sum_{i=1}^{l} p_i \left\|(\boldsymbol{x}_i \,-\, \bar{\boldsymbol{x}})\,(\boldsymbol{x}_i \,-\, \bar{\boldsymbol{x}})^T \right\|_2 \tag{205}$$

$$= \; \sum_{i=1}^{l} p_i \|\boldsymbol{x}_i \,-\, \bar{\boldsymbol{x}}\|^2 \;\leqslant\; \sum_{i=1}^{l} p_i \|\boldsymbol{x}_i \,-\, \boldsymbol{m_x}\|^2 \;\leqslant\; (1 \,-\, \gamma)\, m_{\max}^2 \;.$$

We obtain for the spectral norm of J:

$$\|\mathrm{J}\|_2 \;\leqslant\; \beta \Big( (1 - \gamma)\, \|\mathrm{Var}_{\tilde{p}}[\boldsymbol{x}_{1:l}]\|_2 \tag{206}$$

$$+ \;\frac{\gamma}{1-\gamma} \left\| \left( \sum_{i=1}^{l} p_i\, \boldsymbol{x}_i \right) \left( \sum_{i=1}^{l} p_i\, \boldsymbol{x}_i \right)^T \right\|_2$$

$$+ \;\left\| \sum_{i=l+1}^{N} p_i\, \boldsymbol{x}_i\, \boldsymbol{x}_i^T \right\|_2 + \left\| \left( \sum_{i=l+1}^{N} p_i\, \boldsymbol{x}_i \right) \left( \sum_{i=l+1}^{N} p_i\, \boldsymbol{x}_i \right)^T \right\|_2$$

$$+ \;\left\| \left( \sum_{i=1}^{l} p_i\, \boldsymbol{x}_i \right) \left( \sum_{i=l+1}^{N} p_i\, \boldsymbol{x}_i \right)^T \right\|_2 + \left\| \left( \sum_{i=l+1}^{N} p_i\, \boldsymbol{x}_i \right) \left( \sum_{i=1}^{l} p_i\, \boldsymbol{x}_i \right)^T \right\|_2 \Big)$$

$$\leqslant \; \beta \big( (1 - \gamma)\, \|\mathrm{Var}_{\tilde{p}}[\boldsymbol{x}_{1:l}]\|_2 \,+\, \gamma\,(1 - \gamma)\, M^2 \,+\, \gamma\, M^2 \,+\, \gamma^2\, M^2 \,+$$

$$\gamma\,(1 - \gamma)\, M^2 \,+\, \gamma\,(1 - \gamma)\, M^2 \big)$$

$$= \; \beta \big( (1 - \gamma)\, \|\mathrm{Var}_{\tilde{p}}[\boldsymbol{x}_{1:l}]\|_2 \,+\, \gamma\, 2\,(2 \,-\, \gamma)\, M^2 \big) \;.$$

Combining the previous two estimates immediately leads to Eq. (190).
The function $h(x) = x2(2 - x)$ has the derivative $h'(x) = 4(1 - x)$. Therefore $h(x)$ is monotone increasing for $x < 1$. For $0 \leqslant \gamma \leqslant \epsilon < 1$, we can immediately deduce that $\gamma 2(2 - \gamma) \leqslant \epsilon 2(2 - \epsilon)$. Since $\epsilon$ is larger than $\gamma$, we obtain the following $\epsilon$-bound for $\|\mathrm{J}\|_2$:

$$\|\mathrm{J}\|_2 \;\leqslant\; \beta \big( m_{\max}^2 \,+\, \epsilon\, 2\,(2 \,-\, \epsilon)\, M^2 \big) \;. \tag{207}$$

$\square$

We revisit the bound on $(1 - \gamma)\, \mathrm{Var}_{\tilde{p}}[\boldsymbol{x}_{1:l}]$. The trace $\sum_{k=1}^{d} e_k$ is the sum of the eigenvalues $e_k$. The spectral norm is equal to the largest eigenvalue $e_1$, that is, the largest singular value. We obtain:

$$\|\mathrm{Var}_{\tilde{p}}[\boldsymbol{x}_{1:l}]\|_2 \;=\; \mathrm{Tr} \left( \sum_{i=1}^{l} p_i\, (\boldsymbol{x}_i \,-\, \bar{\boldsymbol{x}})\,(\boldsymbol{x}_i \,-\, \bar{\boldsymbol{x}})^T \right) \,-\, \sum_{k=2}^{d} e_k \tag{208}$$

$$= \; \sum_{i=1}^{l} p_i\, \mathrm{Tr} \left( (\boldsymbol{x}_i \,-\, \bar{\boldsymbol{x}})\,(\boldsymbol{x}_i \,-\, \bar{\boldsymbol{x}})^T \right) \,-\, \sum_{k=2}^{d} e_k$$

$$= \; \sum_{i=1}^{l} p_i \|\boldsymbol{x}_i \,-\, \bar{\boldsymbol{x}}\|^2 \,-\, \sum_{k=2}^{d} e_k \;.$$

Therefore the tightness of the bound depends on eigenvalues which are not the largest. That is variations which are not along the strongest variation weaken the bound.

**Proof of a Fixed Point by Banach Fixed Point Theorem**   Without restricting the generality, we assume that the first $l$ patterns are much more probable (and similar to one another) than the other patterns. Therefore we define:

$$M := \max_{i} \|\boldsymbol{x}_i\| , \tag{209}$$

$$\gamma = \sum_{i=l+1}^{N} p_i \leqslant \epsilon , \tag{210}$$

$$1 - \gamma = \sum_{i=1}^{l} p_i \geq 1 - \epsilon , \tag{211}$$

$$\tilde{p}_i := \frac{p_i}{1 - \gamma} \leqslant p_i/(1 - \epsilon) , \tag{212}$$

$$\sum_{i=1}^{l} \tilde{p}_i = 1 , \tag{213}$$

$$\boldsymbol{m_x} = \frac{1}{l} \sum_{i=1}^{l} \boldsymbol{x}_i , \tag{214}$$

$$m_{\max} = \max_{1 \leqslant i \leqslant l} \|\boldsymbol{x}_i - \boldsymbol{m_x}\| . \tag{215}$$

$M$ is an upper bound on the Euclidean norm of the patterns, which are vectors. $\epsilon$ is an upper bound on the probability $\gamma$ of not choosing one of the first $l$ patterns, while $1 - \epsilon$ is a lower bound the probability $(1 - \gamma)$ of choosing one of the first $l$ patterns. $\boldsymbol{m_x}$ is the arithmetic mean (the center) of the first $l$ patterns. $m_{\max}$ is the maximal distance of the patterns to the center $\boldsymbol{m_x}$ . $\tilde{\boldsymbol{p}}$ is the probability $\boldsymbol{p}$ normalized for the first $l$ patterns.

**Mapped Vectors Stay in a Compact Environment.**   We show that if $\boldsymbol{m_x}$ is sufficient dissimilar to other $\boldsymbol{x}_j$ with $l < j$ then there is an compact environment of $\boldsymbol{m_x}$ (a sphere) where the fixed point iteration maps this environment into itself. The idea of the proof is to define a sphere around $\boldsymbol{m_x}$ for which the points from the sphere are mapped by $f$ into the sphere.

We first need following lemma which bounds the distance $\|\boldsymbol{m_x} - f(\boldsymbol{\xi})\|$ of a $\boldsymbol{\xi}$ which is close to $\boldsymbol{m_x}$.

**Lemma 9.** *For a query $\boldsymbol{\xi}$ and data $\boldsymbol{X} = (\boldsymbol{x}_1, \ldots, \boldsymbol{x}_N)$, we define*

$$0 \leqslant c = \min_{j,l<j} \left( \boldsymbol{\xi}^T \boldsymbol{m_x} - \boldsymbol{\xi}^T \boldsymbol{x}_j \right) = \boldsymbol{\xi}^T \boldsymbol{m_x} - \max_{j,l<j} \boldsymbol{\xi}^T \boldsymbol{x}_j . \tag{216}$$

*The following holds:*

$$\|\boldsymbol{m_x} - f(\boldsymbol{\xi})\| \leqslant m_{\max} + 2\gamma M \leqslant m_{\max} + 2\epsilon M , \tag{217}$$

*where*

$$M = \max_{i} \|\boldsymbol{x}_i\| , \tag{218}$$

$$\epsilon = (N - l) \exp(-\beta c) . \tag{219}$$

*Proof.* Let $s = \arg\max_{j,j \leqslant l} \boldsymbol{\xi}^T \boldsymbol{x}_j$, therefore $\boldsymbol{\xi}^T \boldsymbol{m_x} = \frac{1}{l} \sum_{i=1}^{l} \boldsymbol{\xi}^T \boldsymbol{x}_i \leqslant \frac{1}{l} \sum_{i=1}^{l} \boldsymbol{\xi}^T \boldsymbol{x}_s = \boldsymbol{\xi}^T \boldsymbol{x}_s$.
For softmax components $j$ with $l < j$ we have

$$[\text{softmax}(\beta \boldsymbol{X}^T \boldsymbol{\xi})]_j = \frac{\exp(\beta (\boldsymbol{\xi}^T \boldsymbol{x}_j - \boldsymbol{\xi}^T \boldsymbol{x}_s))}{1 + \sum_{k,k \neq s} \exp(\beta (\boldsymbol{\xi}^T \boldsymbol{x}_k - \boldsymbol{\xi}^T \boldsymbol{x}_s))} \leqslant \exp(-\beta c) = \frac{\epsilon}{N - l} , \tag{220}$$

since $\boldsymbol{\xi}^T \boldsymbol{x}_s - \boldsymbol{\xi}^T \boldsymbol{x}_j \geq \boldsymbol{\xi}^T \boldsymbol{m_x} - \boldsymbol{\xi}^T \boldsymbol{x}_j$ for each $j$ with $l < j$, therefore $\boldsymbol{\xi}^T \boldsymbol{x}_s - \boldsymbol{\xi}^T \boldsymbol{x}_j \geq c$
The iteration $f$ can be written as

$$f(\boldsymbol{\xi}) = \boldsymbol{X} \text{softmax}(\beta \boldsymbol{X}^T \boldsymbol{\xi}) = \sum_{j=1}^{N} \boldsymbol{x}_j [\text{softmax}(\beta \boldsymbol{X}^T \boldsymbol{\xi})]_j . \tag{221}$$

We set $p_i = [\mathrm{softmax}(\beta \boldsymbol{X}^T \boldsymbol{\xi})]_i$, therefore $\sum_{i=1}^{l} p_i = 1 - \gamma \geq 1 - \epsilon$ and $\sum_{i=l+1}^{N} p_i = \gamma \leqslant \epsilon$. Therefore

$$
\left\| \boldsymbol{m_x} - \sum_{j=1}^{l} \frac{p_j}{1-\gamma} \, \boldsymbol{x}_j \right\|^2 = \left\| \sum_{j=1}^{l} \frac{p_j}{1-\gamma} \, (\boldsymbol{m_x} - \boldsymbol{x}_j) \right\|^2 \tag{222}
$$

$$
= \sum_{j=1,k=1}^{l} \frac{p_j}{1-\gamma} \frac{p_k}{1-\gamma} \, (\boldsymbol{m_x} - \boldsymbol{x}_j)^T (\boldsymbol{m_x} - \boldsymbol{x}_k)
$$

$$
= \frac{1}{2} \sum_{j=1,k=1}^{l} \frac{p_j}{1-\gamma} \frac{p_k}{1-\gamma} \left( \|\boldsymbol{m_x} - \boldsymbol{x}_j\|^2 + \|\boldsymbol{m_x} - \boldsymbol{x}_k\|^2 - \|\boldsymbol{x}_j - \boldsymbol{x}_k\|^2 \right)
$$

$$
= \sum_{j=1}^{l} \frac{p_j}{1-\gamma} \|\boldsymbol{m_x} - \boldsymbol{x}_j\|^2 - \frac{1}{2} \sum_{j=1,k=1}^{l} \frac{p_j}{1-\gamma} \frac{p_k}{1-\gamma} \|\boldsymbol{x}_j - \boldsymbol{x}_k\|^2
$$

$$
\leqslant \sum_{j=1}^{l} \frac{p_j}{1-\gamma} \|\boldsymbol{m_x} - \boldsymbol{x}_j\|^2 \leqslant m_{\max}^2 \, .
$$

It follows that

$$
\left\| \boldsymbol{m_x} - \sum_{j=1}^{l} \frac{p_j}{1-\gamma} \, \boldsymbol{x}_j \right\| \leqslant m_{\max} \tag{223}
$$

We now can bound $\|\boldsymbol{m_x} - f(\boldsymbol{\xi})\|$:

$$
\|\boldsymbol{m_x} - f(\boldsymbol{\xi})\| = \left\| \boldsymbol{m_x} - \sum_{j=1}^{N} p_j \, \boldsymbol{x}_j \right\| \tag{224}
$$

$$
= \left\| \boldsymbol{m_x} - \sum_{j=1}^{l} p_j \, \boldsymbol{x}_j - \sum_{j=l+1}^{N} p_j \, \boldsymbol{x}_j \right\|
$$

$$
= \left\| \boldsymbol{m_x} - \sum_{j=1}^{l} \frac{p_j}{1-\gamma} \, \boldsymbol{x}_j + \frac{\gamma}{1-\gamma} \sum_{j=1}^{l} p_j \, \boldsymbol{x}_j - \sum_{j=l+1}^{N} p_j \, \boldsymbol{x}_j \right\|
$$

$$
\leqslant \left\| \boldsymbol{m_x} - \sum_{j=1}^{l} \frac{p_j}{1-\gamma} \, \boldsymbol{x}_j \right\| + \frac{\gamma}{1-\gamma} \left\| \sum_{j=1}^{l} p_j \, \boldsymbol{x}_j \right\| + \left\| \sum_{j=l+1}^{N} p_j \, \boldsymbol{x}_j \right\|
$$

$$
\leqslant \left\| \boldsymbol{m_x} - \sum_{j=1}^{l} \frac{p_j}{1-\gamma} \, \boldsymbol{x}_j \right\| + \frac{\gamma}{1-\gamma} \sum_{j=1}^{l} p_j \, M + \sum_{j=l+1}^{N} p_j \, M
$$

$$
\leqslant \left\| \boldsymbol{m_x} - \sum_{j=1}^{l} \frac{p_j}{1-\gamma} \, \boldsymbol{x}_j \right\| + 2 \, \gamma \, M
$$

$$
\leqslant m_{\max} + 2 \, \gamma \, M \leqslant m_{\max} + 2 \, \epsilon \, M \, ,
$$

where we applied Eq. (222) in the penultimate inequality. This is the statement of the lemma. $\square$

The separation of the center (the arithmetic mean) $\boldsymbol{m_x}$ of the first $l$ from data $\boldsymbol{X} = (\boldsymbol{x}_{l+1}, \dots, \boldsymbol{x}_N)$ is $\Delta_m$, defined as

$$
\Delta_m = \min_{j, l < j} \left( \boldsymbol{m_x}^T \boldsymbol{m_x} - \boldsymbol{m_x}^T \boldsymbol{x}_j \right) = \boldsymbol{m_x}^T \boldsymbol{m_x} - \max_{j, l < j} \boldsymbol{m_x}^T \boldsymbol{x}_j \, . \tag{225}
$$

The center is separated from the other data $\boldsymbol{x}_j$ with $l < j$ if $0 < \Delta_m$. By the same arguments as in Eq. (129), $\Delta_m$ can also be expressed as

$$
\begin{aligned}
\Delta_m &= \min_{j,l<j} \frac{1}{2} \left( \|\boldsymbol{m}_{\boldsymbol{x}}\|^2 - \|\boldsymbol{x}_j\|^2 + \|\boldsymbol{m}_{\boldsymbol{x}} - \boldsymbol{x}_j\|^2 \right) \\
&= \frac{1}{2} \|\boldsymbol{m}_{\boldsymbol{x}}\|^2 - \frac{1}{2} \max_{j,l<j} \left( \|\boldsymbol{x}_j\|^2 - \|\boldsymbol{m}_{\boldsymbol{x}} - \boldsymbol{x}_j\|^2 \right) .
\end{aligned}
\tag{226}
$$

For $\|\boldsymbol{m}_{\boldsymbol{x}}\| = \|\boldsymbol{x}_j\|$ we have $\Delta_m = 1/2 \min_{j,l<j} \|\boldsymbol{m}_{\boldsymbol{x}} - \boldsymbol{x}_j\|^2$.
Next we define the sphere where we want to apply Banach fixed point theorem.

**Definition B3** (Sphere $S_m$). *The sphere $S_m$ is defined as*

$$
S_m := \left\{ \boldsymbol{\xi} \mid \|\boldsymbol{\xi} - \boldsymbol{m}_{\boldsymbol{x}}\| \leqslant \frac{1}{\beta \, m_{\max}} \right\} .
\tag{227}
$$

**Lemma 10.** *With $\boldsymbol{\xi}$ given, if the assumptions*

   *A1: $\boldsymbol{\xi}$ is inside sphere: $\boldsymbol{\xi} \in S_m$,*
   *A2: the center $\boldsymbol{m}_{\boldsymbol{x}}$ is well separated from other data $\boldsymbol{x}_j$ with $l < j$:*

$$
\Delta_m \geqslant \frac{2\,M}{\beta\,m_{\max}} - \frac{1}{\beta} \ln\left( \frac{1 - \beta\,m_{\max}^2}{2\,\beta\,(N-l)\,M\,\max\{m_{\max},\,2\,M\}} \right) ,
\tag{228}
$$

   *A3: the distance $m_{\max}$ of similar patterns to the center is sufficient small:*

$$
\beta\,m_{\max}^2 \leqslant 1
\tag{229}
$$

*hold, then $f(\boldsymbol{\xi}) \in S_m$. Therefore, under conditions (A2) and (A3), $f$ is a mapping from $S_m$ into $S_m$.*

*Proof.* We need the separation $\tilde{\Delta}_m$ of $\boldsymbol{\xi}$ from the rest of the data, which is the last $N - l$ data points $\boldsymbol{X} = (\boldsymbol{x}_{l+1}, \ldots, \boldsymbol{x}_N)$.

$$
\tilde{\Delta}_m = \min_{j,l<j} \left( \boldsymbol{\xi}^T \boldsymbol{m}_{\boldsymbol{x}} - \boldsymbol{\xi}^T \boldsymbol{x}_j \right) .
\tag{230}
$$

Using the Cauchy-Schwarz inequality, we obtain for $l + 1 \leqslant j \leqslant N$:

$$
\left| \boldsymbol{\xi}^T \boldsymbol{x}_j - \boldsymbol{m}_{\boldsymbol{x}}^T \boldsymbol{x}_j \right| \leqslant \|\boldsymbol{\xi} - \boldsymbol{m}_{\boldsymbol{x}}\| \, \|\boldsymbol{x}_j\| \leqslant \|\boldsymbol{\xi} - \boldsymbol{m}_{\boldsymbol{x}}\| \, M .
\tag{231}
$$

We have the lower bound

$$
\begin{aligned}
\tilde{\Delta}_m &\geqslant \min_{j,l<j} \left( \left( \boldsymbol{m}_{\boldsymbol{x}}^T \boldsymbol{m}_{\boldsymbol{x}} - \|\boldsymbol{\xi} - \boldsymbol{m}_{\boldsymbol{x}}\| \, M \right) - \left( \boldsymbol{m}_{\boldsymbol{x}}^T \boldsymbol{x}_j + \|\boldsymbol{\xi} - \boldsymbol{m}_{\boldsymbol{x}}\| \, M \right) \right) \\
&= -2\,\|\boldsymbol{\xi} - \boldsymbol{m}_{\boldsymbol{x}}\| \, M + \min_{j,l<j} \left( \boldsymbol{m}_{\boldsymbol{x}}^T \boldsymbol{m}_{\boldsymbol{x}} - \boldsymbol{m}_{\boldsymbol{x}}^T \boldsymbol{x}_j \right) = \Delta_m - 2\,\|\boldsymbol{\xi} - \boldsymbol{m}_{\boldsymbol{x}}\| \, M \\
&\geqslant \Delta_m - 2\,\frac{M}{\beta\,m_{\max}} ,
\end{aligned}
\tag{232}
$$

where we used the assumption (A1) of the lemma.
From the proof in Lemma 9 we have

$$
\sum_{i=1}^{l} p_i \geqslant 1 - (N-l)\,\exp(-\beta\,\tilde{\Delta}_m) = 1 - \tilde{\epsilon} ,
\tag{233}
$$

$$
\sum_{i=l+1}^{N} p_i \leqslant (N-l)\,\exp(-\beta\,\tilde{\Delta}_m) = \tilde{\epsilon} .
\tag{234}
$$

Lemma 9 states that

$$
\begin{aligned}
\|\boldsymbol{m}_{\boldsymbol{x}} - f(\boldsymbol{\xi})\| &\leqslant m_{\max} + 2\,\tilde{\epsilon}\,M \\
&\leqslant m_{\max} + 2\,(N-l)\,\exp(-\beta\,\tilde{\Delta}_m)\,M . \\
&\leqslant m_{\max} + 2\,(N-l)\,\exp\left(-\beta\,(\Delta_m - 2\,\frac{M}{\beta\,m_{\max}})\right)\,M .
\end{aligned}
\tag{235}
$$

Therefore we have

$$\|\boldsymbol{m_x} - f(\boldsymbol{\xi})\| \leqslant m_{\max} + 2\,(N-l)\,\exp\left(-\,\beta\,(\Delta_m - 2\,\frac{M}{\beta\,m_{\max}})\right)\,M \tag{236}$$

$$\leqslant m_{\max} + 2\,(N-l)\,\exp\Bigg(-\,\beta\,\Bigg(\frac{2\,M}{\beta\,m_{\max}} -$$

$$\frac{1}{\beta}\,\ln\left(\frac{1\,-\,\beta\,m_{\max}^2}{2\,\beta\,(N-l)\,M\,\max\{m_{\max}\,,\,2\,M\}}\right)\,-\,2\,\frac{M}{\beta\,m_{\max}}\Bigg)\Bigg)\,M$$

$$=\, m_{\max} + 2\,(N-l)\,\frac{1\,-\,\beta\,m_{\max}^2}{2\,\beta\,(N-l)\,M\,\max\{m_{\max}\,,\,2\,M\}}\,M$$

$$\leqslant\, m_{\max} + \frac{1\,-\,\beta\,m_{\max}^2}{\beta\,m_{\max}}\,=\,\frac{1}{\beta\,m_{\max}}\;,$$

where we used assumption (A2) of the lemma. Therefore, $f(\boldsymbol{\xi})$ is a mapping from the sphere $S_m$ into the sphere $S_m$.

$$m_{\max} = \max_{1 \leqslant i \leqslant l}\|\boldsymbol{x}_i - \boldsymbol{m_x}\| \tag{237}$$

$$= \max_{1 \leqslant i \leqslant l}\left\|\boldsymbol{x}_i - 1/l\sum_{j=1}^{l}\boldsymbol{x}_j\right\| \tag{238}$$

$$= \max_{1 \leqslant i \leqslant l}\left\|1/l\sum_{j=1}^{l}(\boldsymbol{x}_i - \boldsymbol{x}_j)\right\| \tag{239}$$

$$\leqslant \max_{1 \leqslant i,j \leqslant l}\|\boldsymbol{x}_i - \boldsymbol{x}_j\| \tag{240}$$

$$\leqslant \max_{1 \leqslant i \leqslant l}\|\boldsymbol{x}_i\| + \max_{1 \leqslant j \leqslant l}\|\boldsymbol{x}_i\| \tag{241}$$

$$\leqslant 2M \tag{242}$$

$\square$

**Contraction Mapping.** For applying Banach fixed point theorem we need to show that $f$ is contraction in the compact environment $S_m$.

**Lemma 11.** *Assume that*

*A1:*

$$\Delta_m \,\geqslant\, \frac{2\,M}{\beta\,m_{\max}}\,-\,\frac{1}{\beta}\,\ln\left(\frac{1\,-\,\beta\,m_{\max}^2}{2\,\beta\,(N-l)\,M\,\max\{m_{\max}\,,\,2\,M\}}\right)\,, \tag{243}$$

*and*

*A2:*

$$\beta\,m_{\max}^2 \,\leqslant\, 1\,, \tag{244}$$

*then $f$ is a contraction mapping in $S_m$.*

*Proof.* The mean value theorem states for the symmetric $J^m = \int_0^1 J(\lambda\boldsymbol{\xi} + (1-\lambda)\boldsymbol{m_x})\,\mathrm{d}\lambda$:

$$f(\boldsymbol{\xi}) \,=\, f(\boldsymbol{m_x}) \,+\, J^m\,(\boldsymbol{\xi} - \boldsymbol{m_x})\,. \tag{245}$$

In complete analogy to Lemma 6, we get:

$$\|f(\boldsymbol{\xi}) - f(\boldsymbol{m_x})\| \,\leqslant\, \|J^m\|_2\,\|\boldsymbol{\xi} - \boldsymbol{m_x}\|\,. \tag{246}$$

We define $\tilde{\boldsymbol{\xi}} = \lambda\boldsymbol{\xi} + (1-\lambda)\boldsymbol{m_x}$ for some $\lambda \in [0,1]$. We need the separation $\tilde{\Delta}_m$ of $\tilde{\boldsymbol{\xi}}$ from the rest of the data, which is the last $N - l$ data points $\boldsymbol{X} = (\boldsymbol{x}_{l+1}, \ldots, \boldsymbol{x}_N)$.

$$\tilde{\Delta}_m \,=\, \min_{j,l<j}\left(\tilde{\boldsymbol{\xi}}^T\boldsymbol{m_x} - \tilde{\boldsymbol{\xi}}^T\boldsymbol{x}_j\right)\,. \tag{247}$$

From the proof in Lemma 9 we have

$$\tilde{\epsilon} \; = \; (N-l) \; \exp(-\beta \, \tilde{\Delta}_m) \, , \tag{248}$$

$$\sum_{i=1}^{l} p_i(\tilde{\boldsymbol{\xi}}) \; \geq \; 1 \; - \; (N-l) \; \exp(-\beta \, \tilde{\Delta}_m) \; = \; 1 \; - \; \tilde{\epsilon} \, , \tag{249}$$

$$\sum_{i=l+1}^{N} p_i(\tilde{\boldsymbol{\xi}}) \; \leq \; (N-l) \; \exp(-\beta \, \tilde{\Delta}_m) \; = \; \tilde{\epsilon} \, . \tag{250}$$

We first compute an upper bound on $\tilde{\epsilon}$. Using the Cauchy-Schwarz inequality, we obtain for $l+1 \leq j \leq N$:

$$\left| \tilde{\boldsymbol{\xi}}^T \boldsymbol{x}_j \; - \; \boldsymbol{m}_{\boldsymbol{x}}^T \boldsymbol{x}_j \right| \; \leq \; \left\| \tilde{\boldsymbol{\xi}} \; - \; \boldsymbol{m}_{\boldsymbol{x}} \right\| \; \|\boldsymbol{x}_j\| \; \leq \; \left\| \tilde{\boldsymbol{\xi}} \; - \; \boldsymbol{m}_{\boldsymbol{x}} \right\| M \, . \tag{251}$$

We have the lower bound on $\tilde{\Delta}_m$:

$$\begin{aligned}
\tilde{\Delta}_m \; &\geq \; \min_{j,l<j} \left( \left( \boldsymbol{m}_{\boldsymbol{x}}^T \boldsymbol{m}_{\boldsymbol{x}} - \left\| \tilde{\boldsymbol{\xi}} - \boldsymbol{m}_{\boldsymbol{x}} \right\| M \right) - \left( \boldsymbol{m}_{\boldsymbol{x}}^T \boldsymbol{x}_j + \left\| \tilde{\boldsymbol{\xi}} - \boldsymbol{m}_{\boldsymbol{x}} \right\| M \right) \right) \\
&= \; -2 \left\| \tilde{\boldsymbol{\xi}} - \boldsymbol{m}_{\boldsymbol{x}} \right\| M + \min_{j,l<j} \left( \boldsymbol{m}_{\boldsymbol{x}}^T \boldsymbol{m}_{\boldsymbol{x}} - \boldsymbol{m}_{\boldsymbol{x}}^T \boldsymbol{x}_j \right) \; = \; \Delta_m - 2 \left\| \tilde{\boldsymbol{\xi}} - \boldsymbol{m}_{\boldsymbol{x}} \right\| M \\
&\geq \; \Delta_m - 2 \|\boldsymbol{\xi} - \boldsymbol{m}_{\boldsymbol{x}}\| M \, .
\end{aligned} \tag{252}$$

where we used $\left\| \tilde{\boldsymbol{\xi}} - \boldsymbol{m}_{\boldsymbol{x}} \right\| = \lambda \|\boldsymbol{\xi} - \boldsymbol{m}_{\boldsymbol{x}}\| \leq \|\boldsymbol{\xi} - \boldsymbol{m}_{\boldsymbol{x}}\|$. We obtain the upper bound on $\tilde{\epsilon}$:

$$\begin{aligned}
\tilde{\epsilon} \; &\leq \; (N-l) \; \exp\left( -\beta \left( \Delta_m - 2 \|\boldsymbol{\xi} - \boldsymbol{m}_{\boldsymbol{x}}\| M \right) \right) \\
&\leq \; (N-l) \; \exp\left( -\beta \left( \Delta_m - \frac{2M}{\beta \, m_{\max}} \right) \right) \, .
\end{aligned} \tag{253}$$

where we used that in the sphere $S_i$ holds:

$$\|\boldsymbol{\xi} \; - \; \boldsymbol{m}_{\boldsymbol{x}}\| \; \leq \; \frac{1}{\beta \, m_{\max}} \, , \tag{254}$$

therefore

$$2 \, \|\boldsymbol{\xi} \; - \; \boldsymbol{m}_{\boldsymbol{x}}\| \, M \; \leq \; \frac{2M}{\beta \, m_{\max}} \, . \tag{255}$$

Next we compute a lower bound on $\tilde{\epsilon}$ and to this end start with the upper bound on $\tilde{\Delta}_m$ using the same arguments as in Eq. (147) in combination with Eq. (255).

$$\begin{aligned}
\tilde{\Delta}_m \; &\geq \; \min_{j,l<j} \left( \left( \boldsymbol{m}_{\boldsymbol{x}}^T \boldsymbol{m}_{\boldsymbol{x}} + \left\| \tilde{\boldsymbol{\xi}} - \boldsymbol{m}_{\boldsymbol{x}} \right\| M \right) - \left( \boldsymbol{m}_{\boldsymbol{x}}^T \boldsymbol{x}_j - \left\| \tilde{\boldsymbol{\xi}} - \boldsymbol{m}_{\boldsymbol{x}} \right\| M \right) \right) \\
&= \; 2 \left\| \tilde{\boldsymbol{\xi}} - \boldsymbol{m}_{\boldsymbol{x}} \right\| M + \min_{j,l<j} \left( \boldsymbol{m}_{\boldsymbol{x}}^T \boldsymbol{m}_{\boldsymbol{x}} - \boldsymbol{m}_{\boldsymbol{x}}^T \boldsymbol{x}_j \right) \; = \; \Delta_m + 2 \left\| \tilde{\boldsymbol{\xi}} - \boldsymbol{m}_{\boldsymbol{x}} \right\| M \\
&\geq \; \Delta_m + 2 \|\boldsymbol{\xi} - \boldsymbol{m}_{\boldsymbol{x}}\| M \, .
\end{aligned} \tag{256}$$

where we used $\left\| \tilde{\boldsymbol{\xi}} - \boldsymbol{m}_{\boldsymbol{x}} \right\| = \lambda \|\boldsymbol{\xi} - \boldsymbol{m}_{\boldsymbol{x}}\| \leq \|\boldsymbol{\xi} - \boldsymbol{m}_{\boldsymbol{x}}\|$. We obtain the lower bound on $\tilde{\epsilon}$:

$$\tilde{\epsilon} \; \geq \; (N-l) \; \exp\left( -\beta \left( \Delta_m + \frac{2M}{\beta \, m_{\max}} \right) \right) \, , \tag{257}$$

where we used that in the sphere $S_i$ holds:

$$\|\boldsymbol{\xi} \; - \; \boldsymbol{m}_{\boldsymbol{x}}\| \; \leq \; \frac{1}{\beta \, m_{\max}} \, , \tag{258}$$

therefore

$$2 \, \|\boldsymbol{\xi} \; - \; \boldsymbol{m}_{\boldsymbol{x}}\| \, M \; \leq \; \frac{2M}{\beta \, m_{\max}} \, . \tag{259}$$

From Lemma 8 we have

$$\left\| \mathrm{J}(\tilde{\boldsymbol{\xi}}) \right\|_2 \;\leqslant\; \beta \left( m_{\mathrm{max}}^2 \;+\; \tilde{\epsilon}\,2\,(2\,-\,\tilde{\epsilon})\,M^2 \right) \tag{260}$$

$$= \;\beta \left( m_{\mathrm{max}}^2 \;+\; \tilde{\epsilon}4\,M^2 \;-\; 2\,\tilde{\epsilon}^2\,M^2 \right)$$

$$\leqslant \;\beta \left( m_{\mathrm{max}}^2 \;+\; (N-l)\,\exp\left( -\,\beta\,\left( \Delta_m \;-\; \frac{2\,M}{\beta\,m_{\mathrm{max}}} \right) \right) 4\,M^2 \;-\right.$$

$$\left. 2\,(N-l)^2\,\exp\left( -\,2\,\beta\,\left( \Delta_m \;+\; \frac{2\,M}{\beta\,m_{\mathrm{max}}} \right) \right) M^2 \right) \;.$$

The bound Eq. (260) holds for the mean $\mathrm{J}^m$, too, since it averages over $\mathrm{J}(\tilde{\boldsymbol{\xi}})$:

$$\left\| \mathrm{J}^m \right\|_2 \;\leqslant\; \beta \left( m_{\mathrm{max}}^2 \;+\; (N-l)\,\exp\left( -\,\beta\,\left( \Delta_m \;-\; \frac{2\,M}{\beta\,m_{\mathrm{max}}} \right) \right) 4\,M^2 \;- \tag{261}\right.$$

$$\left. 2\,(N-l)^2\,\exp\left( -\,2\,\beta\,\left( \Delta_m \;+\; \frac{2\,M}{\beta\,m_{\mathrm{max}}} \right) \right) M^2 \right) \;.$$

The assumption of the lemma is

$$\Delta_m \;\geq\; \frac{2\,M}{\beta\,m_{\mathrm{max}}} \;-\; \frac{1}{\beta}\,\ln\left( \frac{1 \;-\; \beta\,m_{\mathrm{max}}^2}{2\,\beta\,(N-l)\,M\,\max\{m_{\mathrm{max}}\,,\,2\,M\}} \right) \;, \tag{262}$$

Therefore we have

$$\Delta_m \;-\; \frac{2\,M}{\beta\,m_{\mathrm{max}}} \;\geq\; -\,\frac{1}{\beta}\,\ln\left( \frac{1 \;-\; \beta\,m_{\mathrm{max}}^2}{2\,\beta\,(N-l)\,M\,\max\{m_{\mathrm{max}}\,,\,2\,M\}} \right) \;. \tag{263}$$

Therefore the spectral norm $\|\mathrm{J}^m\|_2$ can be bounded by:

$$\|\mathrm{J}^m\|_2 \;\leqslant \tag{264}$$

$$\beta \left( m_{\mathrm{max}}^2 \;+\; (N-l)\,\exp\left( -\,\beta\,\left( -\,\frac{1}{\beta}\,\ln\left( \frac{1 \;-\; \beta\,m_{\mathrm{max}}^2}{2\,\beta\,(N-l)\,M\,\max\{m_{\mathrm{max}}\,,\,2\,M\}} \right) \right) \right) \right.$$

$$4\,M^2 \;-\; 2\,(N-l)^2\,\exp\left( -\,2\,\beta\,\left( \Delta_m \;+\; \frac{2\,M}{\beta\,m_{\mathrm{max}}} \right) \right) M^2 \bigg)$$

$$= \;\beta \left( m_{\mathrm{max}}^2 \;+\; (N-l)\,\exp\left( \ln\left( \frac{1 \;-\; \beta\,m_{\mathrm{max}}^2}{2\,\beta\,(N-l)\,M\,\max\{m_{\mathrm{max}}\,,\,2\,M\}} \right) \right) \right.$$

$$4\,M^2 \;-\; 2\,(N-l)^2\,\exp\left( -\,2\,\beta\,\left( \Delta_m \;+\; \frac{2\,M}{\beta\,m_{\mathrm{max}}} \right) \right) M^2 \bigg)$$

$$= \;\beta \left( m_{\mathrm{max}}^2 \;+\; (N-l)\,\frac{1 \;-\; \beta\,m_{\mathrm{max}}^2}{2\,\beta\,(N-l)\,M\,\max\{m_{\mathrm{max}}\,,\,2\,M\}}\,4\,M^2 \;-\right.$$

$$2\,(N-l)^2\,\exp\left( -\,2\,\beta\,\left( \Delta_m \;+\; \frac{2\,M}{\beta\,m_{\mathrm{max}}} \right) \right) M^2 \bigg)$$

$$= \;\beta m_{\mathrm{max}}^2 \;+\; \frac{1 \;-\; \beta\,m_{\mathrm{max}}^2}{\max\{m_{\mathrm{max}}\,,\,2\,M\}}\,2\,M \;-$$

$$\beta\,2\,(N-l)^2\,\exp\left( -\,2\,\beta\,\left( \Delta_m \;+\; \frac{2\,M}{\beta\,m_{\mathrm{max}}} \right) \right) M^2$$

$$\leqslant \;\beta m_{\mathrm{max}}^2 \;+\; 1 \;-\; \beta\,m_{\mathrm{max}}^2 \;-\; \beta\,2\,(N-l)^2\,\exp\left( -\,2\,\beta\,\left( \Delta_m \;+\; \frac{2\,M}{\beta\,m_{\mathrm{max}}} \right) \right) M^2$$

$$= \;1 \;-\; \beta\,2\,(N-l)^2\,\exp\left( -\,2\,\beta\,\left( \Delta_m \;+\; \frac{2\,M}{\beta\,m_{\mathrm{max}}} \right) \right) M^2 \;<\; 1 \;.$$

For the last but one inequality we used $2M \leqslant \max\{m_{\mathrm{max}}, 2M\}$.
Therefore $f$ is a contraction mapping in $\mathrm{S}_m$. $\qquad\qquad\square$

**Banach Fixed Point Theorem.** Now we have all ingredients to apply Banach fixed point theorem.

**Lemma 12.** *Assume that*

*A1:*

$$\Delta_m \geq \frac{2\,M}{\beta\,m_{\max}} - \frac{1}{\beta}\,\ln\left(\frac{1\,-\,\beta\,m_{\max}^2}{2\,\beta\,(N-l)\,M\,\max\{m_{\max}\,,\,2\,M\}}\right)\,,\tag{265}$$

*and*

*A2:*

$$\beta\,m_{\max}^2\ \leqslant\ 1\,,\tag{266}$$

*then $f$ has a fixed point in $\mathrm{S}_m$.*

*Proof.* We use Banach fixed point theorem: Lemma 10 says that $f$ maps from the compact set $\mathrm{S}_m$ into the same compact set $\mathrm{S}_m$. Lemma 11 says that $f$ is a contraction mapping in $\mathrm{S}_m$. □

**Contraction Mapping with a Fixed Point** We assume that the first $l$ patterns are much more probable (and similar to one another) than the other patterns. Therefore we define:

$$M\ :=\ \max_i \|\boldsymbol{x}_i\|\,,\tag{267}$$

$$\gamma\ =\ \sum_{i=l+1}^{N} p_i\ \leqslant\ \epsilon\,,\tag{268}$$

$$1-\gamma\ =\ \sum_{i=1}^{l} p_i\ \geq\ 1\,-\,\epsilon\,,\tag{269}$$

$$\tilde{p}_i\ :=\ \frac{p_i}{1-\gamma}\ \leqslant\ p_i/(1-\epsilon)\,,\tag{270}$$

$$\sum_{i=1}^{l} \tilde{p}_i\ =\ 1\,,\tag{271}$$

$$\boldsymbol{m_x}\ =\ \frac{1}{l}\,\sum_{i=1}^{l} \boldsymbol{x}_i\,,\tag{272}$$

$$m_{\max}\ =\ \max_{1\leqslant i\leqslant l} \|\boldsymbol{x}_i\,-\,\boldsymbol{m_x}\|\,.\tag{273}$$

$M$ is an upper bound on the Euclidean norm of the patterns, which are vectors. $\epsilon$ is an upper bound on the probability $\gamma$ of not choosing one of the first $l$ patterns, while $1-\epsilon$ is a lower bound the probability $(1-\gamma)$ of choosing one of the first $l$ patterns. $\boldsymbol{m_x}$ is the arithmetic mean (the center) of the first $l$ patterns. $m_{\max}$ is the maximal distance of the patterns to the center $\boldsymbol{m_x}$. $\tilde{p}$ is the probability $\boldsymbol{p}$ normalized for the first $l$ patterns.

The variance of the first $l$ patterns is

$$\mathrm{Var}_{\tilde{p}}[\boldsymbol{x}_{1:l}]\ =\ \sum_{i=1}^{l} \tilde{p}_i\,\boldsymbol{x}_i\,\boldsymbol{x}_i^T\ -\ \left(\sum_{i=1}^{l} \tilde{p}_i\,\boldsymbol{x}_i\right)\left(\sum_{i=1}^{l} \tilde{p}_i\,\boldsymbol{x}_i\right)^T\tag{274}$$

$$=\ \sum_{i=1}^{l} \tilde{p}_i\,\left(\boldsymbol{x}_i\,-\,\sum_{i=1}^{l} \tilde{p}_i\boldsymbol{x}_i\right)\left(\boldsymbol{x}_i\,-\,\sum_{i=1}^{l} \tilde{p}_i\boldsymbol{x}_i\right)^T\,.$$

We have shown that a fixed point exists. We want to know how fast the iteration converges to the fixed point. Let $\boldsymbol{m_x^*}$ be the fixed point of the iteration $f$ in the sphere $\mathrm{S}_m$. Using the mean value theorem, we have with $\mathrm{J}^m = \int_0^1 \mathrm{J}(\lambda\boldsymbol{\xi} + (1-\lambda)\boldsymbol{m_x^*})\,\mathrm{d}\lambda$:

$$\|f(\boldsymbol{\xi})\,-\,\boldsymbol{m_x^*}\|\ =\ \|f(\boldsymbol{\xi})\,-\,f(\boldsymbol{m_x^*})\|\ \leqslant\ \|\mathrm{J}^m\|_2\,\|\boldsymbol{\xi}\,-\,\boldsymbol{m_x^*}\|\tag{275}$$

According to Lemma 8 the following bounds on the norm $\|\mathrm{J}\|_2$ of the Jacobian of the fixed point iteration hold. The $\gamma$-bound for $\|\mathrm{J}\|_2$ is

$$\|\mathrm{J}\|_2\ \leqslant\ \beta\left((1-\gamma)\,m_{\max}^2\ +\ \gamma\,2\,(2\,-\,\gamma)\,M^2\right)\,,\tag{276}$$

while the $\epsilon$-bound for $\|J\|_2$ is:

$$\|J\|_2 \leqslant \beta \left( m_{\max}^2 + \epsilon \, 2 \, (2 - \epsilon) \, M^2 \right) . \tag{277}$$

From the last condition we require for a contraction mapping:

$$\beta \, m_{\max}^2 < 1 . \tag{278}$$

We want to see how large $\epsilon$ is. The separation of center $m_x$ from data $X = (x_{l+1}, \ldots, x_N)$ is

$$\Delta_m = \min_{j,l<j} \left( m_x^T m_x - m_x^T x_j \right) = m_x^T m_x - \max_{j,l<j} m_x^T x_j . \tag{279}$$

We need the separation $\tilde{\Delta}_m$ of $\tilde{x} = \lambda \xi + (1 - \lambda) m_x^*$ from the data.

$$\tilde{\Delta}_m = \min_{j,l<j} \left( \tilde{x}^T m_x - \tilde{x}^T x_j \right) . \tag{280}$$

We compute a lower bound on $\tilde{\Delta}_m$. Using the Cauchy-Schwarz inequality, we obtain for $1 \leqslant j \leqslant N$:

$$\left| \tilde{x}^T x_j - m_x^T x_j \right| \leqslant \|\tilde{x} - m_x\| \, \|x_j\| \leqslant \|\tilde{x} - m_x\| \, M . \tag{281}$$

We have the lower bound

$$\tilde{\Delta}_m \geq \min_{j,l<j} \left( \left( m_x^T m_x - \|\tilde{x} - m_x\| \, M \right) - \left( m_x^T x_j + \|\tilde{x} - m_x\| \, M \right) \right) \tag{282}$$

$$= - 2 \, \|\tilde{x} - m_x\| \, M + \min_{j,l<j} \left( m_x^T m_x - m_x^T x_j \right) = \Delta_m - 2 \, \|\tilde{x} - m_x\| \, M .$$

Since

$$\|\tilde{x} - m_x\| = \|\lambda \xi + (1 - \lambda) m_x^* - m_x\| \tag{283}$$
$$\leqslant \lambda \|\xi - m_x\| + (1 - \lambda) \|m_x^* - m_x\|$$
$$\leqslant \max\{\|\xi - m_x\|, \|m_x^* - m_x\|\} ,$$

we have

$$\tilde{\Delta}_m \geq \Delta_m - 2 \, \max\{\|\xi - m_x\|, \|m_x^* - m_x\|\} \, M . \tag{284}$$

$$\epsilon = (N - l) \exp(- \beta \left( \Delta_m - 2 \, \max\{\|\xi - m_x\|, \|m_x^* - m_x\|\} \, M \right)) . \tag{285}$$

## B2.5  Properties of Fixed Points Near Stored Pattern

In Subsection B2.4.3 many stable states that are fixed points near the stored patterns are considered. We now consider this case. In the fist subsection we investigate the storage capacity if all patterns are sufficiently separated so that metastable states do not appear. In the next subsection we look into the convergence speed and error when retrieving the stored patterns. For metastable states we can do the same analyses if each metastable state is treated as one state like one pattern.

We see a trade-off that is known from classical Hopfield networks and for modern Hopfield networks. Small separation $\Delta_i$ of the pattern $x_i$ from the other patterns gives high storage capacity. However the convergence speed is lower and the retrieval error higher. In contrast, large separation $\Delta_i$ of the pattern $x_i$ from the other pattern gives exponentially fast convergence (one update is sufficient) and exponentially low retrieval error.

### B2.5.1  Exponentially Many Patterns can be Stored

From Subsection B2.4.3 need some definitions. We assume to have $N$ patterns, the separation of pattern $x_i$ from the other patterns $\{x_1, \ldots, x_{i-1}, x_{i+1}, \ldots, x_N\}$ is $\Delta_i$, defined as

$$\Delta_i = \min_{j,j\neq i} \left( x_i^T x_i - x_i^T x_j \right) = x_i^T x_i - \max_{j,j\neq i} x_i^T x_j . \tag{286}$$

The pattern is separated from the other data if $0 < \Delta_i$. The separation $\Delta_i$ can also be expressed as

$$\Delta_i = \min_{j,j\neq i} \frac{1}{2} \left( \|x_i\|^2 - \|x_j\|^2 + \|x_i - x_j\|^2 \right) \tag{287}$$

$$= \frac{1}{2} \|x_i\|^2 - \frac{1}{2} \max_{j,j\neq i} \left( \|x_j\|^2 - \|x_i - x_j\|^2 \right) .$$

For $\|\boldsymbol{x}_i\| = \|\boldsymbol{x}_j\|$ we have $\Delta_i = 1/2 \min_{j,j \neq i} \|\boldsymbol{x}_i - \boldsymbol{x}_j\|^2$. The sphere $\mathrm{S}_i$ with center $\boldsymbol{x}_i$ is defined as

$$\mathrm{S}_i = \left\{ \boldsymbol{\xi} \mid \|\boldsymbol{\xi} - \boldsymbol{x}_i\| \leqslant \frac{1}{\beta \, N \, M} \right\} . \tag{288}$$

The maximal length of a pattern is $M = \max_i \|\boldsymbol{x}_i\|$.

We next define what we mean with storing and retrieving a pattern.

**Definition B4** (Pattern Stored and Retrieved). *We assume that around every pattern $\boldsymbol{x}_i$ a sphere $\mathrm{S}_i$ is given. We say $\boldsymbol{x}_i$ is stored if there is a single fixed point $\boldsymbol{x}_i^* \in \mathrm{S}_i$ to which all points $\boldsymbol{\xi} \in \mathrm{S}_i$ converge, and $\mathrm{S}_i \cap \mathrm{S}_j = \emptyset$ for $i \neq j$. We say $\boldsymbol{x}_i$ is retrieved if iteration (update rule) Eq. (81) converged to the single fixed point $\boldsymbol{x}_i^* \in \mathrm{S}_i$. The retrieval error is $\|\boldsymbol{x}_i - \boldsymbol{x}_i^*\|$.*

For a query $\boldsymbol{\xi} \in \mathrm{S}_i$ to converge to a fixed point $\boldsymbol{x}_i^* \in \mathrm{S}_i$ we required for the application of Banach fixed point theorem and for ensuring a contraction mapping the following inequality:

$$\Delta_i \geqslant \frac{2}{\beta \, N} + \frac{1}{\beta} \, \ln \left( 2 \, (N - 1) \, N \, \beta \, M^2 \right) . \tag{289}$$

This is the assumption in Lemma 7 to ensure a fixed point in sphere $\mathrm{S}_i$. Since replacing $(N-1)N$ by $N^2$ gives

$$\frac{2}{\beta \, N} + \frac{1}{\beta} \, \ln \left( 2 \, N^2 \, \beta \, M^2 \right) > \frac{2}{\beta \, N} + \frac{1}{\beta} \, \ln \left( 2 \, (N - 1) \, N \, \beta \, M^2 \right) , \tag{290}$$

the inequality follows from following master inequality

$$\Delta_i \geqslant \frac{2}{\beta \, N} + \frac{1}{\beta} \, \ln \left( 2 \, N^2 \, \beta \, M^2 \right) , \tag{291}$$

If we assume that $\mathrm{S}_i \cap \mathrm{S}_j \neq \emptyset$ with $i \neq j$, then the triangle inequality with a point from the intersection gives

$$\|\boldsymbol{x}_i - \boldsymbol{x}_j\| \leqslant \frac{2}{\beta \, N \, M} . \tag{292}$$

Therefore we have using the Cauchy-Schwarz inequality:

$$\Delta_i \leqslant \boldsymbol{x}_i^T \, (\boldsymbol{x}_i - \boldsymbol{x}_j) \leqslant \|\boldsymbol{x}_i\| \, \|\boldsymbol{x}_i - \boldsymbol{x}_j\| \leqslant M \, \frac{2}{\beta \, N \, M} = \frac{2}{\beta \, N} . \tag{293}$$

The last inequality is a contraction to Eq. (291) if we assume that

$$1 < 2 \, (N - 1) \, N \, \beta \, M^2 . \tag{294}$$

With this assumption, the spheres $\mathrm{S}_i$ and $\mathrm{S}_j$ do not intersect. Therefore each $\boldsymbol{x}_i$ has its separate fixed point in $\mathrm{S}_i$. We define

$$\Delta_{\min} = \min_{1 \leqslant i \leqslant N} \Delta_i \tag{295}$$

to obtain the master inequality

$$\Delta_{\min} \geqslant \frac{2}{\beta \, N} + \frac{1}{\beta} \, \ln \left( 2 \, N^2 \, \beta \, M^2 \right) . \tag{296}$$

**Patterns on a sphere.** For simplicity and in accordance with the results of the classical Hopfield network, we assume all *patterns being on a sphere* with radius $M$:

$$\forall_i : \|\boldsymbol{x}_i\| = M . \tag{297}$$

Under assumption Eq. (294) we have only to show that the master inequality Eq. (296) is fulfilled for each $\boldsymbol{x}_i$ to have a separate fixed point near each $\boldsymbol{x}_i$.

We defined $\alpha_{ij}$ as the angle between $\boldsymbol{x}_i$ and $\boldsymbol{x}_j$. The minimal angle $\alpha_{\min}$ between two data points is

$$\alpha_{\min} = \min_{1 \leqslant i < j \leqslant N} \alpha_{ij} . \tag{298}$$

On the sphere with radius $M$ we have

$$\Delta_{\min} = \min_{1 \leqslant i < j \leqslant N} M^2 (1 - \cos(\alpha_{ij})) = M^2 (1 - \cos(\alpha_{\min})) , \tag{299}$$

therefore it is sufficient to show the master inequality on the sphere:

$$M^2 (1 - \cos(\alpha_{\min})) \geqslant \frac{2}{\beta \, N} + \frac{1}{\beta} \, \ln \left( 2 \, N^2 \, \beta \, M^2 \right) . \tag{300}$$

Under assumption Eq. (294) we have only to show that the master inequality Eq. (296) is fulfilled for $\Delta_{\min}$. We consider patterns on the sphere, therefore the master inequality Eq. (296) becomes Eq. (300). First we show results when pattern positions on the sphere are constructed and $\Delta_{\min}$ is ensured. Then we move on to random patterns on a sphere, where $\Delta_{\min}$ becomes a random variable.

**Storage Capacity for Patterns Placed on the Sphere.** Next theorem says how many patterns we can stored (fixed point with attraction basin near pattern) if we are allowed to place them on the sphere.

**Theorem B3** (Storage Capacity (M=2): Placed Patterns). *We assume $\beta = 1$ and patterns on the sphere with radius $M$. If $M = 2\sqrt{d-1}$ and the dimension $d$ of the space is $d \geq 4$ or if $M = 1.7\sqrt{d-1}$ and the dimension $d$ of the space is $d \geq 50$, then the number of patterns $N$ that can be stored (fixed point with attraction basin near pattern) is at least*

$$N = 2^{2(d-1)} .\tag{301}$$

*Proof.* For random patterns on the sphere, we have to show that the master inequality Eq. (300) holds:

$$M^2(1 - \cos(\alpha_{\min})) \geq \frac{2}{\beta\,N} + \frac{1}{\beta} \ln\left(2\,N^2\,\beta\,M^2\right) .\tag{302}$$

We now place the patterns equidistant on the sphere where the pattern are separated by an angle $\alpha_{\min}$:

$$\forall i :\ \min_{j,j\neq i} \alpha_{ij} = \alpha_{\min} ,\tag{303}$$

In a $d$-dimensional space we can place

$$N = \left(\frac{2\pi}{\alpha_{\min}}\right)^{d-1}\tag{304}$$

points on the sphere. In a spherical coordinate system a pattern differs from its most closest patterns by an angle $\alpha_{\min}$ and there are $d-1$ angles. Solving for $\alpha_{\min}$ gives

$$\alpha_{\min} = \frac{2\pi}{N^{1/(d-1)}} .\tag{305}$$

The number of patterns that can be stored is determined by the largest $N$ that fulfils

$$M^2 \left(1 - \cos\left(\frac{2\pi}{N^{1/(d-1)}}\right)\right) \geq \frac{2}{\beta\,N} + \frac{1}{\beta} \ln\left(2\,N^2\,\beta\,M^2\right) .\tag{306}$$

We set $N = 2^{2(d-1)}$ and obtain for Eq. (306):

$$M^2 \left(1 - \cos\left(\frac{\pi}{2}\right)\right) \geq \frac{2}{\beta\,2^{3(d-1)}} + \frac{1}{\beta} \ln\left(2\,\beta\,M^2\right) + \frac{1}{\beta}\,4\,(d-1)\ln 2 .\tag{307}$$

This inequality is equivalent to

$$\beta\,M^2 \geq \frac{1}{2^{2(d-1)-1}} + \ln\left(2\,\beta\,M^2\right) + 4\,(d-1)\ln 2 .\tag{308}$$

The last inequality can be fulfilled with $M = K\sqrt{d-1}$ and proper $K$. For $\beta = 1$, $d = 4$ and $K = 2$ the inequality is fulfilled. The left hand side minus the right hand side is $4(d-1) - 1/2^{2(d-1)-1} - \ln(8(d-1)) - 4(d-1)\ln 2$. Its derivative with respect to $d$ is strict positive. Therefore the inequality holds for $d \geq 4$.

For $\beta = 1$, $d = 50$ and $K = 1.7$ the inequality is fulfilled. The left hand side minus the right hand side is $2.89(d-1) - 1/2^{2(d-1)-1} - \ln(5.78(d-1)) - 4(d-1)\ln 2$. Its derivative with respect to $d$ is strict positive. Therefore the inequality holds for $d \geq 50$.

$\square$

If we want to store considerably more patterns, then we have to increase the length of the vectors or the dimension of the space where the vectors live. The next theorem shows results for the number of patterns $N$ with $N = 2^{3(d-1)}$.

**Theorem B4** (Storage Capacity (M=5): Placed Patterns). *We assume $\beta = 1$ and patterns on the sphere with radius $M$. If $M = 5\sqrt{d-1}$ and the dimension $d$ of the space is $d \geq 3$ or if $M = 4\sqrt{d-1}$ and the dimension $d$ of the space is $d \geq 13$, then the number of patterns $N$ that can be stored (fixed point with attraction basin near pattern) is at least*

$$N = 2^{3(d-1)} .\tag{309}$$

*Proof.* We set $N = 2^{3(d-1)}$ and obtain for Eq. (306):

$$M^2 \left(1 - \cos\left(\frac{\pi}{4}\right)\right) \geq \frac{2}{\beta \, 2^{3(d-1)}} + \frac{1}{\beta} \ln\left(2 \, \beta \, M^2\right) + \frac{1}{\beta} \, 6 \, (d-1) \ln 2 \,. \qquad (310)$$

This inequality is equivalent to

$$\beta \, M^2 \left(1 - \frac{\sqrt{2}}{2}\right) \geq \frac{1}{2^{3(d-1)-1}} + \ln\left(2 \, \beta \, M^2\right) + 6 \, (d-1) \ln 2 \,. \qquad (311)$$

The last inequality can be fulfilled with $M = K\sqrt{d-1}$ and proper $K$. For $\beta = 1$, $d = 13$ and $K = 4$ the inequality is fulfilled. The left hand side minus the right hand side is $4.686292(d-1) - 1/2^{3(d-1)-1} - \ln(32(d-1)) - 6(d-1)\ln 2$. Its derivative with respect to $d$ is strict positive. Therefore the inequality holds for $d \geq 13$.

For $\beta = 1$, $d = 3$ and $K = 5$ the inequality is fulfilled. The left hand side minus the right hand side is $7.32233(d-1) - 1/2^{3(d-1)-1} - \ln(50(d-1)) - 6(d-1)\ln 2$. Its derivative with respect to $d$ is strict positive. Therefore the inequality holds for $d \geq 3$.

$\square$

**Storage Capacity for Random Patterns on the Sphere.** Next we investigate random points on the sphere. Under assumption Eq. (294) we have to show that the master inequality Eq. (300) is fulfilled for $\alpha_{\min}$, where now $\alpha_{\min}$ is now a random variable. We use results on the distribution of the minimal angles between random patterns on a sphere according to [12] and [10]. Theorem 2 in [12] gives the distribution of the minimal angle for random patterns on the unit sphere. Proposition 3.5 in [10] gives a lower bound on the probability of the minimal angle being larger than a given constant. We require this proposition to derive the probability of pattern having a minimal angle $\alpha_{\min}$. Proposition 3.6 in[10] gives the expectation of the minimal angle.

We will prove high probability bounds for the expected storage capacity. We need the following tail-bound on $\alpha_{\min}$ (the minimal angle of random patterns on a sphere):

**Lemma 13** ([10])**.** *Let $d$ be the dimension of the pattern space,*

$$\kappa_d := \frac{1}{d\,\sqrt{\pi}} \, \frac{\Gamma((d+1)/2)}{\Gamma(d/2)} \,. \qquad (312)$$

*and $\delta > 0$ such that $\frac{\kappa_{d-1}}{2}\delta^{(d-1)} \leqslant 1$. Then*

$$\Pr(N^{\frac{2}{d-1}}\alpha_{\min} \geq \delta) \geq 1 - \frac{\kappa_{d-1}}{2} \, \delta^{d-1} \,. \qquad (313)$$

*Proof.* The statement of the lemma is Eq. (3-6) from Proposition 3.5 in [10]. $\square$

Next we derive upper and lower bounds on the constant $\kappa_d$ since we require them later for proving storage capacity bounds.

**Lemma 14.** *For $\kappa_d$ defined in Eq. (312) we have the following bounds for every $d \geq 1$:*

$$\frac{1}{\exp(1/6)\,\sqrt{e\,\pi\,d}} \leqslant \kappa_d \leqslant \frac{\exp(1/12)}{\sqrt{2\,\pi\,d}} < 1 \,. \qquad (314)$$

*Proof.* We use for $x > 0$ the following bound related to Stirling's approximation formula for the gamma function, c.f. [35, (5.6.1)]:

$$1 < \Gamma(x)\,(2\,\pi)^{-\frac{1}{2}} x^{\frac{1}{2}-x}\exp(x) < \exp\left(\frac{1}{12\,x}\right) \,. \qquad (315)$$

Using Stirling's formula Eq. (315), we upper bound $\kappa_d$:

$$\kappa_d = \frac{1}{d\,\sqrt{\pi}}\,\frac{\Gamma((d+1)/2)}{\Gamma(d/2)} < \frac{1}{d\,\sqrt{\pi}}\,\frac{\exp\left(\frac{1}{6(d+1)}\right)\exp\left(-\frac{d+1}{2}\right)\left(\frac{d+1}{2}\right)^{\frac{d}{2}}}{\exp\left(-\frac{d}{2}\right)\left(\frac{d}{2}\right)^{\frac{d}{2}-\frac{1}{2}}} \qquad (316)$$

$$= \frac{1}{d\,\sqrt{\pi\,e}}\exp\left(\frac{1}{6(d+1)}\right)\left(1+\frac{1}{d}\right)^{\frac{d}{2}}\sqrt{\frac{d}{2}} \leqslant \frac{\exp\left(\frac{1}{12}\right)}{\sqrt{2\,\pi}\,\sqrt{d}} \,.$$

For the first inequality, we applied Eq. (315), while for the second we used $(1 + \frac{1}{d})^d < e$ for $d \geq 1$. Next, we lower bound $\kappa_d$ by again applying Stirling's formula Eq. (315):

$$\kappa_d = \frac{1}{d\sqrt{\pi}} \frac{\Gamma((d+1)/2)}{\Gamma(d/2)} > \frac{1}{d\sqrt{\pi}} \frac{\exp\left(-\frac{d+1}{2}\right) \left(\frac{d+1}{2}\right)^{\frac{d}{2}}}{\exp\left(\frac{1}{6d}\right) \exp\left(-\frac{d}{2}\right) \left(\frac{d}{2}\right)^{\frac{d}{2}-\frac{1}{2}}} \tag{317}$$

$$= \frac{1}{d\sqrt{\pi e} \exp\left(\frac{1}{6d}\right)} \left(1 + \frac{1}{d}\right)^{\frac{d}{2}} \sqrt{\frac{d}{2}} \geq \frac{1}{\exp\left(\frac{1}{6}\right) \sqrt{e\,\pi\,d}} \,,$$

where the last inequality holds because of monotonicity of $(1 + \frac{1}{d})^d$ and using the fact that for $d = 1$ it takes on the value 2. $\qquad\square$

We require a bound on $\cos$ to bound the master inequality Eq. (300).

**Lemma 15.** *For $0 \leqslant x \leqslant \pi$ the function $\cos$ can be upper bounded by:*

$$\cos(x) = 1 - \frac{x^2}{5} \,. \tag{318}$$

*Proof.* We use the infinite product representation of $\cos$ from [35, (4.22.2)]:

$$\cos(x) = \prod_{n=1}^{\infty} \left(1 - \frac{4\,x^2}{(2n-1)^2\,\pi^2}\right) \,. \tag{319}$$

It holds

$$1 - \frac{4\,x^2}{(2n-1)^2\,\pi^2} \leqslant 1 \tag{320}$$

for $|x| \leqslant \pi$ and $n \geq 2$, we can get the following upper bound on Eq. (319):

$$\cos(x) \leqslant \prod_{n=1}^{2} \left(1 - \frac{4\,x^2}{(2n-1)^2\pi^2}\right) = \left(1 - \frac{4\,x^2}{\pi^2}\right) \left(1 - \frac{4\,x^2}{9\,\pi^2}\right) \tag{321}$$

$$= 1 - \frac{40\,x^2}{9\,\pi^2} + \frac{16\,x^4}{9\,\pi^4} \leqslant 1 - \frac{40\,x^2}{9\,\pi^2} + \frac{16\,x^2}{9\,\pi^2}$$

$$= 1 - \frac{24\,x^2}{9\,\pi^2} \leqslant 1 - \frac{x^2}{5} \,.$$

The last but one inequality uses $x \leqslant \pi$, which implies $x/\pi \leqslant 1$. Thus Eq. (318) is proven.

$\qquad\square$

**Exponential storage capacity: the base $c$ as a function of the parameter $\beta$, the radius of the sphere $M$, the probability $p$, and the dimension $d$ of the space.** We express the number $N$ of stored patterns by an exponential function with base $c > 1$ and an exponent linear in $d$. We derive constraints on he base $c$ as a function of $\beta$, the radius of the sphere $M$, the probability $p$ that all patterns can be stored, and the dimension $d$ of the space. With $\beta > 0$, $K > 0$, and $d \geq 2$ (to ensure a sphere), the following theorem gives our main result.

**Theorem B5** (Storage Capacity (Main): Random Patterns)**.** *We assume a failure probability $0 < p \leqslant 1$ and randomly chosen patterns on the sphere with radius $M = K\sqrt{d-1}$. We define*

$$a := \frac{2}{d-1}\left(1 + \ln(2\,\beta\,K^2\,p\,(d-1))\right), \quad b := \frac{2\,K^2\,\beta}{5}\,,$$

$$c = \frac{b}{W_0(\exp(a + \ln(b)))}\,, \tag{322}$$

*where $W_0$ is the upper branch of the Lambert W function and ensure*

$$c \geq \left(\frac{2}{\sqrt{p}}\right)^{\frac{4}{d-1}} \,. \tag{323}$$

*Then with probability $1 - p$, the number of random patterns that can be stored is*

$$N \geq \sqrt{p}\, c^{\frac{d-1}{4}} \,. \tag{324}$$

*Examples are $c \geq 3.1546$ for $\beta = 1$, $K = 3$, $d = 20$ and $p = 0.001$ ($a + \ln(b) > 1.27$) and $c \geq 1.3718$ for $\beta = 1$ $K = 1$, $d = 75$, and $p = 0.001$ ($a + \ln(b) < -0.94$).*

*Proof.* We consider the probability that the master inequality Eq. (300) is fulfilled:

$$\Pr\left(M^2(1 - \cos(\alpha_{\min}))) \geq \frac{2}{\beta\,N} + \frac{1}{\beta}\,\ln\left(2\,N^2\,\beta\,M^2\right)\right) \geq 1 - p\,. \tag{325}$$

Using Eq. (318), we have:

$$1 - \cos(\alpha_{\min}) \geq \frac{1}{5}\,\alpha_{\min}^2\,. \tag{326}$$

Therefore with probability $1 - p$ the storage capacity is largest $N$ that fulfills

$$\Pr\left(M^2\frac{\alpha_{min}^2}{5} \geq \frac{2}{\beta\,N} + \frac{1}{\beta}\,\ln\left(2\,N^2\,\beta\,M^2\right)\right) \geq 1 - p\,. \tag{327}$$

This inequality is equivalent to

$$\Pr\left(N^{\frac{2}{d-1}}\,\alpha_{min} \geq \frac{\sqrt{5}\,N^{\frac{2}{d-1}}}{M}\left(\frac{2}{\beta\,N} + \frac{1}{\beta}\,\ln\left(2\,N^2\,\beta\,M^2\right)\right)^{\frac{1}{2}}\right) \geq 1 - p\,. \tag{328}$$

We use Eq. (313) to obtain:

$$\Pr\left(N^{\frac{2}{d-1}}\,\alpha_{min} \geq \frac{\sqrt{5}\,N^{\frac{2}{d-1}}}{M}\left(\frac{2}{\beta\,N} + \frac{1}{\beta}\,\ln\left(2\,N^2\,\beta\,M^2\right)\right)^{\frac{1}{2}}\right) \tag{329}$$

$$\geq 1 - \frac{\kappa_{d-1}}{2}\,5^{\frac{d-1}{2}}\,N^2\,M^{-(d-1)}\left(\frac{2}{\beta\,N} + \frac{1}{\beta}\,\ln\left(2\,N^2\,\beta\,M^2\right)\right)^{\frac{d-1}{2}}\,.$$

For Eq. (328) to be fulfilled, it is sufficient that

$$\frac{\kappa_{d-1}}{2}\,5^{\frac{d-1}{2}}\,N^2\,M^{-(d-1)}\left(\frac{2}{\beta\,N} + \frac{1}{\beta}\,\ln\left(2\,N^2\,\beta M^2\right)\right)^{\frac{d-1}{2}} - p \leq 0\,. \tag{330}$$

If we insert the assumption Eq. (323) of the theorem into Eq. (324), then we obtain $N \geq 2$. We now apply the upper bound $\kappa_{d-1}/2 < \kappa_{d-1} < 1$ from Eq. (314) and the upper bound $\frac{2}{\beta N} \leq \frac{1}{\beta}$ from $N \geq 2$ to inequality Eq. (330). In the resulting inequality we insert $N = \sqrt{p}c^{\frac{d-1}{4}}$ to check whether it is fulfilled with this special value of $N$ and obtain:

$$5^{\frac{d-1}{2}}\,p\,c^{\frac{d-1}{2}}\,M^{-(d-1)}\left(\frac{1}{\beta} + \frac{1}{\beta}\,\ln\left(2\,p\,c^{\frac{d-1}{2}}\,\beta M^2\right)\right)^{\frac{d-1}{2}} \leq p\,. \tag{331}$$

Dividing by $p$, inserting $M = K\sqrt{d-1}$, and exponentiation of the left and right side by $\frac{2}{d-1}$ gives:

$$\frac{5\,c}{K^2\,(d-1)}\left(\frac{1}{\beta} + \frac{1}{\beta}\,\ln\left(2\,\beta\,c^{\frac{d-1}{2}}\,p\,K^2\,(d-1)\right)\right) - 1 \leq 0\,. \tag{332}$$

After some algebraic manipulation, this inequality can be written as

$$a\,c + c\,\ln(c) - b \leq 0\,, \tag{333}$$

where we used

$$a := \frac{2}{d-1}\left(1 + \ln(2\,\beta\,K^2\,p\,(d-1))\right)\,, \quad b := \frac{2\,K^2\,\beta}{5}\,.$$

We determine the value $\hat{c}$ of $c$ which makes the inequality Eq. (333) equal to zero. We solve

$$a\,\hat{c} + \hat{c}\,\ln(\hat{c}) - b = 0 \tag{334}$$

for $\hat{c}$:

$$a\,\hat{c} + \hat{c}\,\ln(\hat{c}) - b = 0 \tag{335}$$
$$\Leftrightarrow a + \ln(\hat{c}) = b/\hat{c}$$
$$\Leftrightarrow a + \ln(b) + \ln(\hat{c}/b) = b/\hat{c}$$
$$\Leftrightarrow b/\hat{c} + \ln(b/\hat{c}) = a + \ln(b)$$
$$\Leftrightarrow b/\hat{c}\,\exp(b/\hat{c}) = \exp(a + \ln(b))$$
$$\Leftrightarrow b/\hat{c} = W_0(\exp(a + \ln(b)))$$
$$\Leftrightarrow \hat{c} = \frac{b}{W_0(\exp(a + \ln(b)))}\,,$$

where $W_0$ is the upper branch of the Lambert $W$ function (see Def. B10). Hence, the solution is

$$\hat{c} \; = \; \frac{b}{W_0(\exp(a \; + \; \ln(b)))} \; . \tag{336}$$

The solution exist, since the Lambert function $W_0(x)$ is defined for $-1/e < x$ and we have $0 < \exp(a + \ln(b))$.

Since $\hat{c}$ fulfills inequality Eq. (333) and therefore also Eq. (331), we have a lower bound on the storage capacity $N$:

$$N \; \geqslant \; \sqrt{p}\,\hat{c}^{\frac{d-1}{4}} \; . \tag{337}$$

$\square$

Next we aim at a lower bound on $c$ which does not use the Lambert $W$ function. Therefore we upper bound $W_0(\exp(a + \ln(b))$ to obtain a lower bound on $c$, therefore, also a lower bound on the storage capacity $N$. The lower bound is given in the next corollary.

**Corollary 1.** *We assume a failure probability* $0 < p \leqslant 1$ *and randomly chosen patterns on the sphere with radius* $M = K\sqrt{d-1}$. *We define*

$$a \; := \; \frac{2}{d-1}\left(1 \; + \; \ln(2\,\beta\,K^2\,p\,(d-1))\right), \quad b \; := \; \frac{2\,K^2\,\beta}{5} \; .$$

*Using the omega constant* $\Omega \approx 0.56714329$ *we set*

$$c \; = \; \begin{cases} b\,\ln\left(\dfrac{\Omega\,\exp(a + \ln(b)) + 1}{\Omega\,(1 + \Omega)}\right)^{-1} & \text{for } a \; + \; \ln(b) \; \leqslant \; 0 \, , \\ b\,(a \; + \; \ln(b))^{-\frac{a \, + \, \ln(b)}{a \, + \, \ln(b) \, + \, 1}} & \text{for } a \; + \; \ln(b) \; > \; 0 \end{cases} \tag{338}$$

*and ensure*

$$c \; \geq \; \left(\frac{2}{\sqrt{p}}\right)^{\frac{4}{d-1}} \; . \tag{339}$$

*Then with probability* $1 - p$, *the number of random patterns that can be stored is*

$$N \; \geqslant \; \sqrt{p}\,c^{\frac{d-1}{4}} \; . \tag{340}$$

*Examples are* $c \geq 3.1444$ *for* $\beta = 1$, $K = 3$, $d = 20$ *and* $p = 0.001$ ($a + \ln(b) > 1.27$*) and* $c \geq 1.2585$ *for* $\beta = 1$ $K = 1$, $d = 75$, *and* $p = 0.001$ ($a + \ln(b) < -0.94$*).*

*Proof.* We lower bound the $c$ defined in Theorem B5. According to [26, Theorem 2.3] we have for any real $u$ and $y > \frac{1}{e}$:

$$W_0(\exp(u)) \; \leqslant \; \ln\left(\frac{\exp(u) \; + \; y}{1 \; + \; \ln(y)}\right) \; . \tag{341}$$

To upper bound $W_0(x)$ for $x \in [0, 1]$, we set

$$y \; = \; 1/W_0(1) \; = \; 1/\Omega \; = \; \exp\Omega \; = \; -1/\ln\Omega \; \approx \; 1.76322 \, , \tag{342}$$

where the Omega constant $\Omega$ is

$$\Omega \; = \; \left(\int_{-\infty}^{\infty}\frac{\mathrm{d}t}{\left(e^t \; - \; t\right)^2 \; + \; \pi^2}\right)^{-1} \; - \; 1 \; \approx \; 0.56714329 \, . \tag{343}$$

See for these equations the special values of the Lambert $W$ function in Lemma 31. We have the upper bound on $W_0$:

$$W_0(\exp(u)) \; \leqslant \; \ln\left(\frac{\exp(u) \; + \; 1/\Omega}{1 \; + \; \ln(1/\Omega)}\right) \; = \; \ln\left(\frac{\Omega\,\exp(u) \; + \; 1}{\Omega(1 \; + \; \Omega)}\right) \; . \tag{344}$$

At the right hand side of interval $[0, 1]$, we have $u = 0$ and $\exp(u) = 1$ and get:

$$\ln\left(\frac{\Omega\, 1\, +\, 1}{\Omega(1\, +\, \Omega)}\right)\; =\; \ln\left(\frac{1}{\Omega}\right)\; =\; -\,\ln(\Omega)\; =\; \Omega\; =\; W_0(1)\,. \tag{345}$$

Therefore the bound is tight at the right hand side of of interval $[0, 1]$, that is for $\exp(u) = 1$, i.e. $u = 0$. We have derived an bound for $W_0(\exp(u))$ with $\exp(u) \in [0, 1]$ or, equivalently, $u \in [-\infty, 0]$. We obtain from [26, Corollary 2.6] the following bound on $W_0(\exp(u))$ for $1 < \exp(u)$, or, equivalently $0 < u$:

$$W_0(\exp(u))\; \leqslant\; u^{\frac{u}{1\, +\, u}}\,. \tag{346}$$

A lower bound on $\hat{c}$ is obtained via the upper bounds Eq. (346) and Eq. (344) on $W_0$ as $W_0 > 0$. We set $u = a + \ln(b)$ and obtain

$$W_0(\exp(a\, +\, \ln(b)))\; \leqslant\; \begin{cases} \ln\left(\frac{\Omega\,\exp(a\, +\,\ln(b))\, +\, 1}{\Omega\,(1\, +\,\Omega)}\right)^{-1} & \text{for }\, a\, +\,\ln(b)\, \leqslant\, 0\,, \\ (a\, +\,\ln(b))^{-\frac{a\, +\,\ln(b)}{a\, +\,\ln(b)\, +\, 1}} & \text{for }\, a\, +\,\ln(b)\, >\, 0 \end{cases} \tag{347}$$

We insert this bound into Eq. (336), the solution for $\hat{c}$, to obtain the statement of the theorem.

$\square$

**Exponential storage capacity: the dimension $d$ of the space as a function of the parameter $\beta$, the radius of the sphere $M$, and the probability $p$.** We express the number $N$ of stored patterns by an exponential function with base $c > 1$ and an exponent linear in $d$. We derive constraints on the dimension $d$ of the space as a function of $\beta$, the radius of the sphere $M$, the probability $p$ that all patterns can be stored, and the base of the exponential storage capacity. The following theorem gives this result.

**Theorem B6** (Storage Capacity (d computed): Random Patterns). *We assume a failure probability $0 < p \leqslant 1$ and randomly chosen patterns on the sphere with radius $M = K\sqrt{d-1}$. We define*

$$a\; :=\; \frac{\ln(c)}{2}\; -\; \frac{K^2\,\beta}{5\,c}\,, \quad b\; :=\; 1\, +\,\ln\left(2\,p\,\beta\,K^2\right)\,,$$

$$d\; =\; \begin{cases} 1\, +\,\frac{1}{a}\,W(a\,\exp(-b)) & \text{for }\, a \neq 0\,, \\ 1\, +\,\exp(-b) & \text{for }\, a = 0\,, \end{cases} \tag{348}$$

*where $W$ is the Lambert $W$ function. For $0 < a$ the function $W$ is the upper branch $W_0$ and for $a < 0$ we use the lower branch $W_{-1}$. If we ensure that*

$$c\, \geq\, \left(\frac{2}{\sqrt{p}}\right)^{\frac{4}{d-1}}\,, \quad -\frac{1}{e}\, \leqslant\, a\,\exp(-b)\,, \tag{349}$$

*then with probability $1 - p$, the number of random patterns that can be stored is*

$$N\, \geq\, \sqrt{p}\,c^{\frac{d-1}{4}}\,. \tag{350}$$

*Proof.* We consider the probability that the master inequality Eq. (300) is fulfilled:

$$\Pr\left(M^2(1\, -\,\cos(\alpha_{\min}))\, \geq\, \frac{2}{\beta\,N}\, +\,\frac{1}{\beta}\,\ln\left(2\,N^2\,\beta\,M^2\right)\right)\, \geq\, 1\, -\, p\,. \tag{351}$$

Using Eq. (318), we have:

$$1\, -\,\cos(\alpha_{\min})\, \geq\, \frac{1}{5}\,\alpha_{\min}^2\,. \tag{352}$$

Therefore with probability $1 - p$ the storage capacity is largest $N$ that fulfills

$$\Pr\left(M^2\frac{\alpha_{\min}^2}{5}\, \geq\, \frac{2}{\beta\,N}\, +\,\frac{1}{\beta}\,\ln\left(2\,N^2\,\beta\,M^2\right)\right)\, \geq\, 1\, -\, p\,. \tag{353}$$

This inequality is equivalent to

$$\Pr\left(N^{\frac{2}{d-1}}\,\alpha_{\min}\, \geq\, \frac{\sqrt{5}\,N^{\frac{2}{d-1}}}{M}\left(\frac{2}{\beta\,N}\, +\,\frac{1}{\beta}\,\ln\left(2\,N^2\,\beta\,M^2\right)\right)^{\frac{1}{2}}\right)\, \geq\, 1\, -\, p\,. \tag{354}$$

We use Eq. (313) to obtain:

$$\Pr\left(N^{\frac{2}{d-1}}\,\alpha_{min}\;\geq\;\frac{\sqrt{5}\,N^{\frac{2}{d-1}}}{M}\,\left(\frac{2}{\beta\,N}\;+\;\frac{1}{\beta}\,\ln\left(2\,N^2\,\beta\,M^2\right)\right)^{\frac{1}{2}}\right) \tag{355}$$

$$\geq\;1\;-\;\frac{\kappa_{d-1}}{2}\,5^{\frac{d-1}{2}}\,N^2\,M^{-(d-1)}\,\left(\frac{2}{\beta\,N}\;+\;\frac{1}{\beta}\,\ln\left(2\,N^2\,\beta\,M^2\right)\right)^{\frac{d-1}{2}}\,.$$

For Eq. (354) to be fulfilled, it is sufficient that

$$\frac{\kappa_{d-1}}{2}\,5^{\frac{d-1}{2}}\,N^2\,M^{-(d-1)}\,\left(\frac{2}{\beta\,N}\;+\;\frac{1}{\beta}\,\ln\left(2\,N^2\,\beta M^2\right)\right)^{\frac{d-1}{2}}\;-\;p\;\leqslant\;0\,. \tag{356}$$

If we insert the assumption Eq. (349) of the theorem into Eq. (350), then we obtain $N \geq 2$. We now apply the upper bound $\kappa_{d-1}/2 < \kappa_{d-1} < 1$ from Eq. (314) and the upper bound $\frac{2}{\beta N} \leqslant \frac{1}{\beta}$ from $N \geq 2$ to inequality Eq. (356). In the resulting inequality we insert $N = \sqrt{p}c^{\frac{d-1}{4}}$ to check whether it is fulfilled with this special value of $N$ and obtain:

$$5^{\frac{d-1}{2}}\,p\,c^{\frac{d-1}{2}}\,M^{-(d-1)}\,\left(\frac{1}{\beta}\;+\;\frac{1}{\beta}\,\ln\left(2\,p\,c^{\frac{d-1}{2}}\,\beta M^2\right)\right)^{\frac{d-1}{2}}\;\leqslant\;p\,. \tag{357}$$

Dividing by $p$, inserting $M = K\sqrt{d-1}$, and exponentiation of the left and right side by $\frac{2}{d-1}$ gives:

$$\frac{5\,c}{K^2\,(d-1)}\,\left(\frac{1}{\beta}\;+\;\frac{1}{\beta}\,\ln\left(2\,\beta\,c^{\frac{d-1}{2}}\,p\,K^2\,(d-1)\right)\right)\;-\;1\;\leqslant\;0\,. \tag{358}$$

This inequality Eq. (358) can be reformulated as:

$$1\;+\;\ln\left(2\,p\,\beta\,c^{\frac{d-1}{2}}\,K^2\,(d-1)\right)\;-\;\frac{(d-1)\,K^2\,\beta}{5\,c}\;\leqslant\;0\,. \tag{359}$$

Using

$$a\;:=\;\frac{\ln(c)}{2}\;-\;\frac{K^2\,\beta}{5\,c}\,,\quad b\;:=\;1\;+\;\ln\left(2\,p\,\beta\,K^2\right)\,,$$

$$\tag{360}$$

we write inequality Eq. (359) as

$$\ln(d-1)\;+\;a\,(d-1)\;+\;b\;\leqslant\;0\,. \tag{361}$$

We determine the value $\hat{d}$ of $d$ which makes the inequality Eq. (361) equal to zero. We solve

$$\ln(\hat{d}-1)\;+\;a\,(\hat{d}-1)\;+\;b\;=\;0\,. \tag{362}$$

for $\hat{d}$

For $a \neq 0$ we have

$$\ln(\hat{d}-1)\;+\;a\,(\hat{d}-1)\;+\;b\;=\;0 \tag{363}$$
$$\Leftrightarrow\;a\,(\hat{d}-1)\;+\;\ln(\hat{d}-1)\;=\;-b$$
$$\Leftrightarrow\;(\hat{d}-1)\exp(a\,(\hat{d}-1))\;=\;\exp(-b)$$
$$\Leftrightarrow\;a\,(\hat{d}-1)\exp(a\,(\hat{d}-1))\;=\;a\,\exp(-b)$$
$$\Leftrightarrow\;a\,(\hat{d}-1)\;=\;W(a\,\exp(-b))$$
$$\Leftrightarrow\;\hat{d}\;-\;1\;=\;\frac{1}{a}\,W(a\,\exp(-b))$$
$$\Leftrightarrow\;\hat{d}\;=\;1\;+\;\frac{1}{a}\,W(a\,\exp(-b))\,,$$

where $W$ is the Lambert $W$ function (see Def. B10). For $a > 0$ we have to use the upper branch $W_0$ of the Lambert $W$ function and for $a < 0$ we use the lower branch $W_{-1}$ of the Lambert $W$

function. We have to ensure that $-1/e \leqslant a \exp(-b)$ for a solution to exist. For $a = 0$ we have $\hat{d} = 1 + \exp(-b)$.
Hence, the solution is

$$\hat{d} \;=\; 1 \;+\; \frac{1}{a}\, W(a \exp(-b)) \,. \tag{364}$$

Since $\hat{d}$ fulfills inequality Eq. (358) and therefore also Eq. (357), we have a lower bound on the storage capacity $N$:

$$N \;\geqslant\; \sqrt{p}\,\hat{c}^{\frac{d-1}{4}} \,. \tag{365}$$

$\square$

**Corollary 2.** *We assume a failure probability $0 < p \leqslant 1$ and randomly chosen patterns on the sphere with radius $M = K\sqrt{d-1}$. We define*

$$a \;:=\; \frac{\ln(c)}{2} \;-\; \frac{K^2\,\beta}{5\,c} \,, \quad b \;:=\; 1 \;+\; \ln\left(2\,p\,\beta\,K^2\right) \,,$$
$$d \;=\; 1 \;+\; \frac{1}{a}\,\left(-\ln(-a) \;+\; b\right) \,, \tag{366}$$

*and ensure*

$$c \;\geqslant\; \left(\frac{2}{\sqrt{p}}\right)^{\frac{4}{d-1}} \,, \quad -\frac{1}{e} \;\leqslant\; a\,\exp(-b) \,, \quad a \;<\; 0 \,, \tag{367}$$

*then with probability $1 - p$, the number of random patterns that can be stored is*

$$N \;\geqslant\; \sqrt{p}\,c^{\frac{d-1}{4}} \,. \tag{368}$$

*Setting $\beta = 1$, $K = 3$, $c = 2$ and $p = 0.001$ yields $d < 24$.*

*Proof.* For $a < 0$ the Eq. (348) from Theorem (B6) can be written as

$$d \;=\; 1 \;+\; \frac{W_{-1}(a\exp(-b))}{a} \;=\; 1 \;+\; \frac{W_{-1}(-\exp\left(-(-\ln(-a) + b - 1) - 1\right))}{a} \tag{369}$$

From [2, Theorem 3.1] we get the following bound on $W_{-1}$:

$$-\frac{e}{e-1}\,(u+1) \;<\; W_{-1}(-\exp(-u-1)) \;<\; -(u+1) \,. \tag{370}$$

for $u > 0$. We apply Eq. (370) to Eq. (369) with $u = -\ln(-a) + b - 1$.
Since $a < 0$ we get

$$d \;>\; 1 \;+\; \frac{-\ln(-a) + b}{a} \,. \tag{371}$$

$\square$

**Storage capacity for the expected minimal separation instead of the probability that all patterns can be stored.**  In contrast to the previous paragraph, we want to argue about the storage capacity for the expected minimal separation. Therefore we will use the following bound on the expectation of $\alpha_{\min}$ (minimal angle), which gives also a bound on the expected of $\Delta_{\min}$ (minimal separation):

**Lemma 16** (Proposition 3.6 in [10]). *We have the following lower bound on the expectation of $\alpha_{\min}$:*

$$\mathrm{E}\left[N^{\frac{2}{d-1}}\,\alpha_{\min}\right] \;\geqslant\; \left(\frac{\Gamma(\frac{d}{2})}{2(d-1)\,\sqrt{\pi}\,\Gamma(\frac{d-1}{2})}\right)^{-\frac{1}{d-1}} \Gamma(1 + \frac{1}{d-1})\,\frac{d^{-\frac{1}{d-1}}}{\Gamma(2 + \frac{1}{d-1})} \;:=\; C_{d-1}. \tag{372}$$

*The bound is valid for all $N \geq 2$ and $d \geq 2$.*

Let us start with some preliminary estimates. First of all we need some asymptotics for the constant $C_{d-1}$ in Eq. (372):

**Lemma 17.** *The following estimate holds for $d \geq 2$:*

$$C_d \geq 1 - \frac{\ln(d+1)}{d} . \tag{373}$$

*Proof.* The recursion formula for the Gamma function is [35, (5.5.1)]:

$$\Gamma(x+1) = x\,\Gamma(x) . \tag{374}$$

We use Eq. (314) and the fact that $d^{\frac{1}{d}} \geq 1$ for $d \geq 1$ to obtain:

$$C_d \geq (2\sqrt{d})^{\frac{1}{d}}\Gamma(1+\frac{1}{d})\,\frac{(d+1)^{-\frac{1}{d}}}{\Gamma(2+\frac{1}{d})} = (2\sqrt{d})^{\frac{1}{d}}\frac{(d+1)^{-\frac{1}{d}}}{1-\frac{1}{d}} > (d+1)^{\frac{1}{d}} \tag{375}$$

$$= \exp(-\frac{1}{d}\ln(d+1)) \geq 1 - \frac{1}{d}\ln(d+1) ,$$

where in the last step we used the elementary inequality $\exp(x) \geq 1+x$, which follows from the mean value theorem. $\qquad\square$

The next theorem states the number of stored patterns for the expected minimal separation.

**Theorem B7** (Storage Capacity (expected separation): Random Patterns)**.** *We assume patterns on the sphere with radius $M = K\sqrt{d-1}$ that are randomly chosen. Then for all values $c \geq 1$ for which*

$$\frac{1}{5}\,(d-1)\,K^2\,c^{-1}(1 - \frac{\ln(d-1)}{(d-1)})^2 \geq \frac{2}{\beta\,c^{\frac{d-1}{4}}} + \frac{1}{\beta}\,\ln\left(2\,c^{\frac{d-1}{2}}\,\beta\,(d-1)\,K^2\right) \tag{376}$$

*holds, the number of stored patterns for the expected minimal separation is at least*

$$N = c^{\frac{d-1}{4}} . \tag{377}$$

*The inequality Eq. (376) is e.g. fulfilled with $\beta = 1$, $K = 3$, $c = 2$ and $d \geq 17$.*

*Proof.* Instead of considering the probability that the master inequality Eq. (300) is fulfilled we now consider whether this inequality is fulfilled for the expected minimal distance. We consider the expectation of the minimal distance $\Delta_{\min}$:

$$\mathrm{E}[\Delta_{\min}] = \mathrm{E}[M^2(1 - \cos(\alpha_{\min}))] = M^2(1 - \mathrm{E}[\cos(\alpha_{\min})]) . \tag{378}$$

For this expectation, the master inequality Eq. (300) becomes

$$M^2(1 - \mathrm{E}[\cos(\alpha_{\min})]) \geq \frac{2}{\beta\,N} + \frac{1}{\beta}\,\ln\left(2\,N^2\,\beta\,M^2\right) . \tag{379}$$

We want to find the largest $N$ that fulfills this inequality.
We apply Eq. (318) and Jensen's inequality to deduce the following lower bound:

$$1 - \mathrm{E}[\cos(\alpha_{\min})] \geq \frac{1}{5}\,\mathrm{E}\left[\alpha_{\min}^2\right] \geq \frac{1}{5}\,\mathrm{E}[\alpha_{\min}]^2 . \tag{380}$$

Now we use Eq. (372) and Eq. (373) to arrive at

$$\mathrm{E}[\alpha_{\min}]^2 \geq N^{-\frac{4}{d-1}}\,\mathrm{E}[N^{\frac{2}{d-1}}\,\alpha_{\min}]^2 \geq N^{-\frac{4}{d-1}}\,C_{d-1}^2 \geq N^{-\frac{4}{d-1}}\,(1 - \frac{\ln(d-1)}{(d-1)})^2 , \tag{381}$$

for sufficiently large $d$. Thus in order to fulfill Eq. (379), it is enough to find values that satisfy Eq. (376). $\qquad\square$

### B2.5.2 Convergence after One Update and Small Retrieval Error

**Theorem B8** (Convergence After One Update)**.** *With query $\boldsymbol{\xi}$, after one update the distance of the new point $f(\boldsymbol{\xi})$ to the fixed point $\boldsymbol{x}_i^*$ is exponentially small in the separation $\Delta_i$. The precise bounds are:*

$$\|f(\boldsymbol{\xi}) - \boldsymbol{x}_i^*\| \leq \|\mathrm{J}^m\|_2\,\|\boldsymbol{\xi} - \boldsymbol{x}_i^*\| , \tag{382}$$

$$\|\mathrm{J}^m\|_2 \leq 2\,\beta\,N\,M^2\,(N-1)\exp(-\,\beta\,(\Delta_i\,-\,2\,\max\{\|\boldsymbol{\xi}-\boldsymbol{x}_i\|, \|\boldsymbol{x}_i^*-\boldsymbol{x}_i\|\}\,M)) . \tag{383}$$

*Proof.* From Eq. (169) we have

$$\|\mathrm{J}^m\|_2 \;\leqslant\; 2\,\beta\,N\,M^2\,(N-1)\exp(-\,\beta\,(\Delta_i\;-\;2\,\max\{\|\boldsymbol{\xi}\;-\;\boldsymbol{x}_i\|,\|\boldsymbol{x}_i^*\;-\;\boldsymbol{x}_i\|\}\,M))\,. \tag{384}$$

After every iteration the mapped point $f(\boldsymbol{\xi})$ is closer to the fixed point $\boldsymbol{x}_i^*$ than the original point $\boldsymbol{x}_i$:

$$\|f(\boldsymbol{\xi})\;-\;\boldsymbol{x}_i^*\|\;\leqslant\;\|\mathrm{J}^m\|_2\,\|\boldsymbol{\xi}\;-\;\boldsymbol{x}_i^*\|\,. \tag{385}$$

$\square$

We want to estimate how large $\Delta_i$ is. For $\boldsymbol{x}_i$ we have:

$$\Delta_i\;=\;\min_{j,j\neq i}\left(\boldsymbol{x}_i^T\boldsymbol{x}_i\;-\;\boldsymbol{x}_i^T\boldsymbol{x}_j\right)\;=\;\boldsymbol{x}_i^T\boldsymbol{x}_i\;-\;\max_{j,j\neq i}\boldsymbol{x}_i^T\boldsymbol{x}_j\,. \tag{386}$$

To estimate how large $\Delta_i$ is, assume vectors $\boldsymbol{x}\in\mathbb{R}^d$ and $\boldsymbol{y}\in\mathbb{R}^d$ that have as components standard normally distributed values. The expected value of the separation of two points with normally distributed components is

$$\mathrm{E}\left[\boldsymbol{x}^T\boldsymbol{x}\;-\;\boldsymbol{x}^T\boldsymbol{y}\right]\;=\;\sum_{j=1}^{d}\mathrm{E}\left[x_j^2\right]\;+\;\sum_{j=1}^{d}\mathrm{E}\left[x_j\right]\sum_{j=1}^{d}\mathrm{E}\left[y_j\right]\;=\;d\,. \tag{387}$$

The variance of the separation of two points with normally distributed components is

$$\mathrm{Var}\left[\boldsymbol{x}^T\boldsymbol{x}\;-\;\boldsymbol{x}^T\boldsymbol{y}\right]\;=\;\mathrm{E}\left[\left(\boldsymbol{x}^T\boldsymbol{x}\;-\;\boldsymbol{x}^T\boldsymbol{y}\right)^2\right]\;-\;d^2 \tag{388}$$

$$=\;\sum_{j=1}^{d}\mathrm{E}\left[x_j^4\right]\;+\;\sum_{j=1,k=1,k\neq j}^{d}\mathrm{E}\left[x_j^2\right]\mathrm{E}\left[x_k^2\right]\;-\;2\sum_{j=1}^{d}\mathrm{E}\left[x_j^3\right]\mathrm{E}\left[y_j\right]\;-$$

$$2\sum_{j=1,k=1,k\neq j}^{d}\mathrm{E}\left[x_j^2\right]\mathrm{E}\left[x_k\right]\mathrm{E}\left[y_k\right]\;+\;\sum_{j=1}^{d}\mathrm{E}\left[x_j^2\right]\mathrm{E}\left[y_j^2\right]\;+$$

$$\sum_{j=1,k=1,k\neq j}^{d}\mathrm{E}\left[x_j\right]\mathrm{E}\left[y_j\right]\mathrm{E}\left[x_k\right]\mathrm{E}\left[y_k\right]\;-\;d^2$$

$$=\;3\,d\;+\;d\,(d-1)\;+\;d\;-\;d^2\;=\;3\,d\,.$$

The expected value for the separation of two random vectors gives:

$$\|\mathrm{J}^m\|_2\;\leqslant\;2\,\beta\,N\,M^2\,(N-1)\exp(-\,\beta\,(d\;-\;2\,\max\{\|\boldsymbol{\xi}\;-\;\boldsymbol{x}_i\|,\|\boldsymbol{x}_i^*\;-\;\boldsymbol{x}_i\|\}\,M))\,. \tag{389}$$

For the exponential storage we set $M=2\sqrt{d-1}$. We see the Lipschitz constant $\|\mathrm{J}^m\|_2$ decreases exponentially with the dimension. Therefore $\|f(\boldsymbol{\xi})\;-\;\boldsymbol{x}_i^*\|$ is exponentially small after just one update. Therefore the fixed point is well retrieved after one update.

The retrieval error decreases exponentially with the separation $\Delta_i$.

**Theorem B9** (Exponentially Small Retrieval Error). *The retrieval error $\|\boldsymbol{x}_i\;-\;\boldsymbol{x}_i^*\|$ of pattern $\boldsymbol{x}_i$ is bounded by*

$$\|\boldsymbol{x}_i\;-\;\boldsymbol{x}_i^*\|\;\leqslant\;2\,(N-1)\;\exp(-\,\beta\,(\Delta_i\;-\;2\,\|\boldsymbol{x}_i^*\;-\;\boldsymbol{x}_i\|\,M))\,M \tag{390}$$

*and for $\|\boldsymbol{x}_i-\boldsymbol{x}_i^*\|\leqslant\frac{1}{2\,\beta\,M}$ by*

$$\|\boldsymbol{x}_i\;-\;\boldsymbol{x}_i^*\|\;\leqslant\;e\,(N-1)\,M\;\exp(-\,\beta\,\Delta_i)\,. \tag{391}$$

*Proof.* We compute the retrieval error which is just $\|\boldsymbol{x}_i\;-\;\boldsymbol{x}_i^*\|$. From Lemma 4 we have

$$\|\boldsymbol{x}_i\;-\;f(\boldsymbol{\xi})\|\;\leqslant\;2\,\epsilon\,M\,, \tag{392}$$

From Eq. (168) we have

$$\epsilon\;=\;(N-1)\exp(-\,\beta\,(\Delta_i\;-\;2\,\max\{\|\boldsymbol{\xi}\;-\;\boldsymbol{x}_i\|,\|\boldsymbol{x}_i^*\;-\;\boldsymbol{x}_i\|\}\,M))\,. \tag{393}$$

We use $\boldsymbol{\xi}=\boldsymbol{x}_i^*$ and get

$$\epsilon\;=\;(N-1)\exp(-\,\beta\,(\Delta_i\;-\;2\,\|\boldsymbol{x}_i^*\;-\;\boldsymbol{x}_i\|\,M))\,. \tag{394}$$

We obtain

$$\|\boldsymbol{x}_i\;-\;\boldsymbol{x}_i^*\|\;\leqslant\;2\,(N-1)\;\exp(-\,\beta\,(\Delta_i\;-\;2\,\|\boldsymbol{x}_i^*\;-\;\boldsymbol{x}_i\|\,M))\,M\,. \tag{395}$$

For $\|\boldsymbol{x}_i-\boldsymbol{x}_i^*\|\leqslant\frac{1}{2\,\beta\,M}$ inequality Eq. (395) gives

$$\|\boldsymbol{x}_i\;-\;\boldsymbol{x}_i^*\|\;\leqslant\;e\,(N-1)\,M\;\exp(-\,\beta\,\Delta_i)\,. \tag{396}$$

$\square$

### B2.6 Learning Associations

#### B2.6.1 Initialization: Random Matrix Theory

For the initial matrices and scaling, the random matrix theory is of interest. For matrix entries with variance $\sigma^2$ we know from the circular law [43] and the Marchenko-Pastur quarter circular law [31, 50, 8] that $1/(\sigma 2\sqrt{N})\boldsymbol{X}$ has a singular value density concentrated at values smaller than one. The maximal singular value of $\boldsymbol{X} \in \mathbb{R}^{N \times n}$ is $s_{\max}(\boldsymbol{X}) \propto \sqrt{N} + \sqrt{n}$ [38]. Furthermore large singular values have lower density according the quarter circular law. Initialization of mappings to the space, where the modern Hopfield networks works, can be based on the largest singular value. Therefore we can estimate the largest possible norm $M$ of the patterns as we used in the theory.

#### B2.6.2 Directly Learning Associations

In the first setting, $\boldsymbol{x}$ is mapped by $\boldsymbol{W}\boldsymbol{x}$ to the query space, where the query $\boldsymbol{\xi}$ lives. With and the largest norm of a pattern

$$M_W = \max_i \left\| \boldsymbol{W}^T \boldsymbol{x}_i \right\|, \tag{397}$$

the energy function E is now

$$\mathrm{E} = -\operatorname{lse}(\beta, \boldsymbol{X}^T \boldsymbol{W}^T \boldsymbol{\xi}) + \frac{1}{2}\boldsymbol{\xi}^T\boldsymbol{\xi} + \beta^{-1}\ln N + \frac{1}{2}M_W^2 \tag{398}$$

$$= -\beta^{-1}\ln\left(\sum_{i=1}^N \exp(\beta \boldsymbol{x}_i^T \boldsymbol{W}^T \boldsymbol{\xi})\right) + \frac{1}{2}\boldsymbol{\xi}^T\boldsymbol{\xi} + \beta^{-1}\ln N + \frac{1}{2}M_W^2 . \tag{399}$$

The derivative of the energy E with respect to $\boldsymbol{\xi}$ is

$$\frac{\partial \mathrm{E}}{\partial \boldsymbol{\xi}} = -\boldsymbol{W}\boldsymbol{X}\operatorname{softmax}(\beta \boldsymbol{X}^T \boldsymbol{W}^T \boldsymbol{\xi}) + \boldsymbol{\xi} = -\boldsymbol{W}\boldsymbol{X}\boldsymbol{p} + \boldsymbol{\xi}, \tag{400}$$

where we used

$$\boldsymbol{p} = \operatorname{softmax}(\beta \boldsymbol{X}^T \boldsymbol{W}^T \boldsymbol{\xi}) . \tag{401}$$

The gradient update rule gives

$$\boldsymbol{\xi}^{\mathrm{new}} = \boldsymbol{W}\boldsymbol{X}\boldsymbol{p} = \boldsymbol{\xi} - \frac{\partial \mathrm{E}}{\partial \boldsymbol{\xi}} . \tag{402}$$

We consider the query $\boldsymbol{\xi}$ with result $\boldsymbol{y}$:

$$\boldsymbol{y} = \boldsymbol{W}\boldsymbol{X}\boldsymbol{p} = \boldsymbol{W}\boldsymbol{X}\operatorname{softmax}(\beta \boldsymbol{X}^T \boldsymbol{W}^T \boldsymbol{\xi}) \tag{403}$$

Since the retrieved vector $\boldsymbol{y}$ is mapped by a weight matrix $\boldsymbol{V}$ to another vector, we consider the simplified update rule:

$$\boldsymbol{y} = \boldsymbol{X}\boldsymbol{p} = \boldsymbol{X}\operatorname{softmax}(\beta \boldsymbol{X}^T \boldsymbol{W}^T \boldsymbol{\xi}) \tag{404}$$

The derivative with respect to $\boldsymbol{W}$ is

$$\frac{\partial \boldsymbol{a}^T \boldsymbol{y}}{\partial \boldsymbol{W}} = \frac{\partial \boldsymbol{y}}{\partial \boldsymbol{W}}\frac{\partial \boldsymbol{a}^T \boldsymbol{y}}{\partial \boldsymbol{y}} = \frac{\partial \boldsymbol{y}}{\partial (\boldsymbol{W}^T \boldsymbol{\xi})}\frac{\partial (\boldsymbol{W}^T \boldsymbol{\xi})}{\partial \boldsymbol{W}}\frac{\partial \boldsymbol{a}^T \boldsymbol{y}}{\partial \boldsymbol{y}} . \tag{405}$$

$$\frac{\partial \boldsymbol{y}}{\partial (\boldsymbol{W}^T \boldsymbol{\xi})} = \beta \, \boldsymbol{X} \left(\operatorname{diag}(\boldsymbol{p}) - \boldsymbol{p}\boldsymbol{p}^T\right) \boldsymbol{X}^T \tag{406}$$

$$\frac{\partial \boldsymbol{a}^T \boldsymbol{y}}{\partial \boldsymbol{y}} = \boldsymbol{a} . \tag{407}$$

We have the product of the 3-dimensional tensor $\frac{\partial (\boldsymbol{W}^T \boldsymbol{\xi})}{\partial \boldsymbol{W}}$ with the vector $\boldsymbol{a}$ which gives a 2-dimensional tensor, i.e. a matrix:

$$\frac{\partial (\boldsymbol{W}^T \boldsymbol{\xi})}{\partial \boldsymbol{W}}\frac{\partial \boldsymbol{a}^T \boldsymbol{y}}{\partial \boldsymbol{y}} = \frac{\partial (\boldsymbol{W}^T \boldsymbol{\xi})}{\partial \boldsymbol{W}}\boldsymbol{a} = \boldsymbol{\xi}^T \boldsymbol{a}\boldsymbol{I} . \tag{408}$$

$$\frac{\partial \boldsymbol{a}^T \boldsymbol{y}}{\partial \boldsymbol{W}} = \beta \, \boldsymbol{X} \left(\operatorname{diag}(\boldsymbol{p}) - \boldsymbol{p}\boldsymbol{p}^T\right) \boldsymbol{X}^T \left(\boldsymbol{\xi}^T \boldsymbol{a}\right) . \tag{409}$$

### B2.6.3 Learning the Mappings to the Association Space

We consider the patterns $\boldsymbol{x}$ that are mapped to $\tilde{\boldsymbol{x}}$ in the association space $\mathbb{R}^d$ by $\tilde{\boldsymbol{x}} = \boldsymbol{W}^K \boldsymbol{x}$. The query $\boldsymbol{\xi}$ is mapped to $\tilde{\boldsymbol{\xi}}$ in the space $\mathbb{R}^d$ by $\tilde{\boldsymbol{\xi}} = \boldsymbol{W}^Q \boldsymbol{\xi}$, too.

With and the largest norm of a pattern

$$M_W = \max_i \left\| \boldsymbol{W}^K \boldsymbol{x}_i \right\|, \tag{410}$$

the energy function E with mappings $\boldsymbol{W}^K$ and $\boldsymbol{W}^Q$ is

$$\mathrm{E} = -\operatorname{lse}(\beta, \boldsymbol{X}^T (\boldsymbol{W}^K)^T \boldsymbol{W}^Q \boldsymbol{\xi}) + \frac{1}{2} \boldsymbol{\xi}^T (\boldsymbol{W}^Q)^T \boldsymbol{W}^Q \boldsymbol{\xi} \tag{411}$$

$$+ \beta^{-1} \ln N + \frac{1}{2} M_W^2$$

$$= -\beta^{-1} \ln \left( \sum_{i=1}^N \exp(\beta \boldsymbol{x}_i^T (\boldsymbol{W}^K)^T \boldsymbol{W}^Q \boldsymbol{\xi}) \right) + \frac{1}{2} \boldsymbol{\xi}^T (\boldsymbol{W}^Q)^T \boldsymbol{W}^Q \boldsymbol{\xi}$$

$$+ \beta^{-1} \ln N + \frac{1}{2} M_W^2.$$

In the association space that is

$$\mathrm{E} = -\operatorname{lse}(\beta, \tilde{\boldsymbol{X}}^T \tilde{\boldsymbol{\xi}}) + \frac{1}{2} \tilde{\boldsymbol{\xi}}^T \tilde{\boldsymbol{\xi}} + \beta^{-1} \ln N + \frac{1}{2} \tilde{\boldsymbol{x}}_{\max}^T \tilde{\boldsymbol{x}}_{\max} \tag{412}$$

$$= -\beta^{-1} \ln \left( \sum_{i=1}^N \exp(\beta \tilde{\boldsymbol{x}}_i^T \tilde{\boldsymbol{\xi}}) \right) + \frac{1}{2} \tilde{\boldsymbol{\xi}}^T \tilde{\boldsymbol{\xi}} + \beta^{-1} \ln N + \frac{1}{2} \tilde{\boldsymbol{x}}_{\max}^T \tilde{\boldsymbol{x}}_{\max}. \tag{413}$$

The derivative of the energy E with respect to $\tilde{\boldsymbol{\xi}}$ is

$$\frac{\partial \mathrm{E}}{\partial \tilde{\boldsymbol{\xi}}} = -\tilde{\boldsymbol{X}} \operatorname{softmax}(\beta \tilde{\boldsymbol{X}}^T \tilde{\boldsymbol{\xi}}) + \tilde{\boldsymbol{\xi}} = -\tilde{\boldsymbol{X}} \boldsymbol{p} + \tilde{\boldsymbol{\xi}}, \tag{414}$$

where we used

$$\boldsymbol{p} = \operatorname{softmax}(\beta \tilde{\boldsymbol{X}}^T \tilde{\boldsymbol{\xi}}). \tag{415}$$

The gradient update rule gives

$$\tilde{\boldsymbol{\xi}}^{\mathrm{new}} = \tilde{\boldsymbol{X}} \boldsymbol{p} = \tilde{\boldsymbol{\xi}} - \frac{\partial \mathrm{E}}{\partial \tilde{\boldsymbol{\xi}}}. \tag{416}$$

We consider the query $\boldsymbol{\xi}$ that is mapped to $\tilde{\boldsymbol{\xi}}$ to obtain $\tilde{\boldsymbol{\xi}}^{\mathrm{new}}$:

$$\tilde{\boldsymbol{\xi}}^{\mathrm{new}} = \boldsymbol{W}^Q \boldsymbol{\xi}^{\mathrm{new}} = \boldsymbol{W}^K \boldsymbol{X} \boldsymbol{p} = \boldsymbol{W}^K \boldsymbol{X} \operatorname{softmax}(\beta \boldsymbol{X}^T (\boldsymbol{W}^K)^T \boldsymbol{W}^Q \boldsymbol{\xi}). \tag{417}$$

Since the retrieved vector is mapped by a weight matrix $\boldsymbol{V}$ to another vector, we consider the simplified update rule. The retrieved vector is now $\boldsymbol{y}$ given by

$$\boldsymbol{y} = \boldsymbol{X} \boldsymbol{p} = \boldsymbol{X} \operatorname{softmax}(\beta \boldsymbol{X}^T (\boldsymbol{W}^K)^T \boldsymbol{W}^Q \boldsymbol{\xi}). \tag{418}$$

The vector $\boldsymbol{y}$ does not live in the association space but in the pattern space of $\boldsymbol{x}$. Only $\boldsymbol{W}^K$ would map it to the association space.

The derivative with respect to $\boldsymbol{W}^Q$ is

$$\frac{\partial \boldsymbol{a}^T \boldsymbol{y}}{\partial \boldsymbol{W}^Q} = \frac{\partial \boldsymbol{y}}{\partial \boldsymbol{W}^Q} \frac{\partial \boldsymbol{a}^T \boldsymbol{y}}{\partial \boldsymbol{y}} = \frac{\partial \boldsymbol{y}}{\partial (\boldsymbol{W}^Q \boldsymbol{\xi})} \frac{\partial (\boldsymbol{W}^Q \boldsymbol{\xi})}{\partial \boldsymbol{W}^Q} \frac{\partial \boldsymbol{a}^T \boldsymbol{y}}{\partial \boldsymbol{y}}. \tag{419}$$

$$\frac{\partial \boldsymbol{y}}{\partial (\boldsymbol{W}^Q \boldsymbol{\xi})} = \beta \boldsymbol{X} \left( \operatorname{diag}(\boldsymbol{p}) - \boldsymbol{p} \boldsymbol{p}^T \right) \boldsymbol{X}^T (\boldsymbol{W}^K)^T \tag{420}$$

$$\frac{\partial \boldsymbol{a}^T \boldsymbol{y}}{\partial \boldsymbol{y}} = \boldsymbol{a}. \tag{421}$$

We have the product of the 3-dimensional tensor $\frac{\partial(\boldsymbol{W}^Q\boldsymbol{\xi})}{\partial \boldsymbol{W}^Q}$ with the vector $\boldsymbol{a}$ which gives a 2-dimensional tensor, i.e. a matrix:

$$\frac{\partial(\boldsymbol{W}^Q\boldsymbol{\xi})}{\partial \boldsymbol{W}^Q}\frac{\partial \boldsymbol{a}^T\boldsymbol{y}}{\partial \boldsymbol{y}} = \frac{\partial(\boldsymbol{W}^Q\boldsymbol{\xi})}{\partial \boldsymbol{W}^Q}\boldsymbol{a} = \boldsymbol{\xi}^T\boldsymbol{a}\boldsymbol{I} \ . \tag{422}$$

$$\frac{\partial \boldsymbol{a}^T\boldsymbol{y}}{\partial \boldsymbol{W}} = \beta\,\boldsymbol{X}\left(\operatorname{diag}(\boldsymbol{p}) - \boldsymbol{p}\boldsymbol{p}^T\right)\boldsymbol{X}^T\,(\boldsymbol{W}^K)^T(\boldsymbol{\xi}^T\boldsymbol{a}) \ . \tag{423}$$

The derivative with respect to $\boldsymbol{W}^K$ is

$$\frac{\partial \boldsymbol{a}^T\boldsymbol{y}}{\partial \boldsymbol{W}^K} = \frac{\partial \boldsymbol{y}}{\partial \boldsymbol{W}^K}\frac{\partial \boldsymbol{a}^T\boldsymbol{y}}{\partial \boldsymbol{y}} = \frac{\partial \boldsymbol{y}}{\partial((\boldsymbol{W}^K)^T\boldsymbol{W}^Q\boldsymbol{\xi})}\frac{\partial((\boldsymbol{W}^K)^T\boldsymbol{W}^Q\boldsymbol{\xi})}{\partial \boldsymbol{W}^K}\frac{\partial \boldsymbol{a}^T\boldsymbol{y}}{\partial \boldsymbol{y}} \ . \tag{424}$$

$$\frac{\partial \boldsymbol{y}}{\partial((\boldsymbol{W}^K)^T\boldsymbol{W}^Q\boldsymbol{\xi})} = \beta\,\boldsymbol{X}\left(\operatorname{diag}(\boldsymbol{p}) - \boldsymbol{p}\boldsymbol{p}^T\right)\boldsymbol{X}^T \tag{425}$$

$$\frac{\partial \boldsymbol{a}^T\boldsymbol{y}}{\partial \boldsymbol{y}} = \boldsymbol{a} \ . \tag{426}$$

We have the product of the 3-dimensional tensor $\frac{\partial(\boldsymbol{W}\boldsymbol{\xi})}{\partial \boldsymbol{W}^K}$ with the vector $\boldsymbol{a}$ which gives a 2-dimensional tensor, i.e. a matrix:

$$\frac{\partial((\boldsymbol{W}^K)^T\boldsymbol{W}^Q\boldsymbol{\xi})}{\partial \boldsymbol{W}^K}\frac{\partial \boldsymbol{a}^T\boldsymbol{y}}{\partial \boldsymbol{y}} = \frac{\partial((\boldsymbol{W}^K)^T\boldsymbol{W}^Q\boldsymbol{\xi})}{\partial \boldsymbol{W}^K}\boldsymbol{a} = (\boldsymbol{W}^Q)^T\boldsymbol{\xi}^T\boldsymbol{a}\boldsymbol{I} \ . \tag{427}$$

$$\frac{\partial \boldsymbol{a}^T\boldsymbol{y}}{\partial \boldsymbol{W}^K} = \beta\,\boldsymbol{X}\left(\operatorname{diag}(\boldsymbol{p}) - \boldsymbol{p}\boldsymbol{p}^T\right)\boldsymbol{X}^T\,((\boldsymbol{W}^Q)^T\boldsymbol{\xi}^T\boldsymbol{a}) \ . \tag{428}$$

## B2.7 Sequential Softmax Associative Memory

### B2.7.1 Infinite Softmax Associative Memory

We have infinite many patterns $\boldsymbol{x}_1, \boldsymbol{x}_2, \ldots$ that are represented by the infinite matrix

$$\boldsymbol{X} = (\boldsymbol{x}_1, \boldsymbol{x}_2, \ldots,) \ . \tag{429}$$

The pattern index is now a time index, that is, we observe $\boldsymbol{x}_t$ at time $t$.
The pattern matrix at time $t$ is

$$\boldsymbol{X}_t = (\boldsymbol{x}_1, \boldsymbol{x}_2, \ldots, \boldsymbol{x}_t) \ . \tag{430}$$

The query at time $t$ is $\boldsymbol{\xi}_t$.
The energy function at time $t$ is $\mathrm{E}_t$

$$\mathrm{E}_t = -\operatorname{lse}(\beta, \boldsymbol{X}_t^T\boldsymbol{\xi}_t) + \frac{1}{2}\boldsymbol{\xi}_t^T\boldsymbol{\xi}_t + \beta^{-1}\ln N + \frac{1}{2}M^2 \tag{431}$$

$$= -\beta^{-1}\ln\left(\sum_{i=1}^{T}\exp(\beta\boldsymbol{x}_i^T\boldsymbol{\xi}_t)\right) + \frac{1}{2}\boldsymbol{\xi}_t^T\boldsymbol{\xi}_t + \beta^{-1}\ln N + \frac{1}{2}M^2 \ . \tag{432}$$

The derivative of the energy $\mathrm{E}_t$ with respect to $\boldsymbol{\xi}_t$ is

$$\frac{\partial \mathrm{E}_t}{\partial \boldsymbol{\xi}_t} = -\boldsymbol{X}_t\operatorname{softmax}(\beta\boldsymbol{X}_t^T\boldsymbol{\xi}_t) + \boldsymbol{\xi}_t = -\boldsymbol{X}_t\boldsymbol{p}_t + \boldsymbol{\xi}_t \ , \tag{433}$$

where we used

$$\boldsymbol{p}_t = \operatorname{softmax}(\beta\boldsymbol{X}_t^T\boldsymbol{\xi}_t) \ . \tag{434}$$

The fixed point iteration is

$$\boldsymbol{\xi}_t^{\mathrm{new}} = \boldsymbol{X}_t\boldsymbol{p}_t = \boldsymbol{\xi}_t - \frac{\partial \mathrm{E}_t}{\partial \boldsymbol{\xi}_t} \ . \tag{435}$$

$$\boldsymbol{\xi}_t^{\text{new}} \; = \; \boldsymbol{X}_t \boldsymbol{p}_t \; = \; \boldsymbol{X}_t \text{softmax}(\beta \boldsymbol{X}_t^T \boldsymbol{\xi}_t) \, . \tag{436}$$

We can use an infinite pattern matrix with an infinite softmax. The pattern matrix at time $t$ is

$$\boldsymbol{X}_t \; = \; (\boldsymbol{x}_1, \boldsymbol{x}_2, \dots, \boldsymbol{x}_t, -\alpha \boldsymbol{\xi}_t, -\alpha \boldsymbol{\xi}_t, \dots) \, , \tag{437}$$

with the query $\boldsymbol{\xi}_t$ and $\alpha \to \infty$. The energy function at time $t$ is $\text{E}_t$

$$\text{E}_t \; = \; - \text{lse}(\beta, \boldsymbol{X}_t^T \boldsymbol{\xi}_t) \; + \; \frac{1}{2} \boldsymbol{\xi}_t^T \boldsymbol{\xi}_t \tag{438}$$

$$= \; - \beta^{-1} \ln \left( \sum_{i=1}^{t} \exp(\beta \boldsymbol{x}_i^T \boldsymbol{\xi}_t) \; + \; \sum_{i=t+1}^{\lfloor \alpha \rfloor} \exp(-\beta \alpha \|\boldsymbol{\xi}_i\|^2) \right) \; + \; \frac{1}{2} \boldsymbol{\xi}_t^T \boldsymbol{\xi}_t \, . \tag{439}$$

For $\alpha \to \infty$ and $\|\boldsymbol{\xi}_t\| \geq k > 0$ this becomes

$$\text{E}_t \; = \; - \text{lse}(\beta, \boldsymbol{X}_t^T \boldsymbol{\xi}_t) \; + \; \frac{1}{2} \boldsymbol{\xi}_t^T \boldsymbol{\xi}_t \tag{440}$$

$$= \; - \beta^{-1} \ln \left( \sum_{i=1}^{t} \exp(\beta \boldsymbol{x}_i^T \boldsymbol{\xi}_t) \right) \; + \; \frac{1}{2} \boldsymbol{\xi}_t^T \boldsymbol{\xi}_t \, . \tag{441}$$

### B2.7.2 Forgetting Softmax Associative Memory

We have infinite many patterns $\boldsymbol{x}_1, \boldsymbol{x}_2, \dots$ that are represented by the infinite matrix

$$\boldsymbol{X} \; = \; (\boldsymbol{x}_1, \boldsymbol{x}_2, \dots, ) \, . \tag{442}$$

The pattern index is now a time index, that is, we observe $\boldsymbol{x}_t$ at time $t$.
The pattern matrix at time $t$ is

$$\boldsymbol{X}_t \; = \; (\boldsymbol{x}_1, \boldsymbol{x}_2, \dots, \boldsymbol{x}_t) \, . \tag{443}$$

The query at time $t$ is $\boldsymbol{\xi}_t$.
The energy function with forgetting parameter $\gamma$ at time $t$ is $\text{E}_t$

$$\text{E}_t \; = \; - \text{lse}(\beta, \boldsymbol{X}_t^T \boldsymbol{\xi}_t \; - \; \gamma(t-1, t-2, \dots, 0)^T) \; + \; \frac{1}{2} \boldsymbol{\xi}_t^T \boldsymbol{\xi}_t \; + \; \beta^{-1} \ln N \; + \; \frac{1}{2} M^2 \tag{444}$$

$$= \; - \beta^{-1} \ln \left( \sum_{i=1}^{T} \exp(\beta \boldsymbol{x}_i^T \boldsymbol{\xi}_t \; - \; \gamma(t-i)) \right) \; + \; \frac{1}{2} \boldsymbol{\xi}_t^T \boldsymbol{\xi}_t \; + \; \beta^{-1} \ln N \; + \; \frac{1}{2} M^2 \, . \tag{445}$$

The derivative of the energy $\text{E}_t$ with respect to $\boldsymbol{\xi}_t$ is

$$\frac{\partial \text{E}_t}{\partial \boldsymbol{\xi}_t} \; = \; - \boldsymbol{X}_t \text{softmax}(\beta \boldsymbol{X}_t^T \boldsymbol{\xi}_t \; - \; \gamma(t-1, t-2, \dots, 0)^T) \; + \; \boldsymbol{\xi}_t \; = \; - \boldsymbol{X}_t \boldsymbol{p}_t \; + \; \boldsymbol{\xi}_t \, , \tag{446}$$

where we used

$$\boldsymbol{p}_t \; = \; \text{softmax}(\beta \boldsymbol{X}_t^T \boldsymbol{\xi}_t) \, . \tag{447}$$

The fixed point iteration is

$$\boldsymbol{\xi}_t^{\text{new}} \; = \; \boldsymbol{X}_t \boldsymbol{p}_t \; = \; \boldsymbol{\xi}_t \; - \; \frac{\partial \text{E}_t}{\partial \boldsymbol{\xi}_t} \, . \tag{448}$$

$$\boldsymbol{\xi}_t^{\text{new}} \; = \; \boldsymbol{X}_t \boldsymbol{p}_t \; = \; \boldsymbol{X}_t \text{softmax}(\beta \boldsymbol{X}_t^T \boldsymbol{\xi}_t) \, . \tag{449}$$

## B3 Properties of Softmax, Log-Sum-Exponential, Legendre Transform, Lambert W Function

For $\beta > 0$, the *softmax* is defined as

**Definition B5** (Softmax)**.**

$$\boldsymbol{p} \; = \; \text{softmax}(\beta \boldsymbol{x}) \tag{450}$$

$$p_i \; = \; [\text{softmax}(\beta \boldsymbol{x})]_i \; = \; \frac{\exp(\beta x_i)}{\sum_k \exp(\beta x_k)} \, . \tag{451}$$

We also need the *log-sum-exp function* (lse), defined as

**Definition B6** (Log-Sum-Exp Function)**.**

$$\mathrm{lse}(\beta, \boldsymbol{x}) \; = \; \beta^{-1} \ln \left( \sum_{i=1}^{N} \exp(\beta x_i) \right) \; . \tag{452}$$

Next, we give the relation between the softmax and the lse function.

**Lemma 18.** *The softmax is the gradient of the* lse*:*

$$\mathrm{softmax}(\beta \boldsymbol{x}) \; = \; \nabla_{\boldsymbol{x}} \mathrm{lse}(\beta, \boldsymbol{x}) \; . \tag{453}$$

In the next lemma we report some important properties of the lse function.

**Lemma 19.** *We define*

$$\mathrm{L} \; := \; \boldsymbol{z}^T \boldsymbol{x} \; - \; \beta^{-1} \sum_{i=1}^{N} z_i \ln z_i \tag{454}$$

*with* $\mathrm{L} \geq \boldsymbol{p}^T \boldsymbol{x}$. *The* lse *is the maximum of* L *on the $N$-dimensional simplex $D$ with $D = \{\boldsymbol{z} \mid \sum_i z_i = 1, 0 \leqslant z_i\}$:*

$$\mathrm{lse}(\beta, \boldsymbol{x}) \; = \; \max_{\boldsymbol{z} \in D} \boldsymbol{z}^T \boldsymbol{x} \; - \; \beta^{-1} \sum_{i=1}^{N} z_i \ln z_i \; . \tag{455}$$

*The softmax $\boldsymbol{p} = \mathrm{softmax}(\beta \boldsymbol{x})$ is the argument of the maximum of* L *on the $N$-dimensional simplex $D$ with $D = \{\boldsymbol{z} \mid \sum_i z_i = 1, 0 \leqslant z_i\}$:*

$$\boldsymbol{p} \; = \; \mathrm{softmax}(\beta \boldsymbol{x}) \; = \; \arg \max_{\boldsymbol{z} \in D} \boldsymbol{z}^T \boldsymbol{x} \; - \; \beta^{-1} \sum_{i=1}^{N} z_i \ln z_i \; . \tag{456}$$

*Proof.* Eq. (455) is obtained from Equation (8) in [22] and Eq. (456) from Equation (11) in [22]. □

From a physical point of view, the lse function represents the "free energy" in statistical thermodynamics [22].

Next we consider the Jacobian of the softmax and its properties.

**Lemma 20.** *The Jacobian $\mathrm{J}_s$ of the softmax $\boldsymbol{p} = \mathrm{softmax}(\beta \boldsymbol{x})$ is*

$$\mathrm{J}_s \; = \; \frac{\partial \mathrm{softmax}(\beta \boldsymbol{x})}{\partial \boldsymbol{x}} \; = \; \beta \left( \mathrm{diag}(\boldsymbol{p}) - \boldsymbol{p}\boldsymbol{p}^T \right) \; , \tag{457}$$

*which gives the elements*

$$[\mathrm{J}_s]_{ij} \; = \; \begin{cases} \beta p_i (1 - p_i) & \textit{for } i = j \\ -\beta p_i p_j & \textit{for } i \neq j \end{cases} \; . \tag{458}$$

Next we show that $\mathrm{J}_s$ has eigenvalue 0.

**Lemma 21.** *The Jacobian $\mathrm{J}_s$ of the softmax function $\boldsymbol{p} = \mathrm{softmax}(\beta \boldsymbol{x})$ has a zero eigenvalue with eigenvector $\boldsymbol{1}$.*

*Proof.*

$$[\mathrm{J}_s \boldsymbol{1}]_i \; = \; \beta \left( p_i(1 - p_i) \; - \sum_{j, j \neq i} p_i p_j \right) \; = \; \beta \, p_i (1 \; - \sum_j p_j) \; = 0 \; . \tag{459}$$

□

Next we show that 0 is the smallest eigenvalue of $\mathrm{J}_s$, therefore $\mathrm{J}_s$ is positive semi-definite but not (strict) positive definite.

**Lemma 22.** *The Jacobian $\mathrm{J}_s$ of the softmax $\boldsymbol{p} = \mathrm{softmax}(\beta \boldsymbol{\xi})$ is symmetric and positive semi-definite.*

*Proof.* For an arbitrary $\boldsymbol{y}$, we have

$$\boldsymbol{y}^T \left( \mathrm{diag}(\boldsymbol{p}) - \boldsymbol{p}\boldsymbol{p}^T \right) \boldsymbol{y} \;=\; \sum_i p_i y_i^2 - \left( \sum_i p_i y_i \right)^2 \tag{460}$$

$$= \left( \sum_i p_i y_i^2 \right) \left( \sum_i p_i \right) - \left( \sum_i p_i y_i \right)^2 \;\geq\; 0 \,.$$

The last inequality hold true because the Cauchy-Schwarz inequality says $(\boldsymbol{a}^T \boldsymbol{a})(\boldsymbol{b}^T \boldsymbol{b}) \geq (\boldsymbol{a}^T \boldsymbol{b})^2$, which is the last inequality with $a_i = y_i \sqrt{p_i}$ and $b_i = \sqrt{p_i}$. Consequently $\left( \mathrm{diag}(\boldsymbol{p}) - \boldsymbol{p}\boldsymbol{p}^T \right)$ is positive semi-definite.

Alternatively $\sum_i p_i y_i^2 - \left( \sum_i p_i y_i \right)^2$ can be viewed as the expected second moment minus the mean squared which gives the variance that is larger equal to zero.

The Jacobian is $0 < \beta$ times a positive semi-definite matrix, which is a positive semi-definite matrix. $\qquad\square$

Moreover, the softmax is a monotonic map, as described in the next lemma.

**Lemma 23.** *The softmax $\boldsymbol{p} = \mathrm{softmax}(\beta \boldsymbol{x})$ is monotone, that is,*

$$\left( \mathrm{softmax}(\beta \boldsymbol{x}) - \mathrm{softmax}(\beta \boldsymbol{x}') \right)^T (\boldsymbol{x} - \boldsymbol{x}') \;\geq\; 0 \,. \tag{461}$$

*Proof.* We use the mean value theorem with the symmetric matrix $\mathrm{J}_s^m = \int_0^1 \mathrm{J}_s(\lambda \boldsymbol{x} + (1-\lambda)\boldsymbol{x}')\, \mathrm{d}\lambda$:

$$\mathrm{softmax}(\boldsymbol{x}) - \mathrm{softmax}(\boldsymbol{x}') \;=\; \mathrm{J}_s^m\, (\boldsymbol{x} - \boldsymbol{x}') \,. \tag{462}$$

Therefore

$$\left( \mathrm{softmax}(\boldsymbol{x}) - \mathrm{softmax}(\boldsymbol{x}') \right)^T (\boldsymbol{x} - \boldsymbol{x}') \;=\; (\boldsymbol{x} - \boldsymbol{x}')^T \, \mathrm{J}_s^m \, (\boldsymbol{x} - \boldsymbol{x}') \;\geq\; 0 \,, \tag{463}$$

since $\mathrm{J}_s^m$ is positive semi-definite. For all $\lambda$ the Jacobians $\mathrm{J}_s(\lambda \boldsymbol{x} + (1-\lambda)\boldsymbol{x}')$ are positive semi-definite according to Lemma 22. Since

$$\boldsymbol{x}^T \mathrm{J}_s^m \boldsymbol{x} \;=\; \int_0^1 \boldsymbol{x}^T \mathrm{J}_s(\lambda \boldsymbol{x} + (1-\lambda)\boldsymbol{x}')\, \boldsymbol{x} \, \mathrm{d}\lambda \;\geq\; 0 \tag{464}$$

is an integral over positive values for every $\boldsymbol{x}$, $\mathrm{J}_s^m$ is positive semi-definite, too. $\qquad\square$

Next we give upper bounds on the norm of $\mathrm{J}_s$.

**Lemma 24.** *For a softmax $\boldsymbol{p} = \mathrm{softmax}(\beta \boldsymbol{x})$ with $m = \max_i p_i(1 - p_i)$, the spectral norm of the Jacobian $\mathrm{J}_s$ of the softmax is bounded:*

$$\|\mathrm{J}_s\|_2 \;\leqslant\; 2\, m\, \beta \,, \tag{465}$$
$$\|\mathrm{J}_s\|_1 \;\leqslant\; 2\, m\, \beta \,, \tag{466}$$
$$\|\mathrm{J}_s\|_\infty \;\leqslant\; 2\, m\, \beta \,. \tag{467}$$

*In particular everywhere holds*

$$\|\mathrm{J}_s\|_2 \;\leqslant\; \frac{1}{2}\, \beta \,. \tag{468}$$

*If $p_{\max} = \max_i p_i \geq 1 - \epsilon \geq 0.5$, then for the spectral norm of the Jacobian holds*

$$\|\mathrm{J}_s\|_2 \;\leqslant\; 2\, \epsilon\, \beta \;-\; 2\, \epsilon^2\, \beta \;<\; 2\, \epsilon\, \beta \,. \tag{469}$$

*Proof.* We consider the maximum absolute column sum norm

$$\|\boldsymbol{A}\|_1 \;=\; \max_j \sum_i |a_{ij}| \tag{470}$$

and the maximum absolute row sum norm

$$\|\boldsymbol{A}\|_\infty \;=\; \max_i \sum_j |a_{ij}| \,. \tag{471}$$

We have for $\boldsymbol{A} = \mathrm{J}_s = \beta \left( \mathrm{diag}(\boldsymbol{p}) - \boldsymbol{p}\boldsymbol{p}^T \right)$

$$\sum_j |a_{ij}| = \beta \left( p_i(1 - p_i) + \sum_{j, j \neq i} p_i p_j \right) = \beta\, p_i\, (1 - 2p_i + \sum_j p_j) \qquad (472)$$

$$= 2\, \beta\, p_i\, (1 - p_i) \leqslant 2\, m\, \beta\,,$$

$$\sum_i |a_{ij}| = \beta \left( p_j\, (1 - p_j) + \sum_{i, i \neq j} p_j p_i \right) = \beta\, p_j\, (1 - 2p_j + \sum_i p_i) \qquad (473)$$

$$= 2\, \beta\, p_j\, (1 - p_j) \leqslant 2\, m\, \beta\,.$$

Therefore we have

$$\|\mathrm{J}_s\|_1 \leqslant 2\, m\, \beta\,, \qquad (474)$$

$$\|\mathrm{J}_s\|_\infty \leqslant 2\, m\, \beta\,, \qquad (475)$$

$$\|\mathrm{J}_s\|_2 \leqslant \sqrt{\|\mathrm{J}_s\|_1 \|\mathrm{J}_s\|_\infty} \leqslant 2\, m\, \beta\,. \qquad (476)$$

The last inequality is a direct consequence of Hölder's inequality.

For $0 \leqslant p_i \leqslant 1$, we have $p_i(1 - p_i) \leqslant 0.25$. Therefore $m \leqslant 0.25$ for all values of $p_i$.

If $p_{\max} \geq 1 - \epsilon \geq 0.5$ ($\epsilon \leqslant 0.5$), then $1 - p_{\max} \leqslant \epsilon$ and for $p_i \neq p_{\max}$ $p_i \leqslant \epsilon$. The derivative $\partial x(1 - x)/\partial x = 1 - 2x > 0$ for $x < 0.5$, therefore $x(1 - x)$ increases with $x$ for $x < 0.5$. Using $x = 1 - p_{\max}$ and for $p_i \neq p_{\max}$ $x = p_i$, we obtain $p_i(1 - p_i) \leqslant \epsilon(1 - \epsilon)$ for all $i$. Consequently, we have $m \leqslant \epsilon(1 - \epsilon)$. $\qquad\square$

Using the bounds on the norm of the Jacobian, we give some Lipschitz properties of the softmax function.

**Lemma 25.** *The softmax function $\boldsymbol{p} = \mathrm{softmax}(\beta \boldsymbol{x})$ is $(\beta/2)$-Lipschitz. The softmax function $\boldsymbol{p} = \mathrm{softmax}(\beta \boldsymbol{x})$ is $(2\beta m)$-Lipschitz in a convex environment $U$ for which $m = \max_{\boldsymbol{x} \in U} \max_i p_i(1 - p_i)$. For $p_{\max} = \min_{\boldsymbol{x} \in U} \max_i p_i = 1 - \epsilon$, the softmax function $\boldsymbol{p} = \mathrm{softmax}(\beta \boldsymbol{x})$ is $(2\beta\epsilon)$-Lipschitz. For $\beta < 2m$, the softmax $\boldsymbol{p} = \mathrm{softmax}(\beta \boldsymbol{x})$ is contractive in $U$ on which $m$ is defined.*

*Proof.* The mean value theorem states for the symmetric matrix $\mathrm{J}_s^m = \int_0^1 \mathrm{J}(\lambda \boldsymbol{x} + (1 - \lambda)\boldsymbol{x}')\, \mathrm{d}\lambda$:

$$\mathrm{softmax}(\boldsymbol{x}) - \mathrm{softmax}(\boldsymbol{x}') = \mathrm{J}_s^m\, (\boldsymbol{x} - \boldsymbol{x}')\,. \qquad (477)$$

According to Lemma 24 for all $\tilde{\boldsymbol{x}} = \lambda \boldsymbol{x} + (1 - \lambda)\boldsymbol{x}'$

$$\|\mathrm{J}_s(\tilde{\boldsymbol{x}})\|_2 \leqslant 2\, \tilde{m}\, \beta\,, \qquad (478)$$

where $\tilde{m} = \max_i \tilde{p}_i(1 - \tilde{p}_i)$. Since $\boldsymbol{x} \in U$ and $\boldsymbol{x}' \in U$ we have $\tilde{\boldsymbol{x}} \in U$, since $U$ is convex. For $m = \max_{\boldsymbol{x} \in U} \max_i p_i(1 - p_i)$ we have $\tilde{m} \leqslant m$ for all $\tilde{m}$. Therefore we have

$$\|\mathrm{J}_s(\tilde{\boldsymbol{x}})\|_2 \leqslant 2\, m\, \beta \qquad (479)$$

which also holds for the mean:

$$\|\mathrm{J}_s^m\|_2 \leqslant 2\, m\, \beta\,. \qquad (480)$$

Therefore

$$\|\mathrm{softmax}(\boldsymbol{x}) - \mathrm{softmax}(\boldsymbol{x}')\| \leqslant \|\mathrm{J}_s^m\|_2\, \|\boldsymbol{x} - \boldsymbol{x}'\| \leqslant 2\, m\, \beta\, \|\boldsymbol{x} - \boldsymbol{x}'\|\,. \qquad (481)$$

From Lemma 24 we know $m \leqslant 1/4$ globally. For $p_{\max} = \min_{\boldsymbol{x} \in U} \max_i p_i = 1 - \epsilon$ we have according to Lemma 24: $m \leqslant \epsilon$. $\qquad\square$

For completeness we present a result about cocoercivity of the softmax:

**Lemma 26.** *For $m = \max_{\boldsymbol{x} \in U} \max_i p_i(1 - p_i)$, softmax function $\boldsymbol{p} = \mathrm{softmax}(\beta \boldsymbol{x})$ is $1/(2m\beta)$-cocoercive in $U$, that is,*

$$(\mathrm{softmax}(\boldsymbol{x}) - \mathrm{softmax}(\boldsymbol{x}'))^T (\boldsymbol{x} - \boldsymbol{x}') \geq \frac{1}{2\, m\, \beta} \|\mathrm{softmax}(\boldsymbol{x}) - \mathrm{softmax}(\boldsymbol{x}')\|. \qquad (482)$$

*In particular the softmax function $\boldsymbol{p} = \mathrm{softmax}(\beta \boldsymbol{x})$ is $(2/\beta)$-cocoercive everywhere. With $p_{\max} = \min_{\boldsymbol{x} \in U} \max_i p_i = 1 - \epsilon$, the softmax function $\boldsymbol{p} = \mathrm{softmax}(\beta \boldsymbol{x})$ is $1/(2\beta\epsilon)$-cocoercive in $U$.*

*Proof.* We apply the Baillon-Haddad theorem (e.g. Theorem 1 in [22]) together with Lemma 25. □

Finally, we introduce the Legendre transform and use it to describe further properties of the lse. We start with the definition of the convex conjugate.

**Definition B7** (Convex Conjugate). *The* Convex Conjugate (Legendre-Fenchel transform) *of a function $f$ from a Hilbert Space $X$ to $[-\infty, \infty]$ is $f^*$ which is defined as*

$$f^*(\boldsymbol{x}^*) = \sup_{\boldsymbol{x} \in X}(\boldsymbol{x}^T\boldsymbol{x}^* - f(\boldsymbol{x})), \quad \boldsymbol{x}^* \in X \tag{483}$$

See page 219 Def. 13.1 in [7] and page 134 in [23]. Next we define the Legendre transform, which is a more restrictive version of the convex conjugate.

**Definition B8** (Legendre Transform). *The* Legendre transform *of a convex function $f$ from a convex set $X \subset \mathbb{R}^n$ to $\mathbb{R}$ ($f : X \to \mathbb{R}$) is $f^*$, which is defined as*

$$f^*(\boldsymbol{x}^*) = \sup_{\boldsymbol{x} \in X}(\boldsymbol{x}^T\boldsymbol{x}^* - f(\boldsymbol{x})), \quad \boldsymbol{x}^* \in X^*, \tag{484}$$

$$X^* = \left\{ \boldsymbol{x}^* \in \mathbb{R}^n \mid \sup_{\boldsymbol{x} \in X}(\boldsymbol{x}^T\boldsymbol{x}^* - f(\boldsymbol{x})) < \infty \right\}. \tag{485}$$

See page 91 in [9].

**Definition B9** (Epi-Sum). *Let $f$ and $g$ be two functions from $X$ to $(-\infty, \infty]$, then the infimal convolution (or epi-sum) of $f$ and $g$ is*

$$f \square g : X \to [-\infty, \infty], \ \boldsymbol{x} \mapsto \inf_{\boldsymbol{y} \in X}(f(\boldsymbol{y}) + g(\boldsymbol{x} - \boldsymbol{y})) \tag{486}$$

See Def. 12.1 in [7].

**Lemma 27.** *Let $f$ and $g$ be functions from $X$ to $(-\infty, \infty]$. Then the following hold:*

1. *Convex Conjugate of norm squared*

$$\left( \frac{1}{2}\|\cdot\|^2 \right)^* = \frac{1}{2}\|\cdot\|^2. \tag{487}$$

2. *Convex Conjugate of a function multiplied by scalar $0 < \alpha \in \mathbb{R}$*

$$(\alpha f)^* = \alpha f^*(./\alpha). \tag{488}$$

3. *Convex Conjugate of the sum of a function and a scalar $\beta \in \mathbb{R}$*

$$(f + \beta)^* = f^* - \beta. \tag{489}$$

4. *Convex Conjugate of affine transformation of the arguments. Let $\boldsymbol{A}$ be a non-singular matrix and $\boldsymbol{b}$ a vector*

$$(f(\boldsymbol{A}\boldsymbol{x} + \boldsymbol{b}))^* = f^*\left(\boldsymbol{A}^{-T}\boldsymbol{x}^*\right) - \boldsymbol{b}^T\boldsymbol{A}^{-T}\boldsymbol{x}^*. \tag{490}$$

5. *Convex Conjugate of epi-sums*

$$(f \square g)^* = f^* + g^*. \tag{491}$$

*Proof.* 1. Since $h(t) := \frac{t^2}{2}$ is a non-negative convex function and $h(t) = 0 \iff t = 0$ we have because of Proposition 11.3.3 in [23] that $h(\|x\|)^* = h^*(\|x^*\|)$. Additionally, by example (a) on page 137 we get for $1 < p < \infty$ and $\frac{1}{p} + \frac{1}{q} = 1$ that $\left(\frac{|t|^p}{p}\right)^* = \frac{|t^*|^q}{q}$. Putting all together we get the desired result. The same result can also be deduced from page 222 Example 13.6 in [7].

2. Follows immediately from the definition since

$$\alpha f^*\left(\frac{\boldsymbol{x}^*}{\alpha}\right) = \alpha \sup_{\boldsymbol{x} \in X}\left(\boldsymbol{x}^T\frac{\boldsymbol{x}^*}{\alpha} - f(\boldsymbol{x})\right) = \sup_{\boldsymbol{x} \in X}(\boldsymbol{x}^T\boldsymbol{x}^* - \alpha f(\boldsymbol{x})) = (\alpha f)^*(\boldsymbol{x}^*)$$

3. $(f + \beta)^* := \sup_{\boldsymbol{x} \in X}\left(\boldsymbol{x}^T\boldsymbol{x}^* - f(\boldsymbol{x}) - \beta\right) =: f^* - \beta$

4.

$$(f(\boldsymbol{Ax}+\boldsymbol{b}))^*(\boldsymbol{x}^*) = \sup_{\boldsymbol{x}\in X} \left(\boldsymbol{x}^T\boldsymbol{x}^* - f(\boldsymbol{Ax}+\boldsymbol{b})\right)$$

$$= \sup_{\boldsymbol{x}\in X} \left((\boldsymbol{Ax}+\boldsymbol{b})^T \boldsymbol{A}^{-T}\boldsymbol{x}^* - f(\boldsymbol{Ax}+\boldsymbol{b})\right) - \boldsymbol{b}^T\boldsymbol{A}^{-T}\boldsymbol{x}^*$$

$$= \sup_{\boldsymbol{y}\in X} \left(\boldsymbol{y}^T\boldsymbol{A}^{-T}\boldsymbol{x}^* - f(\boldsymbol{y})\right) - \boldsymbol{b}^T\boldsymbol{A}^{-T}\boldsymbol{x}^*$$

$$= f^*\left(\boldsymbol{A}^{-T}\boldsymbol{x}^*\right) - \boldsymbol{b}^T\boldsymbol{A}^{-T}\boldsymbol{x}^*$$

5. From Proposition 13.24 (i) in [7] and Proposition 11.4.2 in [23] we get

$$(f\square g)^*(\boldsymbol{x}^*) = \sup_{\boldsymbol{x}\in X} \left(\boldsymbol{x}^T\boldsymbol{x}^* - \inf_{\boldsymbol{y}\in X} \left(f(\boldsymbol{y}) - g(\boldsymbol{x}-\boldsymbol{y})\right)\right)$$

$$= \sup_{\boldsymbol{x},\boldsymbol{y}\in X} \left(\boldsymbol{x}^T\boldsymbol{x}^* - f(\boldsymbol{y}) - g(\boldsymbol{x}-\boldsymbol{y})\right)$$

$$= \sup_{\boldsymbol{x},\boldsymbol{y}\in X} \left(\left(\boldsymbol{y}^T\boldsymbol{x}^* - f(\boldsymbol{y})\right) + \left((\boldsymbol{x}-\boldsymbol{y})^T\boldsymbol{x}^* - g(\boldsymbol{x}-\boldsymbol{y})\right)\right)$$

$$= f^*(\boldsymbol{x}^*) + g^*(\boldsymbol{x}^*)$$

$\square$

**Lemma 28.** *The Legendre transform of the* lse *is the negative entropy function, restricted to the probability simplex and vice versa. For the log-sum exponential*

$$f(\boldsymbol{x}) = \ln\left(\sum_{i=1}^n \exp(x_i)\right), \tag{492}$$

*the Legendre transform is the negative entropy function, restricted to the probability simplex:*

$$f^*(\boldsymbol{x}^*) = \begin{cases} \sum_{i=1}^n x_i^* \ln(x_i^*) & \text{for } 0 \leqslant x_i^* \text{ and } \sum_{i=1}^n x_i^* = 1 \\ \infty & \text{otherwise} \end{cases}. \tag{493}$$

*For the negative entropy function, restricted to the probability simplex:*

$$f(\boldsymbol{x}) = \begin{cases} \sum_{i=1}^n x_i \ln(x_i) & \text{for } 0 \leqslant x_i \text{ and } \sum_{i=1}^n x_i = 1 \\ \infty & \text{otherwise} \end{cases}. \tag{494}$$

*the Legendre transform is the log-sum exponential*

$$f^*(\boldsymbol{x}^*) = \ln\left(\sum_{i=1}^n \exp(x_i^*)\right), \tag{495}$$

*Proof.* See page 93 Example 3.25 in [9] and [22]. If $f$ is a regular convex function (lower semi-continuous convex function), then $f^{**} = f$ according to page 135 Exercise 11.2.3 in [23]. If $f$ is lower semi-continuous and convex, then $f^{**} = f$ according to Theorem 13.37 (Fenchel-Moreau) in [7]. The log-sum-exponential is continuous and convex. $\square$

**Lemma 29.** *Let $\boldsymbol{XX}^T$ be non-singular and $X$ a Hilbert space. We define*

$$X^* = \left\{\boldsymbol{a} \mid 0 \leqslant \boldsymbol{X}^T\left(\boldsymbol{XX}^T\right)^{-1}\boldsymbol{a}, \ \boldsymbol{1}^T\boldsymbol{X}^T\left(\boldsymbol{XX}^T\right)^{-1}\boldsymbol{a} = 1\right\}. \tag{496}$$

*and*

$$X^v = \left\{\boldsymbol{a} \mid \boldsymbol{a} = \boldsymbol{X}^T\boldsymbol{\xi}, \ \boldsymbol{\xi}\in X\right\}. \tag{497}$$

*The Legendre transform of $\mathrm{lse}(\beta, \boldsymbol{X}^T\boldsymbol{\xi})$ with $\boldsymbol{\xi}\in X$ is*

$$\left(\mathrm{lse}(\beta,\boldsymbol{X}^T\boldsymbol{\xi})\right)^*(\boldsymbol{\xi}^*) = \left(\mathrm{lse}(\beta,\boldsymbol{v})\right)^*\left(\boldsymbol{X}^T\left(\boldsymbol{XX}^T\right)^{-1}\boldsymbol{\xi}^*\right), \tag{498}$$

*with $\boldsymbol{\xi}^*\in X^*$ and $\boldsymbol{v}\in X^v$. The domain of $\left(\mathrm{lse}(\beta,\boldsymbol{X}^T\boldsymbol{\xi})\right)^*$ is $X^*$.*
*Furthermore we have*

$$\left(\mathrm{lse}(\beta,\boldsymbol{X}^T\boldsymbol{\xi})\right)^{**} = \mathrm{lse}(\beta,\boldsymbol{X}^T\boldsymbol{\xi}). \tag{499}$$

*Proof.* We use the definition of the Legendre transform:

$$\left(\mathrm{lse}(\beta, \boldsymbol{X}^T\boldsymbol{\xi})\right)^* (\boldsymbol{\xi}^*) \;=\; \sup_{\boldsymbol{\xi}\in X} \boldsymbol{\xi}^T\boldsymbol{\xi}^* \;-\; \mathrm{lse}(\beta, \boldsymbol{X}^T\boldsymbol{\xi}) \tag{500}$$

$$= \; \sup_{\boldsymbol{\xi}\in X} \left(\boldsymbol{X}^T\boldsymbol{\xi}\right)^T \boldsymbol{X}^T \left(\boldsymbol{X}\boldsymbol{X}^T\right)^{-1} \boldsymbol{\xi}^* \;-\; \mathrm{lse}(\beta, \boldsymbol{X}^T\boldsymbol{\xi})$$

$$= \; \sup_{\boldsymbol{v}\in X^v} \boldsymbol{v}^T \boldsymbol{X}^T \left(\boldsymbol{X}\boldsymbol{X}^T\right)^{-1} \boldsymbol{\xi}^* \;-\; \mathrm{lse}(\beta, \boldsymbol{v})$$

$$= \; \sup_{\boldsymbol{v}\in X^v} \boldsymbol{v}^T \boldsymbol{v}^* \;-\; \mathrm{lse}(\beta, \boldsymbol{v})$$

$$= \; (\mathrm{lse}(\beta, \boldsymbol{v}))^* (\boldsymbol{v}^*) \;=\; (\mathrm{lse}(\beta, \boldsymbol{v}))^* \left(\boldsymbol{X}^T \left(\boldsymbol{X}\boldsymbol{X}^T\right)^{-1} \boldsymbol{\xi}^*\right) \;,$$

where we used $\boldsymbol{v}^* = \boldsymbol{X}^T \left(\boldsymbol{X}\boldsymbol{X}^T\right)^{-1} \boldsymbol{\xi}^*$.

According to page 93 Example 3.25 in [9], the equations for the maximum $\max_{\boldsymbol{v}\in X^v} \boldsymbol{v}^T\boldsymbol{v}^* - \mathrm{lse}(\beta, \boldsymbol{v})$ are solvable if and only if $0 < \boldsymbol{v}^* = \boldsymbol{X}^T \left(\boldsymbol{X}\boldsymbol{X}^T\right)^{-1} \boldsymbol{\xi}^*$ and $\boldsymbol{1}^T\boldsymbol{v}^* = \boldsymbol{1}^T \boldsymbol{X}^T \left(\boldsymbol{X}\boldsymbol{X}^T\right)^{-1} \boldsymbol{\xi}^* = 1$. Therefore we assumed $\boldsymbol{\xi}^* \in X^*$.

The domain of $\left(\mathrm{lse}(\beta, \boldsymbol{X}^T\boldsymbol{\xi})\right)^*$ is $X^*$, since on page 93 Example 3.25 in [9] it was shown that outside $X^*$ the $\sup_{\boldsymbol{v}\in X^v} \boldsymbol{v}^T\boldsymbol{v}^* - \mathrm{lse}(\beta, \boldsymbol{v})$ is not bounded.
Using

$$\boldsymbol{p} \;=\; \mathrm{softmax}(\beta\boldsymbol{X}^T\boldsymbol{\xi}) \;, \tag{501}$$

the Hessian of $\mathrm{lse}(\beta, \boldsymbol{X}^T\boldsymbol{\xi})$

$$\frac{\partial^2 \mathrm{lse}(\beta, \boldsymbol{X}^T\boldsymbol{\xi})}{\partial \boldsymbol{\xi}^2} \;=\; \beta \, \boldsymbol{X} \left(\mathrm{diag}(\boldsymbol{p}) - \boldsymbol{p}\boldsymbol{p}^T\right) \boldsymbol{X}^T \tag{502}$$

is positive semi-definite since $\mathrm{diag}(\boldsymbol{p}) - \boldsymbol{p}\boldsymbol{p}^T$ is positive semi-definite according to Lemma 22. Therefore $\mathrm{lse}(\beta, \boldsymbol{X}^T\boldsymbol{\xi})$ is convex and continuous.
If $f$ is a regular convex function (lower semi-continuous convex function), then $f^{**} = f$ according to page 135 Exercise 11.2.3 in [23]. If $f$ is lower semi-continuous and convex, then $f^{**} = f$ according to Theorem 13.37 (Fenchel-Moreau) in [7]. Consequently we have

$$\left(\mathrm{lse}(\beta, \boldsymbol{X}^T\boldsymbol{\xi})\right)^{**} \;=\; \mathrm{lse}(\beta, \boldsymbol{X}^T\boldsymbol{\xi}) \;. \tag{503}$$

$\square$

We introduce the Lambert $W$ function and some of its properties, since it is needed to derive bounds on the storage capacity of our new Hopfield networks.

**Definition B10** (Lambert Function). *The* Lambert $W$ function *is the inverse function of*

$$f(y) \;=\; ye^y \;. \tag{504}$$

*The Lambert $W$ function has an upper branch $W_0$ for $-1 \leqslant y$ and a lower branch $W_{-1}$ for $y \leqslant -1$. We use $W$ if a formula holds for both branches. We have*

$$W(x) \;=\; y \;\Rightarrow\; ye^y \;=\; x \;. \tag{505}$$

We present some identities for the Lambert $W$ function:

**Lemma 30.** *Identities for the Lambert $W$ function are*

$$W(x)\, e^{W(x)} = x \,, \tag{506}$$

$$W(xe^x) = x \,, \tag{507}$$

$$e^{W(x)} = \frac{x}{W(x)} \,, \tag{508}$$

$$e^{-W(x)} = \frac{W(x)}{x} \,, \tag{509}$$

$$e^{nW(x)} = \left(\frac{x}{W(x)}\right)^n \,, \tag{510}$$

$$W_0(x\ln x) = \ln x \quad \text{for } x \geq \frac{1}{e} \,, \tag{511}$$

$$W_{-1}(x\ln x) = \ln x \quad \text{for } x \leqslant \frac{1}{e} \,, \tag{512}$$

$$W(x) = \ln\frac{x}{W(x)} \quad \text{for } x \geq -\frac{1}{e} \,, \tag{513}$$

$$W\left(\frac{n\, x^n}{W(x)^{n-1}}\right) = n\, W(x) \quad \text{for } n, x > 0 \,, \tag{514}$$

$$W(x) + W(y) = W\left(x\, y\left(\frac{1}{W(x)} + \frac{1}{W(y)}\right)\right) \quad \text{for } x, y > 0 \,, \tag{515}$$

$$W_0\left(-\frac{\ln x}{x}\right) = -\ln x \quad \text{for } 0 < x \leqslant e \,, \tag{516}$$

$$W_{-1}\left(-\frac{\ln x}{x}\right) = -\ln x \quad \text{for } x > e \,, \tag{517}$$

$$e^{-W(-\ln x)} = \frac{W(-\ln x)}{-\ln x} \quad \text{for } x \neq 1 \,. \tag{518}$$

We also present some special values for the Lambert $W$ function:

**Lemma 31.**

$$W(0) = 0 \,, \tag{519}$$

$$W(e) = 1 \,, \tag{520}$$

$$W\left(-\frac{1}{e}\right) = -1 \,, \tag{521}$$

$$W\left(e^{1+e}\right) = e \,, \tag{522}$$

$$W(2\ln 2) = \ln 2 \,, \tag{523}$$

$$W(1) = \Omega \,, \tag{524}$$

$$W(1) = e^{-W(1)} = \ln\left(\frac{1}{W(1)}\right) = -\ln W(1) \,, \tag{525}$$

$$W\left(-\frac{\pi}{2}\right) = \frac{i\pi}{2} \,, \tag{526}$$

$$W(-1) \approx -0.31813 + 1.33723i \,, \tag{527}$$

*where the Omega constant $\Omega$ is*

$$\Omega = \left(\int_{-\infty}^{\infty} \frac{\mathrm{d}t}{(e^t - t)^2 + \pi^2}\right)^{-1} - 1 \approx 0.56714329 \,. \tag{528}$$

## B4 Modern Hopfield Networks: Binary States (Krotov and Hopfield)

### B4.1 Modern Hopfield Networks: Introduction

#### B4.1.1 Additional Memory and Attention for Neural Networks

Modern Hopfield networks may serve as additional memory for neural networks. Different approaches have been suggested to equip neural networks with an additional memory beyond recurrent connections. The neural Turing machine (NTM) is a neural network equipped with an external memory and an attention process [24]. The NTM can write to the memory and can read from it. A memory network [48] consists of a memory together with the components: (1) input feature map (converts the incoming input to the internal feature representation) (2) generalization (updates old memories given the input), (3) output feature map (produces a new output), (4) response (converts the output into the response format). Memory networks are generalized to an end-to-end trained model, where the $\arg\max$ memory call is replaced by a differentiable $\text{softmax}$ [40, 41]. Linear Memory Network use a linear autoencoder for sequences as a memory [13].

To enhance RNNs with additional associative memory like Hopfield networks have been proposed [3, 4]. The associative memory stores hidden states of the RNN, retrieves stored states if they are similar to actual ones, and has a forgetting parameter. The forgetting and storing parameters of the RNN associative memory have been generalized to learned matrices [54]. LSTMs with associative memory via Holographic Reduced Representations have been proposed [15].

Recently most approaches to new memories are based on attention. The neural Turing machine (NTM) is equipped with an external memory and an attention process [24]. End to end memory networks (EMN) make the attention scheme of memory networks [48] differentiable by replacing $\arg\max$ through a $\text{softmax}$ [40, 41]. EMN with dot products became very popular and implement a key-value attention [16] for self-attention. An enhancement of EMN is the transformer [45, 46] and its extensions [17]. The transformer had great impact on the natural language processing (NLP) community as new records in NLP benchmarks have been achieved [45, 46]. MEMO uses the transformer attention mechanism for reasoning over longer distances [5]. Current state-of-the-art for language processing is a transformer architecture called "the Bidirectional Encoder Representations from Transformers" (BERT) [19, 20].

#### B4.1.2 Modern Hopfield networks: Overview

The storage capacity of classical binary Hopfield networks [27] has been shown to be very limited. In a $d$-dimensional space, the standard Hopfield model can store $d$ uncorrelated patterns without errors but only $Cd/\ln(d)$ random patterns with $C < 1/2$ for a fixed stable pattern or $C < 1/4$ if all patterns are stable [33]. The same bound holds for nonlinear learning rules [32]. Using tricks-of-trade and allowing small retrieval errors, the storage capacity is about $0.138d$ [14, 25, 44]. If the learning rule is not related to the Hebb rule then up to $d$ patterns can be stored [1]. Using a Hopfield networks with non-zero diagonal matrices, the storage can be increased to $Cd\ln(d)$ [21]. In contrast to the storage capacity, the number of energy minima (spurious states, stable states) of Hopfield networks is exponentially in $d$ [42, 11, 47].

Recent advances in the field of binary Hopfield networks [27] led to new properties of Hopfield networks. The stability of spurious states or metastable states was sensibly reduced by a Hamiltonian treatment for the new relativistic Hopfield model [6]. Recently the storage capacity of Hopfield networks could be increased by new energy functions. Interaction functions of the form $F(x) = x^n$ lead to storage capacity of $\alpha_n d^{n-1}$, where $\alpha_n$ depends on the allowed error probability [28, 29, 18] (see [29] for the non-binary case). Interaction functions of the form $F(x) = x^n$ lead to storage capacity of $\alpha_n \frac{d^{n-1}}{c_n \ln d}$ for $c_n > 2(2n-3)!!$ [18].

Interaction functions of the form $F(x) = \exp(x)$ lead to *exponential* storage capacity of $2^{d/2}$ where all stored pattern are fixed points but the radius of attraction vanishes [18]. It has been shown that the network converges even after one update [18].

### B4.2 Energy and Update Rule for Binary Modern Hopfield Networks

We follow [18] where the goal is to store a set of input data $\boldsymbol{x}_1, \ldots, \boldsymbol{x}_N$ that are represented by the matrix

$$\boldsymbol{X} = (\boldsymbol{x}_1, \ldots, \boldsymbol{x}_N) \ . \tag{529}$$

The $\boldsymbol{x}_i$ is pattern with binary components $x_{ij} \in \{-1, +1\}$ for all $i$ and $j$. $\boldsymbol{\xi}$ is the actual state of the units of the Hopfield model. Krotov and Hopfield [28] defined the energy function E with the

interaction function $F$ that evaluates the dot product between patterns $\boldsymbol{x}_i$ and the actual state $\boldsymbol{\xi}$:

$$\mathrm{E} \ = \ - \sum_{i=1}^{N} F\left(\boldsymbol{\xi}^T \boldsymbol{x}_i\right) \tag{530}$$

with $F(a) = a^n$, where $n = 2$ gives the energy function of the classical Hopfield network. This allows to store $\alpha_n d^{n-1}$ patterns [28]. Krotov and Hopfield [28] suggested for minimizing this energy an asynchronous updating dynamics $T = (T_j)$ for component $\xi_j$:

$$T_j(\boldsymbol{\xi}) \ := \ \mathrm{sgn}\left[\sum_{i=1}^{N}\left(F\left(x_{ij} + \sum_{l \neq j} x_{il}\,\xi_l\right) - F\left(-x_{ij} + \sum_{l \neq j} x_{il}\,\xi_l\right)\right)\right] \tag{531}$$

While Krotov and Hopfield used $F(a) = a^n$, Demircigil et al. [18] went a step further and analyzed the model with the energy function $F(a) = \exp(a)$, which leads to an exponential storage capacity of $N = 2^{d/2}$. Furthermore with a single update the final pattern is recovered with high probability. These statements are given in next theorem.

**Theorem B10** (Storage Capacity for Binary Modern Hopfield Nets (Demircigil et al. 2017)). *Consider the generalized Hopfield model with the dynamics described in Eq. (531) and interaction function $F$ given by $F(x) = e^x$. For a fixed $0 < \alpha < \ln(2)/2$ let $N = \exp(\alpha d) + 1$ and let $\boldsymbol{x}_1, \ldots, \boldsymbol{x}_N$ be $N$ patterns chosen uniformly at random from $\{-1, +1\}^d$. Moreover fix $\varrho \in [0, 1/2)$. For any $i$ and any $\widetilde{\boldsymbol{x}}_i$ taken uniformly at random from the Hamming sphere with radius $\varrho d$ centered in $\boldsymbol{x}_i$, $\mathcal{S}(\boldsymbol{x}_i, \varrho D)$, where $\varrho d$ is assumed to be an integer, it holds that*

$$\mathrm{Pr}\left(\exists i \, \exists j : \, T_j\left(\widetilde{\boldsymbol{x}}_i\right) \, \neq \, x_{ij}\right) \ \to \ 0\,,$$

*if $\alpha$ is chosen in dependence of $\varrho$ such that*

$$\alpha \ < \ \frac{I(1 - 2\varrho)}{2}$$

*with*

$$I : \ a \ \mapsto \ \frac{1}{2}\left((1 + a)\ln(1 + a) \ + \ (1 - a)\ln(1 - a)\right)\,.$$

*Proof.* The proof can be found in [18]. $\qquad\square$

The number of patterns $N = \exp(\alpha d) + 1$ is exponential in the number $d$ of components. The result

$$\mathrm{Pr}\left(\exists i \, \exists j : \, T_j\left(\widetilde{\boldsymbol{x}}_i\right) \, \neq \, x_{ij}\right) \ \to \ 0$$

means that one update for each component is sufficient to recover the pattern with high probability. The constraint $\alpha < \frac{I(1-2\varrho)}{2}$ on $\alpha$ gives the trade-off between the radius of attraction $\varrho N$ and the number $N = \exp(\alpha d) + 1$ of pattern that can be stored.
Theorem B10 in particular implies that

$$\mathrm{Pr}\left(\exists i \, \exists j : \, T_j\left(\boldsymbol{x}_i\right) \, \neq \, x_{ij}\right) \ \to \ 0$$

as $d \to \infty$, i.e. with a probability converging to 1, all the patterns are fixed points of the dynamics. In this case we can have $\alpha \to \frac{I(1)}{2} = \ln(2)/2$.
Krotov and Hopfield define the update dynamics $T_j(\boldsymbol{\xi})$ in Eq. (531) via energy differences of the energy in Eq. (530). First we express the energy in Eq. (530) with $F(a) = \exp(a)$ [18] by the lse function. Then we use the mean value theorem to express the update dynamics $T_j(\boldsymbol{\xi})$ in Eq. (531) by the softmax function. For simplicity, we set $\beta = 1$ in the following. There exists a $v \in [-1, 1]$ with

$$T_j(\boldsymbol{\xi}) \ = \ \mathrm{sgn}\left[\mathrm{E}(\xi_j = 1) \, - \, \mathrm{E}(\xi_j = -1)\right] \ = \ \mathrm{sgn}\left[-\exp(\mathrm{lse}(\xi_j = 1)) \, + \, \exp(\mathrm{lse}(\xi_j = -1))\right] \tag{532}$$

$$= \ \mathrm{sgn}\left[(2\boldsymbol{e}_j)^T \nabla_{\boldsymbol{\xi}} \mathrm{E}(\xi_j = v)\right] \ = \ \mathrm{sgn}\left[\exp(\mathrm{lse}(\xi_j = v))\,(2\boldsymbol{e}_j)^T \frac{\mathrm{lse}(\xi_j = v)}{\partial \boldsymbol{\xi}}\right]$$

$$= \ \mathrm{sgn}\left[\exp(\mathrm{lse}(\xi_j = 1))\,(2\boldsymbol{e}_j)^T \boldsymbol{X}\,\mathrm{softmax}(\boldsymbol{X}^T \boldsymbol{\xi}(\xi_j = v))\right]$$

$$= \ \mathrm{sgn}\left[[\boldsymbol{X}\,\mathrm{softmax}(\boldsymbol{X}^T \boldsymbol{\xi}(\xi_j = v))]_j\right] \ = \ \mathrm{sgn}\left[[\boldsymbol{X}\boldsymbol{p}(\xi_j = v)]_j\right]\,,$$

where $\boldsymbol{e}_j$ is the Cartesian unit vector with a one at position $j$ and zeros elsewhere, $[.]_j$ is the projection to the $j$-th component, and

$$\boldsymbol{p} \ = \ \mathrm{softmax}(\boldsymbol{X}^T \boldsymbol{\xi})\,. \tag{533}$$

## B5 Hopfield Update Rule is Attention of The Transformer

The Hopfield network update rule is the attention mechanism used in the transformer and BERT (see Fig. B2). To see this, we assume patterns $\boldsymbol{y}_i$ that are mapped to the Hopfield space of dimension $d_k$. We set $\boldsymbol{x}_i = \boldsymbol{W}_K^T \boldsymbol{y}_i$, $\boldsymbol{\xi}_i = \boldsymbol{W}_Q^T \boldsymbol{y}_i$, and multiply the result of our update rule with $\boldsymbol{W}_V$. The matrix $\boldsymbol{Y} = (\boldsymbol{y}_1, \ldots, \boldsymbol{y}_N)^T$ combines the $\boldsymbol{y}_i$ as row vectors. We define the matrices $\boldsymbol{X}^T = \boldsymbol{K} = \boldsymbol{Y} \boldsymbol{W}_K$, $\boldsymbol{Q} = \boldsymbol{Y} \boldsymbol{W}_Q$, and $\boldsymbol{V} = \boldsymbol{Y} \boldsymbol{W}_K \boldsymbol{W}_V = \boldsymbol{X}^T \boldsymbol{W}_V$, where $\boldsymbol{W}_K \in \mathbb{R}^{d_y \times d_k}, \boldsymbol{W}_Q \in \mathbb{R}^{d_y \times d_k}, \boldsymbol{W}_V \in \mathbb{R}^{d_k \times d_v}$. For combining all queries in matrix $\boldsymbol{Q}$, $\beta = 1/\sqrt{d_k}$, and $\mathrm{softmax} \in \mathbb{R}^N$ changed to a row vector, we obtain for the update rule Eq. (17) multiplied by $\boldsymbol{W}_V$:

$$\mathrm{softmax}\left(1/\sqrt{d_k}\, \boldsymbol{Q}\,\boldsymbol{K}^T\right)\,\boldsymbol{V}\,. \tag{534}$$

This formula is the transformer attention.

Figure B2: We generalized the energy of binary modern Hopfield networks for allowing continuous states while keeping convergence and storage capacity properties. We defined for the new energy also a new update rule that minimizes the energy. The new update rule is the attention mechanism of the transformer. Formulae are modified to express $\mathrm{softmax}$ as row vector as for transformers. "="-sign means "keeps the properties".

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

[Supplementary Material 2 · Supplement A.pdf]

# SUPPLEMENT A:
# Modern Hopfield Networks and Attention for Immune Repertoire Classification

**Michael Widrich**[*]    **Bernhard Schäfl**[*]    **Milena Pavlović**[†,‡]    **Geir Kjetil Sandve**[‡]

**Sepp Hochreiter**[*,§]                     **Victor Greiff**[†]

**Günter Klambauer**[*]
[*]ELLIS Unit Linz and LIT AI Lab,
Institute for Machine Learning,
Johannes Kepler University Linz, Austria
[†]Department of Immunology, University of Oslo, Norway
[‡]Department of Informatics, University of Oslo, Norway
[§]Institute of Advanced Research in Artificial Intelligence (IARAI)

## Contents

## A1 Introduction

This document is a supplement to the paper "Modern Hopfield Networks and Attention for Immune Repertoire Classification". All datasets and code will be fully released at `https://github.com/ml-jku/DeepRC`. The *CMV dataset* is publicly available at `https://clients.adaptivebiotech.com/pub/Emerson-2017-NatGen`.

## A2 Notation overview

| Definition | Symbol/Notation | Dimension |
|---|---|---|
| bag / input object / repertoire | $X$ | set of $N$ instances |
| sequence / instance | $s_i$ | $d_l \times (20 + 3)$ |
| space of sequences | $\mathcal{S}$ | |
| instance-level representation of sequence $s_i$ | $\boldsymbol{z}_i$ | $d_v$ |
| instance-level representation matrix of $(s_1, \ldots, s_N)$ | $\boldsymbol{Z}$ | $N \times d_v$ |
| repertoire-level representation of bag $X$ | $\boldsymbol{z}$ | $d_v$ |
| attention value of the $i$-th instance | $a_i$ | |
| standard basis vector | $\boldsymbol{e}_m$ | $d_v$ |
| k-mer representation of a sequence $s_i$ | $\boldsymbol{u}_i$ | $d_u$ |
| k-mer representation of a repertoire $X$ | $\boldsymbol{u}$ | $d_u$ |
| pattern | $\boldsymbol{x}_i$ | $d$ or $d_k$ |
| pattern matrix | $\boldsymbol{X}$ | $d \times N$ or $d_k \times N$ |
| query | $\boldsymbol{\xi}$ | $d$ or $d_k$ |
| key matrix | $\boldsymbol{K}$ | $N \times d_k$ |
| query matrix | $\boldsymbol{Q}$ | $N \times d_k$ or $d_q \times d_k$ |
| value matrix | $\boldsymbol{V}$ | $N \times d_v$ |
| key embedding matrix | $\boldsymbol{W}_K$ | $d_y \times d_k$ |
| query embedding matrix | $\boldsymbol{W}_Q$ | $d_y \times d_k$ |
| value embedding matrix | $\boldsymbol{W}_V$ | $d_k \times d_v$ |
| pattern before embedding | $\boldsymbol{y}_i$ | $d_y$ |
| pattern matrix before embedding | $\boldsymbol{Y}$ | $N \times d_y$ |
| number of patterns or instances | $N$ | |
| largest norm of a pattern | $M$ | |
| separation of pattern $\boldsymbol{x}_i$ | $\Delta_i$ | |
| scale/temperature parameter | $\beta$ | |
| sequence embedding function | $h$ | |
| pooling function | $f$ | |
| bag classifier function | $g$ | |
| output function / output layer | $o$ | |
| classifier output / prediction for bag $X$ | $\hat{y}$ | |
| k-mer extraction function | $h_{\mathrm{kmer}}$ | |
| sub-network of DeepRC | $h_1$ | |
| sub-network of DeepRC | $h_2$ | |
| network parameters of $h_1$ | $\boldsymbol{\theta}_1$ | |
| network parameters of $h_2$ | $\boldsymbol{\theta}_2$ | |
| network parameters of $o$ | $\boldsymbol{\theta}_o$ | |
| dimension of a key | $d_k$ | |
| dimension of a value | $d_v$ | |
| dimension of a query | $d_q$ | |
| dimension of a k-mer representation | $d_u$ | |
| dimension of pattern $\boldsymbol{x}_i$ | $d$ | |
| dimension of pattern $\boldsymbol{y}_i$ | $d_y$ | |
| length of input sequence | $d_l$ | |
| witness ratio / motif frequency | $\rho$ | |

Table A1: Symbols and notations used in this paper.

## A3 DeepRC implementation details

The implementation of our method is provided at `https://github.com/ml-jku/DeepRC`.

**Input layer.** For the input layer of the CNN, the characters in the input sequence, i.e. the amino acids (AAs), are encoded in a one-hot vector of length 20. To also provide information about the position of an AA in the sequence, we add 3 additional input features with values in range $[0, 1]$ to encode the position of an AA relative to the sequence. These 3 positional features encode whether the AA is located at the beginning, the center, or the end of the sequence, respectively, as shown in Figure A1. We concatenate these 3 positional features with the one-hot vector of AAs, which results in a feature vector of size 23 per sequence position. Each repertoire, now represented as a bag of feature vectors, is then normalized to unit variance.
We feed the sequences with 23 features per position into the CNN. Sequences of different lengths were zero-padded to the maximum sequence length per batch at the sequence ends.

Figure A1: We use 3 input features with values in range $[0, 1]$ to encode the relative position of each AA in a sequence with respect to the sequence. "feature 1" encodes if an AA is close to the sequence start, "feature 2" to the sequence center, and "feature 3" to the sequence end. For every position in the sequence, the values of all three features sum up to 1.

**Sequence duplicates and abundance.** The cytomegalovirus dataset (*CMV dataset*) (Emerson et al., 2017) provides sequences with an associated pre-processed abundance value per sequence, which indicates the number of occurrences of a sequence in a repertoire. We incorporate this information into the input of DeepRC such that the one-hot AA features of a sequence are multiplied by a scaling factor of $\log(sequence\_abundance)$ before normalization.
Other duplicated sequences are fed as separate sequences into the network, i.e. multiple sequence instances might be identical if the sequence occurs multiple times in the repertoire.

**1D CNN for motif recognition.** In the following, we describe how DeepRC identifies patterns in the individual sequences and reduces each sequence in the input object to a fixed-size feature vector. DeepRC employs 1D convolution layers to extract patterns, where trainable weight kernels are convolved over the sequence positions. In principle, also recurrent neural networks (RNNs) or transformer networks could be used instead of 1D CNNs, however, (a) the computational complexity of the network must be low to be able to process millions of sequences for a single update. Additionally, (b) the learned network should be able to provide insights in the recognized patterns in form of motifs. Both properties (a) and (b) are fulfilled by 1D convolution operations that are used by DeepRC.
We use one 1D CNN layer (Hu et al., 2014) with SELU activation functions (Klambauer et al., 2017) to identify the relevant patterns in the input sequences with a computationally light-weight operation. The larger the kernel size, the more surrounding sequence positions are taken into account, which influences the length of the motifs that can be extracted. We therefore adjust the kernel size during hyperparameter search. In prior works (Ostmeyer et al., 2019), a k-mer size of 4 yielded good predictive performance, which could indicate that a kernel size in the range of 4 may be a proficient choice. For $d_v$ trainable kernels, this produces a feature vector of length $d_v$ at each sequence position. Subsequently, global max-pooling over all sequence positions of a sequence reduces the sequence-representations $z_i$ to vectors of the fixed length $d_v$. Given the challenging size of the input data per repertoire, the computation of the CNN activations and weight updates is performed using

16-bit floating point values. A list of hyperparameters evaluated for DeepRC is given in Table A4. A comparison of RNN-based and CNN-based sequence embedding for motif recognition in a smaller experimental setting is given in Sec. A11.

**Regularization.** During training, we apply random subsampling of repertoire sequences, which can be interpreted as random drop-out (Hinton et al., 2012) on the input sequences or attention weights, to reduce over-fitting and decrease the computational cost. For this, each repertoire is subsampled to $10,000$ input sequences, which are randomly drawn from the respective repertoire.
Additionally, one might employ further regularization techniques, which we only partly investigated further in a smaller experimental setting in Sec. A11 due to high computational demands. Such regularization techniques include $l1$ and $l2$ weight decay, noise in the form of random AA permutations in the input sequences, noise on the attention weights, or random shuffling of sequences between repertoires that belong to the negative class. The last regularization technique assumes that the sequences in positive-class repertoires carry a signal, such as an AA motif corresponding to an immune response, whereas the sequences in negative-class repertoires do not. Hence, the sequences can be shuffled randomly between negative class repertoires without obscuring the signal in the positive class repertoires.

**Reduction of computational cost and memory consumption.** We took measures to address the high computational demands, especially GPU memory consumption, in order to make the large number of experiments feasible:
We train the DeepRC model with a small batch size of $4$ samples and perform computation of inference and updates of the 1D CNN using 16-bit floating point values. The rest of the network is trained using 32-bit floating point values. The Adam parameter for numerical stability was therefore increased from the default value of $\epsilon = 10^{-8}$ to $\epsilon = 10^{-4}$.
During training and evaluation, we apply attention-based subsampling of repertoire sequences to reduce the memory consumption and computational cost. For this, the attention weights computed by the attention network are used to rank the input sequences. Based on this ranking, the repertoire is reduced to the $10\%$ of sequences with the highest attention weights. These top $10\%$ of sequences are then used to compute the weight updates and the prediction for the repertoire.

**Computation time.** Training was performed on various GPU types, mainly `NVIDIA RTX 2080 Ti`. Computation times were highly dependent on the number of sequences in the repertoires and the number and sizes of CNN kernels. A single forward and backward pass with weight update on an `NVIDIA RTX 2080 Ti` GPU took approximately $0.0109$ to $0.0135$ seconds, while requiring approximately $8$ to $11$ GB GPU memory. The average total required time for one update step, including the loading of $4$ samples, their reduction to the $10\%$ of sequences with the highest attention weights, the weight update, the computation of validation scores for early-stopping (pro rata), and the logging of results, was approximately $0.1$ seconds. Taking GPUs with larger memory ($\geq 16$ GB) into account, it is already possible to train DeepRC on larger datasets, possibly with multi-head attention and a larger network architectures (see Sec. A11). Our network implementation is based on `PyTorch 1.3.1` (Paszke et al., 2019).

**Incorporation of additional inputs and metadata.** Additional metadata in the form of sequence-level or repertoire-level features could be incorporated into the input via concatenation with the feature vectors that result from taking the maximum of the 1D CNN outputs w.r.t. the sequence positions. This has the benefit that the attention mechanism and output network can utilize the sequence-level or repertoire-level features for their predictions. Sparse metadata or metadata that is only available during training could be used as auxiliary targets to incorporate the information via gradients into the DeepRC model.

**Limitations.** The current methods are mostly limited by computational complexity, since both hyperparameter and model selection is computationally demanding. For hyperparameter selection, a large number of hyperparameter settings have to be evaluated. For model selection, a single repertoire requires the propagation of many thousands of sequences through a neural network and keeping those quantities in GPU memory in order to perform the attention mechanism and weight update. Thus, increased GPU memory would significantly boost our approach. Increased computational power would also allow for more advanced architectures and attention mechanisms, which may further improve predictive performance. Another limiting factor is over-fitting of the model due to the currently relatively small number of samples (bags) in real-world immunosequencing datasets in comparison to the large number of instances per bag and features per instance.

**Hyperparameters.** For the hyperparameter search of DeepRC for the category "simulated immunosequencing data", we only conducted a full hyperparameter search on the more difficult datasets with motif implantation probabilities below $1\%$, as described in Table A4. This process was repeated for all 5 folds of the 5-fold cross-validation (CV) and the average score on the 5 test sets constitutes the final score of a method.

Table A4 provides an overview of the hyperparameter search, which was conducted as a grid search for each of the datasets in a nested 5-fold CV procedure, as described in section A5.

## A4 Datasets

We aimed at constructing immune repertoire classification scenarios with varying degree of realism and difficulties in order to compare and analyze the suggested machine learning methods. To this end, we either use simulated or experimentally-observed immune receptor sequences and we implant signals, which are sequence motifs (Akbar et al., 2019; Weber et al., 2020), into sequences of repertoires of the positive class. It has been shown previously that interaction of immune receptors with antigens occur via short sequence stretches (Akbar et al., 2019). Thus, implantation of short motif sequences simulating an immune signal is biologically meaningful. Our benchmarking study comprises four different categories of datasets: (a) Simulated immunosequencing data with implanted signals (where the signal is defined as sets of motifs), (b) LSTM-generated immunosequencing data with implanted signals, (c) real-world immunosequencing data with implanted signals, and (d) real-world immunosequencing data. Each of the first three categories consists of multiple datasets with varying difficulty depending on the type of the implanted signal and the ratio of sequences with the implanted signal. The ratio of sequences with the implanted signal, where each sequence carries at most 1 implanted signal, corresponds to the *witness rate* (WR). We consider binary classification tasks to simulate the immune status of healthy and diseased individuals. We randomly generate immune repertoires with varying numbers of sequences, where we implant sequence motifs in the repertoires of the diseased individuals, i.e. the positive class. The sequences of a repertoire are also randomly generated by different procedures (detailed below). Each sequence is composed of 20 different characters, corresponding to amino acids, and has an average length of 14.5 AAs.

### A4.1 Simulated immunosequencing data

In the first category, we aim at investigating the impact of the signal frequency, i.e. the WR, and the signal complexity on the performance of the different methods. To this end, we created 21 datasets, whereas each dataset contains a large number of repertoires with a large number of random AA sequences per repertoire. We then implanted signals in the AA sequences of the positive class repertoires, where the 21 datasets differ in frequency and complexity of the implanted signals. In detail, the AAs were sampled randomly independent of their respective position in the sequence, while the frequencies of AAs, distribution of sequence lengths, and distribution of the number of sequences per repertoire, i.e. the number of instances per bag, are following the respective distributions observed in the real-world *CMV dataset* (Emerson et al., 2017). For this, we first sampled the number of sequences for a repertoire from a Gaussian $\mathcal{N}(\mu = 316k, \sigma = 132k)$ distribution and rounded to the nearest positive integer. We re-sampled if the size was below $5k$. We then generated random sequences of AAs with a length of $\mathcal{N}(\mu = 14.5, \sigma = 1.8)$, again rounded to the nearest positive integers. Each simulated repertoire was then randomly assigned to either the positive or negative class, with $2,500$ repertoires per class. In the repertoires assigned to the positive class, we implanted motifs with an average length of 4 AAs, following the results of the experimental analysis of antigen-binding motifs in antibodies and T-cell receptor sequences by Akbar et al. (2019). We varied the characteristics of the implanted motifs for each of the 21 datasets with respect to the following parameters: (a) $\rho$, the probability of a motif being implanted in a sequence of a positive repertoire, i.e. the average ratio of sequences containing the motif, which is the witness rate. (b) The number of wildcard positions in the motif. A wildcard position contains a random AA, which is randomly sampled for each sequence. Wildcard positions are located in the center of the implanted motif. (c) The number of deletion positions in the implanted motif. A deletion position has a probability of $0.5$ of being removed from the motif. Deletion positions are located in the center of the implanted motifs. In this way, we generated 18 different datasets of variable difficulty containing in total roughly $28.7$ billion sequences. Additionally, we added 3 datasets in which every position has a $20\%$ probability of behaving like a wildcard position to evaluate the performance on motifs with increased noise, resulting in a total of 21 different datasets. See Table A2 for an overview of the properties of the implanted motifs in the 21 datasets.

### A4.2 LSTM-generated data

In the second dataset category, we investigate the impact of the signal frequency and complexity in combination with more plausible immune receptor sequences by taking into account the positional AA distributions and other sequence properties. To this end, we trained an LSTM (Hochreiter & Schmidhuber, 1997) in a standard next character prediction (Graves, 2013) setting to create AA sequences with properties similar to experimentally observed immune receptor sequences.

In the first step, the LSTM model was trained on all immuno-sequences in the *CMV dataset* (Emerson et al., 2017) that contain valid information about sequence abundance and have a known CMV label. Such an LSTM model is able to capture various properties of the sequences, including position-dependent probability distributions and combinations, relationships, and order of AAs. We then used the trained LSTM model to generate $1,000$ repertoires in an autoregressive fashion, starting with a start sequence that was randomly sampled from the trained-on dataset. Based on a visual inspection of the frequencies of 4-mers (see section A8), the similarity of LSTM generated sequences and real sequences was deemed sufficient for the purpose of generating the AA sequences for the datasets in this category. Further details on LSTM training and repertoire generation are given in Section A8.

After generation, each repertoire was assigned to either the positive or negative class, with $500$ repertoires per class. We implanted motifs of length $4$ with varying properties in the center of the sequences of the positive class to obtain $5$ different datasets. Each sequence in the positive repertoires has a probability $\rho$ to carry the motif, which was varied throughout $5$ datasets and corresponds to the WR (see Table A2). Each position in the motif has a probability of $0.9$ to be implanted and consequently a probability of $0.1$ that the original AA in the sequence remains, which can be seen as noise on the motif.

|  | Simulated | LSTM gen. | Real-world |
|---|---|---|---|
| seq. per bag | $N(316k, 132k)$ | $N(285k, 156k)$ | $10k$ |
| repertoires | $5,000$ | $1,000$ | $1,500$ |
| motif noise | $0\%$ | $10\%$ | $*$ |
| wildcards | $\{0; 1; 2\}$ | $0$ | $0$ |
| deletions | $\{0; 1\}$ | $0$ | $0$ |
| mot. freq. $\rho$ (in %) | $\{1; 0.1; 0.01\}$ | $\{10; 1; 0.5; 0.1; 0.05\}$ | $\{1; 0.1\}$ |

Table A2: Properties of simulated repertoires, variations of motifs, and motif frequencies, i.e. the witness rate, for the datasets in categories "simulated immunosequencing data", "LSTM-generated data", and "real-world data with implanted signals". Noise types for $*$ are explained in paragraph "real-world data with implanted signals".

### A4.3 Real-world data with implanted signals

In the third category, we implanted signals into experimentally obtained immuno-sequences, where we considered $4$ dataset variations. Each dataset consists of $750$ repertoires for each of the two classes, where each repertoire consists of $10k$ sequences. In this way, we aim to simulate datasets with a *low sequencing coverage*, which means that only relatively few sequences per repertoire are available. The sequences were randomly sampled from healthy (CMV negative) individuals from the *CMV dataset* (see below paragraph for explanation). Two signal types were considered: (a) **One signal with one motif.** The AA motif LDR was implanted in a certain fraction of sequences. The pattern is randomly altered at one of the three positions with probabilities $0.2$, $0.6$, and $0.2$, respectively. (b) **One signal with multiple motifs.** One of the three possible motifs LDR, CAS, and GL-N was implanted with equal probability. Again, the motifs were randomly altered before implantation. The AA motif LDR changed as described above. The AA motif CAS was altered at the second position with probability $0.6$ and with probability $0.3$ at the first position. The pattern GL-N, where – denotes a gap location, is randomly altered at the first position with probability $0.6$ and the gap has a length of $0$, $1$, or $2$ AAs with equal probability.

Additionally, the datasets differ in the values for $\rho$, the average ratio of sequences carrying a signal, which were chosen as $1\%$ or $0.1\%$. The motifs were implanted at positions $107$, $109$, and $114$ according to the IMGT numbering scheme for immune receptor sequences (Lefranc et al., 2003) with probabilities $0.3$, $0.35$ and $0.2$, respectively. With the remaining $0.15$ chance, the motif is implanted

at any other sequence position. This means that the motif occurrence in the simulated sequences is biased towards the middle of the sequence.

### A4.4 Real-world data: CMV dataset

We used a real-world dataset of 785 repertoires, each of which containing between $4,371$ to $973,081$ (avg. $299,319$) TCR sequences with a length of 1 to 27 (avg. 14.5) AAs, originally collected and provided by Emerson et al. (2017). 340 out of 785 repertoires were labelled as positive for cytomegalovirus (CMV) serostatus, which we consider as the positive class, 420 repertoires with negative CMV serostatus, considered as negative class, and 25 repertoires with unknown status. We changed the number of sequence counts per repertoire from $-1$ to 1 for 3 sequences. Furthermore, we exclude a total of 99 repertoires with unknown CMV status or unknown information about the sequence abundance within a repertoire, reducing the dataset for our analysis to 686 repertoires, 312 of which with positive and 374 with negative CMV status.

### A4.5 Comparison to other MIL datasets

| Dataset | Total number of bags | Total number of instances | Approx. number of features per instance | Avg. number of instances per bag | Source | Dataset reference |
|---|---|---|---|---|---|---|
| Simulated immuno-sequencing data (ours) | 5,000 | 1,597,024,310 x 21 datasets | 14.5x20 AA sequence | 316,000 | this work | |
| LSTM-generated data (ours) | 1,000 | 304,825,671 x 5 datasets | 14.5x20 AA sequence | 285,000 | this work | |
| Real-world data with implanted signals (ours) | 1,500 | 14,715,421 x 4 datasets | 14.5x20 AA sequence | 10,000 | this work | |
| CMV (pre-processed by us) | 785 | 234,965,729 | 14.5x20 AA sequence | 299,000 | this work | Emerson et al. (2017) |
| MNIST bags | 50–500 | 500–50,000 | 28x28x1 image | 100 | Ilse et al. (2018) | |
| Breast Cancer | 58 | approx. 39,000 | 32x32x3 H&E image | 672 | Ilse et al. (2018) | Gelasca et al. (2008) |
| Basal cell carcinomas | 820 | 7,588,767 | 1024x1024x3 H&E image | 9,056 | Kimeswenger et al. (2019) | |
| Birds | 548 | 10,232 | 38 | 9 | Ruiz et al. (2018) | Briggs et al. (2012) |
| Scene | 2,000 | 18,000 | 15 | 9 | Ruiz et al. (2018) | Zhang & Zhang (2007) |
| Reuters | 2,000 | 7,119 | 243 | 4 | Ruiz et al. (2018) | Sebastiani (2002) |
| CK+ | 430 | 7,915 | 4,391 | 18 | Ruiz et al. (2018) | Lucey et al. (2010) |
| UniProt (Geobacter sulfurreducens) | 379 | 1,250 | 216 | 3 | Ruiz et al. (2018) | Wu et al. (2014) |
| MODIS (aerosol data) | 1,364 | 136,400 | 12 | 100 | Uriot (2019) | https://aeronet.gsfc.nasa.gov |
| MISR1 (aerosol data) | 800 | 80,000 | 16 | 100 | Uriot (2019) | https://aeronet.gsfc.nasa.gov |
| MISR2 (aerosol data) | 800 | 80,000 | 12 | 54 | Uriot (2019) | https://aeronet.gsfc.nasa.gov |
| CORN (crop yield) | 525 | 52,500 | 92 | 100 | Uriot (2019) | https://aeronet.gsfc.nasa.gov |
| WHEAT (crop yield) | 525 | 52,500 | 92 | 100 | Uriot (2019) | https://aeronet.gsfc.nasa.gov |

Table A3: MIL datasets with their numbers of bags and numbers of instances. "total number of instances" refers to the total number of instances in the dataset. The simulated and real-world immunosequencing datasets considered in this work contain a by orders of magnitudes larger number of instances per bag than MIL datasets that were considered by machine learning methods up to now.

## A5 Compared methods

We evaluate and compare the performance of DeepRC against a set of machine learning methods that serve as baseline, were suggested, or can readily be adapted to immune repertoire classification. In this section, we describe these compared methods.

### A5.1 Known motif

This method serves as an estimate for the achievable classification performance using prior knowledge about which motif was implanted. Note that this does not necessarily lead to perfect predictive performance since motifs are implanted with a certain amount of noise and could also be present in the negative class by chance. The *known motif* method counts how often the known implanted motif occurs per sequence for each repertoire and uses this count to rank the repertoires. From this ranking, the Area Under the receiver operator Curve (AUC) is computed as performance measure. Probabilistic AA changes in the known motif are not considered for this count, with the exception of gap positions. We consider two versions of this method: (a) **Known motif binary:** counts the occurrence of the known motif in a sequence and (b) **Known motif continuous:** counts the maximum number of overlapping AAs between the known motif and all sequence positions, which corresponds to a convolution operation with a binary kernel followed by max-pooling. Since the implanted signal is not known in the experimentally obtained *CMV dataset*, this method cannot be applied to this dataset.

### A5.2 Support Vector Machine (SVM)

The Support Vector Machine (SVM) approach uses a fixed mapping from a bag of sequences to the corresponding k-mer counts. The function $h_{\mathrm{kmer}}$ maps each sequence $s_i$ to a vector representing the occurrence of k-mers in the sequence. To avoid confusion with the sequence-representation obtained from the CNN layers of DeepRC, we denote $\boldsymbol{u}_i = h_{\mathrm{kmer}}(s_i)$, which is analogous to $\boldsymbol{z}_i$. Specifically, $u_{im} = (h_{\mathrm{kmer}}(s_i))_m = \#\{p_m \in s_i\}$, where $\#\{p_m \in s_i\}$ denotes how often the k-mer pattern $p_m$ occurs in sequence $s_i$. Afterwards, average-pooling is applied to obtain $\boldsymbol{u} = 1/N \sum_{i=1}^{N} \boldsymbol{u}_i$, the *k-mer representation* of the input object $X$. For two input objects $X^{(n)}$ and $X^{(l)}$ with representations $\boldsymbol{u}^{(n)}$ and $\boldsymbol{u}^{(l)}$, respectively, we implement the *MinMax kernel* (Ralaivola et al., 2005) as follows:

$$
\begin{aligned}
k(X^{(n)}, X^{(l)}) &= k_{\mathrm{MinMax}}(\boldsymbol{u}^{(n)}, \boldsymbol{u}^{(l)}) \\
&= \frac{\sum_{m=1}^{d_u} \min(u_m^{(n)}, u_m^{(l)})}{\sum_{m=1}^{d_u} \max(u_m^{(n)}, u_m^{(l)})},
\end{aligned}
\tag{1}
$$

where $u_m^{(n)}$ is the $m$-th element of the vector $u^{(n)}$. The *Jaccard kernel* (Levandowsky & Winter, 1971) is identical to the MinMax kernel except that it operates on binary $\boldsymbol{u}^{(n)}$. We used a standard C-SVM, as introduced by Cortes & Vapnik (1995). The corresponding hyperparameter $C$ is optimized by random search. The settings of the full hyperparameter search as well as the respective value ranges are given in Table A5.

### A5.3 K-Nearest Neighbor (KNN)

The same *k-mer representation* of a repertoire, as introduced above for the SVM baseline, is used for the K-Nearest Neighbor (KNN) approach. As this method clusters samples according to distances between them, the previous kernel definitions cannot be applied directly. It is therefore necessary to transform the MinMax as well as the Jaccard kernel from similarities to distances by constructing the following (Levandowsky & Winter, 1971):

$$
\begin{aligned}
d_{\mathrm{MinMax}}(\boldsymbol{u}^{(n)}, \boldsymbol{u}^{(l)}) &= 1 - k_{\mathrm{MinMax}}(\boldsymbol{u}^{(n)}, \boldsymbol{u}^{(l)}), \\
d_{\mathrm{Jaccard}}(\boldsymbol{u}^{(n)}, \boldsymbol{u}^{(l)}) &= 1 - k_{\mathrm{Jaccard}}(\boldsymbol{u}^{(n)}, \boldsymbol{u}^{(l)}).
\end{aligned}
\tag{2}
$$

The amount of neighbors is treated as the hyperparameter and optimized by an exhaustive grid search. The settings of the full hyperparameter search as well as the respective value ranges are given in Table A6.

### A5.4 Logistic regression

We implemented logistic regression on the *k-mer representation* $\boldsymbol{u}$ of an immune repertoire. The model is trained by gradient descent using the Adam optimizer (Kingma & Ba, 2014). The learning

rate is treated as the hyperparameter and optimized by grid search. Furthermore, we explored two regularization settings using combinations of $l1$ and $l2$ weight decay. The settings of the full hyperparameter search as well as the respective value ranges are given in Table A7.

### A5.5 Burden test

We implemented a burden test (Emerson et al., 2017; Li & Leal, 2008; Wu et al., 2011) in a machine learning setting. The burden test first identifies sequences or k-mers that are associated with the individual's class, i.e., immune status, and then calculates a burden score per individual. Concretely, for each k-mer or sequence, the phi coefficient of the contingency table for absence or presence and positive or negative immune status is calculated. Then, $J$ k-mers or sequences with the highest phi coefficients are selected as the set of associated k-mers or sequences. $J$ is a hyperparameter that is selected on a validation set. Additionally, we consider the type of input features, sequences or k-mers, as a hyperparameter. For inference, a burden score per individual is calculated as the sum of associated k-mers or sequences it carries. This score is used as raw prediction and to rank the individuals. Hence, we have extended the burden test by Emerson et al. (2017) to k-mers and to adaptive thresholds that are adjusted on a validation set.

### A5.6 Logistic MIL (Ostmeyer et al)

The logistic multiple instance learning (MIL) approach for immune repertoire classification (Ostmeyer et al., 2019) applies a logistic regression model to each k-mer representation in a bag. The resulting scores are then summarized by max-pooling to obtain a prediction for the bag. Each amino acid of each k-mer is represented by $5$ features, the so-called Atchley factors (Atchley et al., 2005). As k-mers of length $4$ are used, this gives a total of $4 \times 5 = 20$ features. One additional feature per 4-mer is added, which represents the relative frequency of this 4-mer with respect to its containing bag, resulting in 21 features per 4-mer. Two options for the relative frequency feature exist, which are (a) whether the frequency of the 4-mer ("4MER") or (b) the frequency of the sequence in which the 4-mer appeared ("TCRβ") is used. We optimized the learning rate, batch size, and early stopping parameter on the validation set. The settings of the full hyperparameter search as well as the respective value ranges are given in Table A9.

## A6 Hyperparameter selection

For all competing methods a hyperparameter search was performed, for which we split each of the 5 training sets into an inner training set and inner validation set. The models were trained on the inner training set and evaluated on the inner validation set. The model with the highest AUC score on the inner validation set is then used to calculate the score on the respective test set. Here we report the hyperparameter sets and search strategy that is used for all methods.

**DeepRC.** The set of hyperparameters of DeepRC is shown in Table A4. These hyperparameter combinations are adjusted via a grid search procedure.

| | |
|---|---|
| learning rate | $10^{-4}$ |
| number of kernels ($d_v$) | $\{8; 16; 32; 64^*; 128^*; 256^*\}$ |
| number of CNN layers | $\{1\}$ |
| number of layers in key-NN | $\{2\}$ |
| number of units in key-NN | $\{32\}$ |
| kernel size | $\{5; 7; 9\}$ |
| subsampled seqences | $10,000$ |
| batch size | $4$ |

Table A4: **DeepRC hyperparameter search space.** We apply early stopping, where the model with the best loss on the validation fold after $5 \cdot 10^5$ updates was selected. For this, the model was evaluated against the validation fold every $5 \cdot 10^3$ updates during training. *) Experiments for $\{64; 128; 256\}$ kernels were omitted for simulated datasets with motif implantation probabilities $> 0.1\%$.

**Known motif.** This method does not have hyperparameters and has been applied to all datasets except for the *CMV dataset*.

**SVM.** The corresponding hyperparameter $C$ of the SVM is optimized by randomly drawing $10^3$ values in the range of $[-6; 6]$ according to a uniform distribution. These values act as the exponents of a power of 10 and are applied for each of the two kernel types (see Table A5).

| | |
|---|---|
| $C$ | $10^{\{-6;6\}}$ |
| type of kernel | $\{MinMax; Jaccard\}$ |
| number of trials | $10^3$ |

Table A5: Settings used in the hyperparameter search of the SVM baseline approach. The number of trials defines the quantity of random values of the $C$ penalty term (per type of kernel).

**KNN.** The amount of neighbors is treated as the hyperparameter and optimized by grid search operating in the discrete range of $[1; \max\{N, 10^3\}]$ with a step size of 1. The corresponding tight upper bound is automatically defined by the total amount of samples $N \in \mathbb{N}_{>0}$ in the training set, capped at $10^3$ (see Table A6).

| | |
|---|---|
| number of neighbors | $\{1; \max\{N, 10^3\}\}$ |
| type of kernel | $\{MinMax; Jaccard\}$ |

Table A6: Settings used in the hyperparameter search of the KNN baseline approach. The number of trials (per type of kernel) is automatically defined by the total amount of samples $N \in \mathbb{N}_{>0}$ in the training set, capped at $10^3$.

**Logistic regression.** For this method, we applied a grid search over the hyperparameters listed in Table A7. We varied the learning rate and the strength of the weight decay.

| | |
|---|---|
| learning rate | $10^{-\{1;2;3;4\}}$ |
| batch size | 4 |
| max. updates | $10^5$ |
| coefficient $\beta_1$ (Adam) | 0.9 |
| coefficient $\beta_2$ (Adam) | 0.999 |
| $l1$ weight decay factor | $10^{-7}$ |
| $l2$ weight decay factor | $10^{-\{3;5\}}$ |

Table A7: Settings used in the hyperparameter search of the logistic regression method.

**Burden test.** The burden test selects two hyperparameters: the number of features in the burden set and the type of features, as listed in Table A8. Due to the lack of shared sequences between repertoires in category "simulated immunosequencing data", we omitted the sequence-based burden test for this category.

| | |
|---|---|
| number of features in burden set | $\{50, 100, 150, 250\}$ |
| type of features | $\{4\text{MER}; \text{sequence}^*\}$ |

Table A8: Settings used in the hyperparameter search of the burden test approach. *) Experiments for sequence features were omitted for datasets of category "simulated immunosequencing data" due to the lack of shared sequences between repertoires.

**Logistic MIL.** For this method, we adjusted the learning rate as well as the batch size as hyperparameters by randomly drawing 25 different hyperparameter combinations from a uniform distribution. The corresponding range of the learning rate is $[-4.5; -1.5]$, which acts as the exponent of a power of 10. The batch size lies within the range of $[1; 32]$. For each hyperparameter combination, a model is optimized by gradient descent using Adam, whereas the early stopping parameter is adjusted according to the corresponding validation set (see Table A9).

| | |
|---|---|
| learning rate | $10^{\{-4.5;-1.5\}}$ |
| batch size | $\{1; 32\}$ |
| relative abundance term | $\{4\text{MER}; \text{TCR}\beta\}$ |
| number of trials | 25 |
| max. epochs | $10^2$ |
| coefficient $\beta_1$ (Adam) | 0.9 |
| coefficient $\beta_2$ (Adam) | 0.999 |

Table A9: Settings used in the hyperparameter search of the logistic MIL baseline approach. The number of trials (per type of relative abundance) defines the quantity of combinations of random values of the learning rate as well as the batch size.

## A7 Results

In this section, we report the detailed results on all four categories of datasets (a) simulated immunose-quencing data (Table A10) (b) LSTM-generated data (Table A11), (c) real-world data with implanted signals (Table A12), and (d) real-world data on the *CMV dataset* (Table A13), as discussed in the main paper.

| ID | 0 | 1 | 2 | 3 | 4 | 5 | 6 | 7 | 8 | 9 | 10 | 11 | 12 | 13 | 14 | 15 | 16 | 17 | 18 | 19 | 20 | avg. |
|---|---|---|---|---|---|---|---|---|---|---|---|---|---|---|---|---|---|---|---|---|---|---|
| motif freq. $\rho$ | 1% | 0.1% | 0.01% | 1% | 0.1% | 0.01% | 1% | 0.1% | 0.01% | 1% | 0.1% | 0.01% | 1% | 0.1% | 0.01% | 1% | 0.1% | 0.01% | 0.1% | 0.1% | 0.1% | — |
| implanted motif | SFEN | SFEN | SFEN | $SF^{d}EN$ | $SF^{d}EN$ | $SF^{d}EN$ | SFZN | SFZN | SFZN | $SF^{d}ZN$ | $SF^{d}ZN$ | $SF^{d}ZN$ | SZZN | SZZN | SZZN | $SZ^{d}ZN$ | $SZ^{d}ZN$ | $SZ^{d}ZN$ | $S^{r}F^{r}E^{r}N^{r}$ | $S^{r}F^{r}ZN^{r}$ | $S^{r}ZZN^{r}$ | — |
| DeepRC | **1.000** ±0.000 | **1.000** ±0.000 | 0.703 ±0.271 | **1.000** ±0.000 | **1.000** ±0.000 | 0.600 ±0.218 | **1.000** ±0.000 | **1.000** ±0.000 | 0.509 ±0.029 | **1.000** ±0.000 | **1.000** ±0.001 | 0.492 ±0.017 | **1.000** ±0.001 | **0.997** ±0.002 | 0.487 ±0.023 | **0.999** ±0.001 | **0.942** ±0.048 | 0.492 ±0.013 | **1.000** ±0.000 | **1.000** ±0.000 | **0.947** ±0.028 | **0.865** ±0.211 |
| SVM (MinMax) | **1.000** ±0.000 | **1.000** ±0.000 | 0.764 ±0.016 | **1.000** ±0.000 | **1.000** ±0.000 | 0.603 ±0.021 | **1.000** ±0.000 | 0.998 ±0.002 | **0.539** ±0.024 | **1.000** ±0.000 | 0.994 ±0.004 | **0.529** ±0.016 | **1.000** ±0.000 | 0.741 ±0.024 | **0.513** ±0.006 | **1.000** ±0.000 | 0.706 ±0.013 | 0.503 ±0.013 | **1.000** ±0.000 | 0.941 ±0.004 | 0.640 ±0.041 | 0.832 ±0.203 |
| SVM (Jaccard) | 0.783 ±0.010 | 0.505 ±0.009 | 0.500 ±0.010 | 0.656 ±0.009 | 0.504 ±0.018 | 0.492 ±0.018 | 0.629 ±0.011 | 0.499 ±0.010 | 0.505 ±0.009 | 0.594 ±0.007 | 0.508 ±0.017 | 0.497 ±0.013 | 0.620 ±0.007 | 0.496 ±0.006 | 0.506 ±0.019 | 0.595 ±0.013 | 0.507 ±0.012 | 0.505 ±0.017 | 0.508 ±0.016 | 0.501 ±0.015 | 0.503 ±0.013 | 0.543 ±0.076 |
| KNN (MinMax) | 0.669 ±0.204 | 0.802 ±0.265 | 0.503 ±0.038 | 0.722 ±0.214 | 0.757 ±0.255 | 0.493 ±0.017 | 0.766 ±0.241 | 0.678 ±0.165 | 0.496 ±0.014 | 0.762 ±0.237 | 0.652 ±0.139 | 0.489 ±0.015 | 0.797 ±0.271 | 0.512 ±0.023 | 0.498 ±0.014 | 0.796 ±0.270 | 0.511 ±0.037 | 0.503 ±0.006 | 0.743 ±0.178 | 0.568 ±0.069 | 0.501 ±0.026 | 0.629 ±0.126 |
| KNN (Jaccard) | 0.516 ±0.035 | 0.493 ±0.020 | 0.497 ±0.013 | 0.506 ±0.015 | 0.500 ±0.019 | 0.492 ±0.014 | 0.509 ±0.017 | 0.493 ±0.011 | 0.497 ±0.018 | 0.495 ±0.013 | 0.504 ±0.004 | 0.500 ±0.017 | 0.502 ±0.011 | 0.497 ±0.017 | 0.500 ±0.022 | 0.502 ±0.015 | 0.503 ±0.020 | **0.513** ±0.012 | 0.498 ±0.021 | 0.506 ±0.019 | 0.491 ±0.019 | 0.501 ±0.007 |
| Logistic Regression | **1.000** ±0.000 | **1.000** ±0.000 | **0.797** ±0.026 | **1.000** ±0.000 | **1.000** ±0.000 | **0.614** ±0.011 | **1.000** ±0.000 | 0.997 ±0.002 | **0.539** ±0.010 | **1.000** ±0.000 | 0.994 ±0.004 | 0.523 ±0.015 | **1.000** ±0.000 | 0.723 ±0.020 | 0.507 ±0.020 | **1.000** ±0.000 | 0.701 ±0.007 | 0.505 ±0.027 | **1.000** ±0.001 | 0.946 ±0.004 | 0.632 ±0.004 | 0.832 ±0.204 |
| Logistic MIL (KMER) | **1.000** ±0.000 | **1.000** ±0.000 | 0.509 ±0.039 | **1.000** ±0.000 | 0.476 ±0.216 | 0.489 ±0.023 | **1.000** ±0.000 | 0.544 ±0.038 | 0.517 ±0.018 | **1.000** ±0.000 | 0.529 ±0.043 | 0.483 ±0.007 | 0.579 ±0.042 | 0.498 ±0.017 | 0.502 ±0.018 | 0.550 ±0.051 | 0.488 ±0.009 | 0.498 ±0.005 | 0.867 ±0.206 | 0.554 ±0.009 | 0.509 ±0.015 | 0.662 ±0.216 |
| Logistic MIL (TCRβ) | 0.544 ±0.078 | 0.505 ±0.014 | 0.493 ±0.018 | 0.487 ±0.021 | 0.510 ±0.019 | 0.500 ±0.022 | 0.520 ±0.053 | 0.495 ±0.009 | 0.510 ±0.022 | 0.492 ±0.014 | 0.506 ±0.019 | 0.503 ±0.010 | 0.509 ±0.034 | 0.505 ±0.009 | 0.500 ±0.011 | 0.475 ±0.013 | 0.489 ±0.024 | 0.500 ±0.019 | 0.498 ±0.009 | 0.501 ±0.017 | 0.504 ±0.017 | 0.501 ±0.015 |
| Burden test | 0.770 ±0.013 | 0.523 ±0.013 | 0.510 ±0.014 | 0.666 ±0.011 | 0.510 ±0.009 | 0.509 ±0.007 | 0.652 ±0.008 | 0.508 ±0.011 | 0.505 ±0.012 | 0.583 ±0.012 | 0.508 ±0.007 | 0.509 ±0.014 | 0.564 ±0.017 | 0.508 ±0.010 | 0.507 ±0.020 | 0.536 ±0.012 | 0.508 ±0.016 | 0.504 ±0.016 | 0.510 ±0.010 | 0.507 ±0.012 | 0.504 ±0.013 | 0.543 ±0.070 |
| Known motif b. | 1.000 ±0.000 | 1.000 ±0.000 | 0.973 ±0.004 | 1.000 ±0.000 | 1.000 ±0.000 | 0.865 ±0.004 | 1.000 ±0.000 | 1.000 ±0.000 | 0.700 ±0.020 | 1.000 ±0.000 | 0.989 ±0.002 | 0.609 ±0.017 | 1.000 ±0.000 | 0.946 ±0.010 | 0.570 ±0.024 | 1.000 ±0.000 | 0.834 ±0.016 | 0.532 ±0.020 | 1.000 ±0.000 | 0.982 ±0.004 | 0.870 ±0.004 | 0.899 ±0.158 |
| Known motif c. | 0.999 ±0.001 | 0.720 ±0.014 | 0.529 ±0.020 | 0.999 ±0.001 | 0.698 ±0.013 | 0.534 ±0.017 | 0.999 ±0.001 | 0.694 ±0.012 | 0.532 ±0.012 | 1.000 ±0.001 | 0.696 ±0.018 | 0.527 ±0.018 | 0.997 ±0.002 | 0.666 ±0.010 | 0.520 ±0.009 | 0.998 ±0.002 | 0.668 ±0.012 | 0.509 ±0.013 | 0.685 ±0.018 | 0.657 ±0.007 | 0.635 ±0.007 | 0.727 ±0.189 |

Table A10: AUC estimates based on 5-fold CV for all 21 datasets in category "simulated immunosequencing data". The reported errors are standard deviations across the 5 cross-validation folds except for the last column "avg.", in which they show standard deviations across datasets. Wildcard characters in motifs are indicated by Z, characters with 50% probability of being removed by $^{d}$, characters with 20% probability of behaving like a wildcard Z by $^{r}$.

| ID | 0 | 1 | 2 | 3 | 4 | avg. |
|---|---|---|---|---|---|---|
| motif freq. $\rho$ | 10% | 1% | 0.5% | 0.1% | 0.05% | – |
| implanted motif | $G^rS^rA^rF^r$ | $G^rS^rA^rF^r$ | $G^rS^rA^rF^r$ | $G^rS^rA^rF^r$ | $G^rS^rA^rF^r$ | – |
| DeepRC | **1.000** $\pm$ 0.000 | **1.000** $\pm$ 0.000 | **1.000** $\pm$ 0.000 | **1.000** $\pm$ 0.000 | **0.998** $\pm$ 0.002 | **1.000** $\pm$ 0.001 |
| SVM (MinMax) | **1.000** $\pm$ 0.000 | **1.000** $\pm$ 0.000 | 0.999 $\pm$ 0.001 | 0.999 $\pm$ 0.002 | 0.985 $\pm$ 0.014 | 0.997 $\pm$ 0.007 |
| SVM (Jaccard) | 0.981 $\pm$ 0.041 | **1.000** $\pm$ 0.000 | **1.000** $\pm$ 0.000 | 0.904 $\pm$ 0.036 | 0.768 $\pm$ 0.068 | 0.931 $\pm$ 0.099 |
| KNN (MinMax) | 0.699 $\pm$ 0.272 | 0.717 $\pm$ 0.263 | 0.732 $\pm$ 0.263 | 0.536 $\pm$ 0.156 | 0.516 $\pm$ 0.153 | 0.640 $\pm$ 0.105 |
| KNN (Jaccard) | 0.698 $\pm$ 0.285 | 0.606 $\pm$ 0.237 | 0.523 $\pm$ 0.164 | 0.550 $\pm$ 0.186 | 0.539 $\pm$ 0.194 | 0.583 $\pm$ 0.071 |
| Logistic Regression | **1.000** $\pm$ 0.000 | **1.000** $\pm$ 0.000 | **1.000** $\pm$ 0.000 | 0.697 $\pm$ 0.164 | 0.466 $\pm$ 0.103 | 0.833 $\pm$ 0.243 |
| Logistic MIL (KMER) | 0.997 $\pm$ 0.004 | 0.718 $\pm$ 0.112 | 0.637 $\pm$ 0.144 | 0.571 $\pm$ 0.146 | 0.528 $\pm$ 0.129 | 0.690 $\pm$ 0.186 |
| Logistic MIL (TCR$\beta$) | 0.541 $\pm$ 0.086 | 0.566 $\pm$ 0.162 | 0.468 $\pm$ 0.086 | 0.505 $\pm$ 0.067 | 0.500 $\pm$ 0.121 | 0.516 $\pm$ 0.038 |
| Burden test | **1.000** $\pm$ 0.000 | **1.000** $\pm$ 0.000 | **1.000** $\pm$ 0.000 | 0.999 $\pm$ 0.003 | 0.792 $\pm$ 0.280 | 0.958 $\pm$ 0.093 |
| Known motif b. | 1.000 $\pm$ 0.000 | 1.000 $\pm$ 0.000 | 1.000 $\pm$ 0.000 | 0.999 $\pm$ 0.003 | 0.999 $\pm$ 0.003 | 1.000 $\pm$ 0.001 |
| Known motif c. | 1.000 $\pm$ 0.000 | 1.000 $\pm$ 0.000 | 0.989 $\pm$ 0.011 | 0.722 $\pm$ 0.085 | 0.626 $\pm$ 0.094 | 0.867 $\pm$ 0.180 |

Table A11: AUC estimates based on 5-fold CV for all 5 datasets in category "LSTM-generated data". The reported errors are standard deviations across the 5 cross-validation folds except for the last column "avg.", in which they show standard deviations across datasets. Characters affected by noise, as described in A4, paragraph "LSTM-generated data", are indicated by $^r$.

|  | s.m. 1% | s.m. 0.1% | m.m. 1% | m.m. 0.1% | Avg. |
|---|---|---|---|---|---|
| DeepRC | **1.000** ± 0.000 | **0.984** ± 0.008 | 0.999 ± 0.001 | **0.938** ± 0.009 | **0.980** ± 0.029 |
| SVM (MinMax) | **1.000** ± 0.000 | 0.578 ± 0.020 | **1.000** ± 0.000 | 0.531 ± 0.019 | 0.777 ± 0.258 |
| SVM (Jaccard) | 0.988 ± 0.003 | 0.527 ± 0.016 | **1.000** ± 0.000 | 0.574 ± 0.019 | 0.772 ± 0.257 |
| KNN (MinMax) | 0.744 ± 0.237 | 0.486 ± 0.031 | 0.674 ± 0.182 | 0.500 ± 0.022 | 0.601 ± 0.128 |
| KNN (Jaccard) | 0.652 ± 0.155 | 0.484 ± 0.025 | 0.695 ± 0.200 | 0.508 ± 0.025 | 0.585 ± 0.104 |
| Logistic Regression | **1.000** ± 0.000 | 0.585 ± 0.045 | **1.000** ± 0.000 | 0.512 ± 0.015 | 0.774 ± 0.262 |
| Logistic MIL (KMER) | 0.541 ± 0.074 | 0.506 ± 0.034 | 0.994 ± 0.004 | 0.620 ± 0.153 | 0.665 ± 0.224 |
| Logistic MIL (TCRβ) | 0.503 ± 0.032 | 0.501 ± 0.016 | 0.992 ± 0.003 | 0.782 ± 0.030 | 0.695 ± 0.238 |
| Burden test | **1.000** ± 0.000 | 0.640 ± 0.048 | **1.000** ± 0.000 | 0.891 ± 0.016 | 0.883 ± 0.170 |
| Known motif b. | 1.000 ± 0.000 | 0.704 ± 0.028 | 0.994 ± 0.003 | 0.620 ± 0.038 | 0.830 ± 0.196 |
| Known motif c. | 0.920 ± 0.004 | 0.562 ± 0.028 | 0.647 ± 0.030 | 0.515 ± 0.031 | 0.661 ± 0.181 |

Table A12: AUC estimates based on 5-fold CV for all 4 datasets in category "real-world data with implanted signals". The reported errors are standard deviations across the 5 cross-validation folds except for the last column "avg.", in which they show standard deviations across datasets. **s.m. 1%:** In this dataset, a single motif with a frequency of 1% was implanted. **s.m. 0.1%:** In this dataset, a single motif with a frequency of 0.1% was implanted. **m.m. 1%:** In this dataset, multiple motifs with a frequency of 1% were implanted. **m.m. 0.1%:** In this dataset, multiple motifs with a frequency of 0.1% were implanted. A detailed description of the motifs is provided in section A4, paragraph "Real-world data with implanted signals.".

|  | AUC | F1 score | Balanced accuracy | Accuracy |
|---|---|---|---|---|
| DeepRC | **0.832** ± 0.022 | **0.721** ± 0.030 | **0.734** ± 0.032 | 0.735 ± 0.037 |
| SVM (MinMax) | 0.825 ± 0.022 | 0.680 ± 0.056 | **0.734** ± 0.037 | **0.742** ± 0.031 |
| SVM (Jaccard) | 0.546 ± 0.021 | 0.272 ± 0.184 | 0.523 ± 0.026 | 0.542 ± 0.032 |
| KNN (MinMax) | 0.679 ± 0.076 | 0.000 ± 0.000 | 0.500 ± 0.000 | 0.545 ± 0.044 |
| KNN (Jaccard) | 0.534 ± 0.039 | 0.073 ± 0.101 | 0.508 ± 0.012 | 0.551 ± 0.042 |
| Logistic regression | 0.613 ± 0.044 | 0.405 ± 0.211 | 0.558 ± 0.046 | 0.577 ± 0.058 |
| Logistic MIL (KMER) | 0.582 ± 0.065 | 0.118 ± 0.264 | 0.503 ± 0.007 | 0.515 ± 0.058 |
| Logistic MIL (TCRβ) | 0.515 ± 0.073 | 0.000 ± 0.000 | 0.496 ± 0.008 | 0.541 ± 0.039 |
| Burden test | 0.699 ± 0.041 | - | - | - |

Table A13: Results on the *CMV dataset* (real-world data) in terms of AUC, F1 score, balanced accuracy, and accuracy. For F1 score, balanced accuracy, and accuracy, all methods use their default thresholds. Each entry shows mean and standard deviation across 5 cross-validation folds.

## A8    Repertoire generation via LSTM

We trained a conventional next-character LSTM model (Graves, 2013) based on the implementation in `https://github.com/spro/practical-pytorch` (access date 1st of May, 2020) using `PyTorch 1.3.1` (Paszke et al., 2019). For this, we applied an LSTM model with 100 LSTM blocks in 2 layers, which was trained for $5,000$ epochs using the Adam optimizer (Kingma & Ba, 2014) with learning rate $0.01$, an input batch size of $100$ character chunks, and a character chunk length of $200$. As input we used the immuno-sequences in the `CDR3` column of the *CMV dataset*, where we repeated sequences according to their counts in the repertoires, as specified in the `templates` column of the *CMV dataset*. We excluded repertoires with unknown CMV status and unknown sequence abundance from training.

After training, we generated $1,000$ repertoires using a `temperature` value of $0.8$. The number of sequences per repertoire was sampled from a Gaussian $\mathcal{N}(\mu = 285k, \sigma = 156k)$ distribution, where the whole repertoire was generated by the LSTM at once. That is, the LSTM can base the generation of the individual AA sequences in a repertoire, including the AAs and the lengths of the sequences, on the generated repertoire. A random immuno-sequence from the trained-on repertoires was used as initialization for the generation process. This immuno-sequence was not included in the generated repertoire.

Finally, we randomly assigned 500 of the generated repertoires to the positive (diseased) and 500 to the negative (healthy) class. We then implanted motifs in the positive class repertoires as described in section A4.2.

As illustrated in the comparison of histograms given in Fig. A2, the generated immuno-sequences exhibit a very similar distribution of 4-mers and AAs compared to the original *CMV dataset*.

**Real-world data**

**LSTM-generated data**

**a)**

**b)**

**c)**

**d)**

**e)**

**f)**

Figure A2: Distribution of AAs and k-mers in real-world *CMV dataset* and LSTM-generated data.
**Left:** Histograms of real-world data. **Right:** Histograms of LSTM-generated data. **a)** Frequency of
AAs in sequences of the *CMV dataset*. **b)** Frequency of AAs in sequences of the LSTM-generated
datasets. **c)** Frequency of top 200 4-mers in sequences of the *CMV dataset*. **d)** Frequency of top 200
4-mers in sequences of the LSTM-generated datasets. **e)** Frequency of top 20 4-mers in sequences
of the *CMV dataset*. **f)** Frequency of top 20 4-mers in sequences of the LSTM-generated datasets.
Overall the distributions of AAs and 4-mers are similar in both datasets.

## A9 Interpreting DeepRC

DeepRC allows for two forms of interpretability methods. (a) Due to its attention-based design, a trained model can be used to compute the attention weights of a sequence, which directly indicates its importance. (b) DeepRC furthermore allows for the usage of contribution analysis methods, such as Integrated Gradients (IG) (Sundararajan et al., 2017) or Layer-Wise Relevance Propagation (Montavon et al., 2018; Arras et al., 2019; Montavon et al., 2019; Preuer et al., 2019). We apply IG to identify the input patterns that are relevant for the classification. To identify AA patterns with high contributions in the input sequences, we apply IG to the AAs in the input sequences. Additionally, we apply IG to the kernels of the 1D CNN, which allows us to identify AA motifs with high contributions. In detail, we compute the IG contributions for the AAs and positional features in the kernels for every repertoire in the validation and test set, so as to exclude potential artifacts caused by over-fitting. Averaging the IG values over these repertoires then results in concise AA motifs. We include qualitative visual analyses of the IG method on different datasets below.

Here, we provide examples for the interpretation of trained DeepRC models using Integrated Gradients (IG) (Sundararajan et al., 2017) as contribution analysis method. The following illustrations were created using 50 IG steps, which we found sufficient to achieve stable IG results.

A visual analysis of DeepRC models on the simulated datasets, as illustrated in Tab. A14 and Fig. A3, shows that the implanted motifs can be successfully extracted from the trained model and are straightforward to interpret. In the real-world *CMV dataset*, DeepRC finds complex patterns with high variability in the center regions of the immuno-sequences, as illustrated in figure A4.

| | **Simulated** | | | |
|---|---|---|---|---|
| extracted motif | S F E N | S F E N | S ^ ^ N | S s N |
| implanted motif(s) | SFEN | SF$^d$EN | SZZN | SZ$^d$ZN |
| motif freq. $\rho$ | 0.01% | 0.01% | 0.1% | 0.1% |

| | **LSTM-generated** | **Real-world data with implanted signals** | |
|---|---|---|---|
| extracted motif | GSAF | L D R > | CAS      C$_{A\ S}$ |
| implanted motif(s) | G$^r$S$^r$A$^r$F$^r$ | L$^r$D$^r$R$^r$ | {L$^r$D$^r$R$^r$; C$^r$A$^r$S; G$^r$L-N} |
| motif freq. $\rho$ | 0.05% | 0.1% | 0.1% |

Table A14: Visualization of motifs extracted from trained DeepRC models for datasets from categories "simulated immunosequencing data", "LSTM-generated data", and "real-world data with implanted signals". Motif extraction was performed using Integrated Gradients on the 1D CNN kernels over the validation set and test set repertoires of one CV fold. Wildcard characters are indicated by Z, random noise on characters by $^r$, characters with 50% probability of being removed by $^d$, and gap locations of random lengths of $\{0; 1; 2\}$ by -. Larger characters in the extracted motifs indicate higher contribution, with blue indicating positive contribution and red indicating negative contribution towards the prediction of the diseased class. Contributions to positional encoding are indicated by $<$ (beginning of sequence), $\wedge$ (center of sequence), and $>$ (end of sequence). Only kernels with relatively high contributions are shown, i.e. with contributions roughly greater than the average contribution of all kernels.

Figure A3: Integrated Gradients applied to input sequences of positive class repertoires. Three sequences with the highest contributions to the prediction of their respective repertoires are shown. **a)** Input sequence taken from "simulated immunosequencing data" with implanted motif $SZ^dZ^dN$ and motif implantation probability $0.1\%$. The DeepRC model reacts to the S and N at the $5^{th}$ and $8^{th}$ sequence position, thereby identifying the implanted motif in this sequence. **b)** and **c)** Input sequence taken from "real-world data with implanted signals" with implanted motifs {$L^rD^rR^r$; $C^rA^rS$; $G^rL$-N} and motif implantation probability $0.1\%$. The DeepRC model reacts to the fully implanted motif CAS (b) and to the partly implanted motif AAs C and A at the $5^{th}$ and $7^{th}$ sequence position (c), thereby identifying the implanted motif in the sequences. Wildcard characters in implanted motifs are indicated by Z, characters with $50\%$ probability of being removed by $^d$, and gap locations of random lengths of {$0; 1; 2$} by -. Larger characters in the sequences indicate higher contribution, with blue indicating positive contribution and red indicating negative contribution towards the prediction of the diseased class.

Figure A4: Visualization of the contributions of characters within a sequence via IG. Each sequence was selected from a different repertoire and showed the highest contribution in its repertoire. The model was trained on *CMV dataset*, using a kernel size of 9, 32 kernels and 137 repertoires for early stopping. Larger characters in the extracted motifs indicate higher contribution, with blue indicating positive contribution and red indicating negative contribution towards the prediction of the disease class.

# A10 Attention values for previously associated CMV sequences

| index | sequence | attention | quantile | index | sequence | attention | quantile | index | sequence | attention | quantile | index | sequence | attention | quantile |
|---|---|---|---|---|---|---|---|---|---|---|---|---|---|---|---|
| 1 | CASSGQGAYEQYF | 1.000 | 0.999 | 42 | CASSLGGAGDTQYF | 1.000 | 1.000 | 83 | CASSYVRTGGNYGYTF | 0.967 | 0.932 | 124 | CASSLTGGNSGNTIYF | 0.991 | 0.977 |
| 2 | CASSIGPLEHNEQFF | 0.947 | 0.900 | 43 | CASNRDRGRYEQYF | 0.991 | 0.978 | 84 | CASSLAGVDYEQYF | 0.999 | 0.996 | 125 | CASSRNRGQETQYF | 0.978 | 0.952 |
| 3 | CASSPDRVGQETQYF | 0.995 | 0.987 | 44 | CSVRDNHNQPQHF | 0.965 | 0.929 | 85 | CASSLGAGNQPQHF | 1.000 | 0.999 | 126 | CASSLGQGLAEAFF | 0.996 | 0.989 |
| 4 | CASSLEAEYEQYF | 0.992 | 0.980 | 45 | CASSAQGAYEQYF | 0.998 | 0.995 | 86 | CASSRDRNYGYTF | 0.998 | 0.995 | 127 | CASRTGESGYTF | 0.985 | 0.965 |
| 5 | CASSIEGNQPQHF | 0.993 | 0.983 | 46 | CATSRGTVSYEQYF | 0.990 | 0.975 | 87 | CASGRDTYEQYF | 0.999 | 0.997 | 128 | CASSSDSGGTDTQYF | 0.951 | 0.906 |
| 6 | CATSDGDEQFF | 0.998 | 0.996 | 47 | CASSPPSGLTDTQYF | 0.978 | 0.951 | 88 | CAWSVSDLAKNIQYF | 0.954 | 0.911 | 129 | CASSVDGGRGTEAFF | 0.995 | 0.987 |
| 7 | CASSLVAGGRETQYF | 0.988 | 0.971 | 48 | CASSGDRLYEQYF | 0.998 | 0.994 | 89 | CASSPNQETQYF | 0.999 | 0.996 | 130 | CSVEVRGTDTQYF | 0.955 | 0.912 |
| 8 | CASSRGRQETQYF | 0.997 | 0.993 | 49 | CASSLNRGQETQYF | 0.996 | 0.988 | 90 | CSASDHEQYF | 0.995 | 0.986 | 131 | CASSESGDPSSYEQYF | 0.980 | 0.955 |
| 9 | CASSAGQGVTYEQYF | 0.998 | 0.995 | 50 | CASSLGVGPYNEQFF | 0.986 | 0.967 | 91 | CASSWDRDNSPLHF | 0.918 | 0.855 | 132 | CASSEEAGGSGYTF | 0.982 | 0.959 |
| 10 | CASSQNRGQETQYF | 0.995 | 0.987 | 51 | CATSDSVTNTGELFF | 0.989 | 0.973 | 92 | CASSPGQEAGANVLTF | 0.823 | 0.728 | 133 | CAISESQDRGHEQYF | 0.823 | 0.728 |
| 11 | CASSPQRNTEAFF | 1.000 | 0.999 | 52 | CASSRNRESNQPQHF | 0.968 | 0.934 | 93 | CASSLVAAGRETQYF | 0.959 | 0.919 | 134 | CASSPTGGELFF | 0.989 | 0.974 |
| 12 | CASSLAPGATNEKLFF | 0.976 | 0.949 | 53 | CASSEARTRAFF | 0.927 | 0.869 | 94 | CASSPHRNTEAFF | 0.999 | 0.998 | 135 | CASSVETGGTEAFF | 0.995 | 0.986 |
| 13 | CASSLIGVSSYNEQFF | 0.983 | 0.961 | 54 | CASSYNPYSNQPQHF | 0.892 | 0.819 | 95 | CASRGQGWDEKLFF | 0.994 | 0.984 | 136 | CASASANYGYTF | 0.816 | 0.720 |
| 14 | CSVRDNFNQPQHF | 0.915 | 0.851 | 55 | CASSLGHRDSSYEQYF | 0.987 | 0.969 | 96 | CASSQVETDTQYF | 0.994 | 0.984 | 137 | CASSSRTGEETQYF | 0.996 | 0.988 |
| 15 | CASSQTGGRNQPQHF | 0.997 | 0.992 | 56 | CASSRLAASTDTQYF | 0.992 | 0.979 | 97 | CASRDWDYTDTQYF | 0.994 | 0.984 | 138 | CASSLGRGYEKLFF | 0.985 | 0.965 |
| 16 | CASSLVIGGDTEAFF | 0.966 | 0.931 | 57 | CASSVTGGTDTQYF | 1.000 | 0.999 | 98 | CASSSDRVGQETQYF | 0.980 | 0.955 | 139 | CASSGLNEQFF | 0.994 | 0.984 |
| 17 | CASSLRREKLFF | 0.998 | 0.993 | 58 | CASSPPGQGSDTQYF | 0.975 | 0.946 | 99 | CASSLGDRPDTQYF | 0.940 | 0.889 | 140 | CASSRNRAQETQYF | 0.994 | 0.984 |
| 18 | CASSFHGFNQPQHF | 0.991 | 0.978 | 59 | CATSDSRTGGQETQYF | 0.900 | 0.829 | 100 | CASSLEGQGFGYTF | 0.944 | 0.895 | 141 | CASTPGDEQFF | 0.988 | 0.971 |
| 19 | CATSRDTQGSYGYTF | 0.917 | 0.854 | 60 | CASSSPGRSGANVLTF | 0.995 | 0.986 | 101 | CASSSGQVYGYTF | 0.999 | 0.996 | 142 | CASSLGIDTQYF | 0.997 | 0.991 |
| 20 | CASSRLAGGTDTQYF | 0.999 | 0.998 | 61 | CASSPLSDTQYF | 0.998 | 0.994 | 102 | CASSEEGIQPQHF | 0.998 | 0.994 | 143 | CASSIRTNYYGYTF | 0.996 | 0.990 |
| 21 | CASSFPTSGQETQYF | 0.982 | 0.959 | 62 | CASSLTGGRNQPQHF | 0.999 | 0.997 | 103 | CASSLETYGYTF | 0.998 | 0.995 | 144 | CASSPISNEQFF | 0.967 | 0.933 |
| 22 | CASSPGDEQYF | 0.998 | 0.993 | 63 | CASSIQGYSNQPQHF | 0.993 | 0.983 | 104 | CASSFPGGETQYF | 0.992 | 0.979 | 145 | CASSQNRAQETQYF | 0.984 | 0.962 |
| 23 | CASSLPSGLTDTQYF | 0.994 | 0.985 | 64 | CASSTTGGDGYTF | 0.978 | 0.952 | 105 | CASSSGQVQETQYF | 0.997 | 0.993 | 146 | CASSALGGAGTGELFF | 0.985 | 0.964 |
| 24 | CASSEIPNTEAFF | 0.997 | 0.992 | 65 | CASSVLAGPTDTQYF | 0.951 | 0.906 | 106 | CASSEGARQPQHF | 0.999 | 0.998 | 147 | CASSLAVLPTDTQYF | 0.996 | 0.989 |
| 25 | CASSIWGLDTEAFF | 0.959 | 0.919 | 66 | CASSHRDRNYEQYF | 0.987 | 0.969 | 107 | CSALGHSNQPQHF | 0.926 | 0.867 | 148 | CASSLQAGANEQFF | 0.969 | 0.935 |
| 26 | CASSPGDEQFF | 0.999 | 0.997 | 67 | CASSPSRNTEAFF | 0.999 | 0.998 | 108 | CASSLLWDQPQHF | 0.986 | 0.967 | 149 | CASSTGGAQPQHF | 0.998 | 0.993 |
| 27 | CATSRDSQGSYGYTF | 0.980 | 0.955 | 68 | CASSLGGPGDTQYF | 0.993 | 0.982 | 109 | CASSLVGDGYTF | 1.000 | 1.000 | 150 | CASSLGASGSRTDTQYF | 0.932 | 0.876 |
| 28 | CASSYGGLGSYEQYF | 0.995 | 0.987 | 69 | CASSEARGGVEKLFF | 0.989 | 0.974 | 110 | CASSSRGTGELFF | 0.999 | 0.997 | 151 | CASSRGTGATDTQYF | 0.999 | 0.998 |
| 29 | CASSPSTGTEAFF | 0.997 | 0.992 | 70 | CASSTGTSGSYEQYF | 0.999 | 0.998 | 111 | CATSRVAGETQYF | 0.980 | 0.955 | 152 | CASSYPGETQYF | 0.997 | 0.992 |
| 30 | CSVEEDEGIYGYTF | 0.964 | 0.927 | 71 | CASRSDSGANVLTF | 0.973 | 0.942 | 112 | CASRGQGAGELFF | 0.987 | 0.969 | 153 | CASSLTDTGELFF | 0.994 | 0.984 |
| 31 | CASSPAGLNTEAFF | 0.996 | 0.988 | 72 | CASSLEAENEQFF | 0.973 | 0.943 | 113 | CASSPGGTQYF | 0.999 | 0.996 | 154 | CASRPQGNYGYTF | 0.998 | 0.996 |
| 32 | CASSLGLKGTQYF | 0.964 | 0.928 | 73 | CASSEAPSTSTDTQYF | 0.989 | 0.973 | 114 | CASSLQGINQPQHF | 0.999 | 0.997 | 155 | CASSTSGNTIYF | 1.000 | 0.999 |
| 33 | CASMGGASYEQYF | 0.991 | 0.978 | 74 | CASSLQGADTQYF | 0.997 | 0.991 | 115 | CASSQGRHTDTQYF | 0.960 | 0.921 | 156 | CASSSGTGDEQYF | 1.000 | 1.000 |
| 34 | CASSQVPGQGDNEQFF | 0.983 | 0.961 | 75 | CASSLEGQQPQHF | 0.994 | 0.984 | 116 | CASSPRWQETQYF | 0.991 | 0.978 | 157 | CASSPPAGTNYGYTF | 0.947 | 0.900 |
| 35 | CATSDGDTQYF | 0.996 | 0.989 | 76 | CASSYGGEGYTF | 0.999 | 0.996 | 117 | CASRDRDRVNTEAFF | 0.970 | 0.938 | 158 | CASSPLGGTTEAFF | 0.995 | 0.988 |
| 36 | CATSDGETQYF | 0.998 | 0.994 | 77 | CASSLRGSSYNEQFF | 0.999 | 0.998 | 118 | CASSWDRGTEAFF | 0.999 | 0.999 | 159 | CASSLGWTEAFF | 0.999 | 0.997 |
| 37 | CSVRDNYNQPQHF | 0.998 | 0.993 | 78 | CASSISAGEAFF | 0.992 | 0.979 | 119 | CASSRPGQGNTEAFF | 0.994 | 0.984 | 160 | CATSREGSGYEQYF | 0.987 | 0.969 |
| 38 | CASSLVASGRETQYF | 0.997 | 0.991 | 79 | CASRPTGYEQYF | 0.987 | 0.969 | 120 | CASSPGSGANVLTF | 0.999 | 0.997 | 161 | CASSYAGDGYTF | 0.992 | 0.980 |
| 39 | CSASPGQGASYGYTF | 0.987 | 0.969 | 80 | CAWRGTGNSPLHF | 0.964 | 0.927 | 121 | CASRRGSSYEQYF | 0.999 | 0.998 | 162 | CASSDRGNTGELFF | 0.995 | 0.986 |
| 40 | CASSESGHRNQPQHF | 0.999 | 0.997 | 81 | CASSLGDRAYNEQFF | 0.996 | 0.988 | 122 | CASRTDSGANVLTF | 0.994 | 0.986 | 163 | CSARRGPGELFF | 0.839 | 0.749 |
| 41 | CASSLGHRDPNTGELFF | 0.981 | 0.958 | 82 | CASSLQGYSNQPQHF | 1.000 | 0.999 | 123 | CASSQDPRGTEAFF | 0.950 | 0.905 | 164 | CASSQGLQETQYF | 0.996 | 0.990 |

Table A15: TCRβ sequences that had been discovered by Emerson et al. (2017) with their associated attention values by DeepRC. These sequences have significantly ($p$-value 1.3e-93) higher attention values than other sequences. The column "quantile" provides the quantile values of the empiricial distribution of attention values across all sequences in the dataset.

## A11  DeepRC variations and ablation study

In this section we investigate the impact of different variations of DeepRC on the performance on the *CMV dataset*. We consider both a CNN-based sequence embedding, as used in the main paper, and an LSTM-based sequence embedding. In both cases we vary the number of attention heads and the $\beta$ parameter for the softmax function the attention mechanism (see Eq. 2 in main paper). For the CNN-based sequence embedding we also vary the number of CNN kernels and the kernel sizes used in the 1D CNN. For the LSTM-based sequence embedding we use one one-directional LSTM layer, where the output values at the last sequence position (without padding) are taken as embedding of the sequence. Here we vary the number of LSTM blocks in the LSTM layer. To counter over-fitting due to the increased complexity of these DeepRC variations, we added a $l2$ weight penalty to the training loss. The factor with which the $l2$ weight penalty contributes to the training loss is varied over 3 orders of magnitudes, where suitable value ranges were manually determined on one of the training folds beforehand.

To reduce the computational cost, we do not consider all numbers of kernels that were considered in the main paper. Furthermore, we only compute the AUC scores on 3 of the 5 cross-validation folds. The hyperparameters, which were used in a grid search procedure, are listed in Tab. A16 for the CNN-based sequence embedding and Tab. A17 for the LSTM-based sequence embedding.

**Results.**  We show performance in terms of AUC score with single hyperparameters set to fixed values so as to investigate their influence in Tab. A19 for the CNN-based sequence embedding and Tab. A18 for the LSTM-based sequence embedding. We note that due to restricted computational resources this study was conducted with fewer different numbers of CNN kernels, with the AUC estimated from only 3 of the 5 cross-validation folds, which leads to a slight decrease of performance in comparison to the full hyperparameter search and cross-validation procedure used in the main paper. As can be seen in Tab. A19 and A18, the LSTM-based sequence embedding generalizes slightly better than the CNN-based sequence embedding. The performance of DeepRC, however, remains rather robust w.r.t. the different hyperparameter settings.

| | |
|---|---|
| learning rate | $10^{-4}$ |
| number of attention heads | $\{1; 16; 64\}$ |
| $\beta$ of attention softmax | $\{0.1; 1.0; 10.0\}$ |
| $l2$ weight penalty | $\{1.0; 0.1; 0.01\}$ |
| number of kernels | $\{8; 32; 128\}$ |
| number of CNN layers | $\{1\}$ |
| number of layers in key-NN | $\{2\}$ |
| number of units in key-NN | $\{32\}$ |
| kernel size | $\{5; 7; 9\}$ |
| subsampled seqences | $10,000$ |
| batch size | $4$ |

Table A16: Hyperparameter search space for DeepRC variations with CNN-based sequence embedding. Every $5 \cdot 10^3$ updates, the current model was evaluated against the validation fold. The early stopping hyperparameter was determined by selecting the model with the best loss on the validation fold after $2 \cdot 10^5$ updates.

| | |
|---|---|
| learning rate | $10^{-4}$ |
| number of attention heads | $\{1; 16; 64\}$ |
| $\beta$ of attention softmax | $\{0.1; 1.0; 10.0\}$ |
| $l2$ weight penalty | $\{0.01; 0.001; 0.0001\}$ |
| number of LSTM blocks | $\{8; 32; 128\}$ |
| number of CNN layers | $\{1\}$ |
| number of layers in key-NN | $\{2\}$ |
| number of units in key-NN | $\{32\}$ |
| subsampled seqences | $10,000$ |
| batch size | 4 |

Table A17: Hyperparameter search space for DeepRC variations with LSTM-based sequence embedding. Every $5 \cdot 10^3$ updates, the current model was evaluated against the validation fold. The early stopping hyperparameter was determined by selecting the model with the best loss on the validation fold after $2 \cdot 10^5$ updates.

| Fixed parameter | Test set | | Validation set | | Training set | |
|---|---|---|---|---|---|---|
| | mean | std | mean | std | mean | std |
| beta=0.1 | 0.827 $\pm$ | 0.02 | 0.846 $\pm$ | 0.033 | 0.976 $\pm$ | 0.015 |
| beta=1.0 | 0.82 $\pm$ | 0.012 | 0.853 $\pm$ | 0.031 | 0.979 $\pm$ | 0.016 |
| beta=10.0 | 0.823 $\pm$ | 0.014 | 0.858 $\pm$ | 0.033 | 0.934 $\pm$ | 0.026 |
| heads=1 | 0.838 $\pm$ | 0.033 | 0.856 $\pm$ | 0.029 | 0.966 $\pm$ | 0.012 |
| heads=16 | 0.817 $\pm$ | 0.015 | 0.853 $\pm$ | 0.028 | 0.972 $\pm$ | 0.026 |
| heads=64 | 0.823 $\pm$ | 0.014 | 0.858 $\pm$ | 0.033 | 0.934 $\pm$ | 0.026 |
| lstms=8 | 0.818 $\pm$ | 0.011 | 0.837 $\pm$ | 0.025 | 0.881 $\pm$ | 0.013 |
| lstms=32 | 0.814 $\pm$ | 0.015 | 0.853 $\pm$ | 0.029 | 0.948 $\pm$ | 0.033 |
| lstms=128 | 0.818 $\pm$ | 0.018 | 0.859 $\pm$ | 0.032 | 0.943 $\pm$ | 0.028 |

Table A18: Impact of hyperparameters on DeepRC with LSTM for sequence encoding. Mean ("mean") and standard deviation ("std") for the area under the ROC curve over the first 3 folds of a 5-fold nested cross-validation for different sub-sets of hyperparameters ("sub-set") are shown. The following sub-sets were considered: "full": Full grid search over hyperparameters; "beta=*": Grid search over hyperparameters with reduction to specific value $*$ of beta value of attention softmax; "heads=*": Grid search over hyperparameters with reduction to specific number $*$ of attention heads; "lstms=*": Grid search over hyperparameters with reduction to specific number $*$ of LSTM blocks for sequence embedding.

| Fixed parameter | Test set | | Validation set | | Training set | |
|---|---|---|---|---|---|---|
| | mean | std | mean | std | mean | std |
| beta=0.1 | 0.833 ± | 0.031 | 0.86 ± | 0.025 | 0.94 ± | 0.018 |
| beta=1.0 | 0.799 ± | 0.007 | 0.873 ± | 0.017 | 0.954 ± | 0.005 |
| beta=10.0 | 0.817 ± | 0.02 | 0.87 ± | 0.022 | 0.962 ± | 0.034 |
| heads=1 | 0.822 ± | 0.036 | 0.869 ± | 0.022 | 0.943 ± | 0.032 |
| heads=16 | 0.808 ± | 0.01 | 0.871 ± | 0.025 | 0.965 ± | 0.019 |
| heads=64 | 0.796 ± | 0.039 | 0.864 ± | 0.018 | 0.927 ± | 0.024 |
| ksize=5 | 0.822 ± | 0.036 | 0.866 ± | 0.021 | 0.926 ± | 0.026 |
| ksize=7 | 0.817 ± | 0.02 | 0.87 ± | 0.022 | 0.962 ± | 0.034 |
| ksize=9 | 0.821 ± | 0.016 | 0.869 ± | 0.025 | 0.95 ± | 0.031 |
| kernels=8 | 0.825 ± | 0.024 | 0.86 ± | 0.027 | 0.928 ± | 0.019 |
| kernels=32 | 0.801 ± | 0.001 | 0.877 ± | 0.018 | 0.974 ± | 0.017 |
| kernels=128 | 0.824 ± | 0.027 | 0.864 ± | 0.023 | 0.931 ± | 0.062 |

Table A19: Impact of hyperparameters on DeepRC with 1D CNN for sequence encoding. Mean ("mean") and standard deviation ("std") for the area under the ROC curve over the first 3 folds of a 5-fold nested cross-validation for different sub-sets of hyperparameters ("sub-set") are shown. The following sub-sets were considered: "full": Full grid search over hyperparameters; "beta=*": Grid search over hyperparameters with reduction to specific value ∗ of beta value of attention softmax; "heads=*": Grid search over hyperparameters with reduction to specific number ∗ of attention heads; "ksize=*": Grid search over hyperparameters with reduction to specific kernel size ∗ of 1D CNN kernels for sequence embedding; "kernels=*": Grid search over hyperparameters with reduction to specific number ∗ of 1D CNN kernels for sequence embedding.