[Reviews · NeurIPS 2020]

Review 1

Summary and Contributions: Update after feedback: I have read the other reviews and author feedback. They do not change my view of the paper (accept). For a final version, I would suggest the authors to try to cite as much as possible from their other theoretical paper and tighten the focus of this one more on the application (as they already described in their feedback). ==== In this paper, the authors show an equivalence between the update rule for Hopfield networks and the commonly-used attention mechanism in transformer models. They also show that, under some conditions, the approach can be thought of as storing an exponential number of patterns. An attention-based (alternatively, Hopfield) neural network structure is then given to address multiple instance learning. Immune repertoire datasets are then used for experimental evaluation.

Strengths: The strengths of this work include establishing the theoretical connection between Hopfield networks and attention, as well as the result on the storage capacity. The experimental design is also thorough, in that it includes both state-of-the-art ML methods as well as reasonable bioinformatic sequence-based methods. Appropriate measures of variance are also given for the results.

Weaknesses: The connection between the theoretical developments and the specific NN structure seem tenuous. In particular, it seems that any existing transformer model using attention (e.g., [Yan et al., ACML 2018]) would also enjoy the same guarantees.

Correctness: I did not verify the proofs. The empirical methodology is reasonable.

Clarity: The paper is well-written; however, it would be helpful to define what is exactly meant by “immune repertoire classification” earlier (e.g., in the abstract).

Relation to Prior Work: The prior work is presented.

Reproducibility: Yes

Additional Feedback:


Review 2

Summary and Contributions: This paper uses a deep learning approach to classify the immune repertoire of individuals, a task that is posed as a multiple instance learning (MIL) problem. In this task, a model is learned that uses the set of amino acid sequences from the immune repertoire of an individual to classify the immune status of the individual with respect to one or more specific diseases of interest. For this task, it is desirable to be able to understand which sequences were responsible for the classification of any given individual. To address this task, the authors develop deep repertoire classification, DeepRC, which uses a transformer like attention mechanism to learn a representation of an input immune repertoire, and then learns a model between this representation and immune status labels. To evaluate the performance of the proposed method at this task, the authors carry out experiments that make us of synthetic datasets containing simulated and/or experimental sequences into which sequence motifs are inserted at frequencies between 0.01% and 10%, in addition to a real world dataset for which the immune status is known. Remarkably, an SVM with a minmax kernel was able to nearly match the performance of the modern Hopfield network across all simulated and LSTM-generated datasets. Performance was also matched across real world datasets with inserted signals at frequency 1%, but the simpler methods struggled when the inserted signals were reduced to 0.1%. Furthermore, DeepRC performance on the real world dataset results in an AUC of 0.832 \pm 0.002, compared to 0.825 \pm 0.022 for the SVM.

Strengths: This paper addresses an interesting problem, and carries out experiments that compare a number of approaches for addressing the machine learning task. The paper also contains a large body of theoretical work on the relationship between modern Hopfield networks and the attention mechanism of a transformer model. Both of these bodies of work may be of interest to portions of the NeurIPS community.

Weaknesses: This manuscript contains a highly theoretical analysis of modern Hopfield networks and their relationship to the attention mechanism of a transformer model. It also contains a deep model that addresses the machine learning task of immune repertoire classification. The major issue with this submission is that the connection between the two topics addressed in this paper, (i) classification of immune repertoires, and (ii) equivalence of the update rule of modern Hopfield networks and the attention mechanism of the transformer, is at best unclear. It feels as though two distinct papers have been condensed into one. Overall, combining these two results into one paper results in a main text manuscript that does not provide sufficient detail about either. It is not possible to follow the main text without referring to the page long notation table provided in the supplement. To address this, the extensive three page introduction of the paper should be made significantly shorter. Essential notation definitions should be present in the main text. Moreover, as written the paper is not accessible to people who are not familiar with Hopfield models. No definition or interpretation of the scale/temperature parameters that is present in equations 1 and 2 of the main text is provided, despite the presence of 86 pages of supplementary information. Why is this parameter necessary? What role does it play? Moreover, the term 'pattern' is used extensively, but not defined in the main text. What is the relationship between a pattern and an amino acid sequence? In section 2, patterns are stacked as columns to form a matrix. How do these patterns relate to the data points? What information does the term pattern add? Even in the supplement, details of how the patterns are learned from amino acid sequences (it appears that a pattern is simply a learned fixed length representation) are difficult to discover. The authors should succinctly explain the important points necessary for others to follow this work. Much of the main text is overly verbose and not necessary, and should be removed. In place, many important details should be extracted from the supplement, summarized and presented in the main text. A deep learning baseline that does not include the modern Hopfield model would be a welcome addition. The authors claim that they exploit the high storage capacity of the modern Hopfield model - but make no comment on the storage capacity of other models, either those included as baselines, or those that could be built. Please could the authors provide some evidence that (i) exponential storage capacity is needed, and (ii) the modern Hopfield model is the only means of achieving it. Does the transformer have this capacity? If so, why is the modern Hopfield model discussed? Given the empirical results, it would also be interesting if the authors could comment on the storage capacity of the SVM. Could the SVM approach be improved to obtain better performance? The datasets used for the empirical experiments should also be better described in the main text. Please also describe how the simulated datasets differ from the real world task.

Correctness: I have not checked the theoretical results presented in the extensive supplement to this manuscript. There is a typo in either table 1 or the main text for the DeepRC result on the real world dataset.

Clarity: My comments on how the paper is written are provided in the weaknesses section above.

Relation to Prior Work: There is a great deal of discussion of prior work. It would be useful to empirically compare the performance of existing methods on the experimental tasks presented in this paper.

Reproducibility: Yes

Additional Feedback:


Review 3

Summary and Contributions: The authors show that training a modern Hopfield network (one with continuous weights) gives an equivalent learning rule to the attention mechanism of transformer architectures. They provide a theoretical result for the storage capacity of a modern Hopfield network which shows that exponentially large numbers of patterns can be stored. They then show that a modern Hopfield network can be used within a neural network architecture for immune repertoire classification and provides best performance on a number of benchmarks datasets versus a range of alternative approaches.

Strengths: The result that the modern Hopfield network can store an exponentially large number of patterns is interesting and I’m not aware of a similar result in the literature. The theoretical appendix provides a good level of detail on the theoretical results which are just summarized briefly in the main paper. The mapping of the modern Hopfield learning rule to the attention mechanism of transformer architectures provides an interesting link from the modern to the classical neural networks literature which I think is of interest to the NeurIPS audience beyond this specific application. The benchmarking looks thorough (in terms of both datasets and methods) and the matching performance between the real data and simulated datas suggests that that simulated data is capturing the main features of the real task. The proposed method works very well in the case were a very low percentage of motifs is available (in the artificial case). Also, I don’t think the best performing alternative method (the MM SVM) has been applied to this problem previously and the benchmarking is more generally interesting if that is the case. The application area of classifying immune repertoires is of great interest to computational biologists and provides strong motivation for this class of problem.

Weaknesses: The theory is rather briefly described in the main paper yet it seems a very interesting result to be buried in what is essentially more of a methods/application paper. If these theoretical results are of more general interest to the community than is this the best publication option? Or are these results actually rather trivial and hence just provided as an appendix to this more application-oriented paper? I found it a bit difficult to switch gear from trying to understand the general theoretical issues to drilling down on the application details. Not sure if this is a theory paper motivated by an application (in which case we need more theory in the main text) or an application paper which just happens to need a lot of theoretical underpinning (as it is presented). The best performing SVM is identical in classification performance on many cases and only just misses out on the real data problem. It is therefore not completely clear how transformational the method will be for this specific application (on the other hand, I don’t think this particular SVM has been used for this problem before so this empirical result is also a contribution).

Correctness: As far as I can tell, but I'm afraid I could not go through all the theoretical details in the time I had to review this.

Clarity: I thought it was very clear and the detailed appendices were very thorough.

Relation to Prior Work: Yes, as far as I can tell.

Reproducibility: Yes

Additional Feedback:


Review 4

Summary and Contributions: The authors propose DeepRC, a deep NN architecture to classify patterns among a large set of instances. They apply a transformer-like attention-pooling instead of max-pooling and learn a classifier on the repertoire- rather than on the sequence-representation. Further they show that the attention mechanism of transformer architectures is actually the update rule of modern 
Hopfield networks that can store exponentially many patterns. 
This high storage capacity of modern Hopfield networks is exploited in DeepRC to solve the immune repertoire 
classification problem, a challenging multiple 
instance learning (MIL) problem in computational biology.

Strengths: The application domain of the work is sound and relevant. The paper is well written and well motivated.

Weaknesses: Comments: 1. Referring to Equations 1 and 4, what is the relationship between \eta and Q. 2. All along, Q, K and V are matrices but in Equation 4, \eta and V are 1-d vectors. What is the reasoning for this? 3. DeepTCR: Why is DeepRC not compared with DeepTCR? 4. The number of repertoires are from 700 - 5K: how long is each repertoire? How are duplicate repertoires handled? 5. Given these are large datasets, why is a CV factor of 5 chosen? Was DeepRC checked with higher CV factors? 6. From the results, SVM MinMax is close on the heels of DeepRC. Why would one consider DeepRC over SVMs? 7. Are the DeepRC-identified motifs biologically interpretable? Minor comments: MM stands for both multiple motifs as well as MinMax. There are typos in the last section of the paper.

Correctness: The formulation of the problem is correct.

Clarity: The paper is well written.

Relation to Prior Work: Prior work both in terms of the DL architecture as well as the classification models used for immune repertoires are discussed.

Reproducibility: No

Additional Feedback:

[Author Response · NeurIPS 2020]

We are very grateful to the reviewers for their positive feedback and constructive review of our paper, especially in the light of the interdisciplinary context and the extensive content. We first provide general comments: **(I) Connection of application and theory; extensive content.** We agree and are thankful that the reviewers see the importance of the findings. As reviewer 2 suggests, we have focused this work on the application of the theory to repertoire classification. We pursue a separate paper to investigate the theory in more detail. We will elaborate more on the connection between theory and application: The repertoires with 100,000s of instances require a powerful look-up model. We need a model that can store many patterns on which look-up is performed. The theory justifies the usage of modern Hopfield networks with their storage capacity for this challenging task. As reviewer 4 points out, we exploit this storage capacity to solve the immune repertoire classification problem. **(II) SVM.** We thank the reviewers for their interest in the SVM method and will include a more detailed analysis of it. The SVM method is suitable for fewer samples but fails at higher motif complexity due to rigid k-mer representation. DeepRC performs better at higher motif complexity (end-to-end DL with CNN/LSTM embedding offers high flexibility), e.g. datasets "13", "16" in "Simulated". Although in simple settings DeepRC and SVM are on par, DeepRC clearly outperforms SVM in more difficult settings. The difference is significant at $p$-values of $3\mathrm{e}{-66}$ ("Simulated"), $1\mathrm{e}{-222}$ ("Real-World data with implanted signals"), and $4\mathrm{e}{-121}$ (all 4 dataset categories) with McNemar's test. Nevertheless, we will add more datasets with higher motif complexity to further highlight this trend. **(III) Other compared methods.** We compare to all proposed methods for immune repertoire classification (burden test, logistic MIL) and several methods that we could adapt to this task. We emphasize that a neural net without the Hopfield attention mechanism is already included in the comparison (logistic MIL). Transformers are computationally not feasible because the query matrix would be quadratic in the number of instances. Nevertheless, we will include two other DL models with mean and max pooling instead of attention pooling.

**Reviewer 1** *"[. . . ] the authors show an equivalence between [. . . ] Hopfield networks and [. . . ] transformer models. [. . . ] Immune repertoire datasets are then used for experimental evaluation."* We thank the reviewer for outlining this – we also consider these our main contributions. *"The connection between the theoretical developments and the specific NN structure seem tenuous."* We will present this better (see I) to show that the connection is indeed strong. A fixed query is possible for modern Hopfield networks but not used in transformers. *"In particular, it seems that any existing transformer model using attention (e.g., [Yan et al., ACML 2018]) would also enjoy the same guarantees"* The reviewer is right. However, transformers are computationally not feasible here due to large matrix $\mathbf{Q}$, see III. We will cite and add this. *"The paper is well-written; [. . . ] helpful to define [. . . ] "immune repertoire classification" earlier [. . . ]"* We will define this term earlier in the manuscript.

**Reviewer 2** *SVM comparison.* See II. *Storage capacity of SVM and other methods.* How many patterns can be stored (shattered) in an NN or SVM is best given by the VC dimension. However, this is no working memory that can actively store instances. We are not aware of other models with this property, except transformers/Hopfield nets. *Why modern Hopfield?* Only Hopfield formalism allowed deriving the storage capacity. We are not aware of a rigorous proof for transformer models. *Comparison to existing methods; DL without Hopfield.* See III. *"[. . . ] connection between the two topics [. . . ] both of these bodies of work may be of interest [. . . ]."* See I. We agree the transition is not entirely smooth and will improve on that. *Temperature.* We will introduce the temperature parameter in the main text and add more information on Hopfield nets. *Term "pattern".* We will define and elaborate. A data point (=repertoire) consists of many instances (=AA sequences), each mapped to a fixed-sized vector (=pattern). "Pattern" in terminology of MIL corresponds to an instance or representation thereof. The modern Hopfield net stores patterns. *Description of datasets.* We will add the detailed description of the datasets and differences to the main paper. *Verbose introduction/text; notation.* We will condense the introduction and improve notations in the main text. *Other comments.* Thanks, we will adapt as suggested.

**Reviewer 3** *"The theory is rather briefly described in the main paper [. . . ]."* We will present the theory better (see I). *"[. . . ] matching performance between the real data and simulated data [. . . ]"* Yes, a really good catch: The simulation design is based on recent findings and expertise in computational and experimental immunology. *"If these theoretical results [are] rather trivial[. . . ]?"* No, the theoretical results are far from trivial and a new method (modern Hopfield net) is introduced (see I). *"Not sure if this is a theory paper [. . . ]"* It is an application paper presenting and using a new theory (see I). *"[. . . ] SVM [has] identical [. . . ] performance on many cases [. . . ] this particular SVM [is] also a contribution."* Actually, DeepRC outperforms SVM on tasks with complex motifs (see II); will elaborate. *Other comments.* Thanks, handled as suggested.

**Reviewer 4** *"[. . . ] relationship between [$\xi$] and $\mathbf{Q}$."* A $\mathbf{Q}$ with one row is $\xi^{T}$. *"[. . . ] $\mathbf{Q}$, $\mathbf{K}$ and $\mathbf{V}$ are matrices but in Equation 4, $\xi$ and $\mathbf{V}$ are 1-d vectors."* We will explain this better. In the transformer notation, single queries that are vectors become multiple queries that are matrices. *DeepTCR.* DeepTCR is not a method by itself but a collection of software building blocks. Nevertheless, we will include more compared methods (see III). *"[. . . ] how long is each repertoire?"* Simulated data: 316K, LSTM-generated: 285K, real-world with implanted signals: 10K, real-world: 299K (see Tables A2/A3 for details). *Duplicates.* We will provide details on handling duplicates. *Why CV factor 5? Also for DeepRC?* A CV procedure with 5 folds (also for DeepRC) was the border of computational feasibility. *"SVM MinMax is close on the heels of DeepRC. Why [. . . ] DeepRC over SVMs?"* Actually, there is a large gap between the methods in difficult settings (see II). DeepRC as a DL method requires a larger amount of samples. However, DeepRC, as an end-to-end DL model, can learn to detect more complex/flexible motifs with its CNN/LSTM sequence embedding. *"Are the DeepRC-identified motifs biologically interpretable?"* Yes. They may be interpreted as reproducible (across individuals) traces of immune responses. I.e. they have been useful in fighting a given disease, e.g. recognize pathogens. *Other comments.* Edited as suggested.

[Meta-Review · NeurIPS 2020]

The reviewers find the application compelling, to a timely topic, and with interesting theoretical connections that they now understand will primarily be presented elsewhere, and thus can now be cited in this NeurIPS paper, thereby enabling a cleaner exposition.